# Resistance training tempo selectively modulates corticospinal and reticulospinal excitability in humans

Yonas Akalu[1,2] , Jamie Tallent[1,3] , Ashlyn K. Frazer[1], Ummatul Siddique[1], Mohamad Rostami[1], Glyn Howatson[4,5] , Simon Walker[6] and Dawson J. Kidgell[1]

[1] Monash University Exercise Neuroplasticity Research Unit, School of Primary and Allied Care, Monash University, Frankston, Australia
[2] Department of Human Physiology, School of Medicine, University of Gondar, Gondar, Ethiopia
[3] School of Sport, Rehabilitation and Exercise Sciences, University of Essex, Colchester, UK
[4] Department of Sport, Exercise and Rehabilitation, Faculty of Health and Life Sciences, Northumbria University, Newcastle-upon-Tyne, UK
[5] Water Research Group, North West University, Potchefstroom, South Africa
[6] NeuroMuscular Research Center, Faculty of Sport and Health Sciences, University of Jyväskylä, Jyväskylä, Finland

Handling Editors: Richard Carson & Kathy Ruddy

The peer review history is available in the Supporting Information section of this article (https://doi.org/10.1113/JP289141#support-information-section).

**Abstract figure legend** Three weeks of unilateral resistance training (RT) performed under metronome-paced (MP-RT) or self-paced (SP-RT) conditions increased strength (maximum voluntary force (MVF) and one-repetition maximum (1RM)) but were accompanied by distinct neural adaptations. MP-RT enhanced corticospinal excitability (CSE) with reduced intracortical inhibition, reflected by reduced short-interval intracortical inhibition (SICI). In contrast, SP-RT increased cortico-reticular and reticulospinal excitability, indicated by increased ipsilateral to contralateral

motor-evoked potentials amplitude ratio (ICAR) and enhanced StartReact responses with a greater rate of force development (RFD). Together, these measures illustrate distinct neural adaptations to MP-RT and SP-RT and support a potential role for reticulospinal tract excitability in strength development.

**Abstract**    The neural mechanisms underlying resistance training (RT) adaptations remain incompletely understood, particularly the contribution of cortico-reticular and reticulospinal tracts (RSTs), which have rarely been examined in humans. One factor that may critically shape these adaptations is training pace, an overlooked yet physiologically relevant variable, as externally paced and self-paced contractions impose distinct demands on cortical and subcortical motor circuits. This study examined (1) how RT affects RST excitability and (2) whether different types of RT produce distinct changes in descending motor pathway excitability. Thirty healthy participants were randomized to metronome-paced RT (MP-RT), self-paced RT (SP-RT) or control. Cortical and corticospinal excitability were assessed with transcranial magnetic stimulation, while cortico-reticulospinal and RST excitability were measured using the ipsilateral to contralateral motor-evoked potential amplitude ratio (ICAR) and the StartReact effect. Training consisted of dumbbell exercises performed three times weekly for three weeks, with repeated neurophysiological testing. Both RT protocols improved one-repetition maximum and maximal voluntary force ($P < 0.001$). SP-RT resulted in greater increases in the StartReact effect ($P < 0.001$), rate of force development (0.0453) and cortico-reticulospinal excitability (ICAR; $P = 0.0351$), whereas MP-RT elicited larger increases in corticospinal excitability ($P = 0.00420$) and greater reductions in short-interval intracortical inhibition ($P < 0.001$). This study provides the first evidence in humans that RT modifies cortico-reticular and RST excitability. Importantly, adaptations were pacing-dependent: SP-RT selectively targets cortico-reticular circuit and RST, whereas MP-RT engaged corticospinal and inhibitory intracortical circuits. This pathway-specific plasticity underscores training modality as critical determinant of neural adaptation, with implications for rehabilitation and performance.

(Received 22 April 2025; accepted after revision 24 February 2026; first published online 19 March 2026)

**Corresponding author** Dawson J. Kidgell: Monash Exercise Neuroplasticity Research Unit, Department of Physiotherapy, School of Primary and Allied Health Care, Faculty of Medicine, Nursing and Health Science, Monash University, PO Box 527, Frankston, Melbourne, VIC 3199, Australia.    Email: dawson.kidgell@monash.edu

## Key points

- This study examined the effects of resistance training (RT) on descending motor pathways using transcranial magnetic stimulation and the StartReact protocol.
- Given the inaccessibility of the reticulospinal tract in humans, its excitability was inferred indirectly via the StartReact effect and ipsilateral to contralateral MEP amplitude ratio (ICAR), offering novel insight into cortico-reticulospinal modulation.
- Both metronome-paced and self-paced RT increased muscle strength but induced distinct neural adaptations.
- Self-paced RT engaged cortico-reticular and reticulospinal pathways, as indicated by elevated ICAR and StartReact effect values, while metronome-paced RT increased corticospinal excitability and reduced intracortical inhibition.
- These findings demonstrate, for the first time in humans, that RT pacing can differentially modulate corticospinal and reticulospinal excitability, two parallel descending motor systems with distinct roles. This selective pathway engagement suggests that training tempo is not merely behavioural but a physiologically meaningful factor for directing neuroplasticity.

## Introduction

Resistance training (RT) is one of the most effective strategies for improving and maintaining muscular strength, enhancing performance in tasks requiring force, power or speed, and supporting overall health and quality of life (American College of Sports Medicine, 2009). In the initial weeks of RT (<4 weeks), strength improvements are considered to be predominantly mediated by neural adaptations, although modest muscle architectural changes may also occur (Akima et al., 1999; Blazevich et al., 2007; Seynnes et al., 2007).

Although neural adaptations to short-term RT are well documented (Carroll et al., 2001; Sale, 1988), the precise mechanism underlying early strength gains remains unclear (Škarabot et al., 2021). Proposed mechanisms include increased cortical excitability, reduced intra-cortical inhibition, enhanced subcortical drive, increased spinal excitability and improved inter-muscular coordination through altered activation of antagonist and stabilizing muscles, yet the relative roles of corticospinal and reticulospinal pathways are poorly defined (Aagaard et al., 2002; Carolan & Cafarelli, 1992; Del Vecchio, Casolo, et al., 2019; Sale, 1988; Škarabot et al., 2021). Studies have shown reductions in short-interval intra-cortical inhibition (SICI) and cortical silent period (cSP) following RT (Kidgell et al., 2017; Siddique, Rahman, Frazer, Pearce, et al., 2020), whereas findings for cortico-spinal excitability (CSE) remain inconsistent, with reports of increases, decreases or no change (Ansdell et al., 2020; Beck et al., 2007; Carroll et al., 2002; Colomer-Poveda et al., 2021; Giboin et al., 2018; Griffin & Cafarelli, 2007; Hendy & Kidgell, 2013; Jensen et al., 2005; Leung et al., 2017). Notably, training modality has emerged as a critical determinant influencing the relationship between CSE and strength adaptation (Gómez-Feria et al., 2023).

Among training modalities, pacing, encompassing both synchronization and tempo, has emerged from recent evidence as a factor influencing neuromuscular adaptation (Gómez-Feria et al., 2023; Wilk et al., 2021). Metronome-paced RT (MP-RT), where tempo or repetitions are synchronized to an external cue, promotes attentional focus, precise force control, and enhanced sensory feedback, engaging higher-order cortical regions such as premotor and supplementary

motor areas and facilitating use-dependent corticospinal plasticity (Ackerley et al., 2011; Gerloff et al., 1998; Hortobágyi et al., 1997; Thaut et al., 2002). Consistent with this, MP-RT has been shown to increase CSE and reduce SICI (Ackerley et al., 2011; Gómez-Feria et al., 2023; Gordon et al., 2024; Leung et al., 2017). By contrast, self-paced RT (SP-RT), characterized by internally timed and unsynchronized repetitions, is often performed with faster concentric actions and less controlled eccentrics, reducing time under tension (TUT) and emphasizing force output over temporal precision (Gómez-Feria et al., 2023; Siddique, Rahman, Frazer, Pearce, et al., 2020). Such motor patterns tend to elicit more diffuse cortical and subcortical activation without the targeted facilitation observed in MP-RT (Gómez-Feria et al., 2023; Siddique, Rahman, Frazer, Pearce, et al., 2020), often yielding strength gains without measurable changes in CSE (Siddique, Rahman, Frazer, Leung, et al., 2020). These findings suggest that strength gains following SP-RT may be mediated by non-corticospinal pathways.

One potential candidate for mediating such adaptations is the reticulospinal tract (RST), which has been implicated in gross motor and forceful movements (Baker, 2011; Brownstone & Chopek, 2018), yet remains under-studied in humans because of its brainstem location. To address this, indirect methods such as the StartReact protocol and ipsilateral motor-evoked potentials (iMEPs) have been developed (Maitland & Baker, 2021; Sangari & Perez, 2020; Valls-Solé et al., 1995). Transcranial magnetic stimulation (TMS)-induced iMEPs are thought to reflect cortico-reticulospinal projections due to the RST's ipsilateral fibres, which convey cortical drive to the same-side limb (Alagona et al., 2001; Baker, 2011; Brownstone & Chopek, 2018; Fisher et al., 2013). Emerging evidence highlights RST involvement in human motor output. Functional magnetic resonance imaging has demonstrated increased activation of the reticular formation during force production (Danielson et al., 2024), and behavioural paradigms revealed that trained individuals exhibited shorter response latencies and greater rate of force development (RFD) to startling stimuli compared with untrained controls (Akalu et al., 2024). Similarly, rock climbers displayed enhanced RFD in response to startling stimuli, potentially reflecting reticulospinal contributions among other mechanisms

**Yonas Akalu** is a final-year PhD candidate at Monash University, Melbourne. His research explores the contribution of the reticulospinal tract to strength gains and how different resistance training modalities shape corticospinal and reticulospinal excitability, offering new insights into pathway-specific neural adaptations. His work highlights training pacing as a key neuromodulatory variable. He aims to translate these findings into neurorehabilitation, developing interventions that harness reticulospinal plasticity to restore strength and motor function after neurological injury, such as stroke or spinal cord injury.

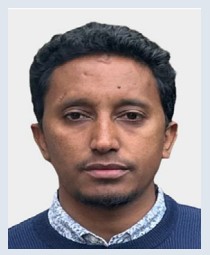

(Colomer-Poveda et al., 2023). In non-human primates, RT enhanced RST- and M1-evoked responses but not corticospinal tract (CST), suggesting plasticity primarily within reticulospinal and intracortical M1 circuits (Glover & Baker, 2020). Collectively, these findings implicate the RST as an important but under-studied substrate of strength adaptations.

Despite these indications, direct evidence of RST plasticity following RT is limited to non-human primates, and no study has examined how different RT modalities modulate RST excitability in humans. Given the proposed role of the RST in force production and its plasticity in non-human primates (Glover & Baker, 2020), this gap is particularly salient in light of evidence suggesting that training modality may influence the neural locus of adaptation. MP-RT appears to preferentially engage corticospinal circuits, whereas SP-RT yields strength gains without corticospinal changes, suggesting that the RST may underpin SP-RT adaptations.

Accordingly, the present study aimed to determine the effect of RT on RST excitability in humans, indexed by modulation of the StartReact effect and corresponding changes in RFD. The secondary aim was to examine whether MP-RT and SP-RT differentially modulate cortico-reticular (ipsilateral to contralateral motor-evoked potential (cMEP) amplitude ratio (ICAR), intracortical (SICI and intracortical facilitation (ICF)), and corticospinal (active motor threshold (AMT), CSE and cSP) excitability. We hypothesized that SP-RT, due to its reliance on internally generated motor commands, would preferentially enhance cortico-reticular and RST excitability, whereas MP-RT would predominantly modulate corticospinal pathways. By extending findings from animal models to human participants, this study provides evidence for pathway-specific plasticity following RT and suggests that the RST may contribute to early strength adaptations in human.

## Methods

### Ethical approval

The study received approval from the Monash University Human Research Ethics Committee (Project number: 40030) and was conducted in accordance with the *Declaration of Helsinki*. Following written and verbal explanations of the experimental protocol, all individuals provided written informed consent.

### Subjects

The minimum required sample size ($N = 30$; 10 per group) was determined *a priori* using G\*Power, version 3.1 (Faul et al., 2007), for a mixed (Group $\times$ Time; 3 groups $\times$ 4 time points) design, with $\alpha = 0.05$, and $1 - \beta = 0.80$. As no published interaction-based effect size was available for the primary outcome (StartReact effect), in a comparable repeated-measures design, sample size determination was informed by a theoretically related secondary outcome variable. Among the secondary outcomes examined, SICI provided an interaction-based effect size derived from a resistance-training study employing a comparable multigroup, multi-time-point repeated-measures design (Leung et al., 2017), making it the closest available methodological match for the present study design. A partial eta-squared value ($\eta^2_p = 0.359$) derived from that study was entered directly into GPower using the 'Determine → Direct' function, which converts $\eta^2_p$ to Cohen's $f$ following Cohen's (1988) recommendations, yielding an effect size of $f = 0.75$. Following institutional ethical approval, 30 healthy, RT-naïve adults (24 males, 6 females) were recruited. All participants were right-handed, as assessed by the Edinburgh Handedness Inventory (mean score: $88 \pm 14$; range: 50–100) (Oldfield, 1971).

Participants underwent comprehensive screening to confirm eligibility for TMS assessments and high-intensity RT. Exclusion criteria included musculoskeletal disorders, neurological conditions, a history of head trauma, concussion, seizures, epilepsy or use of medications affecting synaptic plasticity (e.g. antidepressants). Eligibility was confirmed via the TMS Adult Safety Screening Questionnaire (Keel et al., 2001), ensuring the absence of contraindications such as cranial implants or metallic devices (Rossi et al., 2009).

To mitigate circadian variations in neuromuscular performance, all sessions were conducted in the afternoon (12.00–20.00 h) at a consistent time, allowing a permissible variation of $\pm 2$ h (Douglas et al., 2021; Racinais et al., 2005). Training and testing sessions (three per week) were scheduled at intervals of no less than 48 h and no more than 3 days apart. Participants were instructed to abstain from unfamiliar vigorous physical activity for at least 36 h before each session and to avoid food intake within 2 h of testing. Additionally, to minimize potential influences on TMS responses, participants refrained from consuming caffeine, alcohol or other stimulants on test days and were advised to maintain consistent dietary and sleep habits throughout the study. For female participants, while complete control of hormonal fluctuations across the menstrual cycle was not feasible given the 3 week study duration, sessions were intentionally scheduled outside the early follicular (menstrual) phase. This was done to mitigate potential reductions in neuromuscular performance that have been reported during this phase (McNulty et al., 2020) and to minimize the likelihood of compromised adherence to testing and training protocols due to common menstruation-related symptoms such as fatigue,

dysmenorrhoea and impaired concentration. By doing so, we aimed to reduce variability in performance outcomes and ensure consistent engagement with the intervention. Participants were allocated 1:1:1 using sex-stratified, covariate-adaptive, computer-generated randomization, conducted independently within women ($n = 6$) and men ($n = 24$). Within each sex stratum, age and baseline strength were dichotomized at the sex-specific median, and the system assigned each participant to the group that minimized overall imbalance in group totals and covariate margins, yielding two women and eight men per group, thereby ensuring equal sex distribution and optimizing balance for age and baseline strength: MP-RT ($28 \pm 7$ years), SP-RT ($30 \pm 4$ years) and control ($27 \pm 8$ years).

### Experimental setup

Prior to the study, participants attended a familiarization session to ensure adaptation to experimental procedures. This session included anthropometric assessments (height and body mass), and one-repetition maximum (1RM) testing for dynamic elbow flexor strength (used solely for randomization). Participants were introduced to key methodologies, including 1RM and maximum voluntary force (MVF) measurements, the single-arm dumbbell curl exercise performed with the dominant (right) arm, TMS, surface electromyography (sEMG), peripheral nerve stimulation and the StartReact protocol. Additionally, participants underwent iMEP measurement familiarization to ensure correct execution of controlled bilateral elbow flexion–extension with a barbell. The MP-RT group received specific instruction to synchronize the RT movements with a metronome, adhering to a tempo of three seconds for the concentric (lifting) phase and four seconds for the eccentric (lowering) phase. These instructions were first introduced and practised during the dedicated familiarization session, where participants were trained until they could reliably synchronize with the metronome by maintaining the prescribed cadence.

One week later, baseline assessments were conducted, including muscular strength (1RM and MVF) and neurophysiological measures (cortical, corticospinal, cortico–reticulospinal and reticulospinal excitability). Following baseline testing, participants in the MP-RT and SP-RT groups completed a structured, progressive 3 week supervised unilateral RT programme. Training consisted of four sets of 6–8 repetitions at 70–75% 1RM, performed three times per week, with weekly 1RM reassessments to maintain progressive overload. Strength and neurological assessments were conducted during the first and last sessions of Week 1 and following the final session of each subsequent week.

The control group underwent identical laboratory visits and assessment protocols but did not participate in RT.

Each testing session lasted ∼40 min, while supervised training sessions were ∼15–20 min (Fig. 1).

### Surface electromyography

sEMG recordings were obtained from the dominant arm's biceps brachii at baseline and at the end of Weeks 1, 2 and 3. Participants sat upright in an adjustable chair with back support; hips and knees at 90°; feet flat. The trunk and shoulder girdle were stabilized using non-elastic pelvic and shoulder straps to minimize compensatory movements. The dominant shoulder was neutral (≈0–10° abduction and flexion) with the scapula gently retracted/depressed and the humerus alongside the torso on a support. The dominant elbow was fixed at 90° with the lateral epicondyle aligned with the arm-bar; the forearm was fully supinated (to preferentially load biceps brachii) and the wrist held neutral (0° flexion/extension; 0° radial/ulnar deviation) using a light brace to minimize wrist-flexor contribution. The forearm midline was aligned with the force axis, and the hand and forearm rested on a fixed arm-bar connected to a force transducer (LSB302 S-beam load cell, FUTEK Advanced Sensor Technology Inc., Irvine, CA, USA). Two wide non-compliant nylon straps were applied to the distal forearm (≈2–3 cm proximal to the radial/ulnar styloid processes) and tightened against the arm-bar to prevent translation and pronation–supination without impeding distal circulation. The non-testing arm rested comfortably on the lap. Investigators provided standardized verbal cues and continuous visual monitoring to avoid shoulder elevation, trunk motion, wrist flexion or forearm pronation; any compensated trial was stopped, the position reset and the attempt repeated. The same positioning and stabilization procedures were applied across all testing, including MVF, TMS and StartReact protocol (startReact effect and RFD). To standardize positioning across trials and visits, all measurement settings were documented for each participant and replicated in subsequent trials.

Bipolar Ag-AgCl surface electrodes (Medi-Trace, Graphic Controls LLC, Buffalo, NY, USA) were placed along the muscle belly with a 2 cm interelectrode distance, positioned one-third of the distance between the antecubital fossa and the acromion process (Siddique et al., 2024). Electrodes were aligned along the anatomical axis from the medial acromion to the antecubital fossa to ensure optimal signal acquisition. Skin preparation included cleansing with 70% isopropyl alcohol and gentle abrasion to minimize impedance and enhance signal transmission (Gilmore & Meyers, 1983). Electrodes were secured with adhesive tape to reduce motion artefacts. Additional electrodes were placed on the non-dominant biceps brachii for iMEP recordings, with a grounding

strap positioned around the left wrist to minimize electrical interference. Electrode placement and signal acquisition adhered to European SENIAM guidelines for standardized, non-invasive muscle assessment (Hermens et al., 1999). sEMG signals were recorded using an Octal Bio Amp (ML138) connected to a PowerLab 4/26 data acquisition system, both from ADInstruments (Bella Vista, Australia). Signals were amplified (×1000), band-pass filtered (13–1000 Hz) and digitized in real time at 20 kHz. Data acquisition and analysis were performed using LabChart software (ADInstruments, Bella Vista, Australia). To standardize visual feedback, sEMG and force output were displayed on a monitor positioned 1 m away at eye level, ensuring real-time monitoring and task execution.

## StartReact protocol

The StartReact protocol was used to assess reticulospinal excitability, an indirect method for investigating rapid motor responses triggered by startling stimuli (Tapia et al., 2022). This validated approach infers the involvement of the ponto-medullary reticular formation by measuring reaction time (latency) during a pre-planned motor task performed in response to an imperative visual cue that is unpredictably paired with either a loud, soft or no auditory stimulus (Baker & Perez, 2017; Tapia et al., 2022). The

protocol followed standardized procedures as previously established (Baker & Perez, 2017; Sangari & Perez, 2020).

For the StartReact protocol, in which response latency was the primary outcome and RFD was additionally quantified, participants were positioned and stabilized in the same manner as described for sEMG measurement, with the same electrode placement at the biceps brachii. They were instructed to maintain visual fixation on a light-emitting diode positioned at eye level, 1 m in front of them. Upon light illumination (20 ms duration), participants were required to execute a maximal isometric elbow flexion or push the force transducer as fast and as forcefully as possible, then relax without sustaining maximal force (Anzak et al., 2011; Colomer-Poveda et al., 2023; Del Vecchio, Negro, et al., 2019). To ensure that each contraction was performed with maximal explosive effort, participants received immediate visual feedback after each contraction along with verbal encouragement between trials (Maffiuletti et al., 2016). The visual cue was randomly paired with one of three auditory conditions: a loud startling sound (visual-startling stimuli (VSS)) (115 dB, 500 Hz, 50 ms), a soft sound (visual non-startling stimuli (VnSS)) (80 dB, 500 Hz, 50 ms) or a control condition with no auditory stimulus (visual stimuli (VS)). All auditory stimuli were delivered from a speaker positioned 1 m behind the participant. Each session comprised 30 trials (10 per condition) of rapid isovolumetric elbow flexion, with inter-trial intervals

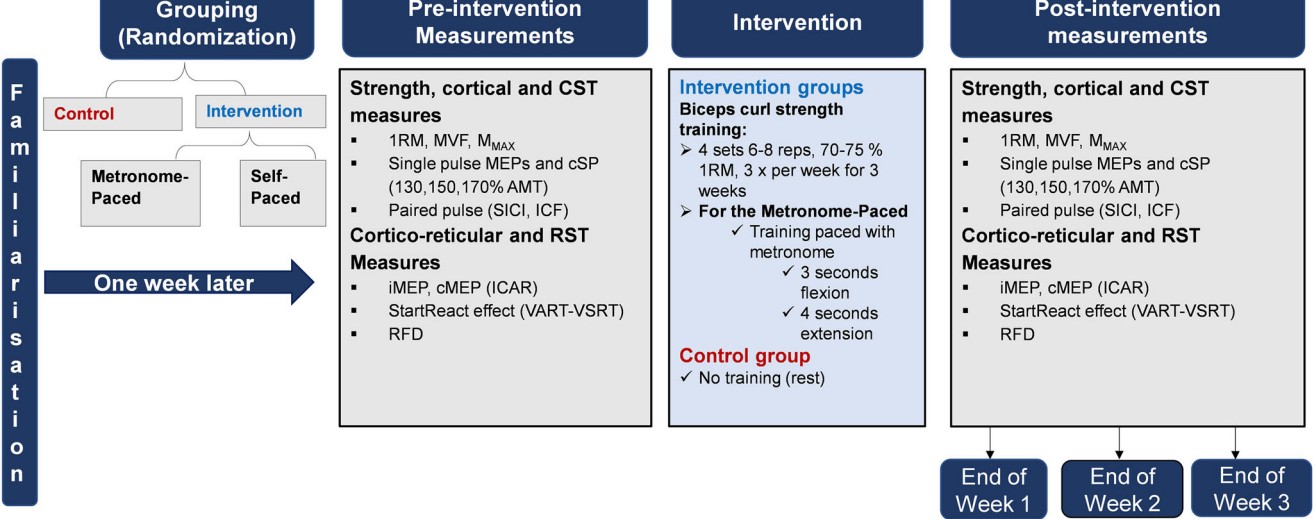

**Figure 1. Schematic representation of the experimental design**
Measurements of cortical, corticospinal, cortico-reticular and reticulospinal excitability were obtained at baseline and after each training week (Weeks 1–3) of MP-RT and SP-RT. 1RM, one-repetition maximum; AMT, active motor threshold; cMEP, contralateral motor-evoked potential; cSP, cortical silent period; ICAR, ipsilateral to contralateral motor-evoked potential amplitude ratio; ICF, intracortical facilitation; iMEP, ipsilateral motor-evoked potential; $M_{MAX}$, maximum M-wave; MEPs, motor-evoked potentials; MP-RT, metronome-paced resistance training; MVF, maximum voluntary force; RFD, rate of force development; RST, reticulospinal tract; SICI, short-interval intracortical inhibition; SP-RT, self-paced resistance training; VART, visual-auditory reaction time; VSRT, visual-startle reaction time.

ranging from 3 to 6 s to minimize predictability and reduce the likelihood of habituation. Participants were continuously monitored to ensure the correct execution of all rapid isometric elbow contractions. They were explicitly instructed to avoid any anticipatory muscle tension or counter-movements before the imperative 'go' signal (i.e. LED illumination). Additionally, to confirm task proficiency or to get used to producing the force as quickly as possible, they performed three maximal isometric biceps flexion contractions in response to the visual stimulus before the formal testing phase (Škarabot et al., 2022). RFD was analysed as the slope of the force–time curve from the onset up to 50 ms and from 50 to 100 ms of contraction following prior methodologies (Akalu et al., 2024; Colomer-Poveda et al., 2023).

Prior to the experimental trials, participants underwent a familiarization phase to ensure appropriate responses to the startle stimulus, as per established protocols (Colomer-Poveda et al., 2023; Sangari & Perez, 2020).

The StartReact protocol, measurement of reaction time and RFD, was conducted at baseline, at the end of Week 1, Week 2 and Week 3 over 3 weeks.

To examine test–retest reliability before commencing the main study, a pilot assessment was conducted in eight healthy adults at two time points 1 week apart. These participants did not take part in the main trial. The intraclass correlation coefficients (ICCs) for visual–auditory reaction time (VART) and visual–startle reaction time (VSRT) were 0.60 and 0.86, respectively, consistent with previous reliability reports (Colomer-Poveda et al., 2023). Given the moderate reproducibility observed for VART, reproducibility of StartReact measurements was subsequently reassessed within the main study across four time points (Baseline, Week 1, Week 2 and Week 3) to provide a more comprehensive evaluation of reliability.

### Maximum compound action potential

To standardize TMS responses, the maximal compound muscle action potential (M-wave), a direct muscle response, was recorded from the right biceps brachii via electrical stimulation of the brachial plexus at Erb's point. A DS7A Bipolar constant current stimulator (DS7A, Digitimer, Hertfordshire, UK) delivered pulses of 200 μs duration through surface electrodes (diameter: 3.2 cm, Axelgaard Manufacturing Co., LTD). The cathode was positioned in the supraclavicular fossa (Erb's point), while the anode was placed over the acromion. To elicit a direct muscle response, low-intensity stimulation was initially applied while participants sustained a submaximal isometric contraction at 10% of MVF. This approach ensured reliable M-wave recordings by maintaining stable conduction velocity and sarcolemma excitability under low-intensity contraction conditions

(Rodriguez-Falces & Place, 2021). The stimulus intensity was systematically increased in 5 mA increments until a plateau in M-wave amplitude was reached, marking the maximal M-wave ($M_{MAX}$), determined from the peak-to-peak sEMG amplitude. To verify the accuracy of this measurement, the stimulation intensity was further increased by 20%, ensuring no additional increase in M-wave amplitude, thereby confirming the attainment of $M_{MAX}$. Each stimulus was administered at intervals of 6–9 s. This methodology has demonstrated high reliability (ICC = 0.92) (Walker et al., 2013). $M_{MAX}$ measurements were performed at baseline and reassessed at end of Week 1 (after three sessions), Week 2 (after six sessions) and Week 3 (after nine sessions) to evaluate potential changes in muscle excitability that could influence motor-evoked potential (MEP) amplitudes.

### Strength measurements

Given the structural similarity between the unilateral 1RM assessment and the RT protocol, it was hypothesized that the 1RM test is influenced by learning effects (Rutherford & Jones, 1986). To control for this, MVF testing was incorporated to distinguish training-induced strength adaptations from potential learning effects. Isometric (MVF) and dynamic (1RM) strength assessments of the dominant biceps brachii were conducted at baseline, following 1, 2 and 3 weeks of RT.

For unilateral MVF assessment, participants adopted the same seated position described for sEMG recordings, with the elbow fixed at 90° flexion while exerting force against a transducer positioned on a height-adjustable table. Force was recorded using a force transducer, sampled at 2000 Hz. Signals were baseline-corrected and low-pass filtered at 12 Hz using a fourth-order zero-lag Butterworth digital filter to remove high-frequency noise while preserving physiological force fluctuations (Hewett et al., 2017). Participants performed a 3 s isometric contraction with standardized verbal encouragement, while real-time force feedback was displayed on a monitor 1 m ahead. Trials were discarded if participants applied submaximal effort (Gandevia, 2001). Two trials were conducted, with a 3 min inter-trial rest to minimize fatigue, and the highest recorded force was designated as MVF. If inter-trial variability exceeded 5%, an additional trial was conducted to enhance reliability (Lahouti et al., 2019; Škarabot et al., 2022). The final MVF value was used to establish a 10% target force level for $M_{MAX}$ and subsequent TMS neurophysiological assessments.

For unilateral 1RM testing, participants stood upright with their back against a wall and the opposite arm placed behind their back to prevent trunk and shoulder movement. The dominant shoulder was positioned in adduction and neutral flexion/extension, with the upper

arm maintained alongside the torso throughout the test. The forearm was fully supinated, and each attempt began from full elbow extension (∼0°) and continued through the full elbow-flexion range of motion until the dumbbell approached the shoulder. Range of motion and technique were visually assessed by the investigator, and any trial with shoulder displacement or incomplete range of motion was repeated. Initial load estimation was based on self-reported strength. Each attempt was performed as a maximal-effort elbow flexion, with standardized verbal encouragement provided to promote consistent maximal performance. Successful lifts resulted in progressive weight increments of 0.25–0.5 kg, with 3 min rest intervals between attempts to prevent fatigue and ensure maximal force output. The heaviest successfully lifted load, performed with full range of motion and proper technique, was recorded as 1RM (Jensen et al., 2005). This protocol demonstrated high reliability (ICC = 0.980) (Kidgell et al., 2010). The final 1RM value was used to prescribe training loads for the MP-RT and SP-RT groups.

### Resistance training protocol

Following baseline assessments, participants in the intervention groups (MP-RT and SP-RT) completed nine sessions of high-load unilateral RT over 3 weeks (three sessions per week). Training for both groups consisted of the same supervised unilateral dumbbell biceps curl described for 1RM testing, performed under identical positioning and stabilization procedures to minimize non-target contributions. Training loads were prescribed at 70–75% of each participant's 1RM, in accordance with established recommendations for novice strength training (American College of Sports Medicine, 2009), and were progressively adjusted across sessions by increasing the load 2–5% once participants successfully completed all repetitions with proper technique. Repetitions were executed either to a metronome (MP-RT) or at a self-selected pace (SP-RT), with each set performed continuously without unloading between repetitions to maintain constant muscle tension throughout the movement. Training was conducted under the same movement control and monitoring procedures as described for 1RM testing, ensuring correct range of motion and the absence of compensatory movements. To enhance joint mobility, neuromuscular activation and tissue elasticity while minimizing fatigue, a 3–4 min warm-up was performed before each session, consisting of light dumbbell biceps curls (∼40% of 1RM) with slow eccentric control and dynamic arm swings (McCrary et al., 2015).

The MP-RT group performed four sets of 6–8 repetitions of supervised unilateral biceps curls at 70–75% 1RM. Repetitions were performed to a standardized

tempo, paced by an audible metronome, with a 3 s concentric (lifting) and 4 s eccentric (lowering) phase, and 2 min inter-set rest for recovery. This cadence was selected to reliably synchronize with the metronome and ensure controlled execution, minimize momentum and enhance reproducibility across participants. To ensure proper adherence, the investigator closely supervised performance, provided verbal feedback and immediately corrected and repeated any trial that deviated from the prescribed cadence or metronome beat.

Participants in the SP-RT group performed the same supervised unilateral dumbbell curl protocol (four sets of 6–8 repetitions at 70–75% 1RM, 2 min rest) but without an externally imposed tempo. They were instructed to flex their elbow and lift the dumbbell as they would during a standard biceps curl, at a natural self-selected pace, while maintaining correct form and full range of motion. All sessions were supervised by the investigator, with verbal feedback provided to minimize compensatory movements and ensure consistency of execution. On the other hand, the control group remained seated at rest for 15 min per session, matching the duration of the training period for the intervention groups.

TUT during RT was measured across the four training sets using a digital stopwatch (Wessel et al., 1997). For each set, timing commenced at the start of the first repetition and stopped at the end of the final repetition, with the elapsed time recorded as the set TUT. To ensure consistency, all measurements were conducted by the same investigator.

### Transcranial magnetic stimulation

A non-invasive brain stimulation technique, transcranial magnetic stimulation (TMS), was used to examine cortical, corticospinal and cortico-reticulospinal excitability at baseline, after Week 1, Week 2 and Week 3. MEPs were recorded from the biceps brachii of the dominant arm through TMS of the corresponding motor cortical representation. Stimulation was delivered using a circular coil connected to two Magstim 200$^2$ stimulators (Magstim Co. Ltd., Whitland, Dyfed, UK). To ensure precise control over stimulation parameters, both stimulators were utilized for single- and paired-pulse paradigms, maintaining equivalent intensities across conditions. The stimulation protocol was configured to preserve identical stimulation intensities between single- and paired-pulse applications, ensuring methodological consistency in assessing CSE (Do et al., 2020).

The TMS procedure involved positioning a 90 mm circular coil at a 45° angle to the sagittal plane, with the handle directed posteriorly to induce a posterior-to-anterior cortical current flow. The optimal stimulation site for eliciting MEPs in the biceps brachii

was identified by systematically adjusting the coil position in 0.5 cm increments across the motor cortex until the site producing the maximal peak-to-peak MEP amplitude at a given stimulus intensity was located. This motor hotspot was marked on the scalp to ensure consistent coil placement across all testing sessions. Once the motor hotspot was established, it was marked on the scalp with a permanent marker and re-boldened at each session to ensure consistent coil placement across the 3 week intervention. Active motor threshold (AMT) was defined as the lowest stimulus intensity that evoked MEPs >200 µV in at least five of 10 consecutive trials (Akalu et al., 2024; Cohen et al., 1998), with stimulator output adjusted in 1% increments. Baseline AMT was re-evaluated at the end of Weeks 1, 2 and 3 to monitor potential changes and were adjusted if required over the training period. Each assessment included 10 trials to ensure measurement reliability and consistency. All TMS pulses were delivered at fixed 10 s intervals, with isometric contractions initiated by a soft metronome cue; upon hearing the sound, participants increased to the target 10% MVF level and stimulation was applied once a stable level was achieved (Kidgell et al., 2010). All single- and paired-pulse stimuli were delivered during a submaximal voluntary contraction of right biceps brachii at 10% of the participant's MVF, following established neuromodulation methodologies (Rossini et al., 2015; Weier et al., 2012). To account for potential variations in pre-stimulus muscle activation, root mean square EMG (rmsEMG) data were extracted from a 100 ms epoch immediately preceding each TMS pulse.

CSE was assessed using a single-pulse TMS protocol, wherein area under the recruitment curve (AURC) was calculated by analysing MEP amplitudes elicited at stimulation intensities of 130%, 150% and 170% of AMT (Mason et al., 2020; Woodhead et al., 2024). These intensities were selected to sample the steep-to-plateau region of the corticospinal input–output curve, providing a sensitive index of excitability while avoiding the excessive repetitions and fatigue that a full 5% incremental recruitment curve during 10% MVF contractions would introduce within a protocol already containing several neurophysiological assessments. In addition, corticospinal inhibition was examined through the cSP, which reflects GABA$_B$ receptor-mediated inhibitory neurotransmission. It is defined as the duration between MEP onset and the re-emergence of EMG activity (Damron et al., 2008). To quantify cSP modulation across intensities, the AURC was calculated, providing an integrated measure of corticospinal inhibition dynamics, using the duration of cSP for the stimulation intensities corresponding to 130%, 150% and 170% of the AMT.

To investigate intracortical inhibitory and excitatory mechanisms, paired-pulse TMS paradigms were applied. SICI, which reflects GABA$_A$ receptor-mediated inhibitory neurotransmission, was assessed by delivering a sub-threshold conditioning stimulus (CS) at 80% of AMT, followed by a suprathreshold test stimulus (TS) at 130% of AMT, with a 3 ms interstimulus interval. ICF, indicative of glutamatergic excitatory activity, was evaluated using identical CS and TS intensities but with a 10 ms inter-stimulus interval. To mitigate the effects of neuro-muscular fatigue, a standardized 1 min rest interval was incorporated between testing blocks.

## iMEP measurement

Participants were seated on a Nirvana Preacher Curl Bench (RitFit, Memphis, TN, USA) and instructed to perform controlled bilateral elbow flexion and extension movements while lifting and lowering a barbell at a standardized tempo of 5 s per repetition. Bilateral activation may enhance iMEP detection, as studies suggest iMEPs are more pronounced during bimanual cooperative tasks than non-cooperative ones (Altermatt et al., 2023). Throughout the movement, TMS was delivered using a monophasic current waveform through a flat 70 mm figure-of-eight magnetic coil (D70 Alpha Flat Coated coil) connected to two Magstim 200$^2$ stimulators (Magstim Co. Ltd., Whitland, Dyfed, UK). For iMEP and cMEP recordings, the figure-of-eight coil was preferred to provide greater focality and minimize cross-hemispheric spread, ensuring that responses reflected true ipsilateral rather than contralateral activation. The coil was positioned tangentially on the scalp at a 45° angle to the midsagittal plane, generating a posterior-to-anterior current flow.

The optimal stimulation site was initially located approximately 3 cm lateral to the left of the vertex and systematically adjusted in ∼0.5 cm increments until the maximal cMEP response was elicited at 50% of the maximum stimulator output. Fine-tuning adjustments of 1–2 cm in the anteroposterior direction were performed to precisely localize the optimal cortical representation. Once the site eliciting the largest cMEP amplitude was identified, it was designated as the motor hotspot and marked to ensure consistent coil positioning across trials.

At the identified motor hotspot, TMS was applied at 100% of the maximum stimulator output to evoke iMEPs in the biceps brachii of the stimulated hemisphere and cMEPs in the opposite biceps (Seusing et al., 2024). Stimulation was manually triggered by the experimenter and time-locked to the concentric phase of the movement when the elbow reached 110 ± 2° of flexion (Hu et al., 2024). To ensure precision in movement execution, real-time monitoring of elbow joint kinematics was displayed on a computer screen positioned 1 m in front of the participant. Joint angle tracking was performed using electro-mechanical goniometers

(MLTS700, ADInstruments, Bella Vista, Australia) affixed to the lateral aspect of the elbow joint, following the manufacturer's specifications (Biometrics, 1997).

Each participant completed two sets of five repetitions, with a standardized 2 min rest interval between sets, yielding a total of 10 TMS pulses per session to elicit iMEPs and cMEPs. To standardize resistance across individuals, the test load was individualized to 30% of each participant's unilateral 1RM. None of the participants reported symptoms of muscular fatigue or difficulty completing the full set of 10 repetitions.

## Data analysis

All force, peripheral nerve stimulation, TMS and StartReact data were recorded and analysed using LabChart 8 software (ADInstruments, Bella Vista, Australia). Each recording was examined in high resolution to accurately identify the onset and termination of the MEP or M-wave. Pre-stimulus rmsEMG activity of the dominant biceps brachii was assessed within 100 ms prior to each TMS stimulus. Trials were retained when pre-stimulus rmsEMG fell within $10 \pm 2\%$ of the participant's maximal rmsEMG (computed from a stable MVF plateau), ensuring background activation close to the 10% MVF target; otherwise, the trial was discarded and repeated. MVF and 1RM's rmsEMG were recorded and training load volume for each participant was quantified using the formula: Training Volume–Load = sets × repetitions × load. Total RT TUT was calculated by summing the TUT of all four sets within each session and then across the nine training sessions (i.e. Total TUT = Σ set TUT × 4 sets × 9 sessions). The peak-to-peak amplitude of $M_{MAX}$ and MEPs recorded from the contra-lateral biceps brachii was measured offline within a 10–50 ms window post-stimulation at each of the three stimulation intensities (130%, 150% and 170% of AMT). To account for potential peripheral changes, MEP amplitudes were normalized to $M_{MAX}$ and scaled by a factor of 100. Therefore, MEPs recorded in mV were expressed as a percentage of $M_{MAX}$. The cSP was assessed at each stimulation intensity by determining the duration between MEP onset and the return of continuous sEMG activity, which was visually inspected and manually marked. This was achieved by setting horizontal cursors at the maximum and minimum pre-stimulus sEMG levels and identifying the time point when sEMG activity crossed these thresholds following the silent period. To minimize variability and enhance the reliability of cortical inhibition measurements, the duration of the cSP was normalized to the corresponding MEP amplitude (Orth & Rothwell, 2004).

The total AURC was computed for the two conditions separately: cSP, calculated from cSP duration recruitment curves across the three stimulation intensities; and CSE, based on MEP amplitude recruitment curves in the dominant (right) biceps brachii. Both AURC measures were derived using the trapezoidal integration method, and the experimenter remained blinded to group allocation throughout the analysis. The trapezoidal integration formula used for AURC calculation was:

$$\text{AURC} = \sum_{i=1}^{n-1} \frac{(y_i + y_{i+1})}{2} \Delta x, \tag{1}$$

where $y_i$ and $y_{i+1}$ represent MEP amplitudes or cSP duration at successive points, and $(\Delta x)$ is the interval between these points. This formula calculates the AURC by averaging the MEP amplitudes or cSP duration values at two successive points, $y_i$ and $y_{i+1}$, and then multiplying by the interval width $\Delta x$, thereby providing an accurate estimation of recruitment dynamics across intensities.

SICI and ICF were expressed as a percentage of the unconditioned single-pulse MEP amplitude (Leung et al., 2017; Siddique, Rahman, Frazer, Leung, et al., 2020). Specifically, the MEP amplitudes for SICI and ICF were divided by the MEP amplitude elicited through single-pulse TMS at 130% of AMT and subsequently multiplied by 100.

To evaluate cortico-reticular excitability, iMEPs and cMEPs were recorded from the left and right biceps brachii muscles, respectively. The latency difference between these responses was analysed to determine whether the iMEP reflected true reticulospinal activation or was instead the result of direct transcallosal transmission. Specifically, an iMEP - cMEP latency difference of less than 5 ms was considered indicative of direct cortico-spinal activation of the contralateral hemisphere, rather than a true iMEP response (Ziemann et al., 1999). In contrast, latency differences ranging from 5 to 13 ms were considered characteristic of subcortical transmission via the RST (Maitland & Baker, 2021; Ziemann et al., 1999). Based on these criteria, data points in which cMEP onset preceded iMEP by less than 5 ms were excluded from further analysis to eliminate potential artefacts related to transcallosal activation (Maitland & Baker, 2021).

To quantify the relative excitability of corticospinal and reticulospinal pathways during contraction, we employed the iMEP-to-cMEP amplitude ratio (ICAR) as previously described (Bawa et al., 2004). This metric serves as an index of the cortico-reticular (iMEP) excitability relative to CSE (cMEP) during bilateral upper limb contraction. Since ICAR represents the ratio of iMEP to cMEP, an increase in ICAR does not necessarily indicate an absolute increase in iMEP but may instead result from a relative reduction in cMEP amplitude. To distinguish these potential contributions, we further analysed the independent changes in iMEP and cMEP amplitudes across conditions.

A customized macro in the LabChart software (ADInstruments, Bella Vista, Australia) was utilized for processing and analysing StartReact data. To ensure accuracy in detecting sEMG and force onset, each trial was individually reviewed for potential artefact-related errors, including those caused by electrical noise or pre-stimulus muscle activity. In cases where the software misidentified force or sEMG onset, manual corrections were performed to eliminate detection inaccuracies. The StartReact protocol was used to assess reaction times following startling and non-startling stimuli, specifically: VSRT, the latency from the onset of the light and startling auditory stimulus to the initiation of the sEMG burst in the biceps brachii muscle. Similarly, VART was the time interval between VnSS and the onset of the sEMG burst. sEMG onset was identified when the rectified signal exceeded $\pm 3$ SD above baseline pre-stimulus activity, measured over a 200 ms epoch (Baker & Perez, 2017; Colomer-Poveda et al., 2023). Reaction times exceeding 700 ms were excluded from analysis to mitigate outlier influence (Akalu et al., 2024; Sangari & Perez, 2020).

To evaluate RST excitability, we analysed the extent to which startling stimuli modulated reaction times, as RST activation is hypothesized to play a role in this facilitation (Tapia et al., 2022; Valls-Solé et al., 1999). This RST-mediated enhancement of reaction time has been previously validated (Tapia et al., 2022) and is believed to result from the subcortical release of stored movement plans upon exposure to a startling stimulus (Rothwell, 2006). Reticulospinal activity was examined by computing the StartReact effect, defined as the difference between VART and VSRT (StartReact effect = VART − VSRT) (Carlsen & Maslovat, 2019; Germann & Baker, 2021; Sangari & Perez, 2019). This metric serves as a primary indicator of RST functionality, reflecting the rapid, involuntary initiation of pre-programmed movements and its role in facilitating reaction time acceleration in response to startling stimuli (Carlsen & Maslovat, 2019; Germann & Baker, 2021; Sangari & Perez, 2019). The difference between VART and VSRT provides insight into reticulospinal contribution to movement execution and aligns with previous studies suggesting that RST activation underlies the StartReact effect (Baker & Perez, 2017; Carlsen et al., 2004; Valls-Solé et al., 1999).

RFD was calculated over the 0–50 and 50–100 ms intervals following contraction onset. Force signals were sampled at 2000 Hz, baseline-corrected and low-pass filtered at 12 Hz using a fourth-order, zero-lag Butterworth filter to minimize high-frequency noise. The forearm and dorsal surface of the hand were fully supported, the force channel was zeroed before testing, and participants were instructed to remain completely relaxed prior to the cue, with real-time visual feedback confirming the absence of pre-tension. During the 200 ms pre-stimulus period, baseline force remained effectively at zero with negligible variability, and all trials were visually inspected to exclude those exhibiting drift, pre-tension or counter-movement (Del Vecchio et al., 2018; Lecce et al., 2025). Contraction onset (time = 0 ms) was defined as the first point at which the force signal exceeded three SDs above the pre-stimulus baseline (Akalu et al., 2024; Colomer-Poveda et al., 2023; Dos'Santos et al., 2017). RFD values were then calculated relative to this detected onset

## Statistical analysis

All statistical analyses were performed using R (version 4.4.1; R Foundation for Statistical Computing, Vienna, Austria, 2024). Details regarding the R environment, including package versions, are available upon request. GraphPad Prism version 10.1.2 (324) for Windows (GraphPad Software Inc., San Diego, CA, USA) was used to create graphs.

Descriptive statistics were computed for age, body mass, MVF, 1RM and $M_{MAX}$, with results expressed as means $\pm$ SD. Prior to statistical modelling, data were screened for outliers. Normality assumptions were assessed using visual inspection of Q–Q plots and histograms of model residuals, complemented by the Shapiro–Wilk test applied to residuals. Homogeneity of variance was evaluated using Levene's test and residual *versus* fitted value plots. To assess baseline differences across groups, a one-way analysis of variance (ANOVA) was performed for the normally distributed baseline data (M-wave duration, MVF, 1RM, ICF, SICI, RFD and ICAR) and the Kruskal–Wallis test for the non-normally distributed baseline data (StartReact effect, cSP, CSE, ICF, single-pulse rmsEMG, paired-pulse rmsEMG and StartReact rmsEMG).

To evaluate changes in cortico-reticulospinal (ICAR, StartReact effect and RFD), cortical (SICI and ICF) and corticospinal (AMT, cSP and CSE) excitability across groups and time points, while accounting for potential missing data and baseline variability, both generalized linear mixed model (GLMM) and a linear mixed model for repeated measures ($LMM_{RM}$) were employed (Wilkinson et al., 2023). Variables that did not conform to normal distribution despite transformation (MVF, 1RM, $M_{MAX}$, AMT, SICI, ICF, cMEP, cSP, paired-pulse rmsEMG, StartReact effect and RFD) were analysed using a GLMM. Datasets that met normality criteria (ICAR, CSE, iMEP, M-wave duration and single-pulse rmsEMG) were analysed using an $LMM_{RM}$.

The models incorporated the fixed effects: type of RT (MP-RT, SP-RT and Control), time (Baseline, end of Week 1, Week 2 and Week 3) and, for StartReact data alone, sound conditions (VS, VnSS and VSS). Subjects

were modelled as a random variable to account for inter-individual variability. Repeated measures factors: StartReact sound conditions (VS, VnSS and VSS) and time (end of Week 1, Week 2 and Week 3) were included to account for within-subject dependencies. Where a significant Group × Time interaction was detected in the mixed-effects models, we conducted interaction contrasts to formally compare the change from baseline between groups. Estimated marginal means for each Group × Time combinations were obtained using the emmeans package, and contrasts were specified to compute estimated marginal differences (EMDs) in change from baseline. Contrasts were evaluated using Wald $z$ tests for GLMMs and Satterthwaite-adjusted $t$ tests for LMMs. All multiple comparisons were corrected using the Bonferroni adjustment.

To address potential experiment-wise $\alpha$ inflation arising from the inclusion of multiple outcome variables or statistical tests, the Benjamini–Hochberg (BH) false discovery rate (FDR) correction (Benjamini & Hochberg, 1995) was applied to the primary Group × Time $P$ values within each neurophysiological domain: intra-cortical (SICI and ICF), corticospinal (AMT stimulator intensity, CSE and cSP) and cortico-reticulospinal (ICAR, StartReact effect and RFD). This approach effectively controls the FDR while preserving statistical power. As a further sensitivity analysis, global Holm (Holm, 1979) and Benjamini–Yekutieli (BY) corrections (Benjamini & Yekutieli, 2001) were applied across all main outcome variables to ensure robustness under more conservative family-wise error control and dependency-robust FDR correction, respectively.

The generalized eta-squared ($\eta^2 G$) statistic, which accounts for both fixed and random effects (Olejnik & Algina, 2003), was employed to evaluate effect sizes for group, time, sound condition and their interactions. To improve precision and interpretability, 95% confidence intervals were also estimated for all effect sizes (Williams et al., 2023). Effect sizes were categorized as follows: small: $\eta^2 G \geq 0.01$; medium: $\eta^2 G \geq 0.06$; and large: $\eta^2 G \geq 0.14$ (Cohen, 2013). Statistical significance was set at $\alpha = 0.05$, and only statistically significant findings are reported.

Test–retest reliability of key neurophysiological measures was assessed using ICC and coefficient of variation (CV) to evaluate the stability and relative variability of measurements across the four time points within the control group. ICCs were calculated to assess test–retest reproducibility using a two-way mixed-effects model, single measurement and absolute agreement definition. ICC values were interpreted as poor (<0.50), moderate (0.50–0.75), good (0.75–0.90) or excellent (>0.90) reliability (Koo & Li, 2016). Similarly, CVs were interpreted as very good (<10%), good (10–20%), fair (20–30%) and poor (>30%) reliability (Dutra et al., 2023; Rosner, 2006).

## Results

### Baseline characteristics and test–retest reproducibility

All participants completed the intervention without dropouts or adverse effects. Baseline assessments revealed no between-group differences in body mass ($P = 0.815$), MVF ($P = 0.832$), 1RM strength ($P = 0.552$), $M_{MAX}$ ($P = 0.814$), M-wave duration ($P = 0.708$), AMT stimulus intensity ($P = 0.341$), ICF ($P = 0.377$), SICI ($P = 0.369$), cSP ($P = 0.737$), StartReact effect ($P = 0.8755$), StartReact rmsEMG (VS: $P = 0.111$; VnSS: $P = 0.096$, VSS: $P = 0.137$), ICAR ($P = 0.709$) or CSE ($P = 0.326$). Additionally, pre-stimulus rmsEMG values for single- and paired-pulse TMS remained stable across baseline and post-intervention measurements (Table 1).

Test–retest reliability within the control group varied across measures, with several outcomes demonstrating good-to-excellent reliability (ICC ≥ 0.80), whereas others showed moderate reliability based on ICC and CV estimates (Table 2). Excellent reliability was observed for cSP AURC, the StartReact effect and RFD during the first 50 ms under VSS (ICC ≥ 0.92). ICAR and ICF demonstrated good reliability (ICC range: 0.72–0.77), while SICI showed moderate reliability (ICC = 0.61). The CVs further supported these findings, with low variability for cSP, ICF and SICI (CV ≤ 7.2%), and acceptable reproducibility for CSE and StartReact effect (CV ≤ 20%) (Table 2).

### Change in muscle strength (1RM and maximal voluntary force)

No main effects of Group ($\chi^2_{(2)} = 0.92$, $P = 0.632$, $\eta^2 G = 0.01$, 95% CI (0.00, 0.04)) or Time ($\chi^2_{(3)} = 1.50$, $P = 0.681$, $\eta^2 G = 0.01$, 95% CI (0.00, 0.05)) were observed for 1RM strength. However, a significant Group × Time interaction was detected ($\chi^2_{(6)} = 23.38$, $P < 0.001$, $\eta^2 G = 0.36$, 95% CI (0.34, 0.38)).

Between-group interaction contrasts for change from Baseline showed that compared with the Control group, the MP-RT group had a significantly greater change in 1RM from Baseline to Week 2 (EMD = 0.19, SE = 0.05, $P = 0.00250$) and Week 3 (EMD = 0.21, SE = 0.05, $P < 0.001$). Similarly, the SP-RT group showed a greater increase than the Control group from Baseline to Week 2 (EMD = 0.16, SE = 0.05, $P = 0.0182$) and Week 3 (EMD = 0.16, SE = 0.05, $P = 0.0140$). There was no significant difference in 1RM change from baseline between the SP-RT and MP-RT groups at any time point ($P = 1.000$) (Fig. 2A).

A significant main effect of Time ($\chi^2_{(3)} = 27.34$, $P < 0.001$, $\eta^2 G = 0.01$, 95% CI (0.00, 0.09)) and a Group × Time interaction ($\chi^2_{(6)} = 27.65$, $P < 0.001$, $\eta^2 G = 0.32$, 95% CI (0.27, 0.38)) were observed for MVF.

**Table 1. Demographic and neuromuscular variables at baseline and across 3 weeks of training**

| Variable | Control (n = 10) | | | | Metronome-paced (n = 10) | | | | Self-paced (n = 10) | | | |
|---|---|---|---|---|---|---|---|---|---|---|---|---|
| | Baseline | Week 1 | Week 2 | Week 3 | Baseline | Week 1 | Week 2 | Week 3 | Baseline | Week 1 | Week 2 | Week 3 |
| Age (years)† | 27.4 ± 7.47 | NA | NA | NA | 27.7 ± 7.15 | NA | NA | NA | 30.3 ± 3.68 | NA | NA | NA |
| Body mass (kg)† | 70.1 ± 14.39 | NA | NA | NA | 68.75 ± 10.86 | NA | NA | NA | 72.4 ± 12.10 | NA | NA | NA |
| Training volume-load (total) (kg) | NA | | | | 2171.28 ± 630.98 | | | | 3318.21 ± 794.41^^ | | | |
| TUT (total) (s) | NA | | | | 1662.12 ± 169.37^^^ | | | | 815.83 ± 314.32 | | | |
| 1RM (kg) | 13.85 ± 3.31 | 14.02 ± 3.27 | 14.02 ± 3.22 | 14.15 ± 3.18 | 12.91 ± 4.14 | 14.07 ± 4.12 | 15.12 ± 4.15****## | 16.05 ± 4.32***##### | 14.75 ± 3.48 | 15.91 ± 3.6 | 16.85 ± 3.59*# | 17.7 ± 3.8****# |
| MVF (N) | 107.3 ± 35.39 | 106.2 ± 36.76 | 107.1 ± 34.73 | 108.27 ± 35.84 | 103.03 ± 37.74 | 104.96 ± 32.53 | 119 ± 27.46***## | 122.34 ± 38.32***## | 110 ± 31.4 | 118.17 ± 34.87 | 121.12 ± 34.74 | 130.5 ± 38.1****# |
| $M_{MAX}$ (mV) | 11.74 ± 3.63 | 11.12 ± 3.63 | 11.66 ± 3.68 | 12.22 ± 3.91 | 12.66 ± 3.92 | 12.38 ± 3.76 | 11.85 ± 3.79 | 12.12 ± 3.80 | 11.77 ± 3.28 | 11.35 ± 3.48 | 11.54 ± 3.62 | 11.61 ± 3.04 |
| M-wave duration (ms) | 20.66 ± 2.13 | 21.09 ± 2.57 | 20.44 ± 2.42 | 20.53 ± 2.5 | 21.56 ± 3.09 | 20.88 ± 3.15 | 20.62 ± 2.82 | 20.22 ± 3.28 | 21.46 ± 2.64 | 19.73 ± 2.37 | 20.03 ± 2.3 | 19.26 ± 1.94 |
| AMT SI (%) | 38 ± 8 | 38 ± 8 | 38 ± 9 | 38 ± 9 | 41 ± 9 | 39 ± 7** | 38 ± 7*** | 38 ± 7*** | 36 ± 5 | 36 ± 6 | 35 ± 6 | 35 ± 6 |
| 1RM pre-stimulus rmsEMG (%$M_{MAX}$) | 0.61 ± 0.26 | 0.6 ± 0.26 | 0.59 ± 0.24 | 0.6 ± 0.25 | 0.79 ± 0.28 | 0.8 ± 0.33 | 0.76 ± 0.35 | 0.86 ± 0.34 | 0.66 ± 0.23 | 0.69 ± 0.17 | 0.64 ± 0.19 | 0.66 ± 0.21 |
| MVF pre-stimulus rmsEMG (%$M_{MAX}$) | 0.49 ± 0.16 | 0.48 ± 0.26 | 0.52 ± 0.22 | 0.49 ± 0.16 | 0.6 ± 0.41 | 0.5 ± 0.39 | 0.55 ± 0.36 | 0.64 ± 0.51 | 0.56 ± 0.28 | 0.57 ± 0.27 | 0.56 ± 0.26 | 0.56 ± 0.27 |
| sTMS pre-stimulus rmsEMG (%$M_{MAX}$) | 0.09 ± 0.04 | 0.07 ± 0.04 | 0.09 ± 0.04 | 0.09 ± 0.04 | 0.12 ± 0.06 | 0.12 ± 0.04 | 0.12 ± 0.05 | 0.14 ± 0.06 | 0.1 ± 0.04 | 0.1 ± 0.12 | 0.09 ± 0.04 | 0.09 ± 0.04 |
| ppTMS pre-stimulus rmsEMG (%$M_{MAX}$) | 0.09 ± 0.03 | 0.08 ± 0.05 | 0.09 ± 0.06 | 0.1 ± 0.04 | 0.11 ± 0.06 | 0.12 ± 0.04 | 0.12 ± 0.06 | 0.13 ± 0.06 | 0.09 ± 0.04 | 0.08 ± 0.04 | 0.08 ± 0.04 | 0.08 ± 0.05 |

Values are means ± SD; †single-time measurements; $*P$ = 0.0172, $**P$ = 0.00410, $***P$ < 0.001 versus baseline (within group). Between-group interaction: #MP-RT and SP-RT compared with control, #$P$ = 0.0182 (Week 2, 1RM), #$P$ = 0.0140 (Week 3, 1RM), #$P$ = 0.0025 (1RM), ##$P$ = 0.0010 (MVF, Week 3), ###$P$ = 0.0041 (MVF, Week 2), ##$P$ = 0.0041 (MVF, Week 3), ###$P$ = 0.00216, ^$P$ MP-RT versus SP-RT, ^^$P$ < 0.001. AMT, active motor threshold; $M_{MAX}$, maximum muscle compound action potential; MVF, maximum voluntary force; ppTMS, paired-pulse transcranial magnetic stimulation; RM, repetition maximum; rmsEMG, root mean square electromyography; SI, stimulation intensity; sTMS, single-pulse transcranial magnetic stimulation; TUT, time under tension.

**Table 2. Test–retest reproducibility of neurophysiological measures in the control group across four time points (ICC and CV values) (*n* = 10)**

| Outcome variable | | | CV (%) | ICC with 95% CI and *P* value |
|---|---|---|---|---|
| StartReact effect (ms) | | | 12.5 | 0.92 [0.82, 0.98], $P < 0.001$ |
| VART (ms) | | | 25.5 | 0.95 [0.88, 0.99], $P < 0.001$ |
| VSRT (ms) | | | 21.3 | 0.88 [0.72, 0.96], $P < 0.001$ |
| RFD (N s$^{-1}$) | VS | 0–50 ms | 16.3 | 0.82 [0.95, 1.09], $P < 0.001$ |
| | | 50–100 ms | 12.7 | 0.76 [0.50, 0.93], $P < 0.001$ |
| | VnSS | 0–50 ms | 17.3 | 0.91 [0.77, 0.98], $P < 0.001$ |
| | | 50–100 ms | 4.7 | 0.85 [0.66, 0.96], $P < 0.001$ |
| | VSS | 0–50 ms | 11.3 | 0.97 [0.91, 0.96], $P < 0.001$ |
| | | 50–100 ms | 7.7 | 0.76 [0.51, 0.92], $P < 0.001$ |
| ICAR | | | 22 | 0.72 [0.42, 0.92], $P < 0.001$ |
| CSE (AURC, a.u.) | | | 20 | 0.73 [0.46, 0.91], $P < 0.001$ |
| cSP (AURC, a.u.) | | | 6 | 0.92 [0.82, 0.97], $P < 0.001$ |
| ICF (% test response) | | | 4.3 | 0.77 [0.52, 0.93], $P < 0.001$ |
| SICI (% test response) | | | 7.2 | 0.61 [0.30, 0.86], $P < 0.001$ |

a.u., arbitrary units; AURC, area under the recruitment curve; cSP, cortical silent period; CV, coefficient of variation; ICAR, ipsilateral to contralateral motor-evoked potential amplitude ratio (unitless, ratio of mV values); ICC, intraclass correlation coefficient; ICF, intracortical facilitation; SICI, short-interval intracortical inhibition; VART, visual-auditory reaction time; VnSS, visual non-startling stimuli; VS, visual stimuli; VSRT, visual-startle reaction time; VSS, visual-startling stimuli.

Between-group interaction contrasts for change from Baseline showed that the MP-RT group exhibited greater increases in MVF than Control at Week 2 (EMD = 0.20 ± 0.05, $z = 3.87$, $P = 0.00100$), and this difference remained significant at Week 3 (EMD = 0.18 ± 0.05, $z = 3.51$, $P = 0.00410$). The SP-RT group also showed a greater increase in MVF than Control from Baseline to Week 3 (EMD = 0.16, SE = 0.05, $z = 3.16$, $P = 0.0142$). No differences in change from Baseline were detected between MP-RT and SP-RT (all $P = 1.000$) (Fig. 2*B*) (Table 1).

### Subcortical excitability and behavioural measures

**Reticulospinal and cortico-reticulospinal excitability.** The StartReact effect analysis revealed no main effects of Group ($\chi^2_{(2)} = 0.50$, $P = 0.779$, $\eta^2 G = 0.01$, 95% CI (0.00, 0.10)) or Time ($\chi^2_{(3)} = 2.48$, $P = 0.478$, $\eta^2 G = 0.05$, 95% CI (0.00, 0.15)). However, there was a Group × Time interaction ($\chi^2_{(6)} = 42.74$, $P < 0.001$, $\eta^2 G = 0.50$, 95% CI (0.46, 0.54)). Between-group interaction contrasts revealed that, relative to baseline, the SP-RT group showed greater increases in the StartReact effect compared with the MP-RT group at Week 1 (EMD = 0.57, SE = 0.18, $z = 3.10$, $P = 0.0173$), Week 2 (EMD = 1.07, SE = 0.19, $z = 5.75$, $P < 0.001$) and Week 3 (EMD = 0.85, SE = 0.19, $z = 4.60$, $P < 0.001$). The SP-RT group also demonstrated greater increases from baseline relative to the Control group at Week 1 (EMD = 0.68, SE = 0.18, $z = 3.68$, $P = 0.0020$), Week 2 (EMD = 0.64, SE = 0.18, $z = 3.48$, $P = 0.00440$) and Week 3 (EMD = 0.63, SE = 0.18, $z = 3.42$, $P = 0.00570$) (Table 3, Fig. 3*A–C*).

**Rate of force development.** Analysis of RFD during the first 50 ms revealed a significant Group × Type of Sound × Time interaction ($\chi^2_{(12)} = 25.20$, $P = 0.0179$, $\eta^2 G = 0.052$, 95% CI (0.023, 0.081)). Significant main effects of Type of Sound ($\chi^2_{(2)} = 280.12$, $P < 0.001$, $\eta^2 G = 0.49$), Time ($\chi^2_{(3)} = 8.34$, $P = 0.0395$, $\eta^2 G = 0.03$) and Group × Type of Sound ($\chi^2_{(4)} = 71.17$, $P < 0.001$, $\eta^2 G = 0.19$), as well as Type of Sound × Time ($\chi^2_{(6)} = 21.29$, $P = 0.00163$, $\eta^2 G = 0.07$) and Group × Time ($\chi^2_{(6)} = 17.37$, $P = 0.00803$, $\eta^2 G = 0.06$) interactions, were also observed.

Within-group interaction contrasts demonstrated that in the SP-RT group the increase in RFD from baseline was greater for VSS compared with both VnSS and VS at Week 1 (VSS *vs.* VnSS: EMD = 0.64, SE = 0.18, $z = 3.53$, $P < 0.001$; VSS *vs.* VS: EMD = 0.61, SE = 0.18, $z = 3.34$, $P = 0.0017$), Week 2 (EMD = 0.88, SE = 0.18, $z = 4.83$, $P < 0.001$; EMD = 0.60, SE = 0.18, $z = 3.28$, $P = 0.002$) and Week 3 (EMD = 0.76, SE = 0.18, $z = 4.15$, $P < 0.001$; EMD = 0.67, SE = 0.18, $z = 3.70$, $P < 0.001$). No stimulus-specific differences were detected within the MP-RT or Control groups at any time point (all $P \geq 0.378$).

Between-group interaction contrasts showed that for the ΔVSS − ΔVnSS comparison, SP-RT exhibited greater enhancement than Control at Week 2 (EMD = 0.80, SE = 0.26, $z = 3.11$, $P = 0.00560$) and Week 3 (EMD = 0.68, SE = 0.26, $z = 2.63$, $P = 0.0256$), and greater enhancement than MP-RT at Week 2 (EMD = 0.68, SE = 0.26, $z = 2.64$, $P = 0.0246$). For the ΔVSS − ΔVS contrast at Week 3, SP-RT also demonstrated greater enhancement than Control (EMD = 0.63, SE = 0.26, $z = 2.44$, $P = 0.0448$) and MP-RT (EMD = 0.62, SE = 0.26, $z = $

**Table 3. StartReact and TMS measures across baseline and 3 weeks of training**

| Variable | Control group (n = 10) | | | | Metronome-paced resistance training group (n = 10) | | | | Self-paced resistance training group (n = 10) | | | |
|---|---|---|---|---|---|---|---|---|---|---|---|---|
| | Baseline | Week 1 | Week 2 | Week 3 | Baseline | Week 1 | Week 2 | Week 3 | Baseline | Week 1 | Week 2 | Week 3 |
| StartReact effect (ms) | 19.68 ± 16.04 | 19.43 ± 17.08 | 20.34 ± 16.72 | 20.58 ± 16.53 | 20.35 ± 14.62 | 20.08 ± 12.42 | 14.89 ± 13.24 | 19.88 ± 14.28 | 16.56 ± 12.84 | 29.12 ± 17.68***## | 31.74 ± 20.41****##^ | 32.14 ± 19.51 ***##^^^ |
| VART (ms) | 121.81 ± 16.71 | 128.27 ± 21.17 | 123.04 ± 20.44 | 124.18 ± 21.62 | 122.62 ± 14.93 | 115.22 ± 17.99 | 110.46 ± 17.67** | 109.10 ± 18.12*** | 116.73 ± 27.60 | 112.24 ± 24.59 | 115.68 ± 27.56 | 111.56 ± 25.75 |
| VSRT (ms) | 102.13 ± 8.81 | 108.64 ± 14.49 | 102.97 ± 11.94 | 102.35 ± 10.51 | 102.27 ± 13.44 | 95.14 ± 10.57 | 95.04 ± 9.92 | 89.23 ± 9.91* | 103.71 ± 14.82 | 83.11 ± 14.21*** | 83.93 ± 14.32*** | 79.41 ± 12.42*** |
| ICAR | 0.25 ± 0.10 | 0.26 ± 0.13 | 0.28 ± 0.14 | 0.26 ± 0.08 | 0.31 ± 0.19 | 0.27 ± 0.16 | 0.26 ± 0.18 | 0.30 ± 0.20 | 0.34 ± 0.31 | 0.55 ± 0.26 * | 0.61 ± 0.31 ***#^ | 0.59 ± 0.28 ***##^ |
| cSP (AURC, a.u.) | 125.87 ± 86.49 | 130.95 ± 89.34 | 130.62 ± 88.04 | 126.26 ± 94.23 | 85.11 ± 32.60 | 43.10 ± 26.64***### | 45.01 ± 31.66***### | 53.42 ± 61.23 ***## | 107.28 ± 49.73 | 94.03 ± 75.63 | 88.14 ± 48.84 | 82.25 ± 37.91 |
| CSE (AURC, a.u.) | 825.88 ± 300.60 | 743.99 ± 379.62 | 784.76 ± 298.17 | 789.03 ± 268.00 | 881.65 ± 490.37 | 1348.13 ± 712.07**# | 1381.05 ± 780.17** | 1581.75 ± 637.03 ***##^^ | 824.55 ± 710.94 | 754.32 ± 475.35 | 710.99 ± 447.94 | 796.2 ± 514.84 |
| SICI (% test response) | 76.52 ± 8.50 | 76.18 ± 11.17 | 76.87 ± 13.81 | 76.22 ± 9.26 | 71.18 ± 18.53 | 83.11 ± 17.03 | 90.32 ± 6.32 ***## | 95.29 ± 3.89 ***### | 80.04 ± 15.32 | 75.80 ± 17.31 | 82.91 ± 13.13 | 75.07 ± 19.89 |
| ICF (% test response) | 126.23 ± 14.49 | 126.09 ± 13.77 | 126.11 ± 12.72 | 126.49 ± 14.75 | 118.61 ± 18.59 | 148.46 ± 42.98 * | 158.82 ± 49.44 ** | 169.19 ± 62.74 ***# | 123.57 ± 17.24 | 131.41 ± 28.85 | 134.23 ± 20.80 | 133.86 ± 52.86 |
| iMEP ($\%M_{MAX}$) | 13.93 ± 8.56 | 15.63 ± 13.4 | 14.27 ± 9.12 | 13.01 ± 7.41 | 19.08 ± 6.71 | 18.75 ± 9.4 | 18.02 ± 10.88 | 21.41 ± 13.25 | 15.1 ± 11.51 | 23.24 ± 9.07 ** | 25.3 ± 12.75 ** | 22.64 ± 8.70 ** |
| cMEP ($\%M_{MAX}$) | 57.26 ± 27.4 | 58.91 ± 30.62 | 54.54 ± 24.71 | 54.01 ± 32.52 | 50.68 ± 24.77 | 76.15 ± 28.86 | 78.22 ± 17.60* | 72.73 ± 18.69* | 53.97 ± 28.72 | 47.57 ± 25.23 | 44.31 ± 14.57 | 46.02 ± 24.56 |

Values are means ± SD; *P = 0.0106–0.0265, **P = 0.00170–0.00540, ***P < 0.001 versus baseline (within group). Between-group interactions: #MP-RT and SP-RT compared with control, #P = 0.0156–0.0384, ##P = 0.00210–0.00900, ###P < 0.001; ^MP-RT versus SP-RT, ^P = 0.0173–0.0351, ^^P = 0.00160–0.00560, ^^^P < 0.001. Exact P values are reported in Table 6. a.u., arbitrary units; AURC, area under the recruitment curve; cMEP, contralateral motor-evoked potential; CSE, corticospinal excitability; cSP, cortical silent period; ICAR, ipsilateral to contralateral motor-evoked potential amplitude ratio (iMEP/cMEP, unitless); ICF, intracortical facilitation; iMEP, ipsilateral motor-evoked potential; $M_{MAX}$, maximum compound action potential; MP-RT, metronome-paced resistance training; SICI, short-interval intracortical inhibition; SP-RT, self-paced resistance training; VART, visual-auditory reaction time; VSRT, visual-startle reaction time.

2.43, $P = 0.0453$). No other between-group differences in stimulus-specific change in RFD were observed (all $P \geq 0.0659$) (Table 4).

Analysis of RFD during the 50–100 ms interval did not reveal a significant Group × Time × Type of Sound interaction ($\chi^2_{(12)} = 8.12$, $P = 0.776$, $\eta^2 G = 0.06$, 95% CI (0.00, 0.12)). There were no main effects of Group ($\chi^2_{(2)} = 0.36$, $P = 0.840$, $\eta^2 G = 0.01$, 95% CI (0.00, 0.04)) or Time ($\chi^2_{(3)} = 3.07$, $P = 0.382$, $\eta^2 G = 0.03$, 95% CI (0.00, 0.06)). However, a significant main effect of Type of Sound was observed ($\chi^2_{(2)} = 7.30$, $P = 0.0259$, $\eta^2 G = 0.48$, 95% CI (0.48, 0.49)). None of the two-way interaction effects reached statistical significance, including Group × Time ($\chi^2_{(6)} = 4.87$, $P = 0.560$, $\eta^2 G = 0.08$, 95% CI (0.03, 0.12)), Group × Type of Sound ($\chi^2_{(4)} = 1.24$, $P = 0.872$, $\eta^2 G = 0.04$, 95% CI (0.00, 0.08)), and Time × Type of Sound ($\chi^2_{(6)} = 1.23$, $P = 0.975$, $\eta^2 G = 0.03$, 95% CI (0.00, 0.08)). Within the SP-RT group under the VSS condition, RFD

during 50–100 ms increased from Baseline to Week 3 (EMD = 0.32, SE = 0.09, $z = 3.42$, $P = 0.0038$) (Table 4).

Cortico-reticulospinal excitability was also examined using ICAR. Significant main effects of Group ($F_{(2, 24)} = 5.80$, $P = 0.00882$, $\eta^2 G = 0.02$, 95% CI (0.00, 0.12)), Time ($F_{(3, 72)} = 2.86$, $P = 0.0426$, $\eta^2 G = 0.01$, 95% CI (0.00, 0.13)), and Group × Time interactions ($F_{(6, 72)} = 3.81$, $P = 0.00237$, $\eta^2 G = 0.38$, 95% CI (0.31, 0.45)) were observed. The between-group interaction contrasts (change from baseline) revealed that the SP-RT group showed greater increases in ICAR compared with the MP-RT group at Week 2 (EMD = 0.32, SE = 0.09, $t_{(82.3)} = 3.58$, $P = 0.00520$) and Week 3 (EMD = 0.26, SE = 0.09, $t_{(82.3)} = 2.97$, $P = 0.0351$). The SP-RT group also exhibited greater increases than the Control group at Week 2 (EMD = 0.24, SE = 0.08, $t_{(82.3)} = 3.10$, $P = 0.0241$) and Week 3 (EMD = 0.25, SE = 0.08, $t_{(82.3)} = 3.18$, $P = 0.0188$) (Table 3, Fig. 4).

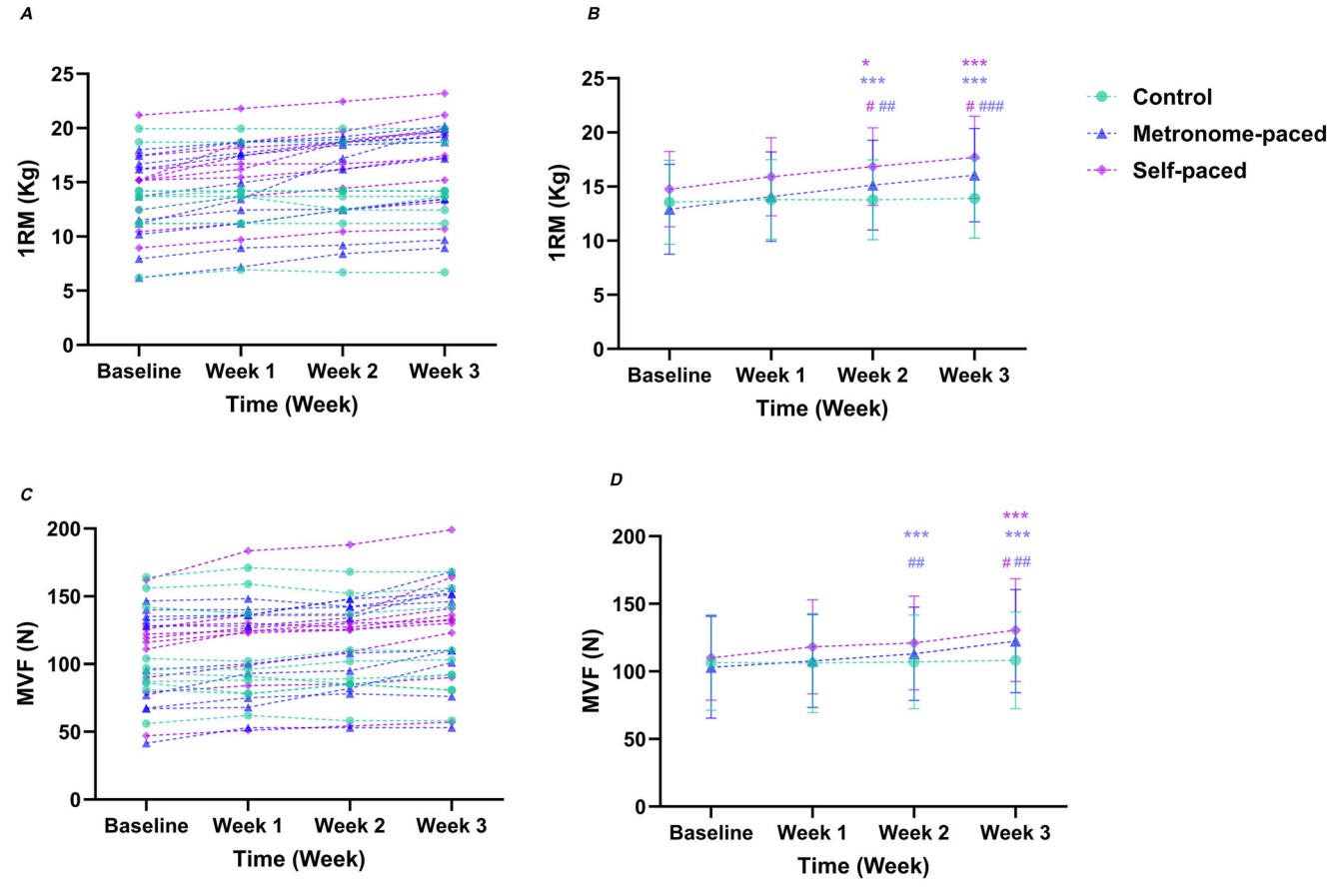

**Figure 2. 1RM (*A*, *B*) and MVF (*C*, *D*) at baseline and after Weeks 1–3 of MP-RT and SP-RT**
*A*, *C*, individual participant data; *B*, *D*, group means ± SD. *$P = 0.0172$, ***$P < 0.001$ *versus* baseline (within group). # Indicates between-group interaction: MP-RT and SP-RT compared with control, #$P = 0.0182$ (Week 2, 1RM, SP-RT), #$P = 0.0140$ (Week 3, 1RM, SP-RT), #$P = 0.0142$ (Week 3, MVF, SP-RT), ##$P = 0.0025$ (1RM, MP-RT), ##$P = 0.0010$ (MVF, Week 2, MP-RT), ##$P = 0.0041$ (MVF, Week 3, MP-RT), ###$P < 0.001$ (MP-RT). Significance markers are shown only on the group means ± SD panels (*B*, *D*). 1RM, one-repetition maximum; MP-RT, metronome-paced resistance training; MVF, maximum voluntary force; SP-RT, self-paced resistance training.

**Table 4. RFD during three stimuli, across baseline and 3 weeks of resistance training**

| Variable | Interval | Type of stimuli | Control group (n = 10) | | | | Metronome-paced resistance training group (n = 10) | | | | Self-paced resistance training group (n = 10) | | | |
|---|---|---|---|---|---|---|---|---|---|---|---|---|---|---|
| | | | Baseline | Week 1 | Week 2 | Week 3 | Baseline | Week 1 | Week 2 | Week 3 | Baseline | Week 1 | Week 2 | Week 3 |
| RFD (N s⁻¹) | 0–50 ms | VS | 1881.11 ± 1073.58 | 1849.38 ± 1041.47 | 1901.04 ± 1333.82 | 1971.41 ± 1769.23 | 1607.04 ± 643.85 | 1585.86 ± 967.79 | 1541.64 ± 886.65 | 1803.87 ± 1013.04 | 1659.01 ± 1327.64 | 1950.88 ± 2478.94 | 1895.79 ± 1472.04 | 1853.17 ± 1734.38 |
| | | VnSS | 1893.71 ± 1268.85 | 1837.26 ± 1339.42 | 1860.47 ± 1282.4 | 1833.86 ± 1264.65 | 1898.89 ± 916.94 | 1707.92 ± 848.72 | 1712.17 ± 994.77 | 1866.33 ± 927.43 | 1796.91 ± 1553.07 | 1695.68 ± 1543.04 | 1616.1 ± 1468.02 | 1724.98 ± 1214.71 |
| | | VSS | 2394.67 ± 1592.14 | 2411.84 ± 1574.32 | 2420.33 ± 1698.45 | 2317.66 ± 1550.11 | 2394.67 ± 1592.14 | 2711.21 ± 1348.84 | 2665.22 ± 1409.31 | 3168.66 ± 2014.92 | 2339.16 ± 1756.1 | 3867.82 ± 2607.93 ***###⌣⌣ | 4025.43 ± 1986.0 ***###⌣⌣ | 4324.44 ± 2621.08 ***###⌣⌣ |
| | 50–100 ms | VS | 703.86 ± 455.65 | 619.42 ± 483.1 | 565.58 ± 201.49 | 610.43 ± 273.17 | 657.14 ± 244.99 | 604.2 ± 238.64 | 662.93 ± 189.38 | 682.68 ± 303.34 | 712.58 ± 222.1 | 721.57 ± 230.04 ***# | 818.65 ± 271.36 | 747.46 ± 219.24 |
| | | VnSS | 741.15 ± 473.59 | 694.27 ± 350.9 | 667.64 ± 224.66 | 665.41 ± 225.91 | 749.95 ± 273.99 | 631.86 ± 204.78 | 684.03 ± 186.91 | 812.43 ± 492.16 | 710.24 ± 206.77 | 731.14 ± 192.19 | 796.59 ± 249.19 | 718.84 ± 202.24 |
| | | VSS | 843.59 ± 367.12 | 788.1 ± 346.97 | 779.64 ± 171.25 | 777.05 ± 175.4 | 834.04 ± 462.98 | 878.27 ± 343.23 | 874.93 ± 267.81 | 867 ± 309.27 | 816.08 ± 189.31 | 1007.53 ± 286.05 | 1008.94 ± 231.27 | 1150.11 ± 443.29*** |

Values are means ± SD; ***$P < 0.001$ *versus* baseline (within group). Between-group interactions conducted separately within each stimulus condition: # SP-RT *versus* control, #$P = 0.0246$, ###$P < 0.001$; ⌣MP-RT *versus* SP-RT, ⌣$P = 0.00560$ (Week 2), ⌣⌣$P = 0.00160$ (Week 2), ⌣⌣$P = 0.00560$ (Week 3). Exact *P* values for all within- and between-group interaction contrasts within each stimulus and for the Group × Time × Stimulus interaction contrasts are provided in Table 6. RFD, rate of force development; VnSS, visual non-startling stimulus; VS, visual stimulus; VSS, visual-startling stimulus.

## Corticospinal excitability and inhibition

**Cortical excitability.** The analysis of AMT stimulus intensity revealed that there was no significant main effect of Group ($\chi^2_{(2)} = 0.53$, $P = 0.768$, $\eta^2 G = 0.01$, 95% CI (0.00, 0.23))). However, there was a main effect of Time ($\chi^2_{(3)} = 28.09$, $P < 0.001$, $\eta^2 G = 0.39$, 95% CI (0.32, 0.47)) and Group × Time interaction ($\chi^2_{(6)} = 18.71$, $P = 0.00468$, $\eta^2 G = 0.30$, 95% CI (0.21, 0.39)).

Between-group interaction contrasts indicated that baseline-to-week changes differed significantly only between the Control and MP-RT groups, with MP-RT showing greater reductions at Week 1 (EMD = −0.06, SE = 0.02, $z = -2.86$, $P = 0.0384$), Week 2 (EMD = −0.07, SE = 0.02, $z = -3.34$, $P = 0.00770$) and Week 3 (EMD = −0.08, SE = 0.02, $z = -4.25$, $P < 0.001$) There was no significant difference in AMT change from baseline between the SP-RT and MP-RT groups ($P = 0.495$).

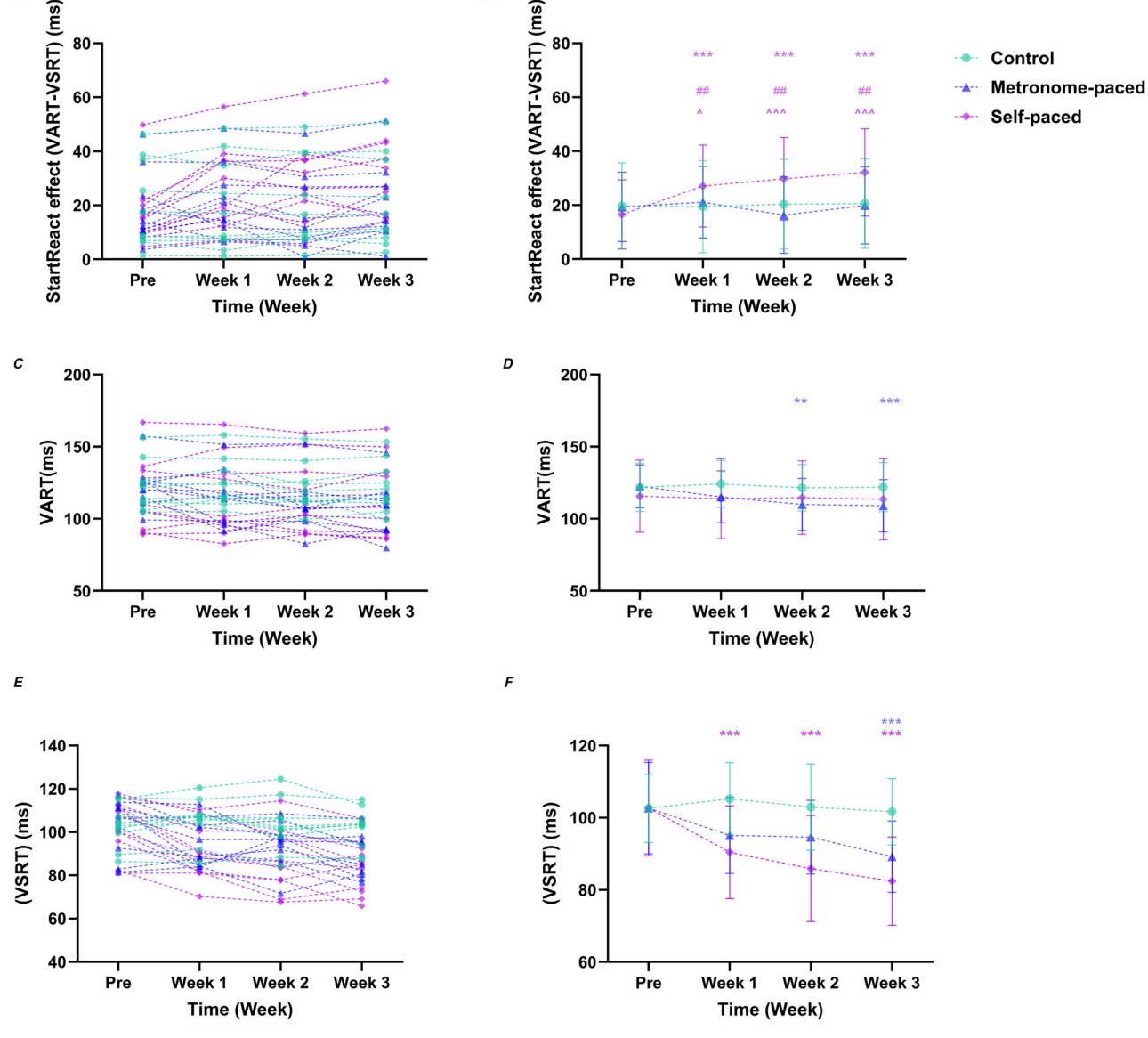

**Figure 3. StartReact effect (*A, B*), VART (*C, D*) and VSRT (*E, F*) at baseline and after Weeks 1–3 of MP-RT and SP-RT**

*A, C, E*, individual participant data; *B, D, F*, group means ± SD. \*\**P* = 0.0018 (MP-RT group), \*\*\**P* < 0.001 *versus* baseline (within group). # Indicates between-group interaction: SP-RT compared with control, ##*P* = 0.00210 (Week 1), ##*P* = 0.00440 (Week 2), ##*P* = 0.00570 (Week 3), ˆ Indicates between-group interaction: MP-RT *versus* SP-RT, ˆ*P* = 0.0173, ˆˆˆ*P* < 0.001. Significance markers are shown only on the group means ± SD panels (*B, D, F*). MP-RT, metronome-paced resistance training; SP-RT, self-paced resistance training; VART, visual-auditory reaction time; VSRT, visual-startle reaction time.

For CSE (AURC), no main effect was found for Time ($F_{(3,81)}$ = 2.51, $P$ = 0.0646, $\eta^2 G$ = 0.04, 95% CI (0.02, 0.07)), but significant effects were observed for Group ($F_{(2,27)}$ = 4.24, $P$ = 0.0251, $\eta^2 G$ = 0.05, 95% CI (0.03, 0.07)) and the Group × Time interaction ($F_{(6,81)}$ = 3.85, $P$ = 0.00199, $\eta^2 G$ = 0.12, 95% CI (0.09, 0.16)). Between-group interaction contrasts revealed that the MP-RT group exhibited a greater increase in CSE from baseline compared with both the control (EMD = 736.95, SE = 189, $t_{(81)}$ = 3.89, $P$ = 0.00360) and SP-RT (EMD = 728.45, SE = 189, $t_{(81)}$ = 3.85, $P$ = 0.00420) groups at Week 3 (Table 3, Fig. 5).

**Cortical silent period.** The AURC for cSP revealed no main effects of Group ($\chi^2_{(2)}$ = 0.38, $P$ = 0.827, $\eta^2 G$ = 0.01, 95% CI (0.00, 0.09)) or Time ($\chi^2_{(3)}$ = 0.30, $P$ = 0.961, $\eta^2 G$ = 0.01, 95% CI (0.00, 0.11)). However, there was a Group × Time interaction ($\chi^2_{(6)}$ = 22.31, $P$ = 0.00107, $\eta^2 G$ = 0.33, 95% CI (0.26, 0.40)).

The interaction contrast analysis comparing change from baseline between groups indicated that MP-RT had a greater reduction in cSP compared with Control at Week 1 (EMD = −0.81, SE = 0.21, $z$ = −3.92, $P$ <0.001), Week 2 (EMD = −0.82, SE = 0.21, $z$ = −4.00, $P$ < 0.001) and Week 3 (EMD = −0.69, SE = 0.21, $z$ = −3.29, $P$ = 0.00900). There was no significant difference in cSP

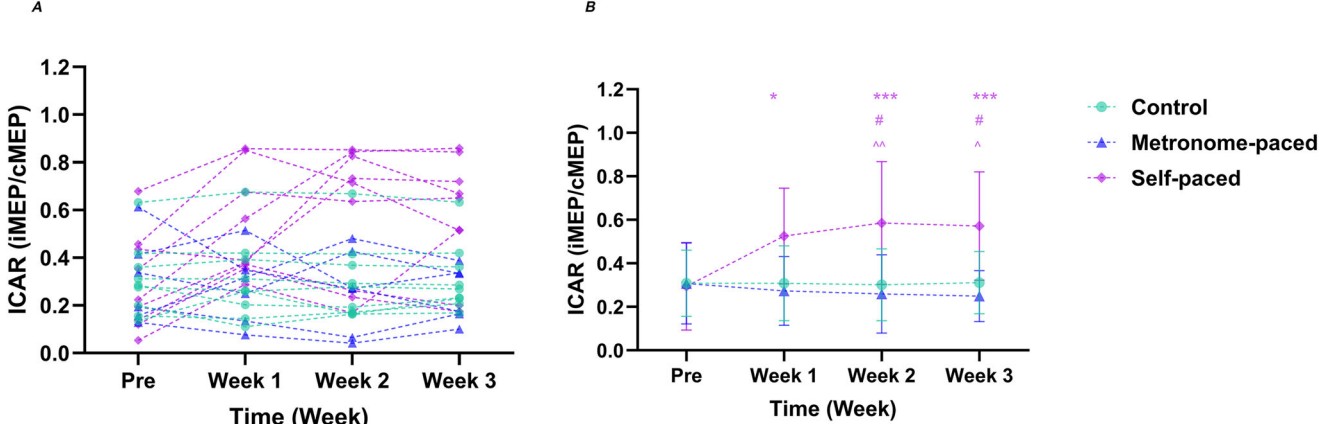

**Figure 4. ICAR at baseline and over 3 weeks of MP-RT and SP-RT**
*A*, individual participant data; *B*, group means ± SD. \**P* = 0.0210, \*\*\**P* < 0.001 *versus* baseline (within group). Between-group differences are denoted as follows: #*P* = 0.0241 (Week 2), #*P* = 0.0188 (Week 3), (SP-RT *vs.* control); ˆ*P* = 0.00520, ˜*P* = 0.0351 (SP-RT *vs.* MP-RT). Significance markers are shown only on the group means ± SD panel (*B*). cMEP, contralateral motor-evoked potential; ICAR, ipsilateral to contralateral motor-evoked potential amplitude ratio (iMEP/cMEP); iMEP, ipsilateral motor-evoked potential; MP-RT, metronome-paced resistance training; SP-RT, self-paced resistance training.

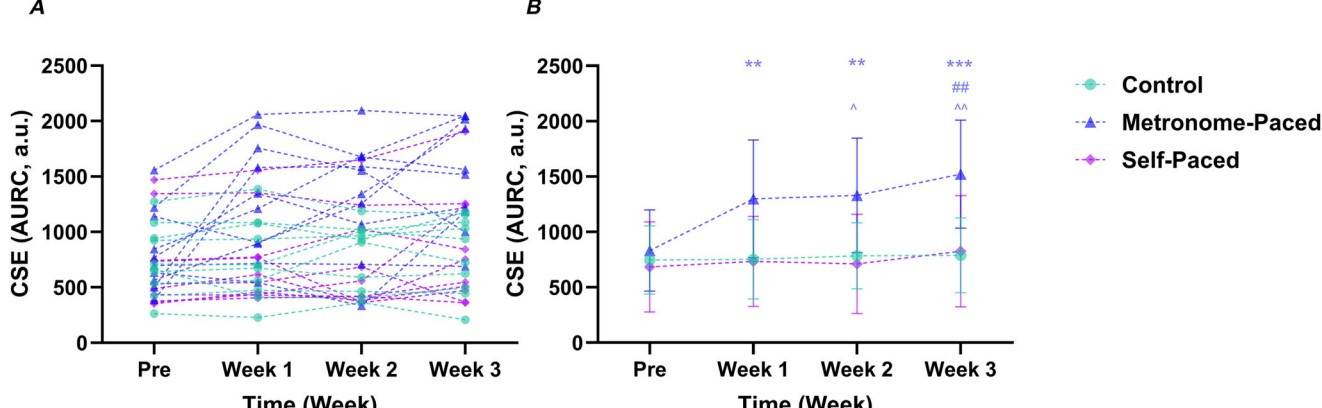

**Figure 5. CSE (AURC) at baseline and after MP-RT and SP-RT across Weeks 1–3**
*A*, individual participant data; *B*, group means ± SD. \*\**P* = 0.00480 (Week 1), \*\**P* = 0.00210 (Week 2), \*\*\**P* < 0.001 *versus* baseline (within group, MP-RT). # Indicates between-group interaction: MP-RT *versus* control, ##*P* = 0.00360; ˆ Indicates between-group interaction: MP-RT *versus* SP-RT, ˆ*P* = 0.0313, ˜*P* = 0.00420. a.u., arbitrary units; AURC, area under the recruitment curve; CSE, corticospinal excitability; MP-RT, metronome-paced resistance training; SP-RT, self-paced resistance training.

change from Baseline to Week 3 between the SP-RT and MP-RT groups ($P = 0.259$) (Table 3, Fig. 6).

**Intracortical excitability.** The SICI analysis revealed no significant main effects for Group ($\chi^2_{(2)} = 1.60$, $P = 0.449$, $\eta^2 G = 0.03$, 95% CI (0.00, 0.12)) or Time ($\chi^2_{(3)} = 1.31$, $P = 0.726$, $\eta^2 G = 0.00$, 95% CI (0.00, 0.12)). However, there was a significant Group × Time interaction ($\chi^2_{(6)} = 19.20$, $P = 0.00384$, $\eta^2 G = 0.43$, 95% CI (0.38, 0.49)). Between-group interaction contrasts revealed that the reduction in SICI from Baseline to Week 3 was greater in the MP-RT group compared with the SP-RT group (EMD $= -0.47$, SE $= 0.10$, $z = -4.84$, $P < 0.001$). The MP-RT group also showed greater reductions in SICI from Baseline compared with the Control group at Week 2 (EMD $= -0.32$, SE $= 0.10$, $z = -3.32$, $P = 0.00820$) and Week 3 (EMD $= -0.40$, SE $= 0.10$, $z = -4.16$, $P < 0.001$) (Table 3, Fig. 7*A*).

The ICF analysis revealed no main effects for Group ($\chi^2_{(2)} = 0.40$, $P = 0.819$, $\eta^2 G = 0.02$, 95% CI (0.00, 0.17)) or Time ($\chi^2_{(3)} = 0.001$, $P = 1.000$, $\eta^2 G = 0.00$, 95% CI (0.00, 0.19)). However, there was a Group × Time interaction ($\chi^2_{(6)} = 12.44$, $P = 0.00528$, $\eta^2 G = 0.33$, 95% CI (0.21, 0.45)). A between-group interaction contrasts indicated that, relative to baseline, the MP-RT group showed a greater increase in ICF than the control group at Week 3 (EMD $= 0.34$, SE $= 0.11$, $z = 3.13$, $P = 0.0156$). No difference in ICF change from baseline between the SP-RT and MP-RT groups ($P = 0.0996$) (Fig. 7*B*).

### Multiplicity control and robustness of findings

All primary Group × Time interaction effects that were significant in the initial analyses remained below the adjusted significance threshold following the BH FDR correction within each neurophysiological domain. Moreover, these effects also remained statistically significant when subjected to the more conservative global Holm and BY corrections applied across all outcome variables (Table 5).

Bonferroni-adjusted *P* values for all between- and within-group interaction contrasts are reported in Table 6.

## Discussion

This study investigated the effects of RT on RST excitability in humans and examined whether MP-RT and SP-RT differentially modulate cortico-reticular, intracortical and corticospinal excitability. The principal finding was that SP-RT elicited greater RST involvement than MP-RT, despite comparable improvements in maximal strength across modalities. Furthermore, the two training approaches produced distinct patterns of change across cortico-reticular, intracortical and corticospinal measures, suggesting that training pacing influences the neural locus of early strength adaptation.

The present findings provide novel evidence that RT modulates cortico-reticular and reticulospinal pathways in humans. Although both training modalities yielded comparable improvements in maximal strength, the neural mechanisms underlying these behavioural gains diverged according to contraction pacing. Specifically, SP-RT was characterized by greater subcortical involvement, whereas MP-RT demonstrated comparatively greater modulation of cortical and corticospinal measures. This divergence suggests that similar strength outcomes can arise from pathway-specific adaptations within the descending motor system.

SP-RT elicited greater cortico-reticulospinal and reticulospinal adaptations than MP-RT, evidenced by

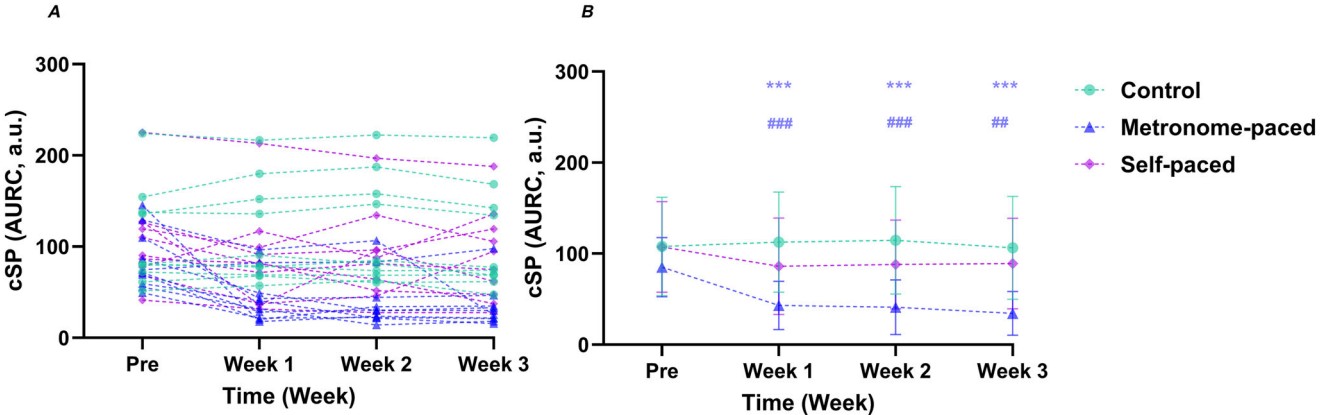

**Figure 6. cSP at baseline and over 3 weeks of MP-RT and SP-RT**
*A*, individual participant data; *B*, group means ± SD. \*\*\**P* < 0.001 *versus* baseline (within group, MP-RT). # Indicates between-group interaction: MP-RT compared with control, ##*P* = 0.00900, ###*P* < 0.001. Significance markers are shown only on the group means ± SD panel (*B*). a.u., arbitrary units; AURC, area under the recruitment curve; cSP, cortical silent period; MP-RT, metronome-paced resistance training; SP-RT, self-paced resistance training.

larger increases in ICAR, larger enhancements in the StartReact effect and greater improvements in early-phase RFD during VSS. These findings are consistent with evidence that the RST contributes to rapid, gross motor output and ballistic force production (Baker, 2011; Glover & Baker, 2020). In contrast, MP-RT elicited greater cortical and corticospinal adaptations than SP-RT, evidenced by greater reductions in SICI and larger increases in CSE, in line with recent reviews comparing externally paced and self-paced training modalities (Gómez-Feria et al., 2023; Gordon et al., 2024).

The divergence in neural adaptations between MP-RT and SP-RT carries important translational implications. MP-RT appears well suited to enhancing corticospinal plasticity, whereas SP-RT, despite reduced TUT, may represent a time-efficient, high-load strategy that engages subcortical circuits. This distinction is particularly relevant in clinical populations such as stroke or spinal cord injury, where corticospinal integrity is often compromised but reticulospinal pathways remain at least partially preserved and capable of supporting recovery. In such cases, SP-RT may offer a practical training option, given its reliance on simple free-weight exercises that can be adapted to individual capacity, although careful supervision, load adjustment and fatigue management would be required. Beyond rehabilitation, these findings also have relevance for healthy and ageing adults: in age-related decline, where corticospinal output diminishes, SP-RT may help preserve strength by engaging reticulospinal pathways, while MP-RT may be preferable for enhancing voluntary cortical drive and fine or precise motor control. Both approaches are accessible, low-cost and based on familiar RT practices, supporting their feasibility across diverse groups. Collectively, these

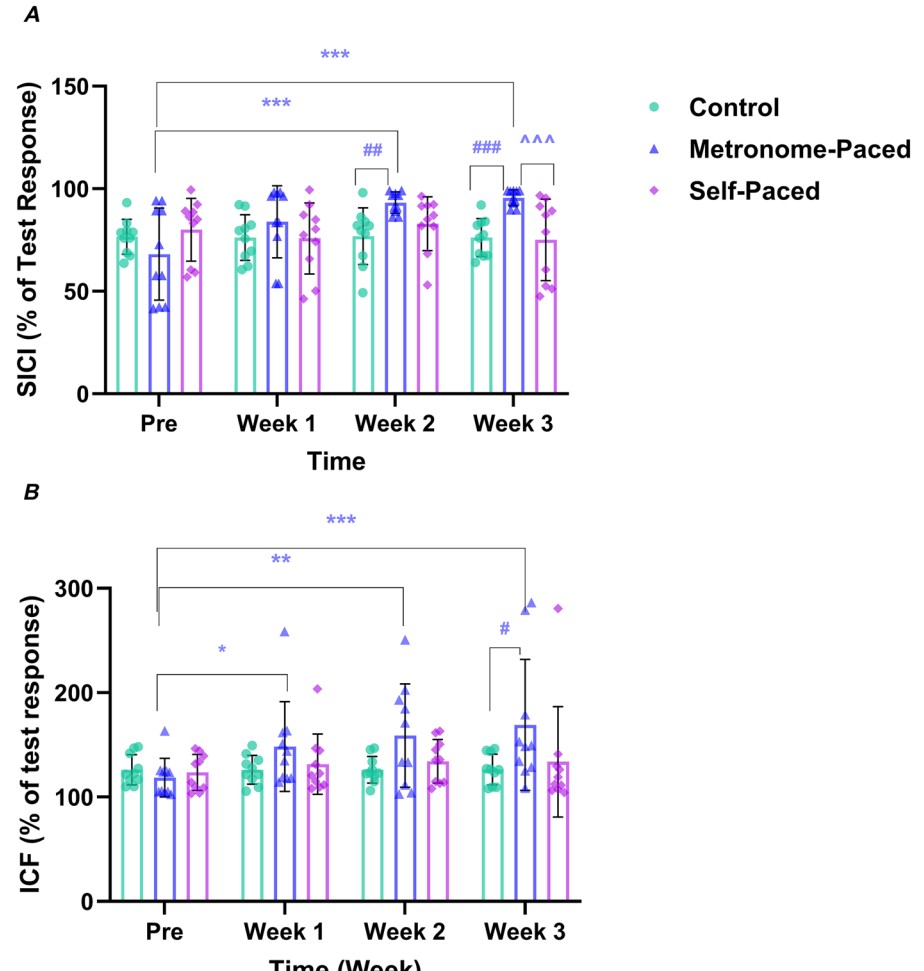

**Figure 7. SICI (*A*) and ICF (*B*) at baseline and over 3 weeks of MP-RT and SP-RT**
Data are means ± SD. *$P = 0.0265$, **$P = 0.00170$, ***$P < 0.001$ *versus* baseline (within group, MP-RT). # Indicates between-group interaction: MP-RT compared with control, #$P = 0.0156$, ##$P = 0.00820$, ###$P < 0.001$. ^ Indicates between-group interaction: MP-RT *versus* SP-RT, ^^^$P < 0.001$. ICF, intracortical facilitation; MP-RT, metronome-paced resistance training; SICI, short-interval intracortical inhibition; SP-RT, self-paced resistance training.

**Table 5.  Summary of multiplicity control analyses across neurophysiological domains**

| Domain | Outcome variable | Unadjusted *P* | BH-adjusted *P* | Holm-adjusted *P* | BY-adjusted *P* |
|---|---|---|---|---|---|
| Cortico-reticulospinal | ICAR | 0.00236 | 0.00472 | 0.0142 | 0.0150 |
| | StartReact effect (ms) | <0.001 | <0.001 | <0.001 | <0.001 |
| | RFD (0–50 ms) N s$^{-1}$ | 0.0179 | 0.0159 | 0.0238 | 0.0379 |
| | RFD (50–100 ms) N s$^{-1}$ | 0.560 | 0.560 | 0.560 | 1.000 |
| Intracortical | SICI (% test response) | 0.00384 | 0.00528 | 0.0192 | 0.0192 |
| | ICF (% test response) | 0.00528 | 0.00528 | 0.0193 | 0.0193 |
| Corticospinal | AMT SI (%) | 0.00468 | 0.00468 | 0.0192 | 0.0192 |
| | CSE (AURC, a.u.) | 0.00199 | 0.00299 | 0.0139 | 0.0150 |
| | cSP (AURC, a.u.) | 0.00106 | 0.00298 | 0.00848 | 0.0135 |

AMT, active motor threshold; a.u., arbitrary unit; AURC, area under recruitment curve; BH, Benjamini–Hochberg; BY, Benjamini–Yekutieli; CSE, corticospinal excitability; cSP, cortical silent period; FDR, false discovery rate; ICAR, ipsilateral to contralateral motor-evoked potential amplitude ratio; ICF, intracortical facilitation; RFD, rate of force development; SI, stimulation intensity; SICI, short-interval intracortical inhibition.

results support the growing evidence that RT protocols induce pathway-specific neuroplastic adaptations and suggest that tailoring contraction tempo to the desired neural target, cortical *versus* subcortical, may enhance the precision of strength-based interventions. Future work should examine the durability and functional transfer of these adaptations, particularly in populations with impaired descending motor pathways.

### Cortico-reticular and reticulospinal changes

Distinct effects of training modality were evident in subcortical circuits, providing new insights into cortico-reticulospinal contributions to strength adaptation. Specifically, SP-RT, in contrast to MP-RT, preferentially engaged cortico-reticular and reticulospinal circuits. This was evidenced by increased ICAR values, enhanced StartReact responses and improved RFD following VSS compared with MP-RT. Although these measures are indirect, they collectively suggest a greater reliance on subcortical mechanisms. These findings align with animal studies showing RST adaptations following RT (Glover & Baker, 2020) and support the view that self-regulated movement tempo may stimulate different neural pathways from externally paced or MP-RT.

The dynamic and less constrained nature of SP-RT likely requires greater reliance on intrinsic motor control systems that integrate proprioceptive feedback and internally generated timing signals (Dean, 2013; Goudini et al., 2019). Repeated sessions over Weeks 1–3 may strengthen synaptic contributions within the reticular formation, thereby enhancing the efficiency of descending subcortical motor command. The enhanced StartReact effect observed after SP-RT supports this interpretation, consistent with the role of the pontomedullary reticular formation in releasing pre-programmed motor commands under startling conditions (Carlsen & Maslovat, 2019; Carlsen et al., 2004; Valls-Solé et al., 1999). By bypassing cortical pathways, this mechanism more efficiently recruits alpha-motoneurons, contributing to the observed improvements in motor output (Carlsen & Maslovat, 2019).

RFD in the earliest contraction window (0–50 ms) showed a stimulus-specific enhancement following SP-RT, with progressively greater increases under VSS compared with both VnSS and VS. Importantly, this enhancement was greater in SP-RT than in MP-RT and Control at Weeks 2 and 3, indicating a training modality-dependent adaptation. In contrast, MP-RT did not demonstrate stimulus-selective facilitation, indicating a modality-specific adaptation of neural drive. The selective enhancement observed following SP-RT is consistent with greater engagement of the RST, which is well suited for ballistic actions by providing a scalable drive for force modulation and coordinated multi-muscle activation (Glover & Baker, 2022). The RST's anatomical organization favours rapid force generation (Atkinson et al., 2022), and its neurons increase firing rates with higher force demands, suggesting SP-RT may better reinforce explosive output than the slower-paced MP-RT. Importantly, the RST remains capable of driving forceful contractions even with corticospinal impairment, underscoring its role in rapid, high-force tasks (Baker, 2011). The faster, self-directed contractions of SP-RT may also facilitate recruitment of fast-twitch fibres and increased neuromuscular drive, consistent with evidence that higher-velocity training enhances RFD through improved motor unit recruitment and firing rates (Maffiuletti et al., 2016). Similarly, fast-eccentric RT appears to be more effective in improving RFD than slower eccentric modalities (Stasinaki et al., 2019).

In contrast, the externally paced and slower tempo of MP-RT is more likely to reinforce corticospinal

**Table 6. Exact *P* values for within- and between-group interaction contrasts across time**

**A. Within-group interaction contrasts**

| Domain | Outcome | | Metronome-paced resistance training (MP-RT) group (*n* = 10) | | | Self-paced resistance training (SP-RT) group (*n* = 10) | | | Control group (*n* = 10) | | |
|---|---|---|---|---|---|---|---|---|---|---|---|
| | | | Baseline →W1 | Baseline →W2 | Baseline →W3 | Baseline →W1 | Baseline →W2 | Baseline →W3 | Baseline →W1 | Baseline →W2 | Baseline →W3 |
| Cortico-reticulospinal | ICAR | | 1.000 | 1.000 | 1.000 | **0.0210** | **<0.001** | **<0.001** | 1.000 | 1.000 | 1.000 |
| | StartReact effect (ms) | | 1.000 | **0.0106** | 1.000 | **<0.001** | **<0.001** | **<0.001** | 1.000 | 1.000 | 1.000 |
| | RFD 0–50 ms (N s$^{-1}$) | VS | 1.000 | 1.000 | 1.000 | 1.000 | 0.676 | 0.968 | 1.000 | 1.000 | 1.000 |
| | | VnSS | 1.000 | 1.000 | 1.000 | 1.000 | 0.875 | 1.000 | 1.000 | 1.000 | 1.000 |
| | | VSS | 1.000 | 1.000 | 0.367 | **<0.001** | **<0.001** | **<0.001** | 1.000 | 1.000 | 1.000 |
| | | VSS − VnSS | 0.5588 | 0.5404 | 0.3784 | **<0.001** | **<0.001** | **<0.001** | 0.9132 | 1.000 | 1.000 |
| | | VSS − VS | 0.5604 | 0.7658 | 0.6861 | **0.00170** | **0.00210** | **<0.001** | 1.000 | 1.000 | 1.000 |
| | RFD 50–100 ms (N s$^{-1}$) | VS | 1.000 | 1.000 | 1.000 | 1.000 | 0.406 | 1.000 | 0.400 | 0.391 | 0.873 |
| | | VnSS | 0.336 | 1.000 | 1.000 | 1.000 | 0.852 | 1.000 | 1.000 | 1.000 | 1.000 |
| | | VSS | 1.000 | 1.000 | 1.000 | 0.0736 | **0.0812** | **0.00190** | 1.000 | 1.000 | 1.000 |
| Intracortical | SICI (% test response) | | 0.5116 | **<0.001** | **<0.001** | 1.000 | 1.000 | 1.000 | 1.000 | 1.000 | 1.000 |
| | ICF (% test response) | | **0.0265** | **0.00170** | **<0.001** | 1.000 | 0.764 | 1.000 | 1.000 | 1.000 | 1.000 |
| Corticospinal | CSE (AURC, a.u.) | | **0.00480** | **0.00210** | **< 0.001** | 1.000 | 1.000 | 1.000 | 1.000 | 1.000 | 1.000 |
| | cSP (AURC, a.u.) | | **<0.001** | **<0.001** | **<0.001** | 0.828 | 0.515 | 0.4831 | 1.000 | 1.000 | 1.000 |
| | AMT SI (%) | | **<0.001** | **<0.001** | **<0.001** | 0.304 | **0.0216** | **0.00540** | 1.000 | 1.000 | 1.000 |

**B. Between-group interaction contrasts**

| Domain | Outcome | | MP-RT *vs.* SP-RT | | | MP-RT *vs.* Control | | | SP-RT *vs.* Control | | |
|---|---|---|---|---|---|---|---|---|---|---|---|
| | | | Δ Baseline →W1 | Δ Baseline →W2 | Δ Baseline →W3 | Δ Baseline →W1 | Δ Baseline →W2 | Δ Baseline →W3 | Δ Baseline →W1 | Δ Baseline →W2 | Δ Baseline →W3 |
| Cortico-reticulospinal | ICAR | | 0.0676 | **0.00520** | **0.0351** | 1.000 | 1.000 | 1.000 | 0.132 | **0.0241** | **0.0188** |
| | StartReact effect (ms) | | **0.0173** | **<0.001** | **<0.001** | 1.000 | 0.184 | 1.000 | **0.00210** | **0.00440** | **0.00570** |
| | RFD 0–50 ms (N s$^{-1}$) | VS | 1.000 | 1.000 | 1.000 | 1.000 | 1.000 | 1.000 | 1.000 | 1.000 | 1.000 |
| | | VnSS | 1.000 | 1.000 | 1.000 | 1.000 | 1.000 | 1.000 | 1.000 | 1.000 | 1.000 |
| | | VSS | 0.0719 | **0.00160** | **0.00560** | 1.000 | 1.000 | 1.000 | **0.0246** | **<0.001** | **<0.001** |
| | | ΔVSS − ΔVnSS | 0.2470 | **0.0246** | 0.1319 | 1.000 | 1.000 | 1.000 | 0.1598 | **0.0056** | **0.0256** |
| | | ΔVSS − ΔVS | 0.327 | 0.164 | **0.0453** | 1.000 | 1.000 | 1.000 | 0.1138 | 0.0941 | **0.0448** |
| | RFD 50–100 ms (N s$^{-1}$) | VS | 1.000 | 1.000 | 1.000 | 1.000 | 1.000 | 1.000 | 1.000 | 0.302 | 1.000 |
| | | VnSS | 1.000 | 1.000 | 1.000 | 1.000 | 1.000 | 1.000 | 1.000 | 0.302 | 1.000 |
| | | VSS | 1.000 | 1.000 | 0.494 | 1.000 | 1.000 | 1.000 | 0.297 | 0.678 | 0.0659 |
| Intracortical | SICI (% test response) | | 0.116 | 0.0552 | **<0.001** | 0.472 | **0.00820** | **<0.001** | 1.000 | 1.000 | 1.0000 |
| | ICF (% test response) | | 1.000 | 0.691 | 0.0996 | 0.388 | 0.0932 | **0.0156** | 1.000 | 1.000 | 1.000 |
| Corticospinal | CSE (AURC, a.u.) | | 0.104 | **0.0313** | **0.00420** | 0.0870 | 0.0980 | **0.00360** | 1.000 | 1.000 | 1.000 |
| | cSP (AURC, a.u.) | | 0.0688 | 0.0816 | 0.259 | **<0.001** | **<0.001** | **0.00900** | 1.000 | 1.000 | 1.000 |
| | AMT SI (%) | | 0.849 | 0.788 | 0.495 | **0.0384** | **0.00770** | **<0.001** | 1.000 | 0.934 | 0.177 |

All *P* values are Bonferroni-adjusted. Δ denotes change. AMT, active motor threshold; a.u., arbitrary units; AURC, area under the recruitment curve; cMEP, contralateral motor-evoked potential; CSE, corticospinal excitability; cSP, cortical silent period; ICAR, ipsilateral to contralateral motor-evoked potential amplitude ratio (iMEP/cMEP, unitless); ICF, intracortical facilitation; iMEP, ipsilateral motor-evoked potential; M$_{MAX}$, maximum compound action potential; MP-RT, metronome-paced resistance training; RFD, rate of force development; SI, stimulation intensity; SICI, short-interval intracortical inhibition; SP-RT, self-paced resistance training; VART, visual-auditory reaction time; VnSS, visual non-startling stimulus; VS, visual stimulus; VSRT, visual-startle reaction time; VSS, visual-startling stimulus.

contributions, supporting precise force modulation for tasks that demand higher cognitive effort and fine motor control rather than ballistic output (Glover & Baker, 2022; Kantak et al., 2012). The RST's role in enhancing rapid force is particularly evident during startling stimuli. Loud sounds preferentially activate the pontomedullary reticular formation, providing strong excitatory drive to motoneurons through mono- and disynaptic pathways (Carlsen et al., 2004; Koch et al., 1992), with minimal corticospinal involvement and consistent evidence for subcortical mediation of the StartReact effect in non-human primates and humans (Neumann et al., 2025; Tapia et al., 2022). This accelerates motor unit recruitment and discharge, increasing RFD (Del Vecchio, Negro, et al., 2019; Škarabot et al., 2022). Although predominantly subcortical, cortical contributions to

StartReact cannot be excluded (Marinovic & Tresilian, 2016; Škarabot et al., 2022) and may even amplify RST output via cortico-reticular connections (Darling et al., 2018; Fregosi et al., 2017).

Finally, the preferential increase in ICAR following SP-RT supports greater engagement of cortico-reticulospinal circuits. Although these markers remain indirect, they suggest a shift in balance towards RST contributions when training is performed at self-selected tempo. Such adaptations may be particularly relevant in rehabilitation, where preserved reticulospinal pathways can support recovery when corticospinal integrity is reduced.

### Intracortical and corticospinal changes

MP-RT produced robust corticospinal adaptations. Increases in CSE and reductions in SICI were observed, aligning with systematic reviews and meta-analyses reporting a decrease in SICI (Gordon et al., 2024), and increase in CSE following MP-RT (Gómez-Feria et al., 2023; Gordon et al., 2024). These effects likely reflect a reduction in GABAergic inhibition alongside increased corticospinal plasticity, consistent with studies showing that complex, attention demanding, externally structured motor tasks enhance cortico-spinal responsiveness (Pearce & Kidgell, 2009). MP-RT, which requires strict synchronization of concentric and eccentric phases, may heighten attentional demands and sensorimotor integration, reinforcing corticospinal plasticity (Ackerley et al., 2011; Leung et al., 2017; Perrey, 2013). Moreover, the slower tempo of MP-RT, particularly during eccentric contractions, prolongs TUT and necessitates greater corticospinal drive to over-come spindle-mediated inhibition, thereby facilitating increased CSE (Duchateau & Enoka, 2016; Lepley et al., 2017). Unlike SP-RT, which is supposed to broadly engage both cortical and subcortical regions, MP-RT is thought to predominantly activate premotor and supplementary motor areas (Gerloff et al., 1998; Leung et al., 2017; Sanes & Donoghue, 2000). This targeted activation likely strengthens corticospinal output by placing higher demands on motor planning and precision. Moreover, because cortical output is modulated by the balance of inhibitory–excitatory circuits (Ni & Chen, 2008; Rothwell et al., 2009), the observed reduction in SICI may reflect a release of cortical inhibition, thereby facilitating cortical representation activation and enhancing excitatory drive to optimize motor performance and strength (Floeter & Rothwell, 1999; Zoghi & Nordstrom, 2007).

Although MP-RT reduced cSP, consistent with previous observations following short-term RT (Coombs et al., 2016; Kidgell & Pearce, 2010; Mason et al., 2017), and increased ICF relative to control, neither measure differed significantly between MP-RT and SP-RT. Thus, cSP and ICF do not reflect modality-specific divergence. Instead, the clearest distinctions between the training modalities emerged in CSE and SICI.

### Limitations

This study is the first to examine how RT pacing influences cortico-reticular and reticulospinal excitability over a 3 week period, providing new insights into pathway-specific adaptations. However, several limitations should be considered when interpreting these findings. Although we controlled for age, sex, handedness, menstrual cycle phase and training time to reduce confounding, other factors such as mechanical loading variation, metabolic responses and hormonal fluctuations may have contributed to the observed changes. Our sample was restricted to healthy, young, RT-naïve participants, which may limit generalisability to trained individuals, older adults or clinical groups. The relatively small sample size, a limitation inherent to intensive neurophysiological studies, should be noted. Although it was determined *a priori* consistent with prior literature and supported by robust analytic approaches, the modest cohort may limit the generalisability; therefore, replication in larger cohorts is warranted to confirm and extend these findings. Sedentary status was self-reported, and unmeasured differences in baseline activity or fitness could have influenced adaptations.

Moreover, RFD was assessed only during visually cued explosive contractions within the StartReact protocol to probe reticulospinal excitability, rather than during self-initiated efforts. Consequently, it remains unclear whether the observed neural adaptations extend to voluntary rapid force production. In addition, although the study was powered *a priori* to detect training-related changes in reticulospinal excitability using a Group × Time interaction framework, with the StartReact effect as the primary indicator, it was not specifically powered to maximize sensitivity for higher-order Group × Time × Stimulus interactions involving RFD, as there are no prior effect-size estimates for such designs. Accordingly, future studies specifically powered to detect higher-order interactions will be important to confirm and extend these findings.

Another limitation of the present study is that lifting velocity or training execution was not verified with detailed kinetic, or kinematic analyses. Instead, we relied on TUT and volume load to confirm that the intended contrast between groups was maintained. Instrumented verification of tempo would strengthen confidence in the interpretation of modality-specific effects. Future studies should directly quantify the lifting velocity or tempo and employ instrumented approaches to

provide more detailed characterization of the mechanical stimulus. Furthermore, we did not manipulate contraction velocity in SP-RT, and whether maximal intent amplifies strength and neurophysiological adaptations remains to be determined.

Although sex was balanced across groups, the unequal distribution within groups remains a limitation. A more even sex representation within each group would have been preferable and could strengthen generalisability.

All measurements were obtained from the biceps brachii, a proximal flexor with relatively strong reticulospinal contributions. Since corticospinal and reticulospinal influence varies across muscle groups, these findings should not be generalized to distal or extensor muscles. Future studies are required to test whether similar adaptations occur in muscles with different descending control. Moreover, it should be noted that RST excitability was assessed using indirect measures, including the StartReact effect and RFD. While these approaches provide valuable insight into reticulospinal drive, they do not permit direct quantification of RST activity. Similarly, iMEPs elicited with TMS index cortico-reticulospinal excitability without specifically isolating RST function. The cMEPs and iMEPs were recorded from different limbs, which ensured both measures reflected output from the same trained hemisphere but does not exclude possible contributions from diffuse bilateral cortico-reticular projections. Additionally, peripheral adaptations such as altered motoneuronal excitability may have influenced MEP characteristics. Future studies incorporating bilateral recordings would provide a more comprehensive assessment of hemispheric and spinal contributions to training-induced plasticity.

Our findings should therefore be interpreted with these limitations in mind. Finally, this study examined short-term neural adaptations, but the long-term retention, progression or reversibility of these changes remains unknown and should be investigated.

Despite these considerations, this study offers important evidence on how different approaches to RT shapes neuromuscular divergent adaptations, underscoring the need for further exploration of its mechanistic and practical implications.

## Conclusion

SP-RT appeared to preferentially engage the RST relative to MP-RT, as reflected by changes in the StartReact effect along with changes in RFD during VSS. This was accompanied by increases in cortico-reticular excitability indexed by ICAR. In contrast, MP-RT produced greater reductions in SICI and larger increases in CSE compared with SP-RT and Control, consistent with enhanced corticospinal and intracortical plasticity. Together, these findings indicate that training pacing influences the neural locus of early strength adaptation, with SP-RT accompanying greater subcortical pathway engagement and MP-RT favouring corticospinal mechanisms. Given the functional relevance of these adaptations, training modality should be considered when designing strength training and rehabilitation programmes.

## Perspective

This study provides novel evidence that contraction tempo during RT induces pathway-specific neural adaptations within the descending motor system. By comparing MP-RT with SP-RT across a 3 week intervention, we observed distinct neuroplastic changes reflecting differential engagement of corticospinal and cortico-reticulospinal circuits. MP-RT, characterized by externally paced contractions, was associated with enhanced CSE, and reduced intracortical inhibition, consistent with disinhibition of motor cortex circuits and enhanced volitional drive. This pattern aligns with early-phase training adaptations and learning-related cortical plasticity. In contrast, SP-RT elicited responses consistent with increased subcortical drive, reflected in augmented StartReact effect, greater RFD and elevated ICAR, suggesting enhanced excitability of reticulospinal pathways. The reliance on cortico-reticulospinal output under self-determined conditions highlights tempo as a means to differentially engage cortical *versus* subcortical levels of the motor hierarchy. Translationally, SP-RT may provide a time-efficient strategy to target subcortical pathways, which could be beneficial in populations with impaired corticospinal integrity (e.g. stroke), whereas MP-RT may be preferable when strengthening cortical output and precise motor control is the objective. Future work should examine the persistence and transferability of these adaptations, and how tempo-driven training parameters can be manipulated to optimize motor system engagement for clinical and performance applications.

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

## Additional information

### Data availability statement

Data supporting the findings of this study are available from the corresponding author upon reasonable request.

### Competing interests

The authors declare that they have no competing interests.

### Author contributions

Y.A. and D.J.K. conceptualized and designed the study; Y.A., D.J.K., U.S. and M.R. conducted the experiments while Y.A., D.J.K. and S.W. analysed the data. The interpretation of findings was performed by Y.A., D.J.K., and S.W. Y.A., A.K.F. and D.J.K. created the figures; Y.A. and D.J.K. drafted the manuscript. The manuscript was further reviewed and edited by Y.A., J.T., A.K.F., U.S., M.R., G.H., S.W. and D.J.K. Finally, all authors approved the final version of the manuscript.

### Funding

Y. Akalu, M. Rostami and U. Siddique were supported by a Monash University Graduate Scholarship. J. Tallent was supported by an International Leverhulme Fellowship Award and S. Walker is supported by funding from the Academy of Finland #350528. This study was funded by an NHMRC Equipment Grant awarded to D.J.K.

### Acknowledgements

The authors gratefully acknowledge Dr Eric J. Frazer for his insightful editorial input and expert review of the manuscript.

Open access publishing facilitated by Monash University, as part of the Wiley - Monash University agreement via the Council of Australasian University Librarians

### Keywords

ICAR, iMEP, resistance training, RST, StartReact protocol

## Supporting information

Additional supporting information can be found online in the Supporting Information section at the end of the HTML view of the article. Supporting information files available:

**Peer Review History**

