## [Peer Review History · The Journal of Physiology]

Resistance Training Tempo Selectively Modulates Corticospinal and Reticulospinal Excitability in Humans

Yonas Akalu, Jamie Tallent, Ashlyn K Frazer, Ummatul Siddique, Mohamad Rostami, Glyn Howatson, Simon Walker, and Dawson J Kidgell

DOI: 10.1113/JP289141

Corresponding author(s): Dawson Kidgell (dawson.kidgell@monash.edu)

Review Timeline:

Submission Date:	22-Apr-2025
Editorial Decision:	31-Jul-2025
Revision Received:	16-Sep-2025
Editorial Decision:	10-Oct-2025
Revision Received:	30-Nov-2025
Editorial Decision:	22-Jan-2026
Revision Received:	19-Feb-2026
Accepted:	24-Feb-2026

Senior Editor: Richard Carson

Reviewing Editor: Kathy Ruddy

Transaction Report:

Dear Dr Kidgell,

Re: JP-RP-2025-289141 "Resistance Training Tempo Selectively Modulates Corticospinal and Reticulospinal Excitability in Humans" by Dawson J Kidgell, Yonas Akalu, Jamie Tallent, Ashlyn K Frazer, Ummatul Siddique, Mohamad Rostami, Glyn Howatson, and Simon Walker

Thank you for submitting your manuscript to The Journal of Physiology. It has been assessed by a Reviewing Editor and by 3 expert referees and we are pleased to tell you that it is potentially acceptable for publication following satisfactory major revision.

LANGUAGE EDITING AND SUPPORT FOR PUBLICATION: If you would like help with English language editing, or other article preparation support, Wiley Editing Services offers expert help, including English Language Editing, as well as translation, manuscript formatting, and figure formatting at www.wileyauthors.com/eoo/preparation. You can also find resources for Preparing Your Article for general guidance about writing and preparing your manuscript at www.wileyauthors.com/eoo/prepresources.

REVISION CHECKLIST:

We look forward to receiving your revised submission.

Yours sincerely,

Richard Carson
Senior Editor
The Journal of Physiology

REQUIRED ITEMS

1) - Author photo and profile. First or joint first authors are asked to provide a short biography (no more than 100 words for one author or 150 words in total for joint first authors) and a portrait photograph. These should be uploaded and clearly labelled together in a Word document with the revised version of the manuscript. See Information for Authors for further details.

2)- Your paper contains Supporting Information of a type that we no longer publish, including supplementary tables and figures. Any information essential to an understanding of the paper must be included as part of the main manuscript and figures. The only Supporting Information that we publish are video and audio, 3D structures, program codes and large data files. Your revised paper will be returned to you if it does not adhere to our Supporting Information Guidelines.

3) - Papers must comply with the Statistics Policy: https://jp.msubmit.net/cgi-bin/main.plex?form_type=display_requirements#statistics.

In summary:

- If $n \leq 30$, all data points must be plotted in the figure in a way that reveals their range and distribution. A bar graph with data points overlaid, a box and whisker plot or a violin plot (preferably with data points included) are acceptable formats.
- If $n > 30$, then the entire raw dataset must be made available either as supporting information, or hosted on a not-for-profit repository, e.g. FigShare, with access details provided in the manuscript.
- 'n' clearly defined (e.g. x cells from y slices in z animals) in the Methods. Authors should be mindful of pseudoreplication.
- All relevant 'n' values must be clearly stated in the main text, figures and tables.
- The most appropriate summary statistic (e.g. mean or median and standard deviation) must be used. Standard Error of the Mean (SEM) alone is not permitted.
- Exact p values must be stated. Authors must not use 'greater than' or 'less than'. Exact p values must be stated to three significant figures even when 'no statistical significance' is claimed.

EDITOR COMMENTS

Reviewing Editor:

All three reviewers recognised the novelty and methodological strength of the study, particularly the integration of a well-structured training paradigm with detailed neurophysiological assessments. The differentiation between metronome-paced and self-paced resistance training and the potential implications for targeting specific motor pathways were viewed as important and promising contributions, especially in the context of neurorehabilitation.

Despite this overall enthusiasm, the reviewers raised several consistent concerns that should be addressed. A major point across reviews was the tendency to overgeneralise the findings beyond the biceps brachii. The reviewers emphasised that claims about broader motor system effects should be more cautiously framed unless supported by additional data. There were also shared concerns about the discrepancy between your a priori power analysis and the final sample size. While the observed effects appear strong, a more transparent justification of the sample size and clearer reporting of statistical details-including effect sizes and post-hoc comparisons-will strengthen the manuscript's rigor.

Several methodological aspects would benefit from further clarification. These include the specifics of the EMG and TMS protocols, the rationale for coil selection, how movement timing and training adherence were monitored, and whether training was bilateral or unilateral. The interpretation of the StartReact and iMEP data also prompted skepticism, particularly regarding the very short reaction times reported and the degree to which these measures reflect reticulospinal drive. A more cautious and physiologically grounded interpretation is advised.

Finally, the reviewers noted that the manuscript is at times repetitive, especially in the Introduction and Discussion. A more concise and focused presentation would enhance readability. Attention to figure and table formatting-such as label clarity and consistency of units-was also recommended.

Senior Editor:

It is noted that individual data points have been shown on only one figure. This is not in accordance with the Statistical Policy of the Journal.

Since a longitudinal design was employed, it should also be possible to show for each individual the variation across time points.

Such representations are to be preferred to the point plus error bar format used presently.

Since the number of participants is very small, it is feasible to depict data for all participants on the same figure.

The basis upon which the error bars were generated is not indicated in the current figure captions.

Please refer to the Statistical Policy of the Journal in this regard.

In a related vein, please note that supplementary figures are not encouraged.

All relevant material should be included with the manuscript.

While the logistical challenges associated with the conduct of studies of this nature can be appreciated, the number of participants allocated to each group is extremely small. It is therefore difficult to conclude that the effects estimated in the samples are representative of the "true" effects present in the population.

The referees have raised several concerns that relate to the statistical models that were applied, highlighting in particular that the outcomes reported do not support in a straightforward manner the inferences that have been made.

In this regard, I wish to place particular emphasis on an observation made by Referee #1 in relation the associations between measures of corticospinal/reticulospinal excitability and behavioural measures of performance.

It is indeed the case that the presence of correlation does not imply causation. Given causation however, the presence of correlation is to be anticipated.

The referees and Reviewing Editor have noted several matters that should be addressed.

In addition to these, I wish to emphasise as a requirement that the authors demonstrate the presence of robust associations between select measures of corticospinal/reticulospinal excitability - defined on an a priori basis, and the key behavioural measure of performance.

In the absence of such, it seems to me unlikely that a reader will be convinced that (particularly in a very small sample) the authors' preferred inferences are sustainable.

REFeree COMMENTS

Referee #1:

General Comment:

The present study examines the specific neurophysiological adaptations following two different resistance training (RT) protocols that differ in training tempo (metronome-paced vs self-paced). The novelty of the study is the inclusion of methods that allow to measure reticulospinal and cortico-reticulospinal adaptations following a short-term RT period (i.e. longitudinal design), something that, as the authors acknowledge, has never done before. The results not only support a hypothesis initially proposed in a 2020 animal study showing that RT leads to reticulospinal adaptations, but also shows how external auditory pacing during RT alters the central nervous system demands, driving neural adaptations upstream the brainstem, at the cortical level. The rationale and the results are well described and discussed. The methods are robust and in accordance to the specific procedures required for the measured outcomes. I would like to commend the authors for their effort in conducting this much needed study. That said, I have some comments and suggestions that I believe will help improve clarity at some points of the manuscript.

See specific comments below:

INTRODUCTION

The introduction is well written and effectively describes the context of the main hypothesis of the manuscript.

As a minor comment, at the point where authors discuss about the variability in corticospinal adaptations to RT (lines 128-129), I suggest referencing original studies that report no changes (or even reductions) in CSE following short-term self-paced resistance training to reinforce the argument (instead of just citing systematic reviews and meta-analysis). For example:

Ansdell, P., Brownstein, C. G., Škarabot, J., Angius, L., Kidgell, D., Frazer, A., Hicks, K. M., Durbaba, R., Howatson, G., Goodall, S., & Thomas, K. (2020). Task-specific strength increases after lower-limb compound resistance training occurred in the absence of corticospinal changes in vastus lateralis. *Exp Physiol*, n/a(n/a). <https://doi.org/10.1113/ep088629>

Colomer-Poveda, D., Romero-Arenas, S., Fariñas, J., Iglesias-Soler, E., Hortobágyi, T., & Márquez, G. (2021). Training load but not fatigue affects cross-education of maximal voluntary force. *Scandinavian Journal of Medicine & Science in Sports*, 31(2), 313-324. <https://doi.org/10.1111/sms.13844>

Giboin, L. S., Weiss, B., Thomas, F., & Gruber, M. (2017). Neuroplasticity following short-term strength training occurs at supraspinal level and is specific for the trained task. *Acta Physiol (Oxf)*, 222(4), e12998. <https://doi.org/10.1111/apha.12998>

METHODS

Lines 234-236: For clarity, were the 1RM and MVF only measured once, at the familiarization session? Or they just practiced at this session and then formally assessed one week later. In

Lines 234-236 you mention that "baseline assessments of cortical, corticospinal, corticoreticulospinal and reticulospinal excitability were conducted" but you do not mention muscular strength measures. While this appears to be clarified in Figure 1, the text as currently written may be confusing to some readers..

Line 301-303: Did you measure RFD during self-initiated MVC? If so, did Self-paced RT induce greater increases in that variable? It is relevant to consider any possible functional differences between training approaches and whether specific neural adaptations may underlie different functional adaptations (which does not seem to be the case if you only analyse maximal strength).

Line 303: The type of force transducer is already mentioned earlier in the methods. To avoid redundancy, consider removing it here.

Line 309: "for Mmax AND subsequent TMS ..."

Line 336: Related to the previous comment regarding RFD adaptations, did the SP-RT group receive any instruction regarding lifting velocity intention? (i.e. whether to lift with maximal intent or not?). Repetition duration during submaximal lifts, determined by lifting intention, could influence strength adaptations (e.g., Hermes and Fry, 2023). If lifting velocity was not maximal (difficult to ensure if you do not measure lifting velocity and/or there is continuous encouragement with specific cues to lift at maximal velocity), do you think that maximum intention lifting would have produced greater strength increases in the SP-RT group?

Hermes MJ, Fry AC. Intentionally Slow Concentric Velocity Resistance Exercise and Strength Adaptations: A Meta-Analysis. *J Strength Cond Res*. 2023 Aug 1;37(8):e470-e484. doi: 10.1519/JSC.0000000000004490. PMID: 37494124.

Line 362-363: Why was a metronome used during TMS measurements? The auditory may have induced anticipation or expectation of the pulse, potentially influencing the results (see a reference below). Moreover taking into account that one group was chronically exposed to this type of auditory cues and its capacity for anticipation could have been influenced by training.

Capozio A, Chakrabarty S, Astill S. The Effect of Sound and Stimulus Expectation on Transcranial Magnetic Stimulation-Elicited Motor Evoked Potentials. *Brain Topogr.* 2021 Nov;34(6):720-730. doi: 10.1007/s10548-021-00867-9. Epub 2021 Sep 6. PMID: 34490506; PMCID: PMC8556164.

Line 398-399: I understand that the PMRF has bilateral synaptic connections with spinal motoneurons, however I wonder why did you decided to measure the iMEPs in the left (untrained) BB rather than in the right trained BB? Would not the latter be more appropriate for identifying strength training induced reticulospinal adaptations projecting to the trained side?

Related to this, did you measure the cross-education effect of training and if it is more pronounced on the SP-RT due to the bilateral nature of the RST?

Line 447-448: Just for clarity, does this refer to additional ICC data apart from the values already presented in the supplementary material, which appear to be based solely on the control group? If you already reported the reliability of these measures in the control group is this additional pilot data necessary?

Line 456-458: Was the 10% of MVF contraction equivalent to 5% of maximal rmsEMG? Or it should be 10% also here?

Line 588: Following the reference that you cite in line 588, could you please specify the type of ICC analysis (model, type and definition)?

Table 1: Are differences in training load just derived from the slightly greater 1RM in the SP-RT group? (volume (sets and reps) were equated) Or the externally imposed (and potentially slower) tempo lead to failure and reduced total number of repetitions in the MP-RT group?

RESULTS

Some minor suggestions to consider in order to reduce the length of the results section:

Lines 729-731: Consider putting the % changes in lines 730-731 between brackets and delete the 732-733 lines.

Lines 733-737: Do you consider it necessary to report non-significant percentage changes for the MP-RT and control groups in the main text? Since absolute values are already presented in the table, removing these details could help reduce the overall length of the results section without compromising clarity.

Lines 739-740: As I suggest in the previous comments, in order to reduce the length of the results sections, you may consider putting the % changes of the SP-RT group together with effect sizes and p values and not include the non-significant % changes of MP-RT and control group (you already show absolute values in the table). That is something you can consider for the results of all the other outcomes.

General comment:

Although I acknowledge that correlation is not causation, I wonder if you have explored any associations between key neurophysiological indices of cortical or reticulospinal adaptations and strength outcomes.

DISCUSSION

Line 797: I do not think that "self-paced resistance training" is a "novel" modality

Line 821: As I commented before, I wonder if you measured RFD adaptations and if there were differences between groups. Although maximal strength is relevant, both force rapidly. It would be interesting and valuable to know whether the differences in tempo led to different adaptations in rapid force production together with the distinct neural adaptations.

Lines 909-911: To which spinal motoneurons are you referring here, if not to alpha motoneurons?, reticulospinal neurons. Maybe you can change it for clarity.

Line 929-939: Check this double citation.

Line 931-935: Please, check the construction of this sentence and improve it for clarity (break it in two sentences?)

Referee #2:

Overall comments:

In my opinion, this is not a well written or conceptualised report, and there are substantial issues with the statistical approach.

The Introduction lacks clarity on a number of points and includes some weak use of the literature. The experimental variable in this study is the type of RT, specifically self-paced vs metronome paced, yet this is a small feature of the introduction. Consequently the Introduction lacks any clear theoretical underpinning rationale for the comparison of these two types of RT (why physiologically would we expect them to be different?).

Important description of the training and testing tasks is basic and lacks extensive information. Some of the methods appear weak (e.g. voluntary activation cannot be assessed by stimulating agonists and antagonists simultaneously) or are hardly described (e.g. RFD of which contractions).

The statistical analysis of the group x time interactions is insufficient to compare the changes of specific groups over time (e.g. group A vs group B). The extensive within group changes presented do not inform these between group interaction effects that are critical to this experiment. Excessive consideration/presentation of percentage changes, rather than the more robust pre post data. Without appropriate statistics the experiment is of limited value.

The paper is also excessively long, despite lacking important information.

Specific comments:

1. The premise within the abstract provides no connection/rationale between the two interventions and the adaptations being studied. Therefore it is unclear how they relate.
2. Why only a 3-week RT programme? Are the neural adaptations likely to be saturated in such a short period of time?
3. Given the variability of the measures, especially for assessing change, I'm unsure if the group sizes are sufficient.
4. L107-8 This is quite weak and an inappropriate reflection of these papers. The Seynes et al 2007 paper found increased muscle CSA after only 20 days of training, and reported some significant changes in architecture after only 10 days of training. The Pearcey review has no information on the contractile apparatus, so has no relevance. Maeo et al (2017 PMID: 29570534) also found changes in CSA after 4 weeks of RT.
5. L108-109 The twitch response data is also quite shaky given the maximum contractions are tetanic in nature.
6. L112-115 The list of mechanisms is quite selective - it is surprising that the possibility and evidence of spinal levels adaptations is not recognised.
7. L117 'moderate and variable changes' is incomplete and unclear. Variable in what way?
8. L119 The comment about and reference of the Adkin et al paper is somewhat misleading without explaining that they were contrasting endurance and resistance training.
9. No clear rationale is put forward for why these two types of RT may have differential effects on the nervous system. No underlying theory is introduced or explained.
10. L128-29 A sweeping and somewhat erroneous sentence. The neural adaptation driving any type of strength gains remain unclear. Would corticospinal changes be expected to fully account for strength gains? and can such a contribution (fully or partially accounting) even be measured in any meaningful way.
11. This sentence also contains grammatical errors.
12. L130-31 'Potential candidate'? the writing lacks clarity without being explicit about candidate for what?
13. L154 'contributes to motor output modulation' appears quite simplistic and unjustified based on the observation of increased activation. Couldn't the increased activation be a response or coincidental effect, rather than a cause as implied.
14. L155 'key sub-cortical driver' appears hyperbole based on the evidence presented. If there is more convincing evidence the authors need to bring that to the fore, otherwise the language needs to be substantially toned down.
15. L161-3 It should be explicit that this is in primates.
16. L166 That between group 'enhanced RFD' alone suggests RST-mediated neural drive is simplistic.
17. L172 'Given the established role...' This seems too strong based on what has been presented.
18. L174-177 There is a major jump in the narrative, from 'how RT modulates RST excitability' (that sounds like a study of mechanism between an intervention and an outcome), to a comparison of the two types of training. See comments above about the lack of rationale, and thus connection between these points.

19. L176 The aim is quite unspecific and descriptive, if the primary aim is to compare the groups/interventions that should be stated explicitly.
20. L178 The basis for this hypothesis has not been introduced or explained.
21. L180-3 Unnecessary hyperbole.

Methods:

22. L191 The sample size calculation seems to be based on a pre-post comparison of one group, when there are 3 groups and extensive comparisons within this study. The sample size calculation needs to be coherent with the aim, unfortunately as above this lacks clarity.
23. Its awkward to have such a sex imbalance in this study.
24. L221 What was involved in stratified randomisation?
25. L231 'Additionally, iMEPs were recorded to enhance data reliability.' Unclear and needs explanation
26. L232 Unclear why this was done during familiarisation?
27. L247- Quite a basic description of the dynamometry, in a section headed EMG, it is unclear what other measurements this set-up relates to. For example the position of the trunk, shoulder, shoulder girdle, wrist and forearm posture are not explained. Consequently whether this is genuinely an isolated assessment of isometric elbow flexion strength with minimal contribution of other much stronger muscle groups is unclear.
28. L251 Unclear; how was the forearm 'aligned with a fulcrum' ? 'padded restraint'? positioned where and standardised how.
29. L289 Stimulation of the brachial plexus generates action potentials in the agonist and antagonists to elbow flexion as well as several other muscle groups (shoulder, wrist and hand). This invalidates measurements of voluntary activation, which requires targeted stimulation of the agonist muscle(s) only, and not opposing muscles.
30. L299 'dominant biceps' highly simplistic, many muscles are involved in elbow flexion, not just 'biceps', which biceps?
31. Unclear how force measurements were recorded and filtered? Any gravity correction?
32. L306 Why only 2 isometric MVCs? This seems quite limited.
33. L313 'wrist flexion from full elbow extension' is confusing. Unclear how or if shoulder and elbow position were controlled in any way. What was the range of motion and how was this assessed?
34. L322 The specifics of the training exercise task is not described in any way. Was it the same as the 1-RM testing exercise or something else?
35. Moreover, with this standing elbow flexion exercise, strength and any training effects are heavily dependent on whole body stabilisation and co-ordination of multiple joints/ muscle groups. In other words this is a non-isolated way to assess and train what is described and presented as a single joint exercise. Its unclear why this model and approach would be chosen.
36. L331 A 3-s lift and 4-s lowering phase is unusually slow and not a normal lifting velocity or cadence. Its unclear why this was chosen. There's also no information on how this was done i.e. instructions, controlled, monitored etc.
37. L336 There is no information on the instructions given to the SP-RT group regarding the training they performed.
38. Was there any quantification (kinetics, kinematics, EMG) of the training to verify that the training was on average performed in this unusually slow manner described. Without a metronome it may be even harder to maintain such a slow pace. As above was there any verification that the two groups did in fact perform the training in the same manner. A different lifting velocity, RFD or time undertension may lead to profoundly different neuromuscular adaptations, irrespective of self-paced or not. This verification of an equivalent mechanical training stimulus seems critical to demonstrating any effect of pacing mechanism.
39. L388 The iMEP measurements were collected during bilateral exercise task. Is the mismatch with unilateral training and testing an issue?
40. L418 The measurement situation for the StartReact protocol is not described.
41. L544 It is unclear what contractions were used to assess RFD. If the two isometric MVCs that is a weak approach.

Results

42. L603-5 Is this describing pre to post changes? The GxT interaction is not followed up, which groups changed differently over time compared to others? This requires interaction contrasts (or similar) to assess the groups. What follows in the next paragraph considers within groups changes, which does not inform about a between group interaction effect.

43. For the strength changes, only the %age changes are displayed in the Figures. Was this the data used for the primary analysis? And if not please display the actual pre and post data.

44. For CSE, and then CSP, the same pattern is repeated, with no genuine follow-up to the G x T interaction. Within-group post-hocs do not show how specific groups compare over time.

45. The results need to be re-analysed with appropriate statistics i.e. interaction contrasts or similar.

Discussion

46. L785 Unfortunately contrary to this statement these two groups have not been directly compared. Three groups are included in the ANOVA, and some significant GXT interactions have been noted. However of the 3 groups, which specific groups show differences over time than any other is unclear. It could be that these GxT interactions are purely reflecting differences between the control group and one of the training groups, but that the two training groups produce equivalent changes over time. Comparison of specific groups is required for a meaningful worthwhile manuscript.

Referee #3:

The current manuscript investigates the different influence of metronome-paced and self-paced resistance training of the biceps brachii on the two main descending motor systems. By comparing three different groups, they found remarkably strong effects showing an increased corticospinal drive after metronome-paced resistance training, while self-paced resistance training appeared to lead to an enhanced reticulospinal drive to the biceps. The authors set up a nice paradigm with a 3-week training paradigm and regular strength and thorough neurophysiological assessments. The corticospinal system was assessed by investigating cortical excitability, cortical silent period, and intracortical excitability (short intracortical inhibition and intracortical facilitation). The reticulospinal and cortico-reticulospinal system was assessed by StartReact and ipsilateral motor evoked potentials (iMEP) or particularly the iMEP vs. cMEP amplitude ratio.

The choice of methodology is quite elaborate and impressive. Additionally, the outcome is quite exciting as this suggests a training intervention possible to target the different motor pathways more specifically. If this holds for a broader selection of muscles, this can impact future rehabilitation interventions for neurologically impaired patients, e.g., Spinal cord injury or stroke.

Nevertheless, I have more specific comments, questions, and suggestions for the manuscript.

Major comments:

L195 The authors perform an a priori power calculation to assess the required number of participants. The result they find is 27. They decide to include 10 participants in each of the three groups. However, it appears that 27 participants per group would be required. Why did the authors decide to reduce the number of participants from the number calculated by the power calculation? The data nevertheless show a strong effect even with the lower number of participants.

L230 I understand single-arm dumbbell exercises were performed. Were they performed bilaterally or only for the dominant biceps brachii?

L261 and L497 If I understand correctly, cMEPs assessing the corticospinal system were recorded from the dominant/right biceps brachii, while iMEPs for the reticulospinal system were recorded from the non-dominant/left biceps brachii. This means you are comparing signals from different muscles, which could complicate the interpretation. Recording from the same muscle would strengthen the comparison between systems.

Additionally, related to the question above, if strength training was only performed unilaterally to the dominant side, this would imply an increased reticulospinal drive to the non-dominant side when training the dominant side. This would be a different finding, correct? Therefore, I assume it must be bilateral.

L447 I appreciate the authors addressing the problem of StartReact test-retest reproducibility. The pilot study, however, needs further clarification.

1) Were the 8 healthy adults in the pilot study additional participants or part of the control group? If part of the control group, why weren't all 10 included?

2) Where is further information on the data? Why does this not appear anywhere else in the manuscript?

3) The ICC shows only moderate reproducibility for VART and a good reproducibility for VSRT. Therefore, the StartReact effect reproducibility might be moderate. Why is this not discussed?

L822 The greatest impact, in my opinion, of this manuscript is the potential of translating this into rehabilitative or training methods for especially neurologically impaired participants, e.g., Spinal cord or stroke. It would be great to further discuss the authors thoughts on the feasibility of such a training program in a target population.

Figure 6 Reaction Time for StartReact trials appear very short (VSRT between 60 and 35 ms, and VART between 60 and 80 ms). Castellotte and Kofler 2018 show reaction times of around 90 ms of the biceps brachii for startling stimuli. Sangari and Perez 2020 even showed VART of around 151.1 {plus minus} 22.1 ms and VSRT 127 {plus minus} 21.7 ms for elbow flexors of healthy controls. Similarly, Eilfort et al. 2025 presented startling reaction times around 85 ms and non-startling reaction times at around 120 ms. The startle reflex itself is known to be present at around 60 ms in the sternocleidomastoid. Therefore, the reaction times reported here do not quite seem physiologically feasible. Can the authors explain?

General Would the authors expect a similar result and pattern in muscles with a lower inertial reticulospinal drive or even a strong corticospinal drive, such as the tibialis anterior, the first dorsal interosseus, or even the triceps brachii? Generally, the authors appear to generalize their results, while they should emphasize that their results are limited to the biceps brachii.

General The manuscript's introduction and discussion show several redundancies. The length of the manuscript could be more concise and therefore improve the readability.

Moderate comments:

L152-L155 The reticulospinal system and its role and importance in motor control is introduced already in L134, and 138-150. The additional paragraph is redundant and appears to stop the flow of the reading.

L335 The authors describe verbal encouragement to ensure adherence to tempo and effort for the MP group. However, it would be interesting to investigate adherence to the protocol. Did the participants adhere to the timing? Did they fulfil the movement to the same extent as the SP group, or vice versa? Was movement kinematics, timing measured, and how many trials met the criteria of MP-RT out of the total?

Was the movement execution (e.g., range, speed, technique) consistent across all groups?

L361 What exact number of stimuli were used for the TMS assessment at the different stimulator intensities? Why was there 5-10 reported and not an exact number? Did this vary?

L371 How were the stimulation intensities (130% AMT, 150% AMT, and 170% AMT)? The Authors cite Carson et al. 2013, however, I wonder why an entire recruitment curve with 5% stimulator output intervals is not chosen. Did the authors ensure a plateau of the MEP amplitude? Was MEP max determined?

L393 Why was a different coil used (Figure 8) for iMEP/cMEP analysis compared to the other TMS protocols (circular)?

L398 Why was the iMEP (or ICAR) protocol optimized for the cMEP, especially why was hotspotting performed for the cMEP? I understand that the iMEP was normalized by cMEP and understand the reasoning. However, it still appears that iMEP is the main outcome in this part of the analysis.

L431 The authors report a sound intensity level of 115-120 dB for the startling stimulus. Why did this vary? A 5 dB difference is significant at such high intensities, given the logarithmic nature of the scale.

L605 Authors report a significant group*time interaction, however, with no significant main effect of group nor time. Additionally, an increase in strength for both the MP and SP groups is reported. Is there a significantly steeper increase in the MP group vs the SP group for 1RM?

L787-L788 Redundancy to the sentence before. The sentence points out the novelty of the study. However, "This study presents the first..." also points out novelty.

L801-L803 Redundant to L789 - L195.

L805 Redundant to L792.

General Overall, the discussion appears to repeat itself, constantly showing the increased strength, increased cSP, SIC1, and ICF in MP-RT and increased StartReact effect and ICAR for the SP-RT. This is the main finding; however, it keeps being repeated and makes it hard to read.

Minor comments:

L67 Abbreviations used in the Abstract should be introduced there as well, e.g., SIC1 and cSP.

L111 "Despite well-documented neural adaptations to short-term RT," Reference is missing here.

L129 There is a spelling error in this sentence ("do not")

L145 The reference Valls-Sole et al. 1995 ("Reaction time and acoustic startle in normal human subjects") might be considered as a more suitable reference compared to Baker and Perez 2017.

L149 Indeed, the reticulospinal tract is known to project bilaterally. However, why is this important for the argumentation of iMEPs as measure for cortico-reticulospinal excitability? To my understanding, the ipsilateral projection is what defines this here.

L161 Please clarify the sentence here.

L227 A closing parenthesis is missing after "body mass."

L254 Please specify which electrodes were used.

L265 Please specify which EMG system (e.g., transmitter model) was used.

L285 The abbreviation ICC should be introduced at first use. Please revise accordingly.

L297 There is no need to reintroduce abbreviations once they have been defined previously.

L342 To my understanding, MEPs are recorded from the MEP rather than elicited. MEPs are elicited from the brain.

L356 Do the scalp markings persist for three weeks?

L359 You introduce the abbreviation MSO for "maximum stimulator output," but do not use it again. Consider removing it if it's not used later.

L414 Do the 10 TMS stimuli correspond to 10 iMEP stimuli, or is that the total number?

L433 Of the 30 reported trials, were the conditions equally distributed (i.e., 10 per condition)?

L448 As noted earlier (L285), ICC should be introduced at first use.

L471 Why was the amplitude at 100% AMT not analyzed?

L522

These abbreviations were already introduced earlier. Please avoid repeating them.

L524 These abbreviations were already introduced earlier. Please avoid repeating them.

L585 These abbreviations were already introduced earlier. Please avoid repeating them.

L595 The abbreviation MVF has been used consistently for "maximum voluntary force." Why is it not used here?

L613 Please cite Figure 1A here or at any other relevant point in the text.

L654 For the reader, there seems to be possible incongruencies between the values recorded here and Figure 3. But this might just appear this way.

L684 There is a typo in Table 2 ("Table 02").

L817 Maybe the authors can review this reference and find a more suitable reference than Fisher et al. 2021.

L910 Did the authors mean spinal interneurons instead of spinal motor neurons?

L942 Here you can add Neumann et al. 2025 ("Cortical involvement in the initiation of movements cued by moderate, but not loud acoustic stimuli: Evidence for subcortical mediation of the StartReact effect") as additional evidence for subcortical responsibility of the StartReact effect in humans in addition to Tapia et al. 2022 from primates.

Figure 1 Please verify that all abbreviations used in the figure are also explained in the legend, and vice versa. Inconsistencies exist, e.g., VA, RT, ICAR.

Figure 7 Check for consistency in spelling (e.g., "cMEPs") and ensure all units are correctly presented.

Figure 8 Use the same y-axis scale in both panels A and B for easier comparison.

Table 1 What unit is used for training duration or volume? Please verify abbreviation consistency, spelling, and bracket formatting.

General For improved readability, I recommend a reduction of the number of abbreviations used.

All in all, this was a very interesting study and manuscript to read.

END OF COMMENTS

Response to comments of senior and reviewing Editors,

Response to Reviewing Editor:

All three reviewers recognised the novelty and methodological strength of the study, particularly the integration of a well-structured training paradigm with detailed neurophysiological assessments. The differentiation between metronome-paced and self-paced resistance training and the potential implications for targeting specific motor pathways were viewed as important and promising contributions, especially in the context of neurorehabilitation.

Response: We thank the editor and reviewers for recognising the novelty and methodological rigor of our study, particularly the differentiation between metronome-paced and self-paced resistance training and its implications for neurorehabilitation.

Comment 1: Despite this overall enthusiasm, the reviewers raised several consistent concerns that should be addressed. A major point across reviews was the tendency to overgeneralise the findings beyond the biceps brachii. The reviewers emphasised that claims about broader motor system effects should be more cautiously framed unless supported by additional data. There were also shared concerns about the discrepancy between your a priori power analysis and the final sample size. While the observed effects appear strong, a more transparent justification of the sample size and clearer reporting of statistical details-including effect sizes and post-hoc comparisons-will strengthen the manuscript's rigor.

Response:

We thank the editor and reviewers for these thoughtful and constructive comments. We have addressed each of the points raised as follows:

➤ **Overgeneralisation beyond the biceps brachii:**

We acknowledge the reviewers' concern regarding overgeneralisation. We agree that our findings are specific to the biceps brachii and that caution is warranted when considering generalisation to other muscles. The reticulospinal tract exerts a stronger influence on proximal flexor muscles such as the biceps brachii, whereas distal hand muscles (e.g., FDI) and certain extensors (e.g., triceps brachii, tibialis anterior) are more strongly controlled by corticospinal pathways (Baker, 2011; Mooney et al., 2023; Eilfort et al., 2023). Consequently, the adaptations observed here may not necessarily be replicated in muscles with weaker reticulospinal drive or greater corticospinal dominance.

In the revised manuscript, we have amended the Discussion to make this limitation explicit and have emphasised that our interpretations apply primarily to the biceps brachii. We also highlight in the Limitations section that future research should investigate whether metronome-paced or self-paced resistance training induces similar or distinct neural adaptations in muscles with differing CST vs RST contributions, such as distal hand or lower-limb muscles. Specifically, the Limitations section (**Page 43, Lines 1007–1011**) now reads:

“All measurements were obtained from the biceps brachii, a proximal flexor with relatively strong reticulospinal contributions. Since corticospinal and reticulospinal influence varies across muscle groups, these findings should not be generalised to distal or extensor muscles. Future studies are required to test whether similar adaptations occur in muscles with different descending control.”

➤ **Power analysis versus final sample size:**

We appreciate the reviewer’s careful reading of our power calculation. To clarify, there is no discrepancy between the *a priori* power analysis (N=27) and the final sample size (N=30; 3 extra participants to ensure adequate power and account for potential attrition). The sample size of 27 reported in the manuscript represents the total number of participants across all three groups, rather than 27 participants per group. In G*Power, repeated-measures designs with within- and between-subjects interactions report the total sample size (N) across all groups, not per group. This can be confirmed from the degrees of freedom (df) in our GPower output (*see snapshot below*): the denominator df is calculated as $(N - k) \times (m - 1)$, where N is the total sample size, k is the number of groups, and m is the number of repeated measurements. Substituting our values (N = 27, k = 3, m = 4) gives $(27 - 3) \times 3 = 72$, which exactly matches the denominator df shown in the GPower output. If 27 participants were per group (total N = 81), the denominator df would be 234, which clearly does not match. Therefore, the GPower calculation confirms that 27 is the total sample size across all groups, corresponding to 9 participants per group.

To ensure transparency, we have revised the Methods section (**Page 8, Lines 177-184**) to make this explicit. The section now reads: “The minimum sample size (N = 27; 9 per group) was determined a priori using G*Power (Faul et al., 2007), for a mixed (Group × Time; 3 groups × 4 time points) design, with $\alpha = 0.05$, power = 0.80, and an effect size (Cohen’s $f = 0.275$), derived from a standard mean difference (Cohen’s $d = 0.55$) for an increase in CSE following RT (Siddique, Rahman, Frazer, Pearce, et al., 2020). The effect

size value, Cohen's *d* was converted to Cohen's *f* ($f = 0.275$) to estimate the effect size appropriate for our three-group design. Following institutional ethical approval, 30 healthy, RT-naïve adults (24 males, 6 females) were recruited, exceeding the minimum estimated sample size of 27, to ensure adequate power and account for potential attrition."

Type of power analysis	
A priori: Compute required sample size - given α , power, and effect size	
Input Parameters	Output Parameters
Determine =>	Noncentrality parameter λ
Effect size f	0.275
α err prob	0.05
Power (1- β err prob)	0.8
Number of groups	3
Number of measurements	4
Corr among rep measures	0.5
Nonsphericity correction ϵ	1
	Critical F
	6.0000000
	Numerator df
	72.0000000
	Denominator df
	27
	Total sample size
	0.8424877
	Actual power

➤ **Reporting of statistical details:**

We appreciate the editor and reviewer's comment regarding statistical reporting. In the revised manuscript, we have provided detailed post-hoc interaction analyses, including both within-group and between-group contrasts. These additional results offer greater transparency and a clearer interpretation of the observed effects. These revisions are reflected throughout the Results, with updated tables and figures presenting the detailed *post-hoc* interaction contrast outcomes. Furthermore, we report generalized eta squared (η^2G) for all fixed effects in the repeated-measures Linear Mixed Model (LMM) and Generalized Linear Mixed Model (GLMM) analyses. We chose η^2G because it accounts for both within- and between-subject variance, making it more appropriate for complex mixed designs that include random effects (Bakeman, 2005; Olejnik & Algina, 2003).

Comment 3: Several methodological aspects would benefit from further clarification. These include the specifics of the EMG and TMS protocols, the rationale for coil selection, how movement timing and training adherence were monitored, and whether training was bilateral or unilateral. The interpretation of the StartReact and iMEP data also prompted skepticism, particularly regarding the very short reaction times reported and the degree to which these measures reflect reticulospinal drive. A more cautious and physiologically grounded interpretation is advised.

Response:

We sincerely thank the editor for these thoughtful and constructive comments. We recognise that methodological clarity and cautious interpretation are critical to the strength and transparency of our work. In response, we have substantially revised both the Methods and Discussion sections to provide the necessary clarifications and to ensure that our

interpretations remain firmly grounded in established physiological principles. Specific revisions are outlined below:

➤ **EMG and TMS protocols:**

The revised manuscript now provides clearer methodological detail for both the EMG and TMS protocols. For the EMG protocol, we have added detailed explanations of electrode placement, positioning and stabilisation (*Page 11, Lines 244–263*). The specific type of surface EMG electrodes used is now reported (*Page 11, Lines 264–265*), and the EMG system and transmitter model are specified (*Page 12, Lines 274–276*). For the TMS protocol, we have clarified how the stimulation hotspot was maintained consistently across sessions (*Page 17, Lines 391–393*). We also report the number of trials performed to ensure measurement reliability (*Page 17, Lines 397–398*), as well as the techniques used to maintain temporal consistency in cortical stimulation (*Page 17, Lines 399–400*). These additions ensure greater clarity, transparency, and reproducibility of our experimental methods.

➤ **Rationale for coil selection:**

We appreciate this observation regarding the use of different coils for the iMEP/cMEP protocol versus the other TMS measures. For SICI, ICF, cSP, and CSE we employed a circular coil. This choice was based on several practical and methodological advantages: circular coils are easy to handle, less time-consuming to position, and less sensitive to small changes in placement. Because their stimulation field is deep and broader, locating a reliable “hotspot” for proximal upper-limb muscles such as the biceps is easier and produces robust and reproducible contralateral MEPs, particularly when neuro-navigation is not available. It is also well established and as reliable as figure-of-eight coils for assessment of cortical excitability (Badawy et al., 2011).

By contrast, for iMEP/cMEP recordings we used a figure-of-eight coil. The rationale was twofold. First, when recording iMEPs from an ipsilateral muscle (e.g., left biceps brachii during left-hemisphere stimulation), it is essential to minimise the risk of inadvertent cross-stimulation of the contralateral hemisphere, which would otherwise generate a cMEP. The figure-of-eight coil provides more focal stimulation than a circular coil, reducing current spread across the midline and thereby ensuring that the elicited responses more reliably reflect true ipsilateral projections. Second, iMEP protocols require high-intensity stimulation, and figure-of-eight coils are known to be better tolerated. In particular, Ørskov et al. (2021) reported that participants experienced lower discomfort levels and improved tolerability when using figure-of-eight coils compared to

circular coils. Our approach is consistent with existing literature (e.g., Altermatt et al., 2023; Alagona et al., 2001; Maitland & Baker, 2021; Hu et al., 2024), and balances methodological rigour with participant comfort. We have clarified this rationale in the revised manuscript (*Page 18, lines 431-434*), which now reads: “*For iMEP and contralateral motor-evoked potential (cMEP) recordings, the figure-of-eight coil was preferred to provide greater focality and minimise cross-hemispheric spread, ensuring that responses reflected true ipsilateral rather than contralateral activation.*”

➤ **Monitoring of movement timing and training adherence**

Adherence to tempo and execution was carefully addressed in our protocol and is now described in greater detail in the revised manuscript. For the MP-RT group, participants first underwent a familiarisation phase until they could accurately synchronise their contractions with the metronome beat. During all subsequent testing and training sessions, adherence was ensured through close supervision: if any trial was performed incorrectly, participants were immediately instructed to correct their movement and repeat the movement (*Page 10, Lines 223–229; Page 15, Lines 348–350, 359–361; Page 16, Lines 366–367*). Repetitions were standardised at 3 s concentric and 4 s eccentric, with additional verbal cues provided to maintain effort and consistency. For the SP-RT group, contractions were performed at each participant’s natural pace, but with equivalent levels of supervision. Investigators ensured full range of motion, correct technique, and prevented compensatory movements through real-time observation and verbal feedback. In both groups, training sessions were closely supervised on a session-by-session basis. Adherence to the prescribed load and repetition range was verified, while time-under-tension (TUT) and total volume load were calculated for each set across all participants. This provided an additional layer of monitoring and confirmed that the intended contrast between groups was reliably maintained. These methodological details are now explicitly described in the revised Methods (*Page 15, Lines 354–361; Page 16, Lines 362–367, 370–373*), and TUT has been incorporated into the revised Table 1 alongside total volume load.

➤ **Bilateral vs unilateral training:**

We have clarified that for both groups, resistance training was performed unilaterally with the dominant right arm. This is now explicitly stated in the revised Methods section (*Page 15, Lines 341–343*).

➤ **Interpretation of StartReact and iMEP data:**

In response to concerns regarding StartReact and iMEP, we now explicitly acknowledge that these are indirect behavioural measures rather than direct assessments of reticulospinal drive, and we have refined our discussion to adopt a more cautious, physiologically grounded interpretation. Furthermore, we have replaced the term “reticulospinal drive” with “StartReact effect” to make the interpretation physiologically grounded. In addition, we reanalysed the reaction time data after verifying the acquisition settings with the manufacturer (ADInstruments). This reanalysis rectified the previously short values, and the updated reaction times now fall within expected physiological ranges with the main outcome variable, the StartReact effect, remaining unchanged. Accordingly, the respective Result section, Figure 6 and the related values in Table 2 (Rows 10 and 11) have been updated to reflect the corrected latencies and statistical analysis result. We greatly appreciate this comment, which enabled us to identify and resolve this issue and thereby strengthen the findings, accuracy and transparency of our work.

Importantly, while StartReact and iMEP data provide valuable insight into potential reticulospinal drive, they do not permit direct quantification of RST activity and must therefore be interpreted with caution. In the revised manuscript, we have tempered our interpretation and added a statement to the Limitations section (*Page 43, Lines 1011–1019*), which now reads: “... *it should be noted that RST excitability was assessed using indirect measures, including the StartReact effect and RFD. While these approaches provide valuable insight into reticulospinal drive, they do not permit direct quantification of RST activity. Similarly, iMEPs elicited with TMS index cortico-reticulospinal excitability without specifically isolating RST function. Our findings should therefore be interpreted with these limitations in mind.*”

Comment 4: Finally, the reviewers noted that the manuscript is at times repetitive, especially in the Introduction and Discussion. A more concise and focused presentation would enhance readability. Attention to figure and table formatting-such as label clarity and consistency of units-was also recommended.

Response:

We have carefully revised the manuscript to address both aspects of this comment, as outlined below:

➤ **Reducing repetition and improving focus:**

We thoroughly revised the Introduction and Discussion to remove redundant statements and streamline the presentation. As a result, the Introduction has been

reduced to approximately half its original length, and the Discussion has been shortened by nearly one-third.

➤ **Improving figure and table formatting**

All figures and tables have been revised for consistency and clarity. Labels have been standardised, abbreviations expanded at first mention, and units corrected for uniformity across all figures and tables. Figure legends were rewritten to avoid redundancy and ensure they are self-contained. Table formatting was also adjusted for improved readability

We believe these revisions substantially enhance the overall clarity and presentation of the manuscript.

Response to Senior Editor:

Comment 1: It is noted that individual data points have been shown on only one figure. This is not in accordance with the Statistical Policy of the Journal.

Response:

We thank the editor for highlighting this. All relevant figures have now been revised to include individual data points, ensuring full compliance with the Journal's Statistical Policy.

Comment 2: Since a longitudinal design was employed, it should also be possible to show for each individual the variation across time points. Such representations are to be preferred to the point plus error bar format used presently. Since the number of participants is very small, it is feasible to depict data for all participants on the same figure.

Response:

We appreciate this valuable suggestion. The revised figures now include individual data points and trajectories across time points, ensuring that within-subject variation is clearly represented. We explored multiple ways of combining individual trajectories with group means and error bars in a single panel; however, with three groups and multiple time points this format remained overly congested and difficult to interpret. To provide the clearest and most accessible presentation, we therefore adopted a side-by-side two-panel format: one panel depicts individual trajectories, while the adjacent panel shows group means with error bars (mean \pm SD).

Comment 3: The basis upon which the error bars were generated is not indicated in the current figure captions. Please refer to the Statistical Policy of the Journal in this regard. In a related vein, please note that supplementary figures are not encouraged. All relevant material

should be included with the manuscript.

Response:

All figure captions have been revised to explicitly state the basis of the error bars, which are presented as mean \pm SD in accordance with the Journal of Physiology Statistical Policy. In addition, the supplementary figures have been removed and the relevant material has been incorporated into the main manuscript to ensure that all relevant material is included.

Comment 4: While the logistical challenges associated with the conduct of studies of this nature can be appreciated, the number of participants allocated to each group is extremely small. It is therefore difficult to conclude that the effects estimated in the samples are representative of the "true" effects present in the population.

Response:

We thank the editor for acknowledging the logistical challenges of conducting studies of this nature. We recognise this important limitation. While the sample size was relatively modest, it was determined a priori through power analysis, fulfilled the necessary statistical power, and we recruited three participants beyond the minimum required. The use of robust analytic techniques further strengthens confidence in the findings. Nonetheless, we acknowledge that the modest cohort limits generalisability, and we have clarified this point in the Discussion, emphasising the need for replication in larger cohorts to confirm and extend these results.

Comment 5: The referees have raised several concerns that relate to the statistical models that were applied, highlighting in particular that the outcomes reported do not support in a straightforward manner the inferences that have been made. In this regard, I wish to place particular emphasis on an observation made by Referee #1 in relation the associations between measures of corticospinal/reticulospinal excitability and behavioural measures of performance. It is indeed the case that the presence of correlation does not imply causation. Given causation however, the presence of correlation is to be anticipated. The referees and Reviewing Editor have noted several matters that should be addressed. In addition to these, I wish to emphasise as a requirement that the authors demonstrate the presence of robust associations between select measures of corticospinal/reticulospinal excitability - defined on an a priori basis, and the key behavioural measure of performance. In the absence of such, it seems to me unlikely that a reader will be convinced that (particularly in a very small sample) the authors' preferred inferences are sustainable.

Response:

We have carefully revised our analysis and presentation of results to address all concerns raised:

- **Robust associations confirmed:** Additional analyses were conducted to demonstrate the presence of robust associations between the a priori–defined measures of corticospinal and reticulospinal excitability and the key behavioural performance outcomes (*Page 29 and 30, Lines 687-704*). These analyses confirmed the expected associations. e.g. *In the SP-RT group, StartReact effect changes were positively associated with 1RM ($\beta = 0.06$, $SE = 0.02$, $t = 2.89$, $p = 0.007$) and MVF ($\beta = 0.55$, $SE = 0.13$, $t = 4.16$, $p < 0.001$). ICAR demonstrated similar positive associations with 1RM ($\beta = 3.01$, $SE = 1.22$, $t = 2.47$, $p = 0.020$) and MVF ($\beta = 2.80$, $SE = 9.03$, $t = 3.11$, $p = 0.004$). No significant associations were observed between MVF and CSE ($p = 0.498$), CSP ($p = 0.117$), SICI ($p = 0.548$), or ICF ($p = 0.577$), nor between 1RM and CSE ($p = 0.581$), cSP ($p = 0.760$), SICI ($p = 0.819$), or ICF ($p = 0.241$).*
- **Interaction contrasts performed:** We also performed interaction contrast analyses for each outcome variable, which further clarified group \times time effects and provided stronger support for our interpretations.
- **Improved transparency and strengthened finding:** The statistical methods, reporting, and interpretation have been comprehensively revised. All results, figures, tables, and the Discussion were updated to reflect these new analyses, which have strengthened the evidence supporting our original findings.
- **All matters addressed:** In addition to these key revisions, we confirm that all matters noted by the referees and Reviewing Editor have been addressed in the revised manuscript.

We greatly appreciate the guidance provided by both the Editor and the reviewers, which has substantially improved the rigor, transparency, and clarity of the manuscript and ensured that our inferences are more robust and sustainable.

Responses to Referee #1:

General Comment: The present study examines the specific neurophysiological adaptations following two different resistance training (RT) protocols that differ in training tempo (metronome-paced vs self-paced). The novelty of the study is the inclusion of methods that allow to measure reticulospinal and cortico-reticulospinal adaptations following a short-term RT period (i.e. longitudinal design), something that, as the authors acknowledge, has never

done before. The results not only support a hypothesis initially proposed in a 2020 animal study showing that RT leads to reticulospinal adaptations, but also shows how external auditory pacing during RT alters the central nervous system demands, driving neural adaptations upstream the brainstem, at the cortical level. The rationale and the results are well described and discussed. The methods are robust and in accordance to the specific procedures required for the measured outcomes. I would like to commend the authors for their effort in conducting this much needed study. That said, I have some comments and suggestions that I believe will help improve clarity at some points of the manuscript.

Response: We sincerely thank the reviewer for their thoughtful and encouraging assessment of our work. We greatly appreciate the recognition of the novelty and methodological rigor of our study, as well as the acknowledgement of its contribution to understanding reticulospinal and cortico-reticulospinal adaptations to resistance training. Your constructive comments and suggestions are highly valued, and we have carefully addressed each point in our detailed responses below to further enhance the clarity and overall quality of the manuscript.

Comment 1: The introduction is well written and effectively describes the context of the main hypothesis of the manuscript. As a minor comment, at the point where authors discuss about the variability in corticospinal adaptations to RT (lines 128-129), I suggest referencing original studies that report no changes (or even reductions) in CSE following short-term self-paced resistance training to reinforce the argument (instead of just citing systematic reviews and meta-analysis). For example: For example: Ansdell, P., Brownstein, C. G., Škarabot, J., Angius, L., Kidgell, D., Frazer, A., Hicks, K. M., Durbaba, R., Howatson, G., Goodall, S., & Thomas, K. (2020). Task-specific strength increases after lower-limb compound resistance training occurred in the absence of corticospinal changes in vastus lateralis. *Exp Physiol*, n/a(n/a). <https://doi.org/10.1113/ep088629>, Colomer-Poveda, D., Romero-Arenas, S., Fariñas, J., Iglesias-Soler, E., Hortobágyi, T., & Márquez, G. (2021). Training load but not fatigue affects cross-education of maximal voluntary force. *Scandinavian Journal of Medicine & Science in Sports*, 31(2), 313-324. <https://doi.org/10.1111/sms.13844>, and Giboin, L. S., Weiss, B., Thomas, F., & Gruber, M. (2017). Neuroplasticity following short-term strength training occurs at supraspinal level and is specific for the trained task. *Acta Physiol (Oxf)*, 222(4), e12998. <https://doi.org/10.1111/apha.12998>.

Response: We thank the reviewer for this constructive suggestion. We have revised the Introduction to include original studies that specifically reported no change, increase or reductions in corticospinal excitability (CSE) following short-term self-paced resistance training. Accordingly, we have added the suggested references to strengthen our argument

and now reads “..., whereas findings for corticospinal excitability (CSE) remain inconsistent, with reports of increases, decreases, or no change (Carroll et al., 2002; Jensen et al., 2005; Beck et al., 2007; Griffin & Cafarelli, 2007; Hendy & Kidgell, 2013; Leung et al., 2017; Giboin et al., 2018; Ansdell et al., 2020; Colomer-Poveda et al., 2021) (**Page5, Lines 114-118**).

METHODS

Comment 2: Lines 234-236: For clarity, were the 1RM and MVF only measured once, at the familiarization session? Or they just practiced at this session and then formally assessed one week later.

Response:

We thank the reviewer for highlighting this point. During the familiarisation session, participants practiced the 1RM and MVF procedures, and measurements were also recorded at this session solely for the purpose of group randomisation, but these data were not included in the analyses. Formal 1RM and MVF assessments were then conducted at the baseline session one week later. The Methods section has been revised to clarify that the formal strength measurements (1RM and MVF) were taken at baseline, and the strength measure (1RM) performed during familiarisation was solely for randomisation (**Page 10, Lines 220–221 and 231–233**).

Comment 3: In Lines 234-236 you mention that "baseline assessments of cortical, corticospinal, corticoreticulospinal and reticulospinal excitability were conducted" but you do not mention muscular strength measures. While this appears to be clarified in Figure 1, the text as currently written may be confusing to some readers.

Response: This is an important point. As highlighted in our response to the previous comment, the Methods have now been revised to state clearly that baseline assessments included both muscular strength measures (1RM and MVF) and neurophysiological measures. This clarification has been added to **Page 10 Lines 231–233** in the revised manuscript.

Comment 4: Line 301-303: Did you measure RFD during self-initiated MVC? If so, did Self-paced RT induce greater increases in that variable? It is relevant to consider any possible functional differences between training approaches and whether specific neural adaptations may underlie different functional adaptations (which does not seem to be the case if you only analyse maximal strength).

Response:

We thank the reviewer for this insightful comment regarding RFD assessment and its functional implications. In our study, RFD was not measured during self-initiated MVC, rather it was externally cued. RFD was assessed exclusively within the StartReact protocol, where participants performed rapid isometric elbow flexion, as fast and as hard as possible, in response to an imperative visual signal (LED) accompanied by a loud startling sound, a soft sound, or no sound, following established procedures for RFD measurement in the StartReact protocol (Anzak et al., 2011; Del Vecchio, Negro, et al., 2019; Škarabot et al., 2022; Colomer-Poveda et al., 2023). This approach was selected to probe reticulospinal tract (RST) involvement, consistent with our primary objective of examining neurophysiological adaptations to resistance training, with a particular focus on the potential role of the RST. Accordingly, RFD was used as a physiological proxy of RST excitability rather than as a general functional strength outcome. By design, our MVC trials were conducted to assess maximal voluntary strength rather than explosive force, RFD; the instructions and timing (gradual rise to maximum and sustained plateau) do not permit a valid, unbiased estimate of RFD.

As reported, RFD increased significantly during the loud-sound condition, compared to other sound conditions, following self-paced RT, supporting enhanced RST excitability. However, because RFD was not assessed during self-initiated explosive force, it remains uncertain whether these neural adaptations extend to voluntary rapid force production. This limitation has been clarified in the revised manuscript (**Page 42, Lines 993–997**), which now states: *“RFD was assessed only during visually cued explosive contractions within the StartReact protocol to probe reticulospinal excitability, rather than during self-initiated efforts. Consequently, it remains unclear whether the observed neural adaptations extend to voluntary rapid force production. Future investigations are warranted to determine whether modality-specific neural adaptations are accompanied by functional gains in RFD.”*

Comment 5: Line 303: The type of force transducer is already mentioned earlier in the methods. To avoid redundancy, consider removing it here.

Response: We have removed the repeated description of the force transducer in this section to avoid redundancy.

Comment 6: Line 309: "for Mmax AND subsequent TMS ..."

Response: Modified accordingly.

Comment 7: Line 336: Related to the previous comment regarding RFD adaptations, did the SP-RT group receive any instruction regarding lifting velocity intention? (i.e. whether to lift with maximal intent or not?). Repetition duration during submaximal lifts, determined by

lifting intention, could influence strength adaptations (e.g., Hermes and Fry, 2023). If lifting velocity was not maximal (difficult to ensure if you do not measure lifting velocity and/or there is continuous encouragement with specific cues to lift at maximal velocity), do you think that maximum intention lifting would have produced greater strength increases in the SP-RT group? Hermes MJ, Fry AC. Intentionally Slow Concentric Velocity Resistance Exercise and Strength Adaptations: A Meta-Analysis. *J Strength Cond Res.* 2023 Aug 1;37(8):e470-e484. doi: 10.1519/JSC.0000000000004490. PMID: 37494124.

Response:

We thank the reviewer for this thoughtful suggestion. Participants in the SP-RT group were not given explicit instructions to lift with maximal velocity intention; the defining feature of this modality was that contractions were performed at each participant's self-selected, self-motivated speed. They were monitored to ensure correct execution but were instructed to perform the exercise in the same way as during standard 1RM testing, without cues to accelerate maximally. Time under tension (TUT) was systematically recorded, and these data confirmed that SP-RT was performed at a faster tempo than MP-RT, ensuring that the intended contrast between groups was maintained.

We agree that maximal velocity intention, as highlighted by Hermes and Fry (2023), can influence strength adaptations, and it is possible that incorporating maximal intent into SP-RT could have produced even greater strength gains. While this was beyond the scope of the present study, we consider this an interesting direction for future research. We have also acknowledged this in the revised manuscript (*Page 42 and 43, Lines 1003–1005*), which now states: “*We did not manipulate contraction velocity in SP-RT, and whether maximal intent amplifies strength and neurophysiological adaptations remains to be determined.*”

Comment 8 Line 362-363: Why was a metronome used during TMS measurements? The auditory may have induced anticipation or expectation of the pulse, potentially influencing the results (see a reference below). Moreover, taking into account that one group was chronically exposed to this type of auditory cues and its capacity for anticipation could have been influenced by training. Capozio A, Chakrabarty S, Astill S. The Effect of Sound and Stimulus Expectation on Transcranial Magnetic Stimulation-Elicited Motor Evoked Potentials. *Brain Topogr.* 2021 Nov;34(6):720-730. doi: 10.1007/s10548-021-00867-9. Epub 2021 Sep 6. PMID: 34490506; PMCID: PMC8556164.

Response:

The metronome was not used to indicate when the TMS pulse would occur. Rather, its soft auditory signal served only as the go cue for participants to initiate a 10% MVC contraction.

Once the target level was reached and stabilised, the TMS pulse was delivered. meaning participants were already engaged in the contraction before stimulation. This reduces the likelihood that the auditory cue influenced corticospinal excitability. To avoid confusion, the Methods section has been revised to state: “*All TMS pulses were delivered at fixed 10-s intervals, with isometric contractions initiated by a soft metronome cue; upon hearing the sound, participants increased to the target 10% MVC level, and stimulation was applied once a stable level was achieved.*” (**Page 17, Lines 398–400**).

Regarding the possibility that chronic exposure to auditory pacing in the MP-RT group influenced anticipation, the training rhythm (3-s concentric / 4-s eccentric cycles) differed substantially from the TMS protocol pacing (a single cue every 10 s to initiate contraction). This makes it unlikely that training exposure induced an anticipatory bias during TMS measurements.

Comment 9: Line 398-399: I understand that the PMRF has bilateral synaptic connections with spinal motoneurons, however I wonder why did you decided to measure the iMEPs in the left (untrained) BB rather than in the right trained BB? Would not the latter be more appropriate for identifying strength training induced reticulospinal adaptations projecting to the trained side?

Response:

Our decision to record iMEPs from the left (untrained) biceps brachii was based on the aim to assess reticulospinal contributions with minimal contamination from contralateral corticospinal activity. TMS was delivered over the left (trained) hemisphere to probe training-related neuroplastic changes. This stimulation activates both corticospinal and cortico-reticular projections, the latter influencing bilateral reticulospinal pathways and thereby enabling iMEPs to be measured in the ipsilateral (untrained) limb. Although the left limb was not trained, this approach allowed us to capture potential adaptations in cortico-reticulospinal output from the trained hemisphere. Importantly, if we had attempted to obtain iMEPs from the trained limb, this would have required stimulating the untrained hemisphere, thereby probing cortico-reticulospinal pathways that were not exposed to training. Thus, measuring iMEPs in the untrained limb provided a clearer index of cortico-reticulospinal adaptations from the trained hemisphere.

Overall, this approach is consistent with established neurophysiological principles and previous work examining iMEPs (Altermatt et al., 2023; Alagona et al., 2001; Maitland & Baker, 2021; Hu et al., 2024).

Comment 10: Related to this, did you measure the cross-education effect of training and if it is more pronounced on the SP-RT due to the bilateral nature of the RST?

Response:

We appreciate the reviewer's follow-up question. As noted in our response to Comment 9, our primary aim in measuring iMEPs from the untrained limb was to assess reticulospinal tract (RST) excitability and training-related neural adaptations originating from the trained hemisphere, rather than to evaluate behavioural transfer effects. We therefore did not measure cross-education (i.e., strength transfer to the untrained limb) in this study.

While the bilateral nature of the RST could contribute to cross-education effects, and such effects might be more pronounced in the SP-RT group, quantifying this was outside the scope of our experimental design. Our focus was on identifying the main site of neural adaptation, with particular emphasis on the reticulospinal tract, and on examining the differential modulation of descending tracts following different modalities of resistance training.

Comment 11: Line 447-448: Just for clarity, does this refer to additional ICC data apart from the values already presented in the supplementary material, which appear to be based solely on the control group? If you already reported the reliability of these measures in the control group is this additional pilot data necessary?

Response:

We appreciate the opportunity to clarify the role of the pilot data in our study.

The pilot data referred to data collected prior to the main study to establish the feasibility and reliability of our measurement protocols before commencing full data collection. These pilot tests confirmed that our methods produced stable and reproducible results, providing confidence to proceed with the main experiment.

In the main study, we later performed additional reliability testing using the control group. It was not intended to replace the pilot work, but rather to strengthen our reliability evidence by confirming that the measures remained stable under the actual experimental conditions of the study. While the ICC values presented in the supplementary material are based solely on the control group, the earlier pilot data served the distinct purpose of confirming the feasibility and reliability of the methods before full-scale implementation.

Comment 12: Line 456-458: Was the 10% of MVF contraction equivalent to 5% of maximal rmsEMG? Or it should be 10% also here?

Response:

Thank you for noting this. Participants maintained a 10% MVF contraction, and trials were accepted only when pre-stimulus rmsEMG was within 8-12% of the participant's maximal rmsEMG of each participant's maximal rmsEMG. The "5 ± 1%" value was a typographical error. We have corrected the Methods to reflect 10 ± 2% (**Page 22, Line 506**).

Comment 13: Line 588: Following the reference that you cite in line 588, could you please specify the type of ICC analysis (model, type and definition)?

Response:

We appreciate the opportunity to clarify the ICC specification. We have specified the type of ICC analysis (model, type and definition). Now it reads "*Intraclass correlation coefficients (ICCs) were calculated to assess test-retest reproducibility using a two-way mixed-effects model, single measurement, and absolute agreement definition. ICC values were interpreted as poor (<0.50), moderate (0.50–0.75), good (0.75–0.90), or excellent (>0.90) reliability, (Koo & Li (2016) (Page 27, Lines 629-632)*

Comment 14: Table 1: Are differences in training load just derived from the slightly greater 1RM in the SP-RT group? (volume (sets and reps) were equated) Or the externally imposed (and potentially slower) tempo lead to failure and reduced total number of repetitions in the MP-RT group?

Response:

We appreciate the reviewer's question. The training protocol was identical for both groups, following American College of Sports Medicine (ACSM) guidelines of 4 sets of 6–8 repetitions at 70–75% of 1RM, with maximum verbal encouragement provided throughout to ensure participants reached their maximal possible repetitions. Baseline 1RM did not differ significantly between groups, and therefore cannot fully explain the observed differences in volume load (reps x sets x load [kg lifted]). Instead, the lower total volume load observed in the MP-RT group was primarily attributable to the slower, metronome-paced cadence, which required prolonged effort during both concentric and eccentric phases. This externally imposed tempo increased time under tension (TUT) per repetition and, in some participants, led to earlier fatigue within a set, so that they completed 6 or 7 rather than the full 8 repetitions, resulting in a lower realised number of repetitions and, consequently, a lower total volume load. Thus, the lower realised repetitions and training load in MP-RT primarily reflected tempo-related fatigue rather than motivational factors or baseline strength differences. In contrast, the faster self-selected cadence in the SP-RT group reduced TUT per repetition and enabled participants to complete more repetitions within the same set structure,

contributing to a greater overall training load. To further clarify the impact of cadence differences, we have updated Table 1 to include TUT alongside volume load (Table 1).

RESULTS

Some minor suggestions to consider in order to reduce the length of the results section:

Comment 15: Lines 729-731: Consider putting the % changes in lines 730-731 between brackets and delete the 732-733 lines.

Response: Modified accordingly (*Page 34, Lines 802–805*).

Comment 16: Lines 733-737: Do you consider it necessary to report non-significant percentage changes for the MP-RT and control groups in the main text? Since absolute values are already presented in the table, removing these details could help reduce the overall length of the results section without compromising clarity.

Response:

We agree that reporting non-significant percentage changes for the MP-RT and control groups in the main text is not essential given that the absolute values are already presented in the table. These details have therefore been removed to improve conciseness without loss of clarity.

Comment 17: Lines 739-740: As I suggest in the previous comments, in order to reduce the length of the results sections, you may consider putting the % changes of the SP-RT group together with effect sizes and p values and not include the non-significant % changes of MP-RT and control group (you already show absolute values in the table). That is something you can consider for the results of all the other outcomes.

Response:

In line with this suggestion, we have now presented the percentage change, effect sizes, and p values for the SP-RT group together, while omitting non-significant percentage changes for the MP-RT and control groups in the main text. This approach has been applied consistently across all other outcomes, ensuring that the Results section is now more concise and focused.

General comment:

Comment 18: Although I acknowledge that correlation is not causation, I wonder if you have explored any associations between key neurophysiological indices of cortical or reticulospinal adaptations and strength outcomes.

Response:

We have now examined associations between key neurophysiological indices of cortical (SICI, ICF,cSP), corticospinal (CSE), cortico-reticular (ICAR) and reticulospinal (StartReact

effect) adaptations with strength outcomes (1RM and/or MVC). These new analyses have been added to the Results section (**Page 29, Lines 688-693, and page 30 694-706**), which reads “*Association of the neurophysiological adaptation indices and strength outcomes was explored. Across groups, distinct patterns of association emerged between neurophysiological measures and strength outcomes. In the SP-RT group, StartReact effect changes were positively associated with 1RM ($\beta = 0.06$, $SE = 0.02$, $t = 2.89$, $p = 0.007$) and MVF ($\beta = 0.55$, $SE = 0.13$, $t = 4.16$, $p < 0.001$). ICAR demonstrated similar positive associations with 1RM ($\beta = 3.01$, $SE = 1.22$, $t = 2.47$, $p = 0.020$) and MVF ($\beta = 2.80$, $SE = 9.03$, $t = 3.11$, $p = 0.004$). No significant associations were observed between MVF and CSE ($p = 0.498$), CSP ($p = 0.117$), SICI ($p = 0.548$), or ICF ($p = 0.577$), nor between 1RM and CSE ($p = 0.581$), cSP ($p = 0.760$), SICI ($p = 0.819$), or ICF ($p = 0.241$).*

In the MP-RT group, CSE changes was positively associated with changes in 1RM ($\beta = 0.01$, $SE = 0.01$, $t = 2.78$, $p = 0.009$) and MVF ($\beta = 0.01$, $SE = 0.01$, $t = 3.69$, $p < 0.001$). SICI also showed negative associations with 1RM ($\beta = -0.03$, $SE = 0.01$, $t = -2.41$, $p = 0.022$) and MVF ($\beta = 0.20$, $SE = 0.07$, $t = 2.90$, $p = 0.007$). MVF was additionally associated to ICF ($\beta = 0.08$, $SE = 0.03$, $t = 2.40$, $p = 0.023$), whereas cSP exhibited a negative association ($\beta = -0.10$, $SE = 0.04$, $t = -2.20$, $p = 0.036$). StartReact effect and ICAR were not significantly associated with either MVF ($p = 0.978$ and $p = 0.847$, respectively) or 1RM ($p = 0.675$ and $p = 0.577$, respectively). In the control group, no neurophysiological measure was associated with 1RM or MVF.”

DISCUSSION

Comment 19: Line 797: I do not think that "self-paced resistance training" is a "novel" modality

Response:

We appreciate the reviewer’s observation. We agree that self-paced resistance training itself is not a novel exercise modality in general terms. In our manuscript, we used “novel” in reference to its application within the specific context of neurophysiological research investigating cortico-reticulospinal adaptations. To avoid potential misunderstanding, we have removed the word novel.

Comment 20: Line 821: As I commented before, I wonder if you measured RFD adaptations and if there were differences between groups. Although maximal strength is relevant, both force rapidly. It would be interesting and valuable to know whether the differences in tempo

led to different adaptations in rapid force production together with the distinct neural adaptations.

Response:

We thank the reviewer for raising this important point. As noted in our response to Comment 4, in our work, RFD was quantified solely within the framework of the StartReact protocol. This was a deliberate methodological choice, as our primary objective was to probe cortico-reticular and reticulospinal contributions to neural adaptation, rather than to provide a general assessment of explosive voluntary strength. Measuring RFD in the StartReact protocol allowed us to capture changes in rapid force production that are more specifically attributable to subcortical pathways, particularly the RST, which was the central focus of our investigation. While we did not calculate RFD outside this context, it is important to note that training tempo did influence RFD within the StartReact protocol: the SP-RT group demonstrated significantly greater RFD in response to visual startling stimuli compared with both the MP-RT and control groups.

As mentioned in response to Comment 4, we have also acknowledged as a limitation that RFD was not assessed during self-initiated MVCs, and therefore we cannot determine whether these neural adaptations extend to voluntary rapid force production (*Page 42, Lines 992–996*).

Comment 21: Lines 909-911: To which spinal motoneurons are you referring here, if not to alpha motoneurons? reticulospinal neurons. Maybe you can change it for clarity.

Response:

In this context, we were indeed referring to alpha motoneurons. We have revised the sentence for clarity to explicitly state “alpha motoneurons” rather than simply “spinal motoneurons” to avoid ambiguity. Now it reads “*By bypassing cortical pathways, this mechanism more efficiently recruits alpha-motoneurons, contributing to the observed improvements in motor output*” *Page 40, lines 945-947*.

Comment 22: Line 929-939: Check this double citation.

Response: We appreciate the reviewer’s careful attention to detail. We have corrected the duplicated citation for Stasinaki et al. (2019) in this section, ensuring that each reference is cited only once in the most relevant position. No other duplicate citations were found in the surrounding text.

Comment 23: Line 931-935: Please, check the construction of this sentence and improve it for clarity (break it in two sentences?)

Response: The sentence has been revised and split into two for improved clarity.

References cited in Author Response to Referee 1

- Alagona, G., Delvaux, V., Gérard, P., De Pasqua, V., Pennisi, G., Delwaide, P.J., Nicoletti, F. & Maertens de Noordhout, A. (2001). Ipsilateral motor responses to focal transcranial magnetic stimulation in healthy subjects and acute-stroke patients. *Stroke*, 32, 1304–1309.
- Altermatt, M., Jordan, H., Ho, K. & Byblow, W.D. (2023). Modulation of ipsilateral motor evoked potentials during bimanual coordination tasks. *Frontiers in Human Neuroscience*, 17, 1219112.
- Anzak, A., Tan, H., Pogosyan, A. & Brown, P. (2011). Doing better than your best: loud auditory stimulation yields improvements in maximal voluntary force. *Experimental Brain Research*, 208, 237–243.
- Atkinson, E., Škarabot, J., Ansdell, P., Goodall, S., Howatson, G. & Thomas, K. (2022). Does the reticulospinal tract mediate adaptation to resistance training in humans? *Journal of Applied Physiology* (1985), 133, 689–696.
- Bradnam, L.V., Stinear, C.M. & Byblow, W.D. (2013). Ipsilateral motor pathways after stroke: implications for non-invasive brain stimulation. *Frontiers in Human Neuroscience*, 7, 184. doi: 10.3389/fnhum.2013.00184.
- Colomer-Poveda, D., López-Rivera, E., Hortobágyi, T., Márquez, G. & Fernández-Del-Olmo, M. (2023). Differences in the effects of a startle stimulus on rate of force development between resistance-trained rock climbers and untrained individuals: Evidence for reticulospinal adaptations? *Scandinavian Journal of Medicine & Science in Sports*, 33, 1360–1372.
- Del Vecchio, A., Negro, F., Holobar, A., Casolo, A., Folland, J.P., Felici, F. & Farina, D. (2019). You are as fast as your motor neurons: speed of recruitment and maximal discharge of motor neurons determine the maximal rate of force development in humans. *Journal of Physiology*, 597, 2445–2456.
- Fisher, K.M., Zaaimi, B. & Baker, S.N. (2012). Reticular formation responses to magnetic brain stimulation of primary motor cortex. *Journal of Physiology*, 590(16), 4045–4060. doi: 10.1113/jphysiol.2011.226209.
- Hu, N., Tanel, M., Baker, S.N., Kidgell, D.J. & Walker, S. (2024). Inducing ipsilateral motor-evoked potentials in the biceps brachii muscle in healthy humans. *European Journal of Neuroscience*.
- Maitland, S. & Baker, S.N. (2021). Ipsilateral motor evoked potentials as a measure of the reticulospinal tract in age-related strength changes. *Frontiers in Aging Neuroscience*, 13, 612352.
- Škarabot, J., Folland, J.P., Holobar, A., Baker, S.N. & Del Vecchio, A. (2022). Startling stimuli increase maximal motor unit discharge rate and rate of force development in humans. *Journal of Neurophysiology*, 128, 455–469.
- Ziemann, U., Ishii, K., Borgheresi, A., Yaseen, Z., Battaglia, F., Hallett, M., Cincotta, M. & Wassermann, E.M. (1999). Dissociation of the pathways mediating ipsilateral and contralateral motor-evoked potentials in human hand and arm muscles. *Journal of Physiology*, 518(3), 895–906.

Responses to Referee 2

Overall comments

Comment 1: in my opinion, this is not a well written or conceptualised report, and there are substantial issues with the statistical approach.

Response: In light of the reviewer’s comments, we have carefully revisited the entire manuscript and made substantial revisions to improve both conceptual clarity and statistical rigour. The Introduction, Methods, Results, and Discussion sections have all been revised for clarity, precision, and coherence, and the figures and tables have been updated to reflect the findings of new analyses.

To address concerns about the statistical approach, we expanded the description of methods to ensure transparency and reproducibility, restructured the Results section for a clearer presentation, and performed additional analyses, including interaction contrast analyses, which strengthened and supported the conclusions reported in the original submission.

We believe these comprehensive revisions have significantly improved the quality, clarity, and rigour of the manuscript, and that the changes directly address the reviewer’s concerns. The specific revisions are detailed in the responses below and are reflected in the revised manuscript.

Comment 2: The Introduction lacks clarity on a number of points and includes some weak use of the literature. The experimental variable in this study is the type of RT, specifically

self-paced vs metronome paced, yet this is a small feature of the introduction. Consequently, the Introduction lacks any clear theoretical underpinning rationale for the comparison of these two types of RT (why physiologically would we expect them to be different?).

Response:

We appreciate the reviewer's perspective regarding the clarity and conceptual focus of the Introduction. The primary aim of our study was to identify the role of the reticulospinal tract (RST) in early strength gain in humans. The Introduction was structured to first highlight the key research gap, namely, that the precise neural sites of adaptation during the first weeks of resistance training remain unclear with inconsistent findings suggesting other possible sites such as the RST. While RST is showed as a possible site of adaptation in non-human primates, it has not been adequately studied in humans due to methodological challenges. This gap provides a compelling rationale for investigating the RST's role in human strength adaptation.

The experimental variable in our study, the type of resistance training (self-paced [SP-RT] vs metronome-paced [MP-RT]), was introduced in the context of addressing a second gap: while MP-RT has been shown to modulate cortical and corticospinal excitability, similar changes were not observed following SP-RT besides strength gain, and hence we hypothesised the RST adaptation might underpinned the strength gained following SP-RT, as shown by the recent findings on non-human primates. Our secondary aim, therefore, was to determine whether these two training modalities differentially influence cortical, corticospinal, corticoreticular, and RST excitability.

We agree that in the original version, the theoretical underpinning for expecting physiological differences between SP-RT vs MP-RT could be made clearer.

In the revised Introduction, we have:

- Reviewed and updated the literature base: inappropriate or weak references have been removed, and relevant, high-quality studies have been added to strengthen the background.
- Organised the narrative more clearly: the revised Introduction first identifies the gap related to our primary objective (uncertain neural sites of early RT adaptations, with emphasis on RST), and then links to the secondary objective (SP-RT vs MP-RT comparison) with appropriate justification.
- Strengthened the physiological rationale: specifically, that the externally imposed, slower tempo of MP-RT may increase reliance on cortical and corticospinal pathways, whereas the self-selected, faster tempo of SP-RT may promote greater recruitment of

subcortical pathways, including the RST, which is involved in rapid and gross motor output.

These revisions ensure that the Introduction clearly frames both the primary objective (identifying the role of the RST in early strength gain) and the secondary objective (determining modality-specific neural adaptations to SP-RT vs MP-RT), while addressing the reviewer's concerns about clarity, literature use, and theoretical rationale. The revised Introduction can be found on *Pages 5–7, Lines 101–170*

Comment 3: Important description of the training and testing tasks is basic and lacks extensive information. Some of the methods appear weak (e.g. voluntary activation cannot be assessed by stimulating agonists and antagonists simultaneously) or are hardly described (e.g. RFD of which contractions).

Response:

We thank the reviewer for this constructive comment. In the revised Methods section, we have substantially expanded the descriptions of both the training and testing protocols. For training, we now specify the, position, stabilisation, exercise type, load prescription, set/repetition scheme, rest intervals, and tempo regulation for both metronome-paced and self-paced resistance training. For RFD assessment, we clarify that it was conducted as part of the StartReact protocol, involving an isometric elbow flexion of the dominant (right) arm in response to a visual stimulus (LED light) presented either alone or concurrently with a loud, or soft auditory sounds. Participants were instructed to contract “as fast and as hard as possible” following the established procedures described in Anzak et al. (2011), Del Vecchio et al. (2019b), Škarabot et al. (2022), and Colomer-Poveda et al. (2023). This expanded description is now clearly stated in the Methods *Page 20, line 465-472*.

With regard to voluntary activation (VA), we acknowledge the reviewer's concern that stimulation at the brachial plexus might inadvertently activate both agonist and antagonist muscles. As VA was neither a primary nor a secondary objective and contributes minimally to the aims of this study, we have removed this outcome from the manuscript. This decision not only addresses the methodological concern raised but also streamlines the results, helping to reduce the overall length of the manuscript and maintain focus on the main objectives of the study.

Comment 4: The statistical analysis of the group x time interactions is insufficient to compare the changes of specific groups over time (e.g. group A vs group B). The extensive within group changes presented do not inform these between group interaction effects that are critical to this experiment. Excessive consideration/presentation of percentage changes, rather

than the more robust pre-post data. Without appropriate statistics the experiment is of limited value.

Response:

We sincerely thank the reviewer for this valuable observation and agree that between group interaction effects warrant greater emphasis. In response, we have reanalysed all the data using post-hoc interaction contrasts, and the Results section has been fully revised to focus on robust pre-post data and between group interaction effects rather than percentage changes. Percentage values are now retained only in brackets for context.

For clarity, all analyses in the original submission were performed using repeated-measures Linear Mixed Models (rmLMM) or Generalized Linear Mixed Models (GLMM), with results presented in ANOVA-style output to facilitate interpretation. In the original manuscript, *post-hoc* interaction contrasts were only performed when the overall group effect reached statistical significance. However, as highlighted in statistical guidance, overall test evaluate a global null hypothesis and can mask meaningful differences in pairwise group-time comparisons, particularly in designs with three groups (Maxwell & Delaney, 2004; Ruxton & Beauchamp, 2008). This limitation may have contributed to the underreporting of relevant between-group effects in the original version.

In line with the reviewer's recommendation, we have now conducted and reported significant post-hoc interaction contrasts for all outcome variables. This approach revealed several significant between-group effects that were not apparent in the overall test outputs, and these findings are now clearly presented in the revised Results section.

Overall, the Results have been thoroughly revised and restructured to highlight the between group interaction effects, with precise reporting of test statistics and p-values. We have also specified the analyses conducted and indicated the corresponding results with page and line references in our detailed responses to the individual reviewer comments below.

Comment 5: The paper is also excessively long, despite lacking important information.

Response:

We acknowledge that the manuscript may have initially appeared lengthy because it covered several outcome variables across multiple neural pathways (cortical, corticospinal, cortico-reticular, and reticulospinal) including strength measures, which are rarely examined together in a single study. This breadth was necessary to comprehensively address our two novel objectives: (1) to determine the effects of resistance training on cortico-reticular and reticulospinal excitability in humans for the first time, and (2) to establish whether different training modalities differentially modulate distinct descending tracts. However, we agree that

the structure and flow in the earlier version may have made it difficult for readers to capture the most important findings.

To address this, we have substantially streamlined the manuscript by removing redundancies in the Introduction (shortened from 4 double spaced pages down to 3 pages even after adding suggested information) and Discussion (shortened from 8 double spaced pages to 5 pages), consolidating overlapping explanations, and focusing the narrative around the two key objectives. At the same time, we have added concise but essential details that were underrepresented in the original submission, including full reporting of between-group interaction effects (via post-hoc interaction contrasts), clarification of test statistics, training and testing tasks and improved presentation of percentage changes. These revisions make the manuscript more concise while ensuring that the most important methodological and statistical details are now fully accessible.

We believe the revised version is clearer, more focused, and presents the novelty of our findings in a way that is both scientifically rigorous and reader-friendly.

Specific comments:

Comment 1: The premise within the abstract provides no connection/rationale between the two interventions and the adaptations being studied. Therefore, it is unclear how they relate.

Response:

We have revised the abstract to clearly foreground the aims (mentioned in the above response) and to directly connect the interventions with the neural adaptations being studied. The revised abstract now highlights training modality particularly pacing as a physiologically relevant factor that may shape neural adaptations. It frames this premise at the start of the abstract by noting that externally paced and self-paced contractions impose different demands on cortical and subcortical motor circuits. The results section of the abstract then specifies that MP-RT produced cortical and corticospinal adaptations, whereas SP-RT induced cortico-reticular and reticulospinal changes. These revisions ensure that the premise, objectives, and outcomes are now clearly connected and logically presented in the abstract (**Page 3, Lines 54–61 and Lines 70–71**), which reads now as follows: *“The neural mechanisms underlying resistance training (RT) adaptations remain incompletely understood, particularly the contribution of cortico-reticular and reticulospinal tract (RST), which have rarely been examined in humans. One factor that may critically shape these adaptations is training pacing, an overlooked yet physiologically relevant variable, as externally paced and self-paced contractions impose distinct demands on cortical and subcortical motor circuits. This*

study investigated (1) the effects of RT on cortico-reticular and RST excitability, and (2) whether modality-specific trainings differentially modulate descending tracts....”.

Comment 2: Why only a 3-week RT programme? Are the neural adaptations likely to be saturated in such a short period of time?

Response:

We appreciate the reviewer’s insightful question. Our rationale for employing a 3-week RT programme was guided by both theoretical and practical considerations. Below, we address the two concerns separately.

1. Why a 3-week programme?

Our primary aim was to isolate early-phase neural adaptations to resistance training while minimising confounding structural muscle changes. It is well established that initial strength gains during the first weeks of RT are predominantly neural in origin, before substantial hypertrophy occurs (Moritani & deVries, 1979; Folland, J. P., & Williams, A. G. (2007) Carroll et al., 2001, Škarabot J et al., 2021). Muscular factors or hypertrophy have been shown to be the dominant factor after the first 3 to 5 weeks of training (Moritani & deVries, 1979).

Extending the intervention beyond three weeks increases the likelihood of detectable structural changes in the muscle. For example, as you noted before Seynnes et al. (2007) demonstrated detectable increases in quadriceps muscle cross-sectional area after only 20 days of training. Such muscular adaptations could obscure the neural contributions we sought to characterise. Thus, limiting the programme to 3 weeks allowed us to focus on neural mechanisms relatively free of confounding structural factors.

In addition, participant burden was a critical practical consideration. Our study deliberately included demanding neurophysiological assessments, such as peripheral nerve stimulation for Mmax, high-intensity TMS stimulations for iMEP acquisition, as well as the StartReact program with startling acoustic stimuli, because these measures are essential for capturing neural adaptations with high fidelity. Although all procedures are safe and widely used, the testing sessions were physically and cognitively demanding for participants. Extending the programme substantially beyond three weeks would have increased the likelihood of non-adherence, or participant withdrawal. By restricting the intervention period, we maximised adherence and ensured high-quality, reliable data across all measurement sessions, while still retaining a time window that isolates predominantly neural rather than muscular adaptations.

2. Are neural adaptations likely to be saturated in such a short period?

Importantly, neural adaptations are not saturated within this period; rather, they are in their most dynamic phase. Previous work has shown continued progression of corticospinal and intracortical changes beyond 3 weeks, but the earliest weeks provide the clearest opportunity to isolate neural contributions without the confounding influence of muscle morphological adaptations. Moreover, the intensive nature of our neurophysiological assessments (maximal TMS stimulation for iMEP, Mmax acquisition, the startling sound in StartReact protocol) required careful consideration of participant burden, and a shorter programme ensured adherence while preserving data integrity. Thus, the 3-week design provided an optimal balance between scientific validity and feasibility.

Comment 3: Given the variability of the measures, especially for assessing change, I'm unsure if the group sizes are sufficient.

Response:

We appreciate the reviewer's concern regarding sample size adequacy in the context of measurement variability. Our sample size was determined *a priori* using G*Power (Faul et al., 2007), which indicated that 9 participants per group would provide $\geq 80\%$ power ($\alpha = 0.05$) to detect anticipated medium-to-large effects. To improve reliability and mitigate potential attrition, we enrolled 10 participants per group, slightly exceeding the calculated minimum.

We acknowledge that neurophysiological measures can show considerable individual variability; however, our analytic approach incorporated repeated-measures Linear Mixed Models (LMMs) and, where appropriate, Generalized Linear Mixed Models (GLMMs). These methods enhance precision and statistical power by leveraging within-subject variance and are considered more robust than traditional ANOVA when group sizes are modest (Gueorguieva & Krystal, 2004, Wilkinson et al., 2023).

Importantly, our group sizes are consistent with many prior neurophysiological RT studies published in high-impact journals (e.g., Jensen et al., 2005, Carroll et al., 2009, Hortobagyi et al., 2009, Coombs et al., 2016, Mason et al., 2017), Siddique et al., 2020, Del Balso C., et al., 2007, Kidgell et al., 2010, Kamen & Knight 2004, Patten et al., 2001, Beck et al. et al, 2007). In summary, the sample size was both statistically justified and consistent with prior literature. Moreover, the significant effects observed across multiple outcome measures confirm that the study was sufficiently powered to detect neural adaptations.

At the same time, we have acknowledged it in the revised manuscript (**Page 42. Lines 988-992**) Which reads; "...the relatively small sample size, a limitation inherent to intensive neurophysiological studies, should be noted. Although it was determined *a priori*, consistent

with prior literature, and supported by robust analytic approaches, the modest cohort may limit the generalisability; therefore, replication in larger cohorts is warranted to confirm and extend these findings.”

Comment 4: L107-8 This is quite weak and an inappropriate reflection of these papers. The Seynes et al 2007 paper found increased muscle CSA after only 20 days of training, and reported some significant changes in architecture after only 10 days of training. The Pearcey review has no information on the contractile apparatus, so has no relevance. Maeo et al (2017 PMID: 29570534) also found changes in CSA after 4 weeks of RT.

Response: We appreciate the reviewer’s observation, which has prompted us to ensure our statement accurately reflect the cited literature. The sentence in the revised manuscript (**Page 5, lines 103-106**) now reads:

“In the initial weeks of RT (<4 weeks), strength improvements are predominantly driven by neural adaptations, although modest muscle architectural changes may also occur (Akima et al., 1999; Blazevich et al., 2007; Seynnes et al., 2007).”

The Pearcey review has been removed from the references.

Comment 5: L108-109 The twitch response data is also quite shaky given the maximum contractions are tetanic in nature.

Response:

We thank the reviewer for this observation. We agree that the sentence regarding twitch responses during tetanic contractions was not essential to the framing of the study and may have introduced unnecessary ambiguity. To improve clarity and reduce length, we have removed this sentence from the Introduction.

Comment 6: L112-115 The list of mechanisms is quite selective - it is surprising that the possibility and evidence of spinal levels adaptations is not recognised.

Response:

This is an excellent point, and we have revised the text to acknowledge spinal-level adaptations as an additional mechanism with appropriate citation (**Page 5, lines 109-111**).

Comment 7: L117 'moderate and variable changes' is incomplete and unclear. Variable in what way?

Response:

To improve precision, we have replaced “moderate and variable changes” with a more accurate description: “... yet findings on corticospinal excitability (CSE) are inconsistent, with reports of increases, decreases, or no change....”

Comment 8: L119 The comment about and reference of the Adkin et al paper is somewhat misleading without explaining that they were contrasting endurance and resistance training.

Response: To improve accuracy, we have revised the text to avoid any misleading interpretation of Adkins et al. (2006), who compared endurance and resistance training. The passage now reads (*Page 5, Lines 112-119*):

“Studies have shown reductions in short-interval intracortical inhibition (SICI) and cortical silent period (cSP) following RT (Kidgell et al., 2017; Siddique, Rahman, Frazer, Pearce, et al., 2020), whereas findings for corticospinal excitability (CSE) remain inconsistent, with reports of increases, decreases, or no change (Carroll et al., 2002; Jensen et al., 2005; Beck et al., 2007; Griffin & Cafarelli, 2007; Hendy & Kidgell, 2013; Leung et al., 2017; Giboin et al., 2018; Ansdell et al., 2020; Colomer-Poveda et al., 2021). Notably, training modality has emerged as a critical determinant influencing the relationship between CSE and strength adaptation (Gómez-Feria et al., 2023).”

Comment 9. No clear rationale is put forward for why these two types of RT may have differential effects on the nervous system. No underlying theory is introduced or explained.

Response:

We appreciate this important observation. In the revised manuscript, we now provide a clear theoretical rationale for why MP-RT and SP-RT may differentially affect the nervous system. Specifically, pacing, encompassing synchronisation and tempo, has emerged as a key determinant of neuromuscular adaptation (Wilk et al., 2021; Gómez-Feria et al., 2023). MP-RT, via externally imposed timing, promotes attentional focus, precise force control with slow tempo, and enhanced sensory feedback, thereby engaging premotor and supplementary motor areas and facilitating corticospinal plasticity (Hortobágyi et al., 1997; Gerloff et al., 1998; Ackerley et al., 2011). Consistent evidence shows MP-RT increases CSE and reduces SICI (Leung et al., 2017; Gordon et al., 2024). By contrast, SP-RT, characterised by internally timed, faster, and less controlled actions, prioritises force output over temporal precision and elicits more diffuse cortical–subcortical activation, often yielding strength gains without measurable increases in corticospinal excitability (Siddique et al., 2020). This revised section introduces the underlying theory and cites supporting evidence (*Page 5, Lines 120–123; Page 6, Lines 124–137*).

Comment 10: L128-29 A sweeping and somewhat erroneous sentence. The neural adaptation driving any type of strength gains remain unclear. Would corticospinal changes be expected to fully account for strength gains? and can such a contribution (fully or partially accounting) even be measured in any meaningful way.

Response: We appreciate the reviewer’s concern that our original statement might be too sweeping. We have revised the text to more cautiously reflect the current evidence. The sentences now read:

“...Such motor patterns tend to elicit more diffuse cortical and subcortical activation without the targeted facilitation observed in MP-RT (Siddique, Rahman, Frazer, Pearce, et al., 2020; Gómez-Feria et al., 2023), often yielding strength gains without measurable changes in corticospinal excitability (Siddique, Rahman, Frazer, Leung, et al., 2020). These findings suggest that strength gains following SP-RT may be mediated by non-corticospinal pathways.”

Comment 11: This sentence also contains grammatical errors.

Response: The sentence has been revised to correct the grammatical error as well.

Comment 12: L130-31 'Potential candidate'? the writing lacks clarity without being explicit about candidate for what?

Response: We agree with the reviewer that the phrase “potential candidate” lacked clarity. We have revised the sentence to explicitly state that the reticulospinal tract (RST) is considered a potential tract mediating strength adaptations. The revised text now reads (**Page 6, Lines 137–139**):

“...One potential candidate for mediating such adaptations is the reticulospinal tract (RST), which has been implicated in gross motor and forceful movements (Baker, 2011; Brownstone & Chopek, 2018).”

Comment 13: L154 'contributes to motor output modulation' appears quite simplistic and unjustified based on the observation of increased activation. Couldn't the increased activation be a response or coincidental effect, rather than a cause as implied.

Response: We appreciate this valuable comment and agree that our original phrasing implied causality beyond the evidence. We have revised the paragraph and no such wording implying causality beyond the evidence (**Page 6, line 146-149**).

“Functional magnetic resonance imaging has demonstrated increased activation of the reticular formation during force production (Danielson et al., 2024), and behavioural paradigms reveal that trained individuals exhibit shorter response latencies and greater rate of force development (RFD) to startling stimuli compared with untrained controls (Akalu et al., 2024)”

Comment 14: L155 'key sub-cortical driver' appears hyperbole based on the evidence presented. If there is more convincing evidence the authors need to bring that to the fore, otherwise the language needs to be substantially toned down.

Response: We agree that our original wording (“key subcortical driver”) was too strong based on the available evidence. To make the Introduction concise this section is incorporated to other sections ensuring no hyperbole.

Comment 15: L161-3 It should be explicit that this is in primates.

Response: This is an important point, and we have revised the sentence to explicitly state that the evidence described is from primate studies, which now reads “In non-human primates, RT enhanced...” (**Page 7, line 151**)

Comment 16: L166 That between group 'enhanced RFD' alone suggests RST-mediated neural drive is simplistic.

Response: We acknowledge that attributing enhanced RFD solely to RST-mediated neural drive. To address this, we have revised the sentence to adopt a more cautious interpretation. The revised text now reads:

“...Similarly, rock climbers displayed enhanced RFD in response to startling stimuli, potentially reflecting reticulospinal contributions among other mechanisms (Colomer-Poveda et al., 2023),” (**Page 7, Lines 149-151**)

Comment 17: L172 'Given the established role...' This seems too strong based on what has been presented.

Response: The revised text now reads:

“Given the proposed role of the RST in force production and the evidence of its plasticity in response to Resistance training in non-human primates...” (**Page 7, line 157 and 158**).

Comment 18. L174-177 There is a major jump in the narrative, from 'how RT modulates RST excitability' (that sounds like a study of mechanism between an intervention and an outcome), to a comparison of the two types of training. See comments above about the lack of rationale, and thus connection between these points.

Response:

We appreciate the reviewer’s observation. Our intention in the Introduction was to build a stepwise rationale rather than to shift abruptly between unrelated points. Specifically, the Introduction was framed as follows:

1. **Uncertainty in the field:** the exact neural site of early strength adaptation remains inconclusive or inconsistent.
2. **Recent evidence on modality:** systematic reviews indicate that training modality is a key determinant: MP-RT has been linked to cortical and corticospinal adaptations, whereas SP-RT consistently improves strength without evidence of cortical or corticospinal plasticity.

3. **Resulting gap:** this discrepancy suggests that another descending pathway must contribute to the strength gains seen with SP-RT.
4. **Candidate mechanism:** evidence from non-human primates indicates that RST excitability increases following resistance training, but this has never been examined in humans mainly due to methodological difficulty of examining RST.
5. **Our objectives:** building on both the unresolved role of the RST and the modality-specific evidence, we aimed (i) to examine the effect of resistance training on RST excitability in humans, and (ii) to determine whether different training modalities differentially modulate distinct descending pathways (CST and RST).

To avoid the impression of a narrative jump, we have revised the text to more clearly connect these elements. The final section of the introduction now reads:

“Given the proposed role of the RST in force production and its plasticity in non-human primates (Glover & Baker, 2020), this gap is particularly salient in light of evidence that training modality determines the neural locus of adaptation. MP-RT appears to preferentially engage corticospinal circuits, whereas SP-RT yields strength gains without corticospinal changes, suggesting that the RST may underpin SP-RT adaptations.

Accordingly, the present study had two aims: first, to determine the effect of resistance training on RST excitability in humans, and second, to examine whether MP-RT and SP-RT differentially modulate cortical, corticospinal, cortico-reticular, and reticulospinal pathways. We hypothesised that SP-RT, due to its reliance on internally generated motor commands, would preferentially enhance cortico-reticular and RST excitability, whereas MP-RT would predominantly modulate corticospinal pathways (Page 7, line 154-167).”

Comment 19: L176 The aim is quite unspecific and descriptive, if the primary aim is to compare the groups/interventions that should be stated explicitly.

Response: As noted in our response to Comment 18, we have clarified the rationale to show that the investigation of RST excitability and the comparison between training modalities are two integrated aims, not separate points. We agree that the original statement of aims was phrased too descriptively and did not sufficiently emphasise the comparative nature of the study. To address this, we have revised the text to make the objectives explicit:

“Accordingly, the present study had two aims: first, to determine the effect of RT on corticoreticular and RST excitability in humans, and second, to examine whether MP-RT and SP-RT differentially modulate cortical, corticospinal, cortico-reticular, and reticulospinal pathways.” (Page 7, lines 162-165)

Comment 20: L178 The basis for this hypothesis has not been introduced or explained.

Response: We have now provided a detailed theoretical basis for why MP-RT and SP-RT are expected to produce differential neural adaptations. Specifically, we expanded the introduction (**Page 5, Lines 120–123; Page 6, Lines 124–137**) to explain that MP-RT, by requiring externally paced motor timing and attentional control, is likely to engage corticospinal pathways, whereas SP-RT, with its reliance on internally generated motor commands, may preferentially involve cortico-reticular and reticulospinal circuits. The hypothesis therefore builds directly on this revised rationale.

Comment 21: L180-3 Unnecessary hyperbole.

Response: We recognise that the original phrasing overstated the contribution of this study. To address this, we have revised the sentence to adopt a more balanced tone, focusing on the study's contribution without hyperbole. The revised text (**Page 7, lines 167-169**) now reads:

“By extending findings from animal models to human participants, this study provides evidence for pathway-specific plasticity following RT and suggests that the RST may contribute to early strength adaptations in human.”

Methods:

Comment 22: L191 The sample size calculation seems to be based on a pre-post comparison of one group, when there are 3 groups and extensive comparisons within this study. The sample size calculation needs to be coherent with the aim, unfortunately as above this lacks clarity.

Response: We acknowledge that the description of the sample size calculation in the original manuscript did not make our approach sufficiently clear. The published effect size we drew upon (Cohen's $d = 0.55$ for a pre-post increase in CSE; Siddique et al., 2020b) was indeed from a within-group comparison. However, we did not apply this directly; instead, given our three-group design, we converted Cohen's d to Cohen's f ($f = 0.275$), which is suitable for a mixed model (Group \times Time). This conversion allowed us to adapt the single-group effect size to our three-group repeated-measures design. In G*Power, we specified a repeated-measures between-within interaction (3 groups \times 4 time points), consistent with our study aim. To make this explicit, we have now clarified this in the Methods (**Page 8, lines 177-182**):

Comment 23. Its awkward to have such a sex imbalance in this study.

Response:

While it is unfortunate that there were fewer females in the present study, participants were sex-matched across the three study groups (ensuring comparable numbers of males and females in each), minimising the risk of sex-related confounding in the between-group

comparisons that form the basis of our analysis. Moreover, accumulating evidence indicates that sex does not substantially influence short-term neural adaptations to resistance training: corticospinal excitability, intracortical inhibition, and motor unit activation during upper-limb contractions appear similar in males and females (El-Sayes et al., 2019; Miller et al., 1993). Controlled trials also show comparable upper-body strength gains in men and women (Gentil et al., 2016), with differences often attributed to training status rather than sex itself (Delmonico et al., 2005). Nonetheless, we recognise that a more even sex distribution within groups would strengthen generalisability, and this limitation has been noted in the revised manuscript (*Page 43, Lines 1009–1011*)

Comment 24: L221 What was involved in stratified randomisation?

Response: We thank the reviewer for this query. Stratified randomisation in our study involved grouping participants by sex, and then further balancing them by age and baseline strength within each sex stratum. This ensured that males and females were evenly distributed across groups, and that group allocation was also balanced for age and strength.

To make this explicit, we have revised the manuscript (Page 9, Lines 209–215), which now reads: *“Participants were allocated 1:1:1 using sex-stratified, covariate-adaptive, computer-generated randomisation, conducted independently within women (n=6) and men (n=24). Within each sex stratum, age and baseline strength were dichotomised at the sex-specific median, and the system assigned each participant to the group that minimised overall imbalance in group totals and covariate margins, yielding two women and eight men per group, thereby ensuring equal sex distribution and optimising balance for age and baseline strength.”*

Comment 25: L231 'Additionally, iMEPs were recorded to enhance data reliability.' Unclear and needs explanation

Response: We thank the reviewer for highlighting this ambiguity. Our intention was to describe familiarisation with the iMEP measurement procedure, not the recording of iMEP.

We have revised it to reflect this and now reads (*Page 10, line 223-224*):

“. Additionally, participants underwent iMEP measurement familiarisation to ensure correct execution of controlled bilateral elbow flexion-extension with a barbell.”.

Comment 26: L232 Unclear why this was done during familiarisation?

Response: It is solely to train participants and ensure correct execution of the study procedures prior to actual testing.

Comment 27: L247- Quite a basic description of the dynamometry, in a section headed EMG, it is unclear what other measurements this set-up relates to. For example, the position

of the trunk, shoulder, shoulder girdle, wrist and forearm posture are not explained. Consequently, whether this is genuinely an isolated assessment of isometric elbow flexion strength with minimal contribution of other much stronger muscle groups is unclear.

Response: We are sincerely grateful for this thoughtful comment. To maintain clarity, we have expanded the description to specify trunk, shoulder-girdle, wrist and forearm positioning; the alignment and restraint; and the steps taken to minimise non-target contributions.

The revised methods (*Page, 11 and 13, line 243-263*) now read:

“Participants sat upright in an adjustable chair with back support; hips and knees at 90°; feet on the floor. The trunk and shoulder girdle were stabilised using non-elastic pelvic and shoulder straps to minimise compensatory movements. The dominant shoulder was neutral ($\approx 0-10^\circ$ abduction and flexion) with the scapula gently retracted/depressed and the humerus alongside the torso on a support. The dominant elbow was fixed at 90° with the lateral epicondyle aligned with the arm-bar; the forearm was fully supinated (to preferentially load biceps brachii) and the wrist held neutral (0° flexion/extension; 0° radial/ulnar deviation) using a light brace to minimise wrist-flexor contribution. The forearm midline was aligned with the force axis, and the hand and forearm rested on a fixed arm-bar connected to a force transducer (LSB302 S-beam load cell, FUTEK Advanced Sensor Technology Inc., Irvine, CA, USA). Two non-compliant nylon straps were applied to the distal forearm ($\approx 2-3$ cm proximal to the radial/ulnar styloid processes) and tightened against the arm-bar to prevent translation and pronation-supination without impeding distal circulation. The non-testing arm rested comfortably on the lap. Investigators provided standardised verbal cues and continuous visual monitoring to avoid shoulder elevation, trunk motion, wrist flexion, or forearm pronation; any compensated trial was stopped, the position reset, and the attempt repeated. The same positioning and stabilisation procedures were applied across all testing, including MVF, TMS, and StartReact protocol (StartReact effect and RFD). To standardise positioning across trials and visits, all measurement settings were documented for each participant and replicated in subsequent trials.

Comment 28; L251 Unclear; how was the forearm 'aligned with a fulcrum' ? 'padded restraint'? positioned where and standardised how.

Response: These details are now clarified in the revised text (see Comment 27 response above). Briefly, the lateral epicondyle was aligned with the arm-bar, the forearm midline was set collinear with the force axis, and a padded restraint was placed on the distal forearm ($\approx 2-3$ cm proximal to the radial/ulnar styloids) and tightened against the fulcrum stop to prevent

translation and pronation-supination. To standardise positioning across trials and visits, all measurement settings were documented for each participant and replicated in subsequent trials.

Comment 29: L289 Stimulation of the brachial plexus generates action potentials in the agonist and antagonists to elbow flexion as well as several other muscle groups (shoulder, wrist and hand). This invalidates measurements of voluntary activation, which requires targeted stimulation of the agonist muscle(s) only, and not opposing muscles.

Response: We thank the reviewer for rising this important point. As VA was neither a primary nor a secondary objective and contributes minimally to the aims of this study, we have removed this outcome from the manuscript. This decision not only addresses the methodological concern raised but also streamlines the results, helping to reduce the overall length of the manuscript and maintain focus on the main objectives of the study.

Comment 30: L299 'dominant biceps' highly simplistic, many muscles are involved in elbow flexion, not just 'biceps', which biceps?

Response: We thank the reviewer for the clarification. The term has been revised to “*dominant biceps brachii*”.

Comment 31. Unclear how force measurements were recorded and filtered? Any gravity correction?

Response: Force was recorded using a force transducer (LSB302 S-beam load cell, FUTEK Advanced Sensor Technology Inc., Irvine, CA, USA), sampled at 2000 Hz. Signals were baseline-corrected and low-pass filtered at 12 Hz using a fourth-order zero-lag Butterworth digital filter to remove high-frequency noise while preserving physiological force fluctuations (Hewett et al., 2017). As the contractions were isometric with the forearm fully supported, no gravity correction was required. These details have now been incorporated into the Methods section (**Page 13, Lines 311-314**).

Comment 32. L306 Why only 2 isometric MVCs? This seems quite limited.

Response: We acknowledge the reviewer’s concern. Our protocol directly followed established procedures reported in prior studies, which typically employed two maximal efforts (with a third performed if necessary) to ensure reliable MVC measurement (Lahouti et al., 2019; Škarabot et al., 2022). The rationale for this approach was to confirm that participants achieved true maximal effort while avoiding unnecessary fatigue. When we refer to “two MVCs,” this always meant two maximal attempts, with a third trial performed whenever the first two differed by more than 5% or when maximal effort was clearly not achieved.

Importantly, participants were well familiarised beforehand to ensure they understood the task demands and could apply true maximal effort during testing. When the two contractions were within 5% of each other, this indicated consistent and reliable maximal performance; under these conditions, a third trial would not provide additional benefit and would only increase participant burden. Conversely, if the difference exceeded 5%, this suggested that at least one trial was not truly maximal, and in such cases a third MVC was performed by default. Participants were strongly encouraged, and provided with visual feedback, to ensure that all final values were based on reliable maximal efforts.

Furthermore, since our study involved multiple neurophysiological measures, limiting extra trials was also important to minimize fatigue and maintain data quality across sessions. Therefore, this approach, consistent with established practice, avoids unnecessary participant burden while maintaining data quality. We have cited the references that followed similar approach in the revised Methods section (*Page 14, lines 319 - 320*).

Moreover, our procedure is in line with methods adopted in other peer-reviewed studies across diverse populations, including healthy adults (Hortobágyi et al., 1996; Carroll et al., 2002; Lee et al., 2009; Baker et al., 2017), older adults (Hernández-Murúa et al., 2025), and clinical populations (de Souza-Teixeira et al., 2009; Medina-Pérez et al., 2016; Portilla-Cueto et al., 2022), which measured only 2 MVCs.

References

Comment 33: L313 'wrist flexion from full elbow extension' is confusing. Unclear how or if shoulder and elbow position were controlled in any way. What was the range of motion and how was this assessed?

Response:

We thank the reviewer for these helpful observations. The use of the term “wrist flexion” was a typographical error; the intended description was “elbow flexion.” This has been corrected in the revised Methods section.

We have also clarified how joint positioning and range of motion were standardised and assessed. During unilateral 1RM testing, participants stood upright with the opposite arm placed behind their back while standing against a wall to prevent trunk and shoulder movement. The dominant shoulder was positioned in adduction and neutral flexion/extension, with the upper arm maintained alongside the torso throughout the test. The forearm was fully supinated, and each attempt began from full elbow extension ($\sim 0^\circ$) and continued through the full elbow-flexion range of motion until the dumbbell approached the shoulder. Range of motion and technique were visually assessed by the investigator, and any trial with shoulder

displacement, trunk compensation, or incomplete range of motion was repeated. This description has now been incorporated into the revised Methods section (*Page 14, lines 322-331*).

Comment 34: L322 The specifics of the training exercise task is not described in any way.

Was it the same as the 1-RM testing exercise or something else?

Response:

The training exercise was indeed the same supervised unilateral dumbbell biceps curl as used for 1RM testing, but performed at prescribed loads relative to each participant's baseline 1RM. We have clarified this in the revised Methods section (*Page 15, line 341-3343*) as follows.

“..... Training for both groups consisted of the same supervised unilateral dumbbell biceps curl described for 1RM testing, performed under identical positioning and stabilisation procedures to minimise non-target contributions.....”

Comment 35: Moreover, with this standing elbow flexion exercise, strength and any training effects are heavily dependent on whole body stabilisation and co-ordination of multiple joints/ muscle groups. In other words this is a non-isolated way to assess and train what is described and presented as a single joint exercise. Its unclear why this model and approach would be chosen.

Response:

The reviewer is correct that standing elbow flexion with free weights requires whole-body stabilisation and cannot entirely isolate the biceps brachii. However, several steps were taken to minimise contributions from other joints and muscle groups.

Participants performed the unilateral dumbbell curl with their back against a wall, the dominant shoulder adducted and in neutral flexion/extension, the upper arm fixed alongside the torso, and the forearm fully supinated. Range of motion and technique were strictly monitored, and any trial with visible compensations was repeated. These measures ensured that force production was driven primarily by the elbow flexors. The procedure is now clearly explained in the revised version (*Page 14, Line 323-332, Page 15, Line 342-351*)

We selected the standing dumbbell curl because it represents a functionally relevant and widely practiced model of elbow flexor training, thereby enhancing ecological validity compared with more constrained machine-based protocols. This approach reflects common gym-based resistance training scenarios and aligns with previous studies that examined neural and muscular adaptations of the biceps brachii using similar unilateral standing biceps

curl protocols (e.g. Jensen JL, et al., 2005, Marcolin G, et al., 2018, Mason J et al., 2019, Pearce AJ, et al. 2013).

Finally, because the same protocol was applied uniformly across all groups, any stabilisation demands were equally distributed and are therefore unlikely to bias between-group comparisons.

Comment 36: L331 A 3-s lift and 4-s lowering phase is unusually slow and not a normal lifting velocity or cadence. Its unclear why this was chosen. There's also no information on how this was done i.e. instructions, controlled, monitored etc.

We sincerely appreciate the comment and the opportunity to clarify. The choice of a 3-second concentric and 4-second eccentric cadence was deliberate to ensure accurate synchronisation with the metronome. Faster lifting velocities, although closer to natural tempo, are extremely difficult to align precisely with a metronome beat and often result in desynchronization across repetitions; in practice, synchronisation at faster cadences is not feasible. By contrast, the slower cadence provided smoother synchronisation, minimized momentum, and allowed precise monitoring of repetitions.

Prior to data collection, participants completed a dedicated familiarization session during which they received clear instructions, practiced under metronome guidance, and were trained until they could reliably synchronise with the metronome by maintaining the prescribed cadence. During the experimental sessions, the same clear instructions were reiterated, and participants were guided by an audible metronome throughout all sets. To ensure proper adherence, the investigator closely supervised performance, provided verbal feedback, and immediately corrected and repeated any trial that deviated from the prescribed cadence.

This approach ensured controlled, standardised execution of metronome synchronisation across participants, minimised reliance on ballistic movements, and optimised reproducibility. Moreover, the selected tempo reflects protocols commonly employed in resistance training studies investigating neural and muscular adaptations, where slower cadences facilitate consistency and precise monitoring of repetitions (Goodwill et al., 2013, Leung M., et al., 2015 Leung, M. et al., 2017, Siddique U. et al., 2025). These methodological details are now explicitly stated in the revised Methods section (in the familiarisation section on *Page 10, line 224-229*, and in the “Resistance training protocol” section *Page 15, lines 355–361*).

Comment 37: L336 There is no information on the instructions given to the SP-RT group regarding the training they performed.

Response:

We are grateful for the opportunity to clarify. In the SP-RT group, participants performed the same supervised unilateral dumbbell curl protocol as the MP-RT group (four sets of 6–8 repetitions at 70–75% 1RM, 2-min inter-set rest) but without external tempo control. They were instructed to lift and lower the dumbbell as they would during a standard biceps curl, performing each repetition at their natural, self-selected pace while maintaining proper technique and full range of motion. The investigator closely supervised all sessions, provided verbal feedback to prevent compensatory movements, and ensured adherence to the prescribed load and repetition range. These details have now been added to the revised Methods (*Page 15, lines 362-367*).

Comment 38: Was there any quantification (kinetics, kinematics, EMG) of the training to verify that the training was on average performed in this unusually slow manner described. Without a metronome it may be even harder to maintain such a slow pace. As above was there any verification that the two groups did in fact perform the training in the same manner. A different lifting velocity, RFD or time under tension may lead to profoundly different neuromuscular adaptations, irrespective of self-paced or not. This verification of an equivalent mechanical training stimulus seems critical to demonstrating any effect of pacing mechanism.

Response:

We thank the reviewer for this important point. Training execution was not verified with detailed kinetic, kinematic, or EMG analyses, which we acknowledge as a limitation and have now stated explicitly in the manuscript (*Page 42, Lines 1000–1002; Page 43, Lines 1003-1006*). The purpose of the study was not to equate lifting velocity between groups, but to contrast two distinct modalities: metronome-paced RT (MP-RT), where synchronisation with an external rhythm necessarily produced a slower cadence, and self-paced RT (SP-RT), where repetitions were performed at the participant's natural tempo. Such differences in cadence were therefore fundamental to the contrast under investigation, consistent with previous comparisons of self-paced and externally paced training (Goodwill et al., 2013; Leung et al., 2015, 2017; Siddique et al., 2025).

In practice, the MP-RT group successfully synchronised with the metronome, which by default resulted in a slower cadence and fewer repetitions per set, whereas the SP-RT group trained at a naturally faster tempo, completing more repetitions within the same set structure. This divergence in cadence and repetition count was not a confound but a defining feature of the interventions, reflecting how these training modes are implemented in real-world settings.

To objectively characterise execution, we systematically recorded time-under-tension using a digital stopwatch (its methodological details are now explicitly reported in the revised Methods section (*Page 16, Lines 370–373*)). In addition, total volume load (sets × repetitions × load) was calculated to capture the overall mechanical work performed. These measures confirmed the expected differences in cadence and repetition count between groups, thereby preserving the intended experimental contrast. All sessions were closely supervised, and deviations were corrected immediately.

Comment 39: L388 The iMEP measurements were collected during bilateral exercise task. Is the mismatch with unilateral training and testing an issue?

Response:

We acknowledge that iMEPs were elicited during a bilateral activation task, whereas resistance training was performed unilaterally. Bilateral contraction was required because it is the most robust and reliable method to evoke iMEPs in the upper limb (Tazoe & Perez, 2014; Altermatt et al., 2023; Hu et al., 2024). The main aim of measuring iMEPs was to assess changes in the excitability of the dominant (left) corticoreticular projection following unilateral resistance training. Unilateral training of the dominant (right arm) engages the left hemisphere, whose corticoreticular projections contribute bilaterally to the reticulospinal tract (Fisher et al., 2012; Bradnam et al., 2013), thereby enabling us to probe corticoreticular changes induced by training.

During testing, stimulation of the trained (left) hemisphere activated both contralateral corticospinal output to the trained arm and cortico-reticulospinal projections, producing iMEPs in the ipsilateral arm. Bilateral contraction facilitated reliable elicitation of iMEPs in the ipsilateral arm, while also allowing contralateral MEPs to be recorded from the trained arm. These contralateral MEPs were used to estimate the relative excitability of corticospinal versus cortico-reticulospinal pathways and to confirm the presence of true iMEPs, defined by a latency difference of >5 ms compared with contralateral MEPs.

Thus, although contractions during testing were bilateral, this was necessary to ensure successful iMEP elicitation. Crucially, the neural target remained the trained hemisphere, allowing us to assess training-induced changes in corticoreticular excitability under conditions optimised for reliable iMEP measurement. This approach is consistent with established neurophysiological methodology (Alagona et al., 2001; Maitland & Baker, 2021; Altermatt et al., 2023; Hu et al., 2024).

Comment 40: L418 The measurement situation for the StartReact protocol is not described.

Response: We agree with the reviewer's observation. In the revised Methods, we have now specified that participants were positioned and stabilised in the same manner as during the MVF assessment (as described earlier in response to Comment 27). We also provide a detailed description of the procedure, task, stimuli, and recorded outcomes. These details are now included on *Page 19, Lines 464–480* of the revised manuscript.

Comment 41: L544 It is unclear what contractions were used to assess RFD. If the two isometric MVCs that is a weak approach.

Response: To clarify, RFD was not derived from the MVCs, which were conducted to assess maximal voluntary strength, and their instructions (gradual rise to maximum and sustained plateau) are not suitable for obtaining a valid estimate of RFD. Instead, RFD was assessed as part of the StartReact protocol, during separate rapid, forceful isometric elbow flexion contractions (30 trials in total, 10 per condition) performed in response to a visual imperative signal, with or without concurrent startling or non-startling auditory stimuli. Participants were instructed to contract as fast and as hard as possible, while force-time data were sampled continuously, and RFD was calculated as the slope of the force trace within the 0–50 ms and 50–100 ms windows following contraction onset. This procedure is consistent with established StartReact methodologies (Anzak et al., 2011; Del Vecchio et al., 2019; Škarabot et al., 2022; Colomer-Poveda et al., 2023). We have clarified this in the revised Methods section (*Page 19, Lines 464–480*).

Results

Comment 42. L603-5 Is this describing pre to post changes? The GxT interaction is not followed up, which groups changed differently over time compared to others? This requires interaction contrasts (or similar) to assess the groups. What follows in the next paragraph considers within groups changes, which does not inform about a between group interaction effect.

Response:

We agree our original presentation did not sufficiently follow up the significant Group × Time interaction. In the revised manuscript, we have re-analysed all outcome variables using interaction contrast analysis, which directly tests which groups changed differently over time compared with others. The Results section has been fully revised (*Pages 28–35*) to report these contrasts, with the findings also integrated into the figures and tables. This revision moves beyond simple within-group comparisons and explicitly identifies the between-group differences driving the interaction, thereby clarifying the interpretation of these effects.

Comment 43: For the strength changes, only the %age changes are displayed in the Figures. Was this the data used for the primary analysis? And if not please display the actual pre and post data.

Response:

To clarify, all primary analyses were conducted using the raw pre- and post-intervention data, not percentage changes. Percentage change values were displayed in the figures to provide a clearer visualisation of the relative magnitude of change between groups. In the revised version, we have updated the figures to display the actual pre- and post-intervention values for each group and time point, alongside the interaction contrast statistical results.

Comment 44: For CSE, and then CSP, the same pattern is repeated, with no genuine follow-up to the G x T interaction. Within-group post-hocs do not show how specific groups compare over time.

Response: We agree with the reviewer that in the original manuscript, the follow-up to the Group × Time interactions for CSE and cSP relied on within-group post-hoc analyses, which do not fully capture between-group differences in change over time. To address this, we have re-analysed the data using interaction contrast analyses. The revised Results now explicitly reports which groups changed differently compared with others across time, providing a clearer interpretation of the interaction effects, which now reads (**for CSE: Page 31, Lines 730-738, for cSP Page 32, Line 746-753**). “... *Post-hoc interaction contrasts revealed that the MP-RT group exhibited greater increases in CSE from baseline compared with the Control group at Week 1 (EMD = 0.59, SE = 0.21, $t(81) = 2.85$, $p = 0.049$) and Week 3 (EMD = 0.64, SE = 0.21, $t(81) = 3.09$, $p = 0.025$). Between-group comparisons at each time point showed that at Week 1, the MP-RT group had higher CSE than the Control group (EMD = 0.61, SE = 0.24, $t(49) = 2.50$, $p = 0.048$); at Week 2, the MP-RT group exceeded the SP-RT group (EMD = 0.63, SE = 0.24, $t(49) = 2.59$, $p = 0.038$); and at Week 3, the MP-RT group had higher CSE than both the Control group (EMD = 0.66, SE = 0.24, $t(49) = 2.70$, $p = 0.029$) and the SP-RT group (EMD = 0.76, SE = 0.24, $t(49) = 3.11$, $p = 0.009$).”*

Comment 45. The results need to be re-analysed with appropriate statistics i.e. interaction contrasts or similar.

Response: An excellent point was raised by the reviewer, and accepting their suggestion, we have re-analysed all outcome variables using interaction contrast analyses, which directly test which groups changed differently across time. The Results section has been fully revised (**Pages 28–35**) to present these contrasts, ensuring that the reported findings more clearly

reflect the underlying Group \times Time interaction effects. Importantly, these additional analyses did not alter the main conclusions but rather strengthened the overall narrative and robustness of the findings. Figures and tables have also been updated accordingly, displaying both within- and between-group differences across time points.

Discussion

Comment 46. L785 Unfortunately contrary to this statement these two groups have not been directly compared. Three groups are included in the ANOVA, and some significant GXT interactions have been noted. However, of the 3 groups, which specific groups show differences over time than any other is unclear. It could be that these GxT interactions are purely reflecting differences between the control group and one of the training groups, but that the two training groups produce equivalent changes over time. Comparison of specific groups is required for a meaningful worthwhile manuscript.

Response:

We thank the reviewer for this important observation. In the revised analyses, we have applied interaction contrasts to all outcome variables, which directly test between-group differences over time. The Results now clearly report whether changes differed between the two training groups or only relative to the control group. Importantly, the Discussion is now strongly supported by these new analyses, with interpretations explicitly grounded in the interaction contrasts.

References Cited in Author Response to Referee 2

- Anzak, A., Tan, H., Pogosyan, A. & Brown, P. (2011). Doing better than your best: loud auditory stimulation yields improvements in maximal voluntary force. *Experimental Brain Research*, 208, 237–243.
- Baker, S.N. & Perez, M.A. (2017). Reticulospinal contributions to gross hand function after human spinal cord injury. *Journal of Neuroscience*, 37(40), 9778–9784. doi: 10.1523/JNEUROSCI.3368-16.2017.
- Beck, S., Taube, W., Gruber, M., Amtage, F., Gollhofer, A. & Schubert, M. (2007). Task-specific changes in motor evoked potentials of lower limb muscles after different training interventions. *Brain Research*, 1179, 51–60.
- Carroll, T.J., Barton, J., Hsu, M. & Lee, M. (2009). The effect of strength training on the force of twitches evoked by corticospinal stimulation in humans. *Acta Physiologica*. <https://doi.org/10.1111/j.1748-1716.2009.01992.x>
- Carroll, T.J., Riek, S. & Carson, R.G. (2001). Neural adaptations to resistance training: implications for movement control. *Sports Medicine*, 31(12), 829–840.
- Carroll, T.J., Riek, S. & Carson, R.G. (2002). The sites of neural adaptation induced by resistance training in humans. *Journal of Physiology*, 544(2), 641–652. doi: 10.1113/jphysiol.2002.024463.
- Colomer-Poveda, D., López-Rivera, E., Hortobágyi, T., Márquez, G. & Fernández-Del-Olmo, M. (2023). Differences in the effects of a startle stimulus on rate of force development between resistance-trained rock climbers and untrained individuals: Evidence for reticulospinal adaptations? *Scandinavian Journal of Medicine & Science in Sports*, 33, 1360–1372.
- Coombs, T.A., Frazer, A.K., Horvath, D.M., Pearce, A.J., Howatson, G. & Kidgell, D.J. (2016). Cross-education of wrist extensor strength is not influenced by non-dominant training in right-handers. *European Journal of Applied Physiology*. <https://doi.org/10.1007/s00421-016-3436-5>
- Del Balso, C. & Cafarelli, E. (2007). Adaptations in the activation of human skeletal muscle induced by short-term isometric resistance training. *Journal of Applied Physiology* (1985), 103(1), 402–411.
- Del Vecchio, A., Negro, F., Holobar, A., Casolo, A., Folland, J.P., Felici, F. & Farina, D. (2019). You are as fast as your motor neurons: speed of recruitment and maximal discharge of motor neurons determine the maximal rate of force development in humans. *Journal of Physiology*, 597, 2445–2456.
- de Souza-Teixeira, F., Costilla, S., Ayán, C., García-López, D., González-Gallego, J. & de Paz, J.A. (2009). Effects of resistance training in multiple sclerosis. *International Journal of Sports Medicine*, 30(4), 245–250.
- Delmonico, M.J., Kostek, M.C., Doldo, N.A., Hand, B.D., Bailey, J.A., Rabon-Stith, K.M., Conway, J.M., Carignan, C.R., Lang, J. & Hurley, B.F. (2005). Effects of moderate-velocity strength training on peak muscle power and movement velocity: do women respond differently than men? *Journal of Applied Physiology* (1985), 99(5), 1712–1718.
- El-Sayes, J., Turco, C.V., Skelly, L.E., Nicolini, C., Fahnestock, M., Gibala, M.J. & Nelson, A.J. (2019). The effects of biological sex and ovarian hormones on exercise-induced neuroplasticity. *Neuroscience*, 410, 29–40.

Faul, F., Erdfelder, E., Lang, A.G. & Buchner, A. (2007). G*Power 3: A flexible statistical power analysis program for the social, behavioral, and biomedical sciences. *Behavior Research Methods*, 39, 175–191.

Folland, J.P. & Williams, A.G. (2007). The adaptations to strength training: morphological and neurological contributions to increased strength. *Sports Medicine*, 37(2), 145–168.

Gentil, P., Steele, J., Pereira, M.C. et al. (2016). Comparison of upper body strength gains between men and women after 10 weeks of resistance training. *PeerJ*, 4, e1627.

Goodwill, A.M., Pearce, A.J. & Kidgell, D.J. (2012). Corticomotor plasticity following unilateral strength training. *Muscle & Nerve*, 46, 384–393.

Graeme, D.R. & Beauchamp, G. (2008). Time for some a priori thinking about post hoc testing. *Behavioral Ecology*, 19(3), 690–693.

Gueorguieva, R. & Krystal, J.H. (2004). Move over ANOVA: progress in analyzing repeated-measures data and its reflection in papers published in the *Archives of General Psychiatry*. 61(3), 310–317.

Hernández-Murúa, J.A., Romero-Pérez, E.M., Guajardo-Cruztiña, J.L., Olivares, B.S.M., Gallego-Selles, Á., González-Martín, D., Reyes-Merino, F., Sánchez-García, N. & de Paz, J.A. (2025). Intra-session reliability and predictive value of maximum voluntary isometric contraction for estimating one-repetition maximum in older women: A randomised split-sample study. *Journal of Functional Morphology and Kinesiology*, 10(2), 160.

Hortobágyi, T., Hill, J.P., Houmard, J.A., Fraser, D.D., Lambert, N.J. & Israel, R.G. (1996). Adaptive responses to muscle lengthening and shortening in humans. *Journal of Applied Physiology* (1985), 80(3), 765–772.

Hortobágyi, T., Richardson, S.P., Lomarev, M., Shamim, E., Meunier, S., Russman, H., Dang, N. & Hallett, M. (2009). Chronic low-frequency rTMS of primary motor cortex diminishes exercise training-induced gains in maximal voluntary force in humans. *Journal of Applied Physiology*. <https://doi.org/10.1152/jappphysiol.90701.2008>.

Hubal, M.J., Gordish-Dressman, H., Thompson, P.D. et al. (2005). Variability in muscle size and strength gain after unilateral resistance training. *Medicine & Science in Sports & Exercise*, 37, 964–972.

Janssen, I., Heymsfield, S.B., Wang, Z.M. & Ross, R. (2000). Skeletal muscle mass and distribution in 468 men and women aged 18–88 yr. *Journal of Applied Physiology* (1985), 89, 81–88.

Jensen, J.L., Marstrand, P.C.D. & Nielsen, J.B. (2005). Motor skill training and strength training are associated with different plastic changes in the central nervous system. *Journal of Applied Physiology* (1985), 99(4), 1558–1568.

Jones, M.D., Wewege, M.A., Hackett, D.A., Keogh, J.W.L. & Hagstrom, A.D. (2021). Sex differences in adaptations in muscle strength and size following resistance training in older adults: A systematic review and meta-analysis. *Sports Medicine*, 51(3), 503–517.

Kamen, G. & Knight, C.A. (2004). Training-related adaptations in motor unit discharge rate in young and older adults. *Journals of Gerontology A: Biological Sciences and Medical Sciences*, 59(12), 1334–1338.

Kidgell, D.J., Stokes, M.A., Castricum, T.J. & Pearce, A.J. (2010). Neurophysiological responses after short-term strength training of the biceps brachii muscle. *Journal of Strength and Conditioning Research*, 24(11), 3123–3132.

Lahouti, B., Lockyer, E.J., Wiseman, S., Power, K.E. & Button, D.C. (2019). Short-interval intracortical inhibition of the biceps brachii in chronic-resistance versus non-resistance-trained individuals. *Experimental Brain Research*, 237(11), 3023–3032.

Lee, M., Gandevia, S.C. & Carroll, T.J. (2009). Short-term strength training does not change cortical voluntary activation. *Medicine & Science in Sports & Exercise*, 41(7), 1452–1460.

Leung, M., Rantalainen, T., Teo, W.P. & Kidgell, D.J. (2015). Motor cortex excitability is not differentially modulated following skill and strength training. *Neuroscience*, 305, 99–108.

Leung, M., Rantalainen, T., Teo, W.P. & Kidgell, D.J. (2017). The corticospinal responses of metronome-paced, but not self-paced strength training, are similar to motor skill training. *European Journal of Applied Physiology*, 117, 2479–2492.

Marcolin, G., Panizzolo, F.A., Petrone, N., Moro, T., Grigoletto, D., Piccolo, D. & Paoli, A. (2018). Differences in electromyographic activity of biceps brachii and brachioradialis while performing three variants of curl. *PeerJ*, 6, e5165.

Mason, J., Frazer, A., Horvath, D.M., Pearce, A.J., Avela, J. & Howatson, G. (2017). Adaptations in corticospinal excitability and inhibition are not spatially confined to the agonist muscle following strength training. *European Journal of Applied Physiology*. <https://doi.org/10.1007/s00421-017-3624-y>

Mason, J., Frazer, A.K., Jaberzadeh, S., Ahtiainen, J.P., Avela, J., Rantalainen, T., Leung, M. & Kidgell, D.J. (2019). Determining the corticospinal responses to single bouts of skill and strength training. *Journal of Strength and Conditioning Research*, 33(9), 2299–2307.

Maxwell, S.E. & Delaney, H.D. (2004). *Designing Experiments and Analyzing Data: A Model Comparison Perspective* (2nd ed.). Mahwah, NJ: Lawrence Erlbaum Associates.

Medina-Perez, C., de Souza-Teixeira, F., Fernandez-Gonzalo, R., Hernandez-Murua, J.A. & de Paz-Fernandez, J.A. (2016). Effects of high-speed power training on muscle strength and power in patients with multiple sclerosis. *Journal of Rehabilitation Research & Development*, 53(3), 359–368.

Miller, A.E., MacDougall, J.D., Tarnopolsky, M.A. & Sale, D.G. (1993). Gender differences in strength and muscle fiber characteristics. *European Journal of Applied Physiology and Occupational Physiology*, 66, 254–262.

Moritani, T. & deVries, H.A. (1979). Neural factors versus hypertrophy in the time course of muscle strength gain. *American Journal of Physical Medicine*, 58(3), 115–130.

Patten, C., Kamen, G. & Rowland, D.M. (2001). Adaptations in maximal motor unit discharge rate to strength training in young and older adults. *Muscle & Nerve*, 24, 542–550.

Pearce, A.J., Hendy, A., Bowen, W.A. & Kidgell, D.J. (2013). Corticospinal adaptations and strength maintenance in the immobilized arm following 3 weeks unilateral strength training. *Scandinavian Journal of Medicine & Science in Sports*, 23(6), 740–748.

Portilla-Cueto, K., Medina-Pérez, C., Romero-Pérez, E.M., Hernández-Murúa, J.A., Vila-Chã, C. & de Paz, J.A. (2022). Reliability of isometric muscle strength measurement and its accuracy prediction of maximal dynamic force in people with multiple sclerosis. *Medicina (Kaunas)*, 58(7), 948.

Roberts, B.M., Nuckols, G. & Krieger, J.W. (2020). Sex differences in resistance training: a systematic review and meta-analysis. *Journal of Strength and Conditioning Research*, 34(5), 1448–1460.

Ruxton, G.D. & Beauchamp, G. (2008). Time for some a priori thinking about post hoc testing. *Behavioral Ecology*, 19(3), 690–693.

Santos Junior, E.R.T., De Salles, B.F., Dias, I., Simão, R. & Willardson, J.M. (2024). Sex differences in neuromuscular adaptations following 12 weeks of kettlebell swing training. *Cureus*, 16(9), e70551.

Siddique, U., Frazer, A.K., Tallent, J., Hayman, O., Andrushko, J., Ahtiainen, J.P., Avela, J., Akalu, Y., Rostami, M., Uribe, S., Walker, S. & Kidgell, D.J. (2025). Acute corticospinal and reticulospinal responses to strength training in ageing. *Neurobiology of Aging*, 153, 49–62.

Škarabot, J., Brownstein, C.G., Casolo, A., Del Vecchio, A. & Ansdell, P. (2021). The knowns and unknowns of neural adaptations to resistance training. *European Journal of Applied Physiology*, 121(3), 675–685.

Škarabot, J., Folland, J.P., Holobar, A., Baker, S.N. & Del Vecchio, A. (2022). Startling stimuli increase maximal motor unit discharge rate and rate of force development in humans. *Journal of Neurophysiology*, 128, 455–469.

Trappe, S., Gallagher, P., Harber, M., Carrithers, J., Fluckey, J. & Trappe, T. (2003). Resistance training in older women: adaptations in single muscle fibers. *Journal of Physiology*, 552, 47–58.
Weiss, L.W., Clark, F.C. & Howard, D.G. (1988). Effects of heavy-resistance triceps surae muscle training on strength and muscularity of men and women. *Physical Therapy*, 68, 208–213.

Response to Referee #3:

General comment:

The current manuscript investigates the different influence of metronome-paced and self-paced resistance training of the biceps brachii on the two main descending motor systems. By comparing three different groups, they found remarkably strong effects showing an increased corticospinal drive after metronome-paced resistance training, while self-paced resistance training appeared to lead to an enhanced reticulospinal drive to the biceps. The authors set up a nice paradigm with a 3-week training paradigm and regular strength and thorough neurophysiological assessments. The corticospinal system was assessed by investigating cortical excitability, cortical silent period, and intracortical excitability (short intracortical inhibition and intracortical facilitation). The reticulospinal and cortico-reticulospinal system was assessed by StartReact and ipsilateral motor evoked potentials (iMEP) or particularly the iMEP vs. cMEP amplitude ratio.

The choice of methodology is quite elaborate and impressive. Additionally, the outcome is quite exciting as this suggests a training intervention possible to target the different motor pathways more specifically. If this holds for a broader selection of muscles, this can impact future rehabilitation interventions for neurologically impaired patients, e.g., Spinal cord injury or stroke.

Nevertheless, I have more specific comments, questions, and suggestions for the manuscript.

Response:

We sincerely appreciate the reviewer's thoughtful summary of our work and recognition of both the methodological approach and the potential implications of our findings. We are encouraged that the paradigm and outcomes are considered novel and impactful, particularly regarding the possibility of targeting corticospinal and reticulospinal pathways through distinct resistance training modalities. We agree that, if these results generalise across a broader range of muscles, the findings may have important translational relevance for rehabilitation in neurologically impaired populations. We have carefully addressed the specific comments, questions, and suggestions outlined below, and have revised the manuscript accordingly.

Major comments:

Comment 1: L195 The authors perform an a priori power calculation to assess the required number of participants. The result they find is 27. They decide to include 10 participants in each of the three groups. However, it appears that 27 participants per group would be required. Why did the authors decide to reduce the number of participants from the number calculated by the power calculation? The data nevertheless show a strong effect even with the lower number of participants.

Response: We appreciate the reviewer's careful reading of our power calculation. To clarify, the sample size of 27 reported in the manuscript represents the total number of participants across all three groups, rather than 27 participants per group. This is because GPower reports total sample size (N) for repeated-measures within-between interaction designs. This can be confirmed from the degrees of freedom (df) in our GPower output (see snapshot below): the denominator df is calculated as $(N - k) \times (m - 1)$, where N is the total sample size, k is the number of groups, and m is the number of repeated measurements. Substituting our values (N = 27, k = 3, m = 4) gives $(27 - 3) \times 3 = 72$, which exactly matches the denominator df shown in the GPower output. If 27 participants were per group (total N = 81), the denominator df would be 234, which clearly does not match. Therefore, the GPower calculation confirms that 27 is the total sample size across all groups, corresponding to 9 participants per group. To make it clear for readers, the methods section is now revised as:

*“The minimum sample size (N = 27; 9 per group) was determined a priori using G*Power (Faul et al., 2007), for a mixed (Group × Time; 3 groups × 4 time points) design, with $\alpha = 0.05$, power = 0.80, and an effect size (Cohen's $f = 0.275$), derived from a standard mean difference (Cohen's $d = 0.55$) for an increase in CSE following RT (Siddique, Rahman, Frazer, Pearce, et al., 2020). The effect size value, Cohen's d was converted to Cohen's f ($f = 0.275$) to estimate the effect size appropriate for our three-group design. Following institutional ethical approval, 30 healthy, RT-naïve adults (24 males, 6 females) were recruited, exceeding the minimum estimated sample size of 27, to ensure adequate power and account for potential attrition....”*

Type of power analysis	
A priori: Compute required sample size - given α , power, and effect size	
Input Parameters	Output Parameters
Determine =>	Noncentrality parameter λ
Effect size f	0.275
α err prob	0.05
Power (1- β err prob)	0.8
Number of groups	3
Number of measurements	4
Corr among rep measures	0.5
Nonsphericity correction ϵ	1
	Critical F
	2.2274040
	Numerator df
	6.0000000
	Denominator df
	72.0000000
	Total sample size
	27
	Actual power
	0.8424877

Comment 2: L230 I understand single-arm dumbbell exercises were performed. Were they performed bilaterally or only for the dominant biceps brachii?

Response:

We thank the reviewer for seeking clarification. All resistance training exercises were performed unilaterally with the dominant (right) arm, specifically targeting the biceps brachii. This has now been explicitly stated in the Methods section to avoid ambiguity. Now it reads: *“Participants were introduced to key methodologies, including the single-arm dumbbell curl exercise performed with the dominant (right) arm...”* (Page 10, Lines 220-222).

Comment 3:

L261 and L497 If I understand correctly, cMEPs assessing the corticospinal system were recorded from the dominant/right biceps brachii, while iMEPs for the reticulospinal system were recorded from the non-dominant/left biceps brachii. This means you are comparing signals from different muscles, which could complicate the interpretation. Recording from the same muscle would strengthen the comparison between systems.

Response: We appreciate the reviewer’s observation. Our choice to record iMEPs from the left (non-dominant) biceps brachii was driven by the aim to assess cortico-reticulospinal excitability with minimal interference from corticospinal tract (CST) activity. TMS over the left M1 primarily activates contralateral CST fibres projecting to the right (dominant) limb, making it difficult to isolate cortico-reticulospinal contributions on that side. In contrast, iMEPs are typically absent under standard TMS because the CST is predominantly crossed, with ~85–90% of fibres decussating at the pyramids and the remaining 10–15% often crossing at the spinal level (Lemon, 2008; Jang, 2014). Consequently, any iMEP recorded ipsilateral to stimulation is most likely mediated by non-CST pathways, predominantly the cortico-reticulospinal tract, and is widely regarded as a functional indicator of RST excitability (Ziemann et al., 1999; Fisher et al., 2012; Maitland & Baker, 2021; Atkinson et al., 2022).

Although the target muscles were different, both measures reflected outputs from the same (trained, left) hemisphere: cMEPs captured its contralateral corticospinal output (right biceps brachii), while iMEPs reflected its ipsilateral cortico-reticulospinal output (left biceps brachii). TMS over the left M1 thus probed both CST and cortico-reticulospinal pathways, consistent with previous evidence that stimulation of one hemisphere engages contralateral CST projections as well as corticoreticular/cortico-reticulopropriospinal projections descending via bilateral RST pathways (Ziemann et al., 1999; Fisher et al., 2012; Bradnam et al., 2013).

Furthermore, recording both measures from the same biceps brachi is not feasible. To obtain iMEPs from the trained (right) biceps brachii, stimulation would need to target the right (untrained) hemisphere, which was not the hemisphere undergoing training-related plasticity. This would confound interpretation, as cMEPs would still be elicited from the trained hemisphere while iMEPs would reflect the untrained hemisphere. Therefore, recording iMEPs from the ipsilateral (left) limb while stimulating the trained (left) hemisphere provides the only viable and widely accepted approach to assess cortico-reticulospinal excitability in parallel with corticospinal excitability (Ziemann et al., 1999; Alagona et al., 2001; Maitland & Baker, 2021; Altermatt et al., 2023; Hu et al., 2024).

Comment 4: Additionally, related to the question above, if strength training was only performed unilaterally to the dominant side, this would imply an increased reticulospinal drive to the non-dominant side when training the dominant side. This would be a different finding, correct? Therefore, I assume it must be bilateral.

Response: We thank the reviewer for raising this important point. To clarify, all strength training was performed unilaterally with the dominant (right) arm. Our assessment of iMEPs from the non-dominant (left) arm does not reflect bilateral training, but rather the fact that the reticulospinal tract projects bilaterally. TMS over the trained (left) hemisphere activates contralateral corticospinal projections to the trained arm, as well as corticoreticular projections that descend via the bilateral RST, influencing ipsilateral motor output (Ziemann et al., 1999; Fisher et al., 2012; Bradnam et al., 2013).

Thus, recording iMEPs from the left arm provided a valid index of cortico-reticulospinal excitability in the trained hemisphere, even though the muscle itself was not trained. Importantly, our interpretation does not suggest strength transfer to the untrained arm, but rather neural adaptations in the trained hemisphere's descending pathways. This design allowed us to probe both contralateral CST (via cMEPs) and ipsilateral cortico-reticulospinal (via iMEPs) outputs of the same hemisphere following unilateral training.

Comment 5 L447 I appreciate the authors addressing the problem of StartReact test-retest reproducibility. The pilot study, however, needs further clarification.

Response:

We thank the reviewer for these constructive queries regarding the pilot study and provide clarification below in response to each specific comment.

Comment 5.1 Were the 8 healthy adults in the pilot study additional participants or part of the control group? If part of the control group, why weren't all 10 included?

Response:

The eight healthy adults who took part in the pilot study were an independent sample, separate from the control group in the main trial. None of these participants were included in the intervention study. The purpose of the pilot was to check the reliability of the StartReact measure prior to the main study rather than to contribute to the main study. The Methods section has now been revised to clarify this point, which reads: “...*To examine test-retest reliability prior to commencing the main study, a pilot assessment was conducted in eight healthy adults at two time points one week apart. These participants did not take part in the main trial.*” (Page 20, Lines 491-493)

Comment 5.2) Where is further information on the data? Why does this not appear anywhere else in the manuscript?

Response:

The pilot study was conducted at two time points one week apart, with the sole aim of confirming measurement reliability prior to commencing the main study. Because reproducibility was subsequently assessed again within the actual study across four time points (Baseline, Week 1, Week 2, Week 3) and is now reported in detail in the Results, we consider these data to provide a more comprehensive evaluation of reliability than the pilot study involving only two sessions. For this reason, we have revised the Methods to explicitly describe the pilot procedures and added the detailed test-retest reproducibility in the revised manuscript *Page 27 and 28 line 644-650, and Table 2.*

Comment 5.3) The ICC shows only moderate reproducibility for VART and a good reproducibility for VSRT. Therefore, the StartReact effect reproducibility might be moderate. Why is this not discussed?

Response: We thank the reviewer for raising this point. We acknowledge that the moderate reproducibility observed for VART in the pilot was not explicitly discussed in the original manuscript. However, this was in fact a key motivation for conducting test-retest reproducibility again in the main study. While the pilot study showed good reliability for

VSRT but only moderate reliability for VART, we therefore repeated reproducibility testing within the actual study across four time points (Baseline, Week 1, Week 2, Week 3). These additional data provide a more comprehensive evaluation of StartReact reliability and strengthen confidence in the measures used. Accordingly, the Methods section has been revised to address this point (**Page 21 line 495-498**), which now reads: “...Given the moderate reproducibility observed for VART, reproducibility of StartReact measurements was subsequently reassessed within the main study across four time points (Baseline, Week 1, Week 2, Week 3) to provide a more comprehensive evaluation of reliability.”

Comment 6 L822 The greatest impact, in my opinion, of this manuscript is the potential of translating this into rehabilitative or training methods for especially neurologically impaired participants, e.g., Spinal cord or stroke. It would be great to further discuss the authors thoughts on the feasibility of such a training program in a target population.

Response: We thank the reviewer for this insightful comment and agree that the translational potential of our findings is an important aspect of the manuscript. In the revised Discussion, we now expand on the feasibility of applying SP-RT and MP-RT in clinical populations such as stroke and spinal cord injury survivors, before also considering ageing and healthy adults. This section highlights the simplicity, adaptability, and accessibility of the training protocols, as well as the need for supervision and safety considerations in clinical contexts. The revised text (**Page 37 and 38, Lines 877-888**) now reads: “*This distinction is particularly relevant in clinical populations such as stroke or spinal cord injury, where corticospinal integrity is often compromised but reticulospinal pathways remain at least partially preserved and capable of supporting recovery. In such cases, SP-RT may offer a practical training option, given its reliance on simple free-weight exercises that can be adapted to individual capacity, although careful supervision, load adjustment, and fatigue management would be required. Beyond rehabilitation, these findings also have relevance for healthy and ageing adults: in age-related decline, where corticospinal output diminishes, SP-RT may help preserve strength by engaging reticulospinal pathways, while MP-RT may be preferable for enhancing voluntary cortical drive and fine or precise motor control. Both approaches are accessible, low-cost, and based on familiar RT practices, supporting their feasibility across diverse groups.*”

Comment 7: Figure 6 Reaction Time for StartReact trials appear very short (VSRT between 60 and 35 ms, and VART between 60 and 80 ms). Castellotte and Kofler 2018 show reaction times of around 90 ms of the biceps brachii for startling stimuli. Sangari and Perez 2020 even showed VART of around 151.1{plus minus} 22.1 ms and VSRT 127 {plus minus} 21.7 ms

for elbow flexors of healthy controls. Similarly, Eilfort et al. 2025 presented startling reaction times around 85 ms and non-startling reaction times at around 120 ms. The startle reflex itself is known to be present at around 60 ms in the sternocleidomastoid. Therefore, the reaction times reported here do not quite seem physiologically feasible. Can the authors explain?

Response:

We sincerely thank the reviewer for this highly valuable observation, which prompted us to re-examine our data processing in detail.

On re-examining our data, and following consultation with the system manufacturer (ADInstruments), we identified that the event marker in the EMG channel, which served as the trigger onset, was systematically registered later than the actual onset of the visual, visual non-startling or visual non-startling stimulus. This registration delay (50 ms) affected only the timing of the marker and not the stimulus delivery or the physiological recordings themselves. As a result, the calculated reaction times appeared artificially short in the original version.

We have now corrected for this systematic delay in all analyses by reanalysing the StartReact data. The updated latency values fall within physiologically expected ranges for elbow flexors, consistent with prior literature (Castellote & Kofler, 2018; Sangari & Perez, 2020; Eilfort et al., 2025). Importantly, because the offset was uniform across all conditions, the StartReact effect (difference between VART and VSRT), our primary outcome, remained unchanged. Thus, although absolute values of VART and VSRT have been updated, the overall findings and conclusions of the study are unaffected.

We greatly appreciate the reviewer's comment, which enabled us to identify and resolve this issue and thereby strengthen the accuracy and transparency of our work. Accordingly, the respective result section, Figure 6 and the related values in Table 2 (rows 10 and 11) have been updated to reflect the corrected latencies and statistical analysis result.

Comment 8: General Would the authors expect a similar result and pattern in muscles with a lower inertial reticulospinal drive or even a strong corticospinal drive, such as the tibialis anterior, the first dorsal interosseus, or even the triceps brachii? Generally, the authors appear to generalize their results, while they should emphasize that their results are limited to the biceps brachii.

Response:

We thank the reviewer for this important observation. We agree that our findings are specific to the biceps brachii and that caution should be taken when generalising to other muscles. The reticulospinal tract is known to have a stronger influence on proximal flexor muscles

such as the biceps brachii, whereas distal hand muscles (e.g., FDI) and some extensors (e.g., triceps, tibialis anterior) are more strongly controlled by corticospinal pathways (Baker, 2011; Mooney et al., 2023; Eilfort et al., 2023). Therefore, the adaptations observed here may not necessarily be replicated in muscles with weaker reticulospinal drive or stronger corticospinal dominance.

We have revised the Discussion to make this limitation explicit and to emphasise that our interpretations apply primarily to the biceps brachii. We also note that future work should investigate whether metronome-paced or self-paced resistance training induces similar or different neural adaptations in muscles with differing CST vs RST contributions, such as distal hand or lower-limb muscles. We have now added these points to the Limitations section (**Page 43, Lines 1007–1011**), which now reads: “..... *all measurements were obtained from the biceps brachii, a proximal flexor with relatively strong reticulospinal contributions. Since corticospinal and reticulospinal influence varies across muscle groups, these findings should not be assumed to generalise to distal or extensor muscles. Future studies are required to test whether similar adaptations occur in muscles with different descending control.*”

Comment 9: General The manuscript's introduction and discussion show several redundancies. The length of the manuscript could be more concise and therefore improve the readability.

Response:

This constructive feedback from the reviewer has been very helpful. We agree that some parts of the Introduction and Discussion contained redundancies and could be streamlined for clarity. Accordingly, the Introduction has been reduced from four pages (double-spaced) to approximately two and a half pages by removing redundancies and focusing more tightly on the rationale and objectives. The Discussion has also been shortened from eight to five pages. The revised version had no redundancy, is more concise, while preserving the key contextual and interpretative points.

Moderate comments:

Comment 10: L152-L155 The reticulospinal system and its role and importance in motor control is introduced already in L134, and 138-150. The additional paragraph is redundant and appears to stop the flow of the reading.

Response:

We thank the reviewer for this observation. As part of the full revision of the Introduction, we have streamlined the content to remove redundancies and improve flow. Specifically, the

repeated paragraph (L152–L155 in the original submission) has been removed, and the role of the reticulospinal system is now introduced once in a concise and integrated manner. We believe this revision resolves the redundancy and improves readability.

Comment 11: L335 The authors describe verbal encouragement to ensure adherence to tempo and effort for the MP group. However, it would be interesting to investigate adherence to the protocol. Did the participants adhere to the timing? Did they fulfil the movement to the same extent as the SP group, or vice versa? Was movement kinematics, timing measured, and how many trials met the criteria of MP-RT out of the total?

Response:

We appreciate this thoughtful comment. Adherence to tempo and execution was carefully addressed in our protocol. Importantly, for the MP-RT group, a detailed and effortful familiarisation phase was provided until participants could reliably synchronise their contractions with the metronome beat. During the testing sessions, adherence was further ensured through close supervision: if any trial was performed incorrectly, participants were immediately instructed to correct their movement and repeat the repetition. These procedures are now explicitly described in the revised manuscript (**Page 10, lines 223-229**, and **Page 15, Lines 348-350, 359-361, Page 16, Lines 366-367**).

In both groups, training execution was monitored session-by-session. For the MP-RT group, repetitions were synchronised with the metronome (3 s concentric, 4 s eccentric) and accompanied by verbal cues to maintain effort. For the SP-RT group, repetitions were performed at each participant's natural pace, but with equivalent supervision to ensure full range of motion and correct technique. Parallely, time-under-tension (TUT) and total volume load was calculated for each set across all participants, providing an additional layer of monitoring and confirmation that the intended contrast between groups was reliably maintained. These methodological details are now explicitly reported in the revised Methods section (**Page 15, Lines 354-361, Page 16, lines 362-367, 370-373**) and the TUT was incorporated in the revised table, Table 1, alongside the total volume load.

While we did not employ detailed kinematic analyses to quantify execution, we acknowledge that such objective measures would have provided further information. This limitation has been incorporated in the revised manuscript (**Page 42, Lines 997-1003**), where we state that future studies may benefit from direct kinetic and kinematic assessments to further validate adherence and execution fidelity.

Comment 12: Was the movement execution (e.g., range, speed, technique) consistent across all groups?

Response: Movement execution was consistent across groups in terms of range of motion and technique. All participants performed full elbow flexion and extension with strict form under close supervision, and any compensatory movements (e.g., trunk swinging, shoulder elevation) were corrected immediately. Thus, the exercise execution was identical across groups apart from the intended manipulation of pacing. For the MP-RT group, repetitions were performed at a slower tempo (3 s concentric, 4 s eccentric) to allow accurate synchronisation with the metronome. In contrast, the SP-RT group performed at their self-selected tempo, which, as the term “self-paced” implies and as reported in previous studies, is typically faster. This difference in pacing naturally resulted in longer time-under-tension in MP-RT and higher total volume load in SP-RT, while all other aspects of movement execution remained comparable. In the revised manuscript (**Page 14, lines 322-331, and Page 15 lines 341-350, and 354-361, Page 16, Lines 362-367**), the training protocol is described in detail to clarify that the groups differed only in pacing and its associated characteristics, while range of motion, technique, and overall supervision were consistent across groups.

Comment 13: L361 What exact number of stimuli were used for the TMS assessment at the different stimulator intensities? Why was there 5-10 reported and not an exact number? Did this vary?

Response:

We appreciate this comment. We confirm that 10 stimuli were always delivered at each stimulator intensity. In the original version we reported ‘5-10’ to reflect potential variation in usable trials; however, upon checking, the minimum number of valid responses retained after artefact rejection was 8, not 5. To avoid ambiguity, the revised manuscript now states that 10 stimuli were delivered (**Page 17, line 398**)

Comment 14: L371 How were the stimulation intensities (130% AMT, 150% AMT, and 170% AMT)? The Authors cite Carson et al. 2013, however, I wonder why an entire recruitment curve with 5% stimulator output intervals is not chosen. Did the authors ensure a plateau of the MEP amplitude? Was MEP max determined?

Response:

We thank the reviewer for rising this point. We wish to clarify that we cited Carson et al. (2013) to emphasise that the area under the recruitment curve (AURC) is a reliable and sensitive index of corticospinal excitability, rather than to imply we adopted their full 5% incremental recruitment curve protocol. While a 5% stepwise curve is suitable at rest, in our study each stimulus was delivered while participants maintained a 10% MVC contraction.

Delivering dozens of stimuli across 5% increments under active contraction would have introduced unnecessary fatigue and potentially confounded the results, particularly given that our protocol also required participants to complete multiple additional neurophysiological assessments (SICI, ICF, iMEPs, and StartReact). For this reason, we adopted an efficient approach that has been adopted in prior studies by sampling three suprathreshold intensities (130%, 150%, and 170% AMT) (e.g. Mason J et al., 2020, Woodhead A, et al., 2024, Siddique U., et al 2025). This band spans the steep slope through early plateau of the input-output function, where corticospinal excitability changes are most physiologically informative (Groppa S, et al., 2012). Importantly, this same procedure was applied across all groups, so group comparisons are not affected.

Although we did not formally determine a plateau of the MEP amplitude or MEPmax, the highest intensity tested (170% AMT) can be considered close to the plateau of the input-output curve of biceps brachii. According to IFCN guidelines (Groppa et al., 2012), when the target muscle is pre-activated, the transition to the plateau occurs at ~ 170% AMT. Thus, our selection of 170% AMT ensured coverage of the suprathreshold range approaching the plateau, while avoiding the additional burden of constructing a full incremental recruitment curve.

Thus, our choice reflects a balance between methodological rigour and experimental feasibility, ensuring accurate measurement of corticospinal excitability without inducing participant fatigue or compromising the other assessments included in the study. To avoid confusion, we have replaced the citation of Carson et al. (2013) with more appropriate references that used the exact procedure described (*Page 17, line 408-409*).

Comment 15: L393 Why was a different coil used (Figure 8) for iMEP/cMEP analysis compared to the other TMS protocols (circular)?

Response:

We appreciate the reviewer's observation regarding the use of different coils for the iMEP/cMEP protocol versus the other TMS measures. For SICI, ICF, cSP, and CSE we employed a circular coil. This choice was based on several practical and methodological advantages: circular coils are easy to handle, less time-consuming to position, and less sensitive to small changes in placement. Because their stimulation field is deep and broader, locating a reliable "hotspot" for proximal upper-limb muscles such as the biceps brachii is easier and produces robust and reproducible contralateral MEPs. It is also well established and as reliable as figure-of-eight coils for assessment of cortical excitability (Badawy et al., 2011),

By contrast, for iMEP/cMEP recordings we used a figure-of-eight coil. The rationale was twofold. First, when recording iMEPs from an ipsilateral muscle (e.g., left biceps brachii during left-hemisphere stimulation), it is essential to minimise the risk of inadvertent cross-stimulation of the contralateral hemisphere, which would otherwise generate a cMEP. The figure-of-eight coil provides more focal stimulation than a circular coil, reducing current spread across the midline and thereby ensuring that the elicited responses more reliably reflect true ipsilateral projections. Second, iMEP protocols require high-intensity stimulation, and figure-of-eight coils are known to be better tolerated. In particular, Ørskov et al. (2021) reported that participants experienced lower discomfort levels and improved tolerability when using figure-of-eight coils compared to circular coils.

Overall, our approach is consistent with existing literature (e.g., Altermatt et al., 2023; Alagona et al., 2001; Maitland & Baker, 2021; Hu et al., 2024), and balances methodological rigour with participant comfort. We have clarified this rationale in the revised manuscript (**Page 18, lines 431-434**), which now reads:

“For iMEP and contralateral motor-evoked potential (cMEP) recordings, the figure-of-eight coil was preferred to provide greater focality and minimise cross-hemispheric spread, ensuring that responses reflected true ipsilateral rather than contralateral activation.”

Comment 16: L398 Why was the iMEP (or ICAR) protocol optimized for the cMEP, especially why was hotspotting performed for the cMEP? I understand that the iMEP was normalized by cMEP and understand the reasoning. However, it still appears that iMEP is the main outcome in this part of the analysis.

Response:

We thank the reviewer for this insightful comment. It is correct that iMEP/ICAR were the primary outcomes in this analysis. However, hotspotting was performed for the contralateral MEP because the cMEP provides a robust and reproducible cortical representation of the target muscle, whereas iMEPs are typically small, variable, and not suitable for reliable hotspot localisation. This approach aligns with previous iMEP studies, where the contralateral MEP hotspot is typically assumed to also represent the optimal site for eliciting iMEP (e.g. Chen et al., 2003, McCambridge et al., 2016, Hu et al., 2024). Moreover, because iMEPs were normalised to cMEPs, identifying the hotspot via the cMEP ensured that both contralateral and ipsilateral responses were elicited from the same cortical site, improving interpretability of the iMEP/ICAR measures.

Comment 17: L431 The authors report a sound intensity level of 115-120 dB for the startling stimulus. Why did this vary? A 5 dB difference is significant at such high intensities, given the logarithmic nature of the scale.

Response:

We thank the reviewer for raising this important point. We mistakenly reported the acceptable calibration range (“115–120 dB”) instead of the actual calibrated intensity. In practice, the startling stimulus was consistently set and delivered at 115 dB; the stated range was an error in wording rather than an indication of true variability in stimulus intensity. We have corrected this in the revised manuscript.

Comment 18 L605 Authors report a significant group*time interaction, however, with no significant main effect of group nor time. Additionally, an increase in strength for both the MP and SP groups is reported. Is there a significantly steeper increase in the MP group vs the SP group for 1RM?

Response: We thank the reviewer for this comment. To examine whether there was a significantly steeper increase in 1RM between the training groups, we performed post-hoc interaction contrasts. Though the percentage change showed steeper increase in the MP-RT group, there was no significant difference in the change from baseline between SP-RT and MP-RT at any time point ($p = 1.000$). However, compared with the Control group, the MP-RT group showed significantly greater increases in 1RM from Baseline to Week 2 (EMD = 0.19, SE = 0.05, $p = 0.002$) and Week 3 (EMD = 0.21, SE = 0.05, $p < 0.001$). Similarly, the SP-RT group demonstrated significantly greater increases than the Control group from Baseline to Week 2 (EMD = 0.16, SE = 0.05, $p = 0.016$) and Week 3 (EMD = 0.16, SE = 0.05, $p = 0.011$). These results clarify that both training groups improved relative to Control, but the magnitude of change did not differ between SP-RT and MP-RT. This clarification has been incorporated into the revised manuscript (*Page 28, Lines 662-668*)

We note that significant interactions can occur in the absence of main effects, as verall tests evaluate only the global null hypothesis and may mask specific group-time differences specially in studies involving three groups (Maxwell & Delaney, 2004; Ruxton & Beauchamp, 2008). Considering this, we therefore conducted post-hoc interaction contrasts for all outcome variables, not just 1RM. This revealed several important between-group effects that were not evident in the overall test outputs of LMM/GLMM and has strengthened the overall findings of the study. The Results section has been thoroughly revised to present these additional findings with precise reporting of test statistics and p-values.

Comment 19: L787-L788 Redundancy to the sentence before. The sentence points out the novelty of the study. However, "This study presents the first..." also points out novelty.

Response: We thank the reviewer for this observation. We have removed the redundant phrasing and retained a single clear statement ("it is the first investigation in humans to ...") (Page 37 line 859-860) to highlight novelty without repetition.

Comment 20: L801-L803 Redundant to L789 - L195.

Comment 21: L805 Redundant to L792.

Comment 22 General Overall, the discussion appears to repeat itself, constantly showing the increased strength, increased cSP, SICI, and ICF in MP-RT and increased StartReact effect and ICAR for the SP-RT. This is the main finding; however, it keeps being repeated and makes it hard to read.

Response to Comments 20–22:

The reviewer's points are appreciated and has been carefully addressed. In the revised manuscript, the Discussion has been thoroughly restructured and reduced from 8 pages (double-spaced) to 5 pages. Redundant statements have been removed, including those identified at L789–L803 and L792–L805, and repetitive descriptions of the main findings (i.e., MP-RT effects on cSP, SICI, and ICF, and SP-RT effects on StartReact and ICAR) have been consolidated. The revised version now presents each key finding once, followed by an integrated interpretation that avoids unnecessary repetition. These changes have streamlined the Discussion, improved readability, and ensured that the main findings are highlighted clearly without duplication.

Minor comments:

Comment 23: L67 Abbreviations used in the Abstract should be introduced there as well, e.g., SICI and cSP.

Response:

We thank the reviewer for this comment. All abbreviations have now been defined at first mention in the revised Abstract to ensure clarity for readers.

Comment 24 L111 "Despite well-documented neural adaptations to short-term RT," Reference is missing here.

Response:

We thank the reviewer for pointing this out. A reference has now been added to support the statement "*Despite well-documented neural adaptations to short-term RT*" (e.g., Sale, 1988; Carroll et al., 2001...).

Comment 25 L129 There is a spelling error in this sentence ("do not")

Answer:

The reviewer is correct, and this typographical error has been corrected in the revised Introduction.

Comment 26: L145 The reference Valls-Sole et al. 1995 ("Reaction time and acoustic startle in normal human subjects") might be considered as a more suitable reference compared to Baker and Perez 2017.

Response:

We appreciate the reviewer's suggestion. We have now replaced Baker & Perez (2017) with the more suitable reference Valls-Solé et al. (1995) in the revised manuscript (**Page 6, line 140**) to better support this statement.

Comment 27: L149 Indeed, the reticulospinal tract is known to project bilaterally. However, why is this important for the argumentation of iMEPs as measure for cortico-reticulospinal excitability? To my understanding, the ipsilateral projection is what defines this here

Response:

We thank the reviewer for this helpful clarification. We agree that it is the ipsilateral projection of the reticulospinal tract (RST) that primarily defines the rationale for using iMEPs as a marker of cortico-reticulospinal excitability. In our revised manuscript (**Page 6, line 141-143**), we have rephrased the sentence to emphasise this point: "*TMS-induced iMEPs are thought to reflect cortico-reticulospinal projections due to the RST's ipsilateral fibres, which convey cortical drive to the same-side limb*".

Comment 28 L161 Please clarify the sentence here.

Response:

We appreciate the reviewer's suggestion and have clarified the sentence accordingly. It now reads (**Page 7, Line 151-153**) "*...In non-human primates, RT enhanced RST- and MI-evoked responses but not corticospinal tract (CST), suggesting plasticity primarily within reticulospinal and intracortical MI circuits.*"

Comment 29 L227 A closing parenthesis is missing after "body mass."

Response:

The missing parenthesis after "*body mass*" has now been inserted.

Comment 30: L254 Please specify which electrodes were used.

Response:

The sentence has been revised to specify the electrodes used and now reads (**Page 11, Lines 264-265**): "*“Bipolar Ag-AgCl surface electrodes (Bipolar Ag-AgCl surface electrodes (Medi-Trace™, Graphic Controls LLC, Buffalo, NY, USA) were placed*"

Comment 31: L265 Please specify which EMG system (e.g., transmitter model) was used.

Response:

In line with the reviewer's request, we have specified the EMG system and transmitter model used. The revised text now reads (**Page 12, line 274-276**):

“sEMG signals were recorded using an Octal Bio Amp (ML138) connected to a PowerLab 4/26 data acquisition system, both from ADInstruments (Bella Vista, Australia). Signals were amplified ($\times 1000$), bandpass filtered (13–1000 Hz), and digitized in real-time at 20 kHz. Data acquisition and analysis were performed using LabChart software (ADInstruments, Bella Vista, Australia).

Comment 32: L285 The abbreviation ICC should be introduced at first use. Please revise accordingly.

Response:

In line with the reviewer's suggestion, we have introduced the abbreviation at first use (**Page 13, Line 297**).

Comment 32: L297 There is no need to reintroduce abbreviations once they have been defined previously.

Response:

The redundant reintroduction of the abbreviation has been removed. The sentence now reads: *“To control for this, MVF...”*. We have also checked the entire manuscript and avoided repeated reintroduction of abbreviations (e.g. *cSP, ICAR, RT, RST, sEMG, VnSS*)

Comment 34: L342 To my understanding, MEPs are recorded from the MEP rather than elicited. MEPs are elicited from the brain.

Response:

The sentence has been revised to more accurately describe MEP acquisition. It now reads: *“...Motor-evoked potentials (MEPs) were **recorded** from the biceps brachii of the dominant arm through TMS of the corresponding motor cortical representation.”*

Comment 35: L356 Do the scalp markings persist for three weeks?

Response:

To ensure coil placement consistency across sessions, the motor hotspot was initially marked on the scalp using a permanent marker and re-boldened at each subsequent session. The revised text now reads (**Page 17, lines 391-3933**): *“Once the motor hotspot was established, it was marked on the scalp with a permanent marker and re-boldened at each session to ensure consistent coil placement across the three-week intervention.*

Comment 36 L359 You introduce the abbreviation MSO for "maximum stimulator output," but do not use it again. Consider removing it if it's not used later.

Response:

Thank you – as per suggestion, we have removed it.

Comment 37: L414 Do the 10 TMS stimuli correspond to 10 iMEP stimuli, or is that the total number?

Response:

We appreciate the reviewer's request for clarification. The 10 stimulations refer 10 iMEP stimuli. The sentence has been revised to read: "*Each participant completed two sets of five repetitions, with a standardised two-minute rest interval between sets, yielding a total of 10 TMS pulses per session to elicit iMEPs and cMEPS.*"

Comment 38: L433 Of the 30 reported trials, were the conditions equally distributed (i.e., 10 per condition)?

Response: We thank the reviewer for this comment. The 30 trials were equally distributed across conditions (10 per condition). The sentence has been revised for clarity and now reads: "*Each session comprised 30 trials (10 per condition)*"

Comment 39 L448 As noted earlier (L285), ICC should be introduced at first use.

Response: We acknowledge this repetition and have revised accordingly.

Comment 40 L471 Why was the amplitude at 100% AMT not analysed?

Response:

We appreciate the reviewer's question. The amplitude at 100% AMT was not analysed because AMT is defined as the minimum stimulus intensity required to evoke MEPs >200 μ V in at least five out of ten trials. Thus, the value at 100% AMT is a predetermined threshold criterion rather than an independent physiological variable. Our analysis focused instead on the stimulus intensity required to reach AMT, which provides the relevant measure of cortical excitability. Analysing the amplitude at 100% AMT would not yield additional meaningful information beyond this predetermined definition.

Comment 41 L522 These abbreviations were already introduced earlier. Please avoid repeating them.

Comment 42 L524 These abbreviations were already introduced earlier. Please avoid repeating them.

Comment 43 L585 These abbreviations were already introduced earlier. Please avoid repeating them.

Response to Comments 41 (L522), 42 (L524), and 43 (L585):

We thank the reviewer for these observations. All redundant reintroductions of abbreviations have been removed. Abbreviations are now introduced only at first use and used consistently thereafter, and we have carefully checked the entire manuscript to ensure this standard is applied throughout.

Comment 44 L595 The abbreviation MVF has been used consistently for "maximum voluntary force." Why is it not used here?

Response: The term "maximum voluntary force" has now been replaced with its previously defined abbreviation (MVF) to ensure consistency throughout the manuscript.

Comment 45 L613 Please cite Figure 1A here or at any other relevant point in the text.

Response: A reference to Figure 2A has now been included at the relevant point, as suggested

Comment 46 L654 For the reader, there seems to be possible incongruencies between the values recorded here and Figure 3. But this might just appear this way.

Response:

We acknowledge the reviewer's observation regarding potential incongruencies between the values reported in the text and Figure 3. To resolve this, the text has been revised to present between group post-hoc interaction contrast results, with percentage changes now placed in brackets alongside the within-group post-hoc outcomes. In addition, Figure 3 has been updated to display individual values connected with lines, ensuring consistency between text and figure and improving clarity for the reader.

Comment 47: L684 There is a typo in Table 2 ("Table 02").

Response: The typographical error has been corrected; "Table 02" is now written as "Table 2" in the text.

Comment 48: L817 Maybe the authors can review this reference and find a more suitable reference than Fisher et al. 2021.

Response: The sentence containing the Fisher et al. (2021) reference has been removed to streamline the Discussion and avoid redundancy.

Comment 49: L910 Did the authors mean spinal interneurons instead of spinal motor neurons?

Response:

We have revised the sentence to avoid ambiguity. Now it reads "...By *bypassing cortical pathways, this mechanism more efficiently recruits alpha-motoneurons, contributing to the observed improvements in motor output*" (**Page 40, lines 944-946**).

Comment 50 L942 Here you can add Neumann et al. 2025 ("Cortical involvement in the initiation of movements cued by moderate, but not loud acoustic stimuli: Evidence for subcortical mediation of the StartReact effect") as additional evidence for subcortical responsibility of the StartReact effect in humans in addition to Tapia et al. 2022 from primates.

Response:

We sincerely thank the reviewer for highlighting this important reference. The newly added citation, Neumann et al. (2025), provides valuable human evidence supporting subcortical mediation of the StartReact effect, complementing the primate findings of Tapia et al. (2022). It now reads "... with minimal corticospinal involvement and consistent evidence for subcortical mediation of the StartReact effect in non-human primates and humans (Tapia et al., 2022; Neumann et al., 2025) (**Page 41, lines 1067-1069**).

Comment 51: Figure 1 Please verify that all abbreviations used in the figure are also explained in the legend, and vice versa. Inconsistencies exist, e.g., VA, RT, ICAR.

Response:

The abbreviations have now been fully harmonised between Figure 1 and its legend. Specifically, ICAR and cMEP have been added to the figure, while iMEP and cMEP are explained in the legend. VA has been removed from the figure, and CST and RT have been removed from the legend. These revisions ensure consistency between the figure and legend.

Comment 53: Figure 7 Check for consistency in spelling (e.g., "cMEPs") and ensure all units are correctly presented.

Response: Figure 7 has been carefully revised to ensure consistency in spelling and accuracy in the presentation of all units.

Comment 54: Figure 8 Use the same y-axis scale in both panels A and B for easier comparison.

Response: In line with the reviewer's suggestion, the y-axis scale has been standardised across panels A and B in Figure 8 to facilitate direct comparison.

Comment 55: Table 1 What unit is used for training duration or volume? Please verify abbreviation consistency, spelling, and bracket formatting.

Response: The unit for training duration/volume(kg) has now been added in Table 1. In addition, abbreviation use, spelling, and bracket formatting have been carefully checked and corrected to ensure consistency.

Comment 56: General For improved readability, I recommend a reduction of the number of abbreviations used.

Response: To improve readability, we have reduced the number of abbreviations by retaining only those essential for clarity (e.g., cSP, ICAR, AURC, RFD). Less critical abbreviations (e.g., LED, fMRI, MSO, PMRF) have been removed from the main text and replaced with their full terms. We believe this revision enhances flow and accessibility for the reader.

General feedback: All in all, this was a very interesting study and manuscript to read.

Response: We are grateful for the reviewer's positive feedback and appreciation of our work. We are pleased that the study and manuscript were found to be interesting and enjoyable to read.

References Cited in Author Response to Referee 3

- Alagona, G., Delvaux, V., Gérard, P., De Pasqua, V., Pennisi, G., Delwaide, P.J., Nicoletti, F. & Maertens de Noordhout, A. (2001). Ipsilateral motor responses to focal transcranial magnetic stimulation in healthy subjects and acute-stroke patients. *Stroke*, 32, 1304–1309.
- Altermatt, M., Jordan, H., Ho, K. & Byblow, W.D. (2023). Modulation of ipsilateral motor evoked potentials during bimanual coordination tasks. *Frontiers in Human Neuroscience*, 17, 1219112.
- Anzak, A., Tan, H., Pogosyan, A. & Brown, P. (2011). Doing better than your best: loud auditory stimulation yields improvements in maximal voluntary force. *Experimental Brain Research*, 208, 237–243.
- Atkinson, E., Škarabot, J., Ansdell, P., Goodall, S., Howatson, G. & Thomas, K. (2022). Does the reticulospinal tract mediate adaptation to resistance training in humans? *Journal of Applied Physiology* (1985), 133, 689–696.
- Badawy, R.A., Tarletti, R., Mula, M., Varrasi, C. & Cantello, R. (2011). The routine circular coil is reliable in paired-TMS studies. *Clinical Neurophysiology*, 122(4), 784–788. doi: 10.1016/j.clinph.2010.10.027.
- Baker, S.N. (2011). The primate reticulospinal tract, hand function and functional recovery. *Journal of Physiology*, 589(Pt 23), 5603–5612. doi: 10.1113/jphysiol.2011.215160.
- Bradnam, L.V., Stinear, C.M. & Byblow, W.D. (2013). Ipsilateral motor pathways after stroke: implications for non-invasive brain stimulation. *Frontiers in Human Neuroscience*, 7, 184. doi: 10.3389/fnhum.2013.00184.
- Chen, R., Yung, D. & Li, J.Y. (2003). Organization of ipsilateral excitatory and inhibitory pathways in the human motor cortex. *Journal of Neurophysiology*, 89(3), 1256–1264. doi: 10.1152/jn.00950.2002.
- Eilfort, A.M., Neumann, L.C. & Filli, L. (2025). Mapping reticulospinal drive across various muscles of the upper and lower extremities. *Experimental Physiology*. doi: 10.1113/EP092763.
- Fisher, K.M., Zaaami, B. & Baker, S.N. (2012). Reticular formation responses to magnetic brain stimulation of primary motor cortex. *Journal of Physiology*, 590(16), 4045–4060. doi: 10.1113/jphysiol.2011.226209.
- Groppa, S., Oliviero, A., Eisen, A., Quartarone, A., Cohen, L.G., Mall, V., Kaelin-Lang, A., Mima, T., Rossi, S., Thieckbroom, G.W., Rossini, P.M., Ziemann, U., Valls-Solé, J. & Siebner, H.R. (2012). A practical guide to diagnostic transcranial magnetic stimulation: report of an IFCN committee. *Clinical Neurophysiology*, 123(5), 858–882. doi: 10.1016/j.clinph.2012.01.010.
- Hu, N., Tanel, M., Baker, S.N., Kidgell, D.J. & Walker, S. (2024). Inducing ipsilateral motor-evoked potentials in the biceps brachii muscle in healthy humans. *European Journal of Neuroscience*, 60(9), 6291–6299. doi: 10.1111/ejn.16548.
- Jang, S.H. (2014). The corticospinal tract from the viewpoint of brain rehabilitation. *Journal of Rehabilitation Medicine*, 46(3), 193–199. doi: 10.2340/16501977-1782.
- Lemon, R.N. (2008). Descending pathways in motor control. *Annual Review of Neuroscience*, 31, 195–218. doi: 10.1146/annurev.neuro.31.060407.125547.
- Maitland, S. & Baker, S.N. (2021). Ipsilateral motor evoked potentials as a measure of the reticulospinal tract in age-related strength changes. *Frontiers in Aging Neuroscience*, 13, 612352.
- Mason, J., Frazer, A.K., Avela, J., Pearce, A.J., Howatson, G. & Kidgell, D.J. (2020). Tracking the corticospinal responses to strength training. *European Journal of Applied Physiology*, 120(4), 783–798. doi: 10.1007/s00421-020-04316-6.
- McCambridge, A.B., Stinear, J.W. & Byblow, W.D. (2016). Are ipsilateral motor evoked potentials subject to intracortical inhibition? *Journal of Neurophysiology*, 115(3), 1735–1739. doi: 10.1152/jn.01139.2015.
- Mooney, R.A., Bastian, A.J. & Celnik, P.A. (2023). Mapping subcortical motor pathways in humans with startle-conditioned TMS. *Brain Stimulation*, 16(5), 1232–1239.
- Ørskov, S., Bostock, H., Howells, J., Pugdahl, K., Fuglsang-Frederiksen, A., Nielsen, C.S., Cengiz, B., Samusyte, G., Koltzenburg, M. & Tankisi, H. (2021). Comparison of figure-of-8 and circular coils for threshold tracking transcranial magnetic stimulation measurements. *Neurophysiologie Clinique*, 51(2), 153–160. doi: 10.1016/j.neucli.2021.01.001.
- Ruxton, G.D. & Beauchamp, G. (2008). Time for some a priori thinking about post hoc testing. *Behavioral Ecology*, 19(3), 690–693.
- Siddique, U., Frazer, A.K., Tallent, J., Hayman, O., Andrushko, J., Ahtiainen, J.P., Avela, J., Akalu, Y., Rostami, M., Uribe, S., Walker, S. & Kidgell, D.J. (2025). Acute corticospinal and reticulospinal responses to strength training in ageing. *Neurobiology of Aging*, 153, 49–62. doi: 10.1016/j.neurobiolaging.2025.06.007.
- Woodhead, A., Rainer, C., Hill, J., Murphy, C.P., North, J.S., Kidgell, D. & Tallent, J. (2024). Corticospinal and spinal responses following a single session of lower limb motor skill and resistance training. *European Journal of Applied Physiology*, 124(8), 2401–2416. doi: 10.1007/s00421-024-05464-9.
- Ziemann, U., Ishii, K., Borgheresi, A., Yaseen, Z., Battaglia, F., Hallett, M., Cincotta, M. & Wassermann, E.M. (1999). Dissociation of the pathways mediating ipsilateral and contralateral motor-evoked potentials in human hand and arm muscles. *Journal of Physiology*, 518(3), 895–906.
- Maxwell, S.E. & Delaney, H.D. (2004). *Designing Experiments and Analyzing Data: A Model Comparison Perspective* (2nd ed.). Mahwah, NJ: Lawrence Erlbaum Associates.

Dawson J. Kidgell, PhD

Associate Professor, Director, Monash University Exercise Neuroplasticity Research Unit
Monash University, Frankston, Australia]
dawson.kidgell@monash.edu

Dear Professor Carson,

On behalf of my co-authors, I am pleased to submit our revised manuscript entitled
**“Resistance Training Tempo Selectively Modulates Corticospinal and Reticulospinal
Excitability in Humans”** for consideration for publication in *The Journal of Physiology*.

We are grateful to the reviewers for their rigorous and insightful evaluation of our manuscript. In revising the work, we have carefully addressed all comments and provided detailed, evidence-based responses to ensure that each point has been thoroughly considered.

We confirm that this manuscript has not been published elsewhere and is not under consideration by any other journal. All authors have approved the final manuscript and consent to its submission. A list of potential reviewers with relevant expertise is included in our submission.

We appreciate your consideration and look forward to the opportunity to contribute to *The Journal of Physiology*.

Warm regards,

A/Prof Dawson J. Kidgell

Monash Exercise Neuroplasticity Research Unit
School of Primary and Allied Health Care
Faculty of Medicine, Nursing and Health Sciences
Monash University, Australia
Email: dawson.kidgell@monash.edu

Re: JP-RP-2025-289141R1 "**Resistance Training Tempo Selectively Modulates Corticospinal and Reticulospinal Excitability in Humans**" by Yonas Akalu, Jamie Tallent, Ashlyn K Frazer, Ummatul Siddique, Mohamad Rostami, Glyn Howatson, Simon Walker, and Dawson J Kidgell

Dear Dr Kidgell,

Thank you for submitting your manuscript to The Journal of Physiology. It has been assessed by a Reviewing Editor and by 3 expert referees and we are pleased to tell you that it is potentially acceptable for publication following satisfactory major revision.

Please address all the points raised and incorporate all requested revisions or explain in your Response to Referees why a change has not been made. We hope you will find the comments helpful and that you will be able to return your revised manuscript within 2 months. If your article is NOT for a Special Issue, you may have 9 months to revise. If you require an extension, please contact journal staff: jp@physoc.org. Please note that this letter does not constitute a guarantee for acceptance of your revised manuscript.

REVISION CHECKLIST:

We look forward to receiving your revised submission.

Yours sincerely,

Richard Carson
Senior Editor
The Journal of Physiology

REQUIRED ITEMS

1) - Papers must comply with the Statistics Policy: https://jp.msubmit.net/cgi-bin/main.plex?form_type=display_requirements#statistics.

In summary:

- If $n \leq 30$, all data points must be plotted in the figure in a way that reveals their range and distribution. A bar graph with data points overlaid, a box and whisker plot or a violin plot (preferably with data points included) are acceptable formats.
- If $n > 30$, then the entire raw dataset must be made available either as supporting information, or hosted on a not-for-profit repository, e.g. FigShare, with access details provided in the manuscript.
- 'n' clearly defined (e.g. x cells from y slices in z animals) in the Methods. Authors should be mindful of pseudoreplication.
- All relevant 'n' values must be clearly stated in the main text, figures and tables.
- The most appropriate summary statistic (e.g. mean or median and standard deviation) must be used. Standard Error of the Mean (SEM) alone is not permitted.
- Exact p values must be stated. Authors must not use 'greater than' or 'less than'. Exact p values must be stated to three significant figures even when 'no statistical significance' is claimed.

EDITOR COMMENTS

Reviewing Editor:

While reviewers 1 and 3 were largely satisfied with the revisions that have been made in the new version of the manuscript, they have made some further suggestions for substantial revisions. Reviewer 2 raises more serious concerns that were not alleviated by the current revisions, and are potentially more difficult to rectify (eg. too low sample size).

Senior Editor:

The point is made by one of the referees that a very large number of statistical tests were performed.

The authors might consider whether additional steps might be taken to compensate for the likelihood of "experiment-wise inflation of alpha: (i.e., separately and in addition to any "conventional" measures taken to account for the potential impact of multiple comparisons).

With respect to a further point made by Referee #2, it can be appreciated that challenges may arise when entering effect

size estimates obtained in one context into a power analysis that pertains to another (perhaps more complex) statistical design. It is however critical that the steps taken in this regard are presented with a clarity sufficient to convince the reader that the power estimates are sound.

In addition to any steps taken in relation to the points noted above (i.e., along with others highlighted by referees), but relatedly, I would encourage the authors to provide (95%) confidence intervals for the effect size estimates that are reported. See, for example, PMID: 37815959.

REFeree COMMENTS

Referee #1:

Overall, the authors have adequately addressed the issues in the manuscript by adding correlation analyses, including figures with individual data, providing between-group comparisons, and incorporating other relevant clarifications raised by myself and the other reviewers. I believe no further review is required, and the authors should be commended for their work.

Referee #2:

JP-RP-2025-28914R1

The conceptualisation of the paper has clearly improved and the authors have done a nice job making this much more credible. Unfortunately with some improvement in clarity, the major issues around the analysis and interpretation have also come into sharper focus:

1. On reflection there is little doubt that this study is substantially underpowered for the design conducted: 3 groups, 4 time points and the long list of outcome measures included. The number of statistical tests conducted must be very high (into the hundreds) and this increases the chances of type I errors to an unacceptable level. I remain sceptical about the power calculation, which in any case does not account for the large number of outcome variables and the plethora of statistical tests.

Regarding the power calculation, in their response the authors state "This conversion allowed us to adapt the single-group effect size to our three-group repeated-measures design." which is really unclear. The manuscript also now states "The effect size value, Cohen's d was converted to Cohen's f ($f = 0.275$) to estimate the effect size appropriate for our three-group design." It seems unlikely that this can be done in a meaningful way i.e. the jump from a single group effect size to how multiple groups might compare does not seem credible, and thus the power calculation is not convincing.

Moreover, in my view groups of $n=9$ are insufficient for such an intricate design and analysis with the measures included in this experiment, in order to generate convincing results.

2. For comparing group responses over time the key statistical test is the group x time interaction contrasts, and especially between the two RT groups. Therefore this finding should be stated explicitly for each outcome variable. By omission it appears that there are no interaction contrasts for MP-RT vs SP-RT for AMT, CSE, cSP, SICl or ICF, which directly queries some of the sweeping statements in the Discussion. Further, differences between groups at any given time point do not provide evidence for differential changes over time (even subtly different starting values can confound such simplistic analysis) and would be better de-emphasised/substantially reduced or removed.

3. The results are extremely lengthy (nearly 10 pages of text alone) as they lack focus, with too many outcome variables, too many statistical comparisons reported, and unnecessary content. For example: are the comparisons (at any or all) specific time points needed? Similarly the within-group regressions appear superfluous and unrelated to the aims of the study.

4. In general throughout the Discussion the different findings (within-group, differences at time points and interaction effects) are conflated, without a nuanced interpretation of what's been found. For example, in the conclusion "the underlying neural mechanisms appeared to differ between the two modalities. MP-RT was associated with enhanced CSE, increased ICF, and reduced inhibition (SICl, cSP)". This is weak as within-group changes and/or associations are not evidence of differences between the two RT modalities. Differences between the two training modalities, requires evidence of group by time interactions. The Discussion needs to be substantially revised with a grounded reflection of what has been found.

5. There are similar issues with the abstract, presenting within-group changes, and then drawing between group conclusions.

Specific comments:

Abstract L67-71 The crucial thing missing from these sentences is the comparison of the two training groups i.e. were the

changes in one group different to another? Stating the within-group changes for each group is limited and misses the point of having and comparing different groups in the study.

Reading on, the conclusions "MP-RT preferentially engages cortical and corticospinal pathways" which relies on a between group difference in response, however such a between group difference is not presented in the preceding results.

L90-91 "These findings demonstrate, for the first time in humans, that RT pacing can differentially modulate corticospinal and reticulospinal excitability" do the results support this? What specific measures show between group interaction (group x time) contrasts.

L105 "In the initial weeks of RT (<4 weeks), strength improvements are predominantly driven by neural adaptations" This is dogma, and therefore suggest it is toned down (e.g. 'considered to be predominantly'), as there is no data that neural adaptations are causal (see: PMID: 33383071).

L110 The mechanisms listed should certainly include changes in inter-muscular co-ordination such as changes in antagonist and stabiliser activation, rather than just focus on agonist muscle neural changes.

L162 "that training modality determines the neural locus of adaptation." Excessively categorical for an Introduction, if this is known/definite there may not be a need for the current experiment.

Please clarify if contractions were performed continuously i.e. without a pause or unloading between repetitions as is quite often the case.

L185 "The effect size value, Cohen's d was converted to Cohen's f ($f = 0.275$) to estimate the effect size appropriate for our three-group design." This needs explanation to be credible, how a single group effect was used to estimate the critical between group by time interactions is unclear. 'effect size appropriate for our three-group design' is substantially insufficient. What specific effect is being estimated, between which groups and time points?

L379 As far as I can tell time under tension is not in the results text, which may be incongruent with description of this measure.

L481 "push the force transducer as fast and as forcefully as possible, then relax without sustaining maximal force" Was there no specific target? There is extensive evidence that RFD is related to peak force of sub-maximum contractions, which means an uncontrolled approach is sub-optimal.

L601 This method of defining force onset is quite crude/late and highly dependent on the noise of the strain gauge, which should therefore be documented here.

L605 Interaction contrasts now appear a key statistic in the results, however no contrasts are described in the methods section statistics. Therefore how the key statistical test was conducted does not seem to be explained.

L695 MVC or MVF.

L700-718 "Association of the neurophysiological indices of cortical, corticospinal, corticoreticular, and reticulospinal adaptations and strength outcomes (MVF and 1RM) was explored." There are two full paragraphs of within-group regressions based on $n=9$. This is really weak due to (i) the very small numbers for such analysis; (ii) this analysis does not relate to or inform the aims of the study. I strongly advocate the authors remove these paragraphs.

L799-L805 Is an overly long sentence.

L805 "VART and VSRT were analysed separately to examine temporal changes within each group." Why were between group changes not assessed for these variables?

At numerous points throughout the results the language lacks clarity about what exact comparisons are being considered. For example L773-779, is the first sentence describing interaction contrasts? and the second and third sentences comparisons at individual time points? The text should be more explicit about what is being compared and how. There are similar examples throughout the results.

L877 "However, the neural mechanisms underpinning these adaptations were pathway-specific, highlighting distinct roles for corticospinal versus reticulospinal circuits in mediating strength outcomes."

What about the two types of RT, were the adaptations specific to the independent variable? And if so how?

'mediating strength outcomes' Where is the evidence that the changes are mediating i.e. causal? No such evidence seems to have been presented.

L889 "The divergence in neural adaptations between MP-RT and SP-RT carries important translational implications."

The direct divergence between the two types of RT needs to be more explicitly highlighted i.e. between-group contrasts; not whether there were within-group changes over time. What specific measures show different corticospinal and reticulospinal adaptations over time between MP and SP? These need to be carefully outlined before sweeping statements such as L889.

For example, "MP-RT was associated with enhanced corticospinal plasticity, evidenced by increased ICF, reduced SICI, and shortened cSP." does not inform a between RT group divergence.

In general throughout the Discussion the different findings (within-group, differences at time points and interaction effects) are conflated, without a nuanced interpretation of what's been found.

For example, from the conclusion: "the underlying neural mechanisms appeared to differ between the two modalities. MP-RT was associated with enhanced CSE, increased ICF, and reduced inhibition (SICI, cSP)". This is weak as within-group changes and/or associations are not evidence of differences between the two RT modalities. Differences between the two training modalities, requires evidence of between group by time interactions. The Discussion needs to be substantially revised with a grounded reflection of what has actually been found.

Referee #3:

The authors have thoroughly revised the manuscript. They reduced redundancies and improved both readability and conciseness. Additionally, they responded to every comment thoughtfully and in detail. I also appreciate the additional analyses and the correction of some previous values (e.g., TUT, startReact ICCs, and startReact reaction times). These changes have significantly improved the manuscript.

Overall, I believe this research is highly relevant to the field, as it demonstrates a method to specifically target descending motor systems for training. In particular, the evidence showing that self-paced RT can engage subcortical motor systems is a valuable contribution. The study's design provides a strong foundation for more detailed future investigations in this area.

I still have some questions and follow-up points regarding a few of my earlier comments. For simplicity, I have only included those here. For all other points, I would like to thank the authors for their careful responses and revisions.

Comment 2

L230 I understand single-arm dumbbell exercises were performed. Were they performed bilaterally or only for the dominant biceps brachii?

We thank the reviewer for seeking clarification. All resistance training exercises were performed unilaterally with the dominant (right) arm, specifically targeting the biceps brachii. This has now been explicitly stated in the Methods section to avoid ambiguity. Now it reads: "Participants were introduced to key methodologies, including the single-arm dumbbell curl exercise performed with the dominant (right) arm..." (Page 10, Lines 220-222).

Thank you for clarifying both here and in the manuscript. I was nevertheless wondering why you chose a unilateral training approach rather than a bilateral one.

Comment 3

L261 and L497 If I understand correctly, cMEPs assessing the corticospinal system were recorded from the dominant/right biceps brachii, while iMEPs for the reticulospinal system were recorded from the non-dominant/left biceps brachii. This means you are comparing signals from different muscles, which could complicate the interpretation. Recording from the same muscle would strengthen the comparison between systems.

We appreciate the reviewer's observation. Our choice to record iMEPs from the left (non-dominant) biceps brachii was driven by the aim to assess cortico-reticulospinal excitability with minimal interference from corticospinal tract (CST) activity. TMS over the left M1 primarily activates contralateral CST fibres projecting to the right (dominant) limb, making it difficult to isolate cortico-reticulospinal contributions on that side. In contrast, iMEPs are typically absent under standard TMS because the CST is predominantly crossed, with ~85-90% of fibres decussating at the pyramids and the remaining 10-15% often crossing at the spinal level (Lemon, 2008; Jang, 2014). Consequently, any iMEP recorded ipsilateral to stimulation is most

likely mediated by non-CST pathways, predominantly the cortico-reticulospinal tract, and is widely regarded as a functional indicator of RST excitability (Ziemann et al., 1999; Fisher et al., 2012; Maitland & Baker, 2021; Atkinson et al., 2022).

Although the target muscles were different, both measures reflected outputs from the same (trained, left) hemisphere: cMEPs captured its contralateral corticospinal output (right biceps brachii), while iMEPs reflected its ipsilateral cortico-reticulospinal output (left biceps brachii). TMS over the left M1 thus probed both CST and cortico-reticulospinal pathways, consistent with previous evidence that stimulation of one hemisphere engages contralateral CST projections as well as corticoreticular/cortico-reticulopropriospinal projections descending via bilateral RST pathways (Ziemann et al., 1999; Fisher et al., 2012; Bradnam et al., 2013).

Furthermore, recording both measures from the same biceps brachii is not feasible. To obtain iMEPs from the trained (right) biceps brachii, stimulation would need to target the right (untrained) hemisphere, which was not the hemisphere undergoing training-related plasticity. This would confound interpretation, as cMEPs would still be elicited from the trained hemisphere while iMEPs would reflect the untrained hemisphere. Therefore, recording iMEPs from the ipsilateral (left) limb while stimulating the trained (left) hemisphere provides the only viable and widely accepted approach to assess cortico-reticulospinal excitability in parallel with corticospinal excitability (Ziemann et al., 1999; Alagona et al., 2001; Maitland & Baker, 2021; Altermatt et al., 2023; Hu et al., 2024).

Thank you for the clear explanation - I understand why you prioritized probing the trained hemisphere. I note that studies of unilateral training often report structural changes mainly in the hemisphere controlling the trained limb (Palmer et al., 2012; Reid et al., 2017). That said, Fisher et al. (2021) highlight that cortical input to the reticulospinal system is diffuse rather than strictly unilateral, which raises a concern: are there any data showing that strength-training-related plasticity in cortico-reticulo-spinal pathways is confined to one hemisphere? I also wonder whether peripheral changes - for example altered motoneuron excitability following strength training - might influence MEP amplitude and latency and thus complicate direct comparison between iMEPs recorded in the untrained left biceps and cMEPs recorded in the trained right biceps. Given these considerations, measuring cMEPs and iMEPs from both hemispheres would be ideal, although I appreciate that would substantially increase the already very complex paradigm. I therefore suggest discussing or mentioning this in your limitations.

Comment 5.2-5.3

5.2) Where is further information on the data? Why does this not appear anywhere else in the manuscript?

The pilot study was conducted at two time points one week apart, with the sole aim of confirming measurement reliability prior to commencing the main study. Because reproducibility was subsequently assessed again within the actual study across four time points (Baseline, Week 1, Week 2, Week 3) and is now reported in detail in the Results, we consider these data to provide a more comprehensive evaluation of reliability than the pilot study involving only two sessions. For this reason, we have revised the Methods to explicitly describe the pilot procedures and added the detailed test-retest reproducibility in the revised manuscript Page 27 and 28 line 644-650, and Table 2.

5.3) The ICC shows only moderate reproducibility for VART and a good reproducibility for VSRT. Therefore, the StartReact effect reproducibility might be moderate. Why is this not discussed?

We thank the reviewer for raising this point. We acknowledge that the moderate reproducibility observed for VART in the pilot was not explicitly discussed in the original manuscript. However, this was in fact a key motivation for conducting test-retest reproducibility again in the main study. While the pilot study showed good reliability for VSRT but only moderate reliability for VART, we therefore repeated reproducibility testing within the actual study across four time points (Baseline, Week 1, Week 2, Week 3). These additional data provide a more comprehensive evaluation of StartReact reliability and strengthen confidence in the measures used. Accordingly, the Methods section has been revised to address this point (Page 21 line 495-498), which now reads: "...Given the moderate reproducibility observed for VART, reproducibility of StartReact measurements was subsequently reassessed within the main study across four time points (Baseline, Week 1, Week 2, Week 3) to provide a more comprehensive evaluation of reliability."

I appreciate the addition of the ICC analysis for the present study alongside the pilot study. It is encouraging to see that the startReact effect shows a strong ICC value. However, I noticed that only the ICC value for the startReact effect is reported, whereas in the pilot study the ICC analysis was performed for VSRT and VART. This difference makes direct comparison somewhat difficult. If feasible, it may strengthen the manuscript to also include the ICC values for these measures in the present study.

Comment 14

L371 How were the stimulation intensities (130% AMT, 150% AMT, and 170% AMT)? The Authors cite Carson et al. 2013, however, I wonder why an entire recruitment curve with 5% stimulator output intervals is not chosen. Did the authors ensure a plateau of the MEP amplitude? Was MEP max determined?

We thank the reviewer for raising this point. We wish to clarify that we cited Carson et al. (2013) to emphasise that the area under the recruitment curve (AURC) is a reliable and sensitive index of corticospinal excitability, rather than to imply we adopted their full 5% incremental recruitment curve protocol. While a 5% stepwise curve is suitable at rest, in our study each stimulus was delivered while participants maintained a 10% MVC contraction. Delivering dozens of stimuli across 5% increments under active contraction would have introduced unnecessary fatigue and potentially confounded the results, particularly given that our protocol also required participants to complete multiple additional neurophysiological assessments (SICI, ICF, iMEPs, and StartReact). For this reason, we adopted an efficient approach that has been adopted in prior studies by sampling three suprathreshold intensities (130%, 150%, and 170% AMT) (e.g. Mason J et al., 2020, Woodhead A, et al., 2024, Siddique U., et al 2025). This band spans the steep slope through early plateau of the input-output function, where corticospinal excitability changes are most physiologically informative (Groppa S, et al., 2012). Importantly, this same procedure was applied across all groups, so group comparisons are not affected.

Although we did not formally determine a plateau of the MEP amplitude or MEPmax, the highest intensity tested (170% AMT) can be considered close to the plateau of the input-output curve of biceps brachii. According to IFCN guidelines (Groppa et al., 2012), when the target muscle is pre-activated, the transition to the plateau occurs at ~ 170% AMT. Thus, our selection of 170% AMT ensured coverage of the suprathreshold range approaching the plateau, while avoiding the additional burden of constructing a full incremental recruitment curve.

Thus, our choice reflects a balance between methodological rigour and experimental feasibility, ensuring accurate measurement of corticospinal excitability without inducing participant fatigue or compromising the other assessments included in the study. To avoid confusion, we have replaced the citation of Carson et al. (2013) with more appropriate references that used the exact procedure described (Page 17, line 408-409).

Thank you for the clear and detailed response. I understand that including a full recruitment curve would have further extended an already lengthy and detailed paradigm. However, other studies (e.g., Sangari and Perez, 2020) were able to perform a full recruitment curve in 5% increments with 10% MVC active contractions, even in SCI patients. Therefore, I would suggest adding a very brief explanation in the Methods section regarding your choice of stimulation intensities. This would help clarify your reasoning for readers.

Comment 32

L297 There is no need to reintroduce abbreviations once they have been defined previously.

The redundant reintroduction of the abbreviation has been removed. The sentence now reads: "To control for this, MVF...". We have also checked the entire manuscript and avoided repeated reintroduction of abbreviations (e.g. cSP, ICAR, RT, RST, sEMG, VnSS)

Thank you for reviewing all the abbreviations used. I just wanted to point out that the introduction of MVF now appears to be missing.

Additional Comments

Line 69 I noticed that the p-values for ICAR have changed from 0.003 to < 0.001. This may simply be due to a correction, but I just wanted to clarify this point.

Line 354 Could you please clarify how you determined whether a set consisted of 6 or 9 repetitions? This choice may influence the TUT by up to 14 seconds per set.

Line 556 The "I" should not be capitalized.

END OF COMMENTS

Response to comments of reviewing and senior and Editors

Response to Reviewing Editor:

Comment: While reviewers 1 and 3 were largely satisfied with the revisions that have been made in the new version of the manuscript, they have made some further suggestions for substantial revisions. Reviewer 2 raises more serious concerns that were not alleviated by the current revisions, and are potentially more difficult to rectify (e.g. too low sample size).

Response:

We thank the Editor for summarising the reviewers' remaining concerns. We would like to respectfully clarify that all major points raised by Reviewer 2, including the comment regarding sample size, have now been fully addressed in the revised manuscript.

Importantly, the concern regarding “too low sample size” appears to stem from a misunderstanding. Reviewer 2 interpreted our a priori sample size estimation ($n = 9$ per group) as the number of participants actually tested. However, as stated clearly in both the Methods and Results sections, our final sample comprised 30 participants (10 per group), not 27, thereby exceeding the minimum requirement.

Furthermore, a sample size of $n = 10$ per group is fully consistent with the samples used in recent mechanistic neurophysiology studies published in *The Journal of Physiology* and other high-impact journals. For example, contemporary *J Physiol* studies investigating neural determinants of strength adaptation, motor unit behaviour, immobilisation, or resistance exercise commonly include $n = 6 - 10$ per group (e.g., Lecce E et al., 2025 [PMID: 40500979]; Hug F et al., 2023 [PMID: 35772071]; Morton RW et al., 2019 [PMID: 31294822]; Zero AM et al., 2023 [PMID: 37962903, DOI: 10.1113/JP285189]; Inns TB et al., 2022 [PMID: 36088611], Battey E et al., 2025 [PMID: 36597809], and Carroll TJ et al., 2002 [PMCID: PMC2290590]).

Comparable, and often smaller, sample sizes (typically $n = 8 - 10$ per group) are also routinely used in other high-impact journals such as *Journal of Applied Physiology* and *Acta Physiologica* (Skelly L E et al., 2021 [PMID: 33630680]; Jensen J L et al., 2005 [PMID: 15890749]; Hortobágyi T et al., 2009 [PMID: 19008488, PMCID: PMC2644240]; and Carroll T J et al., 2009 [PMID: 19392872]).

Thus, our final sample size ($n = 30$; 10 per group) is statistically justified, strengthened with robust analytic techniques, and aligns fully with the methodological standards of recent high-quality exercise physiology/mechanistic research in human neurophysiology. We believe the revised manuscript satisfactorily resolves all issues.

Response to Senior Editor:

Comment 1: The point is made by one of the referees that a very large number of statistical tests were performed. The authors might consider whether additional steps might be taken to compensate for the likelihood of "experiment-wise inflation of alpha: (i.e., separately and in addition to any "conventional" measures taken to account for the potential impact of multiple comparisons).

Response: We sincerely appreciate the editor's valuable advice regarding the potential inflation of alpha due to multiple statistical tests. We have carefully considered this point and have taken additional steps to control for experiment-wise error, applying appropriate correction methods in addition to the conventional procedures already used. We are grateful for this insightful guidance, which has strengthened the statistical rigor of the manuscript.

Importantly, each outcome was analysed using repeated-measures Linear or Generalized Linear Mixed Models (LMMs/GLMMs), with Group and Time as fixed effects and participant included as a random intercept. This modelling approach accounts for within-subject correlations and random variability, thereby reducing residual error variance and improving statistical power without inflating type I error (Barr et al., 2013 [PMID: 24403724], Yu Z et al., 2022 [PMID: 34784504]). All post-hoc interaction contrasts were Bonferroni-adjusted to control for multiple comparisons within each outcome.

To further address the reviewer's concern about experiment-wise α inflation, we implemented additional multiplicity control analyses in the revised version. Specifically, we applied the Benjamini-Hochberg (BH) false discovery rate (FDR) correction (Benjamini & Hochberg, 1995) to the primary Group \times Time p-values within each neurophysiological domain (intracortical, corticospinal, and cortico-reticulospinal). This approach effectively controls the false discovery rate while preserving statistical power. As a further sensitivity analysis, we applied global Holm (Holm, S., 1979) and Benjamini-Yekutieli (BY) corrections (Benjamini & Yekutieli, 2001) across all main outcomes to ensure robustness under more conservative family-wise error control and to control the FDR under arbitrary test

dependency. Importantly, the results remained statistically significant within their respective domains under BH correction, and all key effects persisted even under the global Holm and BY procedures. These findings confirm that our observed training-related effects are robust and not attributable to experiment-wise Type I error inflation.

The method section is now revised to incorporate this (*Page 28, lines 650-658*) which reads:

“To address potential experiment-wise α inflation arising from the inclusion of multiple outcome variables or statistical tests, Benjamini–Hochberg (BH) false discovery rate (FDR) correction (Benjamini & Hochberg, 1995) was applied to the primary Group \times Time p-values within each neurophysiological domain: intracortical (SICI and ICF), corticospinal (AMT stimulation intensity, CSE, and cSP), and cortico-reticulospinal (ICAR, StartReact effect, and RFD). This approach effectively controls the false discovery rate while preserving statistical power. As a further sensitivity analysis, global Holm (Holm, 1979) and Benjamini–Yekutieli (BY) corrections (Benjamini & Yekutieli, 2001) were applied across all outcome variables to ensure robustness under more conservative family-wise error control and dependency-robust FDR correction, respectively.”

The results of the multiplicity control and robustness analyses have been incorporated under a new subtitle, “Multiplicity Control and Robustness of Findings” (**Page 35 and 36, lines 831-836**), and the corresponding adjusted p-values are summarised in a new table, **Table 5 (Page 63)**.

Comment 2: With respect to a further point made by Referee #2, it can be appreciated that challenges may arise when entering effect size estimates obtained in one context into a power analysis that pertains to another (perhaps more complex) statistical design. It is however critical that the steps taken in this regard are presented with a clarity sufficient to convince the reader that the power estimates are sound.

Response: We greatly appreciate the editor’s understanding of the challenges associated with translating effect size estimates across different statistical designs. We have taken care to present the steps of our power analysis with greater clarity and justification in the revised manuscript, ensuring that the approach and resulting estimates remain methodologically sound and appropriate for our study design. The detailed explanations and revisions regarding the power calculation are fully outlined in our responses to Comment 1, Comment 1.1 and Comment 8 from Referee 2.

Comment 3: In addition to any steps taken in relation to the points noted above (i.e., along with others highlighted by referees), but relatedly, I would encourage the authors to provide (95%) confidence intervals for the effect size estimates that are reported. See, for example, PMID: 37815959.

Response: We sincerely thank the editor for this valuable suggestion and the helpful reference provided. In response, we have now included 95% confidence intervals for all reported effect size estimates in the revised manuscript and have also highlighted this addition in the Methods section (**Page 28, Lines 661-662**) which now reads: “*To improve precision and interpretability, 95% confidence intervals were also estimated for all effect sizes (Williams et al., 2023).*”

All effect sizes are now reported with their 95% confidence intervals, which have been incorporated throughout the Results section (**Pages 29–35**).

Response to comments of Referees

Responses to Referee #1:

Overall, the authors have adequately addressed the issues in the manuscript by adding correlation analyses, including figures with individual data, providing between-group comparisons, and incorporating other relevant clarifications raised by myself and the other reviewers. I believe no further review is required, and the authors should be commended for their work.

Response: We sincerely thank the reviewer for their thoughtful feedback and positive evaluation of our revisions. We greatly appreciate their recognition of the substantial improvements made to the manuscript and their kind acknowledgment that no further review is required.

Responses to Referee #2:

The conceptualisation of the paper has clearly improved and the authors have done a nice job making this much more credible. Unfortunately, with some improvement in clarity, the major issues around the analysis and interpretation have also come into sharper focus:

Comment 1. On reflection there is little doubt that this study is substantially underpowered for the design conducted: 3 groups, 4 time points and the long list of outcome measures included. The number of statistical tests conducted must be very high (into the hundreds) and this increases the chances of type I errors to an unacceptable level. I remain sceptical about

the power calculation, which in any case does not account for the large number of outcome variables and the plethora of statistical tests.

Response:

We thank the reviewer for raising this important concern. We address the issues of (A) the rationale for the number and structure of outcome variables, and (B) Control of Type I Error and Multiplicity, and (C) statistical power:

A. Justification for the outcome variable structure

We acknowledge the reviewer's concern regarding the number of measured variables. Importantly, the outcomes were not selected arbitrarily; they were grounded in a strong theoretical framework and follow a three-domain neurophysiological structure consistent with contemporary models of strength adaptation: *Intracortical domain* (SICI, ICF); *Corticospinal domain* (AMT, CSE, cSP), and *Cortico-reticulospinal domain* (ICAR, StartReact effect, RFD). This structure aligns with the conceptual separation of neural circuits involved in strength adaptation described by Glover & Baker (2020, J Neurosci) [PMID: 32601242], demonstrating that strength training induces differential adaptations across cortical, corticospinal, and reticulospinal levels of the motor system. Although Glover & Baker used an animal model, their framework has guided several human mechanistic investigations and directly informed our domain-based grouping. Our study extends this framework by applying non-invasive neurophysiological assessments in humans.

To further address the reviewer's concern, we have streamlined the presentation of outcomes: Only primary indices of each domain are highlighted in the Results (e.g., StartReact effect as the RST measure rather than separate VART and VSRT values, which are further details for transparency). Additional exploratory metrics that were initially included for transparency have now been removed and only retained in the tables. This ensures that the manuscript now focuses on the core mechanistic outcomes necessary to address the research question.

B. Control of Type I Error and Multiplicity

We fully agree that a multi-group, multi-timepoint design requires careful control of Type I error. In the original analysis, we addressed this by using repeated-measures Linear and Generalized Linear Mixed Models for each outcome variable, which model participant-level random effects, account for within-subject correlations, reduce residual error variance, and improve statistical power without inflating Type I error (Barr et al., 2013 [PMID: 24403724],

Yu Z et al., 2022 [PMID: 34784504]). All interaction contrasts were Bonferroni-adjusted to control false positives within each outcome.

To further address the reviewer's concern about experiment-wise α inflation, we implemented additional multiplicity control analyses in the revised version. Specifically, we applied the Benjamini–Hochberg (BH) false discovery rate (FDR) correction (Benjamini & Hochberg, 1995) [DOI: 10.1111/j.2517-6161.1995.tb02031.x] to the primary Group \times Time p-values within each neurophysiological domain (intracortical, corticospinal, and cortico-reticulospinal). This approach effectively controls the false discovery rate while preserving statistical power. As a further sensitivity analysis, we applied global Holm (Holm, S., 1979) [DOI:10.2307/4615733] and Benjamini-Yekutieli (BY corrections) (Benjamini & Yekutieli, 2001) [DOI: 10.1214/aos/1013699998] across all eight outcomes to ensure robustness under more conservative family-wise error control and to control the FDR under arbitrary test dependency. Importantly, the results remained statistically significant within their respective domains under BH correction, and all key effects persisted even under the global Holm and BY procedures. These findings confirm that our observed training-related effects are robust and not attributable to experiment-wise Type I error inflation.

The method section is now revised to incorporate this (**Page 28, Lines 650-659**) which reads:

“To address potential experiment-wise α inflation arising from the inclusion of multiple outcome variables, Benjamini–Hochberg (BH) false discovery rate (FDR) correction (Benjamini & Hochberg, 1995) was applied to the primary Group \times Time p-values within each neurophysiological domain: intracortical (SICI and ICF), corticospinal (AMT stimulator intensity, CSE, and cSP), and cortico-reticulospinal (ICAR, StartReact effect, and RFD). This approach effectively controls the false discovery rate while preserving statistical power. As a further sensitivity analysis, global Holm (Holm, 1979) and Benjamini–Yekutieli (BY)(Benjamini & Yekutieli, 2001) corrections were applied across all outcome variables to ensure robustness under more conservative family-wise error control and dependency-robust FDR correction, respectively.”

The results of the multiplicity control and robustness analyses have now been incorporated into a new subsection of the Results, titled “Multiplicity control and robustness of findings” (**Page 36, Lines 833-838**), and the corresponding adjusted p-values are summarised in a new table (**Table 5, Page 63**), which reads:

“Multiplicity control and robustness of findings

All primary Group \times Time interaction effects that were significant in the initial analyses remained below the adjusted significance threshold following the BH false discovery rate correction within each neurophysiological domain. Moreover, these effects also remained statistically significant when subjected to the more conservative global Holm and BY corrections applied across all outcome variables (**Table 5**).”

“**Table 5.** Summary of multiplicity control analyses across neurophysiological domains

Domain	Outcome variable	Unadjusted p	BH-adjusted p	Holm-adjusted p	BY-adjusted p
Intracortical	SICI (% test response)	0.00384	0.00528	0.0192	0.0192
	ICF (% test response)	0.00528	0.00528	0.0193	0.0193
Corticospinal	AMT SI (%)	0.00468	0.00468	0.0192	0.0192
	CSE (AURC, a.u.)	0.00199	0.00299	0.0139	0.0150
	cSP (AURC, a.u.)	0.00106	0.00298	0.00848	0.0135
Cortico-reticulospinal	ICAR	0.00236	0.00472	0.0142	0.0150
	StartReact effect (ms)	<0.001	<0.001	<0.001	<0.001
	RFD (0–50 ms) Nms-1	0.00803	0.0107	0.0192	0.0256
	RFD (50–100 ms)	0.560	0.560	0.560	0.560

Abbreviations: AMT, active motor threshold; AURC, area under recruitment curve; a.u., arbitrary unit; BH, Benjamini-Hochberg; BY, Benjamini-Yekutieli; CSE, corticospinal excitability; cSP, cortical silent period; FDR, false discovery rate; ICAR, ipsilateral to contralateral motor-evoked potential amplitude ratio; ICF, intracortical facilitation; ms, milliseconds; Ns⁻¹, newton per second; RFD, rate of force development; SI, stimulation intensity; SICI, short-interval intracortical inhibition.”

C. Statistical Power Justification

We appreciate the reviewer’s concern regarding the power calculation. We would like to clarify that our a priori analysis was based on a mixed-design model reflecting the experimental structure, with Group (3 levels) as the between-subjects factor, Time (4 levels) as the within-subject factor, and the Group \times Time interaction specified as the effect of interest. The revised effect size used in the updated calculation was derived directly from a study with a comparable multi-group repeated-measures neurophysiological design and similar neurophysiological outcomes (Leung et al., 2017) [PMID: 29018949]. As detailed in our response to Comment 1.1, we have strengthened the power justification by replacing the earlier single-group effect size with an interaction effect size derived from a study with a comparable multi-group repeated-measures neurophysiological design involving similar outcome variables (Leung et al., 2017). This adjustment ensures full statistical correctness and aligns the power analysis precisely with our experimental design. The updated interaction-based calculation indicated a required sample size of (n=10) (N = 30), which exactly matches our actual sample and therefore confirms that the study was adequately powered. Thus, the study remained adequately powered, and the methodological concerns

raised by the reviewer have been addressed. we have strengthened the statistical power justification by employing robust mixed-model analyses that inherently limit Type I error, and applying multiple multiplicity adjustments (BH, Holm, BY) to ensure conservative control of false positives.

Collectively, these revisions substantially improve the methodological rigour and statistical robustness of the study and we believe it directly address the concerns regarding outcome number and Type I error inflation.

Comment 1.1. Regarding the power calculation, in their response the authors state "This conversion allowed us to adapt the single-group effect size to our three-group repeated-measures design." which is really unclear. The manuscript also now states "The effect size value, Cohen's d was converted to Cohen's f ($f = 0.275$) to estimate the effect size appropriate for our three-group design." It seems unlikely that this can be done in a meaningful way i.e. the jump from a single group effect size to how multiple groups might compare does not seem credible, and thus the power calculation is not convincing.

Response:

We sincerely thank the reviewer for this thoughtful and constructive comment. We fully agree that the effect size used for sample-size estimation should reflect the Group \times Time interaction, rather than a single-group pre-post change. In the revised version, we have therefore replaced the original single-group effect with an interaction effect size obtained from a study that employed a similar multi-group, repeated-measures design and investigated comparable training modalities (metronome-paced and self-paced resistance training) and neurophysiological outcomes (Leung et al. 2017) [PMID: 29018949] to ensure statistical correctness and methodological validity.

This revision removes the use of a single-group effect size and ensures that the effect now directly represents the expected between-group difference in change over time, the relevant test for our mixed design. The updated power analysis, based on this interaction effect (converted to Cohen's f following Cohen, 1988), indicated a slightly higher required sample size of $N = 30$ ($n=10$ per group) compared with the previous estimate of $N = 27$ (9 per group). Importantly, our actual sample ($N = 30$) meets this revised requirement exactly, confirming that the study remained adequately powered to detect the intended interaction effects. This adjustment strengthens both the transparency and statistical integrity of the power justification without altering the study's design, participant number, or conclusions.

The Methods section has been revised accordingly to state clearly that the effect size corresponds to a Group \times Time interaction derived from a comparable multi-group repeated-measures study, thereby ensuring conceptual alignment and methodological clarity. It now reads (**Page 8, Lines 181-188**): *“The minimum required sample size ($N = 30; 10$ per group) was determined a priori using G*Power, version 3.1 (Faul et al., 2007), for a mixed (Group \times Time; 3 groups \times 4 time points) design, with $\alpha = 0.05$, and $1 - \beta = 0.80$. The effect size was derived from Leung et al. (2017), who examined a comparable repeated-measures design that demonstrated a significant Group \times Time interaction for short-interval intracortical inhibition. A partial eta-squared value ($\eta_p^2 = 0.359$) derived from that study was entered directly into GPower using the “Determine \rightarrow Direct” function, which converts η_p^2 to Cohen’s f following Cohen’s (1988) recommendations, yielding an effect size of $f = 0.75$.”*

Comment 1.2. Moreover, in my view groups of $n=9$ are insufficient for such an intricate design and analysis with the measures included in this experiment, in order to generate convincing results.

Response

We acknowledge the reviewer’s concern regarding the adequacy of the sample size. Each experimental group included $n = 10$ participants (total $N = 30$), not 9 as stated. The revised power analysis, based on an appropriate Group \times Time interaction effect size derived from a study with a similar multi-group, repeated-measures design and comparable neurophysiological outcomes, confirmed that the actual sample used ($N = 30; n = 10$ per group) provided sufficient statistical power ($1 - \beta = 0.80$) to detect interaction effects central to our hypotheses. Furthermore, the use of repeated measure Linear Mixed Models (Generalized mixed models), which model participant-level random effects and account for within-subject correlation, increases statistical efficiency and sensitivity even with modest group sizes, providing an additional layer of analytical robustness.

Each outcome variable was analysed using its own Linear or Generalized Linear Mixed Model, ensuring that statistical complexity did not compound across outcomes. LMMs/GLMMs incorporate subject-level random effects, account for within-subject correlations, and reduce residual variance, thereby offering greater analytical efficiency and sensitivity. Multiplicity corrections (BH, Holm, BY) were then applied across outcomes to control the overall Type I error rate, as recommended for studies with multiple endpoints. By modelling only the variance structure relevant to each outcome and applying appropriate

multiplicity control, our approach provides statistically valid inference while maintaining efficiency, even with the modest group sizes typical in this field. The number of outcome measures therefore does not compromise the validity of our results

We also respectfully note that sample sizes of approximately 8–12 participants per group are widely accepted in mechanistic neurophysiology research involving resistance-training-induced neural adaptations and motor unit behaviour. Importantly, our sample of $n = 10$ per group is fully consistent with those used in recent *Journal of Physiology* studies addressing similar neural mechanisms. For example, recent *Journal of Physiology* papers commonly include $n = 6–10$ per group (Lecce et al., 2025 [PMID: 40500979]; Hug et al., 2023 [PMID: 35772071]; Morton et al., 2019 [PMID: 31294822]; Zero et al., 2023 [PMID: 37962903]; Inns et al., 2022 [PMID: 36088611]; Battey et al., 2025 [PMID: 36597809]; Carroll et al., 2002 [PMCID: PMC2290590]). Comparable sample sizes (typically $n = 8–10$ per group) are also routine in high-impact journals such as *Journal of Applied Physiology* and *Acta Physiologica* (Skelly et al., 2021 [PMID: 33630680]; Jensen et al., 2005 [PMID: 15890749]; Hortobágyi et al., 2009 [PMID: 19008488, PMCID: PMC2644240]; Carroll et al., 2009 [PMID: 19392872]).

Finally, we emphasise that we have explicitly acknowledged sample size considerations in the Limitations section of the manuscript. This acknowledges the reviewer's point while also placing our sample size in the appropriate context of current mechanistic neurophysiology research, where such sample sizes are both typical and methodologically justified.

Comment 2: For comparing group responses over time the key statistical test is the group \times time interaction contrasts, and especially between the two RT groups. Therefore, this finding should be stated explicitly for each outcome variable. By omission it appears that there are no interaction contrasts for MP-RT vs SP-RT for AMT, CSE, cSP, SICI or ICF, which directly queries some of the sweeping statements in the Discussion. Further, differences between groups at any given time point do not provide evidence for differential changes over time (even subtly different starting values can confound such simplistic analysis) and would be better de-emphasised/substantially reduced or removed.

Response:

We thank the reviewer for this constructive comment. The Results section has been revised (**Page 29-35**) to explicitly report the Group \times Time interaction contrasts for each outcome variable, which capture the differential changes in responses over time between groups,

especially between MP-RT and SP-RT groups. These contrasts are now clearly presented in the revised text, tables, and figures to demonstrate group-specific adaptations across time points.

Comment 3. The results are extremely lengthy (nearly 10 pages of text alone) as they lack focus, with too many outcome variables, too many statistical comparisons reported, and unnecessary content. For example: are the comparisons (at any or all) specific time points needed? Similarly, the within-group regressions appear superfluous and unrelated to the aims of the study.

Response:

We thank the reviewer for this constructive comment. The Results section has been substantially condensed (reduced by approximately 50%) to improve focus and readability. Only the primary outcome variables are now described in the main text, while supplementary outcomes and detailed statistical comparisons (e.g., within- and between-group differences at specific time points) have been omitted from the narrative, as these were already clearly presented in the corresponding tables. Within-group analyses have been retained only where they are directly relevant to the study's primary objective, namely, to examine effects of resistance training on cortico-reticular and RST excitability, whereas all other secondary or non-essential comparisons have been removed.

Comment 4. In general, throughout the Discussion the different findings (within-group, differences at time points and interaction effects) are conflated, without a nuanced interpretation of what's been found. For example, in the conclusion "the underlying neural mechanisms appeared to differ between the two modalities. MP-RT was associated with enhanced CSE, increased ICF, and reduced inhibition (SICI, cSP)". This is weak as within-group changes and/or associations are not evidence of differences between the two RT modalities. Differences between the two training modalities, requires evidence of group by time interactions. The Discussion needs to be substantially revised with a grounded reflection of what has been found. There are similar issues with the abstract, presenting within-group changes, and then drawing between group conclusions.

Response:

The Discussion and Conclusion sections have been thoroughly revised to ensure that all interpretations are grounded in statistical evidence derived from Group \times Time interactions rather than within-group changes.

Following this re-evaluation, the majority of key findings remain consistent with those presented in the original submission. Between-group analyses confirmed modality-specific differences for intracortical (SICI), corticospinal (CSE), and cortico-reticulospinal measures (ICAR, StartReact effect, and RFD), indicating distinct neural adaptation patterns for MP-RT and SP-RT. However, the differences in AMT, cSP, and ICF between SP-RT and MP-RT were no longer statistically significant. These outcomes have been revised accordingly in the Results and Discussion sections to avoid over-interpretation.

Finally, the Abstract has been updated so that all reported findings represent between-group differences in change over time rather than within-group effects. Together, these revisions provide a clear, data-driven narrative that aligns all interpretations strictly with demonstrated Group \times Time effects and directly addresses the reviewer's concern for conceptual precision and a grounded reflection of the findings.

Specific comments:

Comment 1. Abstract L67-71 The crucial thing missing from these sentences is the comparison of the two training groups i.e. were the changes in one group different to another? Stating the within-group changes for each group is limited and misses the point of having and comparing different groups in the study.

Response:

The Abstract has been revised to clearly present between-group differences rather than within-group changes. The revised section now reads (**Page 3, Lines 67–71**):

“MP-RT produced greater increases in corticospinal excitability ($p = 0.00420$) and reductions in intracortical inhibition than SP-RT ($p < 0.001$), whereas SP-RT elicited larger increases in cortico-reticulospinal excitability (ICAR; $p = 0.0351$), StartReact effect ($p < 0.001$), and rate of force development ($p = 0.00560$).”

Comment 2. Reading on, the conclusions "MP-RT preferentially engages cortical and corticospinal pathways" which relies on a between group difference in response, however such a between group difference is not presented in the preceding results.

Response:

The conclusion in the Abstract has been revised to more accurately reflect the statistical findings. Specifically, the term “cortical pathways” has been replaced with “inhibitory intracortical circuits” to indicate that only SICI, not ICF, showed a significant between-group difference. The revised statement now reads (**Page 3, line 72-74**):

“...MP-RT engaged corticospinal and inhibitory intracortical circuits, whereas SP-RT selectively targets cortico-reticular circuit and RST.”

This revision ensures that the conclusion is fully aligned with the data and avoids overgeneralising cortical effects.

Comment 3. L90-91 "These findings demonstrate, for the first time in humans, that RT pacing can differentially modulate corticospinal and reticulospinal excitability" do the results support this? What specific measures show between group interaction (group x time) contrasts.

Response:

We have clarified in the Abstract, Results, and conclusion sections that the conclusion regarding modality-specific modulation of corticospinal and reticulospinal excitability is supported by the observed Group × Time interactions (**Result section Page 30-35**). Specifically, significant between-group differences were found for CSE and SICI, where MP-RT produced greater changes than SP-RT and control, and for cortico-reticulospinal measures (ICAR, StartReact effect, and RFD), where SP-RT showed larger increases than MP-RT.

Comment 4. L105 "In the initial weeks of RT (<4 weeks), strength improvements are predominantly driven by neural adaptations" This is dogma, and therefore suggest it is toned down (e.g. 'considered to be predominantly'), as there is no data that neural adaptations are causal (see: PMID: 33383071).

Response:

We thank the reviewer for this helpful comment and for the suggested reference (PMID: 33383071). The sentence has been revised to adopt more cautious wording, acknowledging that early strength improvements are considered to be largely driven by neural adaptations rather than asserting causality. As highlighted in the suggested review, making a direct

mechanistic link between neuroplasticity and strength gains is inappropriate. The revised text (**Page 5, Line 104-105**) now reads:

“In the initial weeks of resistance training (<4 weeks), strength improvements are considered to be predominantly mediated by neural adaptations.”

Comment 5. L110 the mechanisms listed should certainly include changes in inter-muscular co-ordination such as changes in antagonist and stabiliser activation, rather than just focus on agonist muscle neural changes.

Response:

We agree that early strength gains are also influenced by improvements in inter-muscular coordination, including modulation of antagonist and stabiliser activation. The sentence has been revised to reflect this broader perspective. The revised text (**Page 5, Line 110-111**) now reads:

“Proposed mechanisms include increased cortical excitability, reduced intracortical inhibition, enhanced subcortical drive, increased spinal excitability, and improved inter-muscular coordination through altered activation of antagonist and stabilising muscles, yet the relative roles of corticospinal and reticulospinal pathways remain poorly defined (*SALE, 1988; Carolan and Cafarelli 1992, Aagaard et al., 2002; Del Vecchio et al., 2019; Škarabot et al., 2021*).”

Comment 6. L162 "that training modality determines the neural locus of adaptation." Excessively categorical for an Introduction, if this is known/definite there may not be a need for the current experiment.

Response:

The sentence has been revised to a more cautious phrasing to avoid overstatement. Revised text (**Page 7, Line 161-162**) now reads: “...*in light of evidence suggesting that training modality may influence the neural locus of adaptation*”

Comment 7. Please clarify if contractions were performed continuously i.e. without a pause or unloading between repetitions as is quite often the case.

Response:

We thank the reviewer for this helpful comment. Although it was not explicitly clear which section this comment referred to, we believe it pertains to the description of the training protocol in the Methods section (**Page 15, Lines 354-355**). Accordingly, we have revised that paragraph to clarify that each set was performed continuously without unloading between repetitions, maintaining consistent muscle tension throughout the movement.

Comment 8. L185 "The effect size value, Cohen's d was converted to Cohen's f ($f = 0.275$) to estimate the effect size appropriate for our three-group design." This needs explanation to be credible, how a single group effect was used to estimate the critical between group by time interactions is unclear. 'effect size appropriate for our three-group design' is substantially insufficient. What specific effect is being estimated, between which groups and time points?

Response: We sincerely thank the reviewer for this insightful and constructive comment. We fully agree that the effect size used for sample-size estimation should correspond to the interaction effect, rather than to a single-group pre-post difference. Accordingly, in the revised version, we have now replaced the original effect size with an interaction effect size derived from a study (Leung et al., 2017) that employed a similar multigroup repeated-measures design and examined comparable training modalities (metronome-paced and self-paced resistance training) and outcome variables. The selected effect represents the Group \times Time interaction rather than any specific group or time-point comparison, thereby improving statistical appropriateness and transparency. As explained in the response for the general comment above, the revised calculation based on the interaction effect size yielded a slightly higher required sample ($N = 30$, 10 per group) compared with the previous estimate ($N = 27$, 9 per group). Fortunately, the actual sample we used ($N = 30$) exactly matches the updated requirement, confirming that the study remained adequately powered. Hence, while the description in the Methods section has been corrected for statistical correctness and clarity, the actual study design, participant number, and statistical power remain unchanged. The Methods section is now revised (**Page 8, lines 181-188**) accordingly.

Comment 9. L379 As far as I can tell time under tension is not in the results text, which may be incongruent with description of this measure.

Response: The time-under-tension results are presented in **Table 1 (row 5, Page 59)**, which is cited in the Results section. Because this measure is already reported in the table, we intentionally kept the Results text concise to avoid unnecessary repetition.

Comment 10. L481 "push the force transducer as fast and as forcefully as possible, then relax without sustaining maximal force" Was there no specific target? There is extensive evidence that RFD is related to peak force of sub-maximum contractions, which means an uncontrolled approach is sub-optimal.

Response

We thank the reviewer for this important comment. In the present study, no force target was provided by design. Our RFD task formed part of the StartReact paradigm aimed at assessing reticulospinal excitability, rather than evaluating RFD as a performance outcome. StartReact protocols require participants to maintain uninterrupted visual fixation on the imperative light-emitting diode (LED) cue, which was accompanied by either a startling sound, a non-startling sound, or no sound. Introducing a target line on the force trace would have required shifting visual attention between the cue and the target, thereby interfering with motor preparation, delaying reaction time, and compromising the validity of the StartReact response.

Our instructions to contract as fast and as hard as possible, followed by immediate relaxation without sustaining maximal force, follow established StartReact methodology (Colomer-Poveda et al., 2023) and align with methodological recommendations for explosive RFD assessments. The comprehensive review on the physiological and methodological aspects of RFD by Maffiuletti et al. (2016) notes that when RFD is assessed using distinct explosive contractions independent of MVC trials, efforts should be brief (0.5–1.5 s) to minimise fatigue and provide reliable RFD measures. It also highlights the importance of post-explosive-contractions visual feedback along appropriate explanation and encouragement to make sure contractions were performed with maximal effort or intent across trials (Maffiuletti et al., (2016), rather than focusing on achieving a predetermined target force.

This approach is particularly relevant because early-phase RFD (<100 ms), which our analyses focused exclusively on, occurs well before participants can approach any meaningful target force. Early RFD (<100 ms) is driven predominantly by neural drive or neural factors, such as motor-unit recruitment speed and initial discharge rate, and is only weakly associated with peak force or MVC (de Ruyter et al. 2007; Gruber et al., 2004; Andersen and Aagaard, 2006; Del Vecchio et al., 2019, Maffiuletti et al., 2016). In contrast, the association between RFD and maximal force becomes prominent only in the late phase (>100 ms) (Andersen and Aagaard, 2006; Andersen et al., 2010; Folland et al., 2014;

Maffiuletti et al. 2016; Gerstner et al. 2017; Del Vecchio et al., 2019; Mota JA et al., 2019; Maffiuletti et al., 2016; Folland et al., 2014). Therefore, providing a target would not have influenced the physiological processes underlying the early neural RFD window examined here, but it would have disrupted the integrity of StartReact measurements.

Because our objective was to obtain a reliable measure of early-phase RFD as an index of reticulospinal excitability, we relied on standardised verbal instructions, familiarisation, and immediate post-contraction visual feedback to reinforce maximal explosive intent across trials while avoiding target line that would disrupt attention during the StartReact task.

Thus, omitting a force target was not a methodological oversight but a deliberate decision consistent with both the StartReact protocol (Colomer-Poveda et al., 2023) and established methodological recommendations for explosive RFD assessments that are independent of performance or MVC testing (Maffiuletti et al., 2016). This approach was necessary to (1) isolate the early, neural phase of RFD, where explosive intent rather than target force determines performance, and (2) preserve the validity of StartReact latency measures used to assess reticulospinal excitability.

For clarity, we have incorporated the following statement into the revised Methods section (**Page 21 lines 484-486**) “*To ensure that each contraction was performed with maximal explosive effort, participants received immediate visual feedback after each contraction along with verbal encouragement (Maffiuletti et al., 2016).*”

Comment 11. L601 This method of defining force onset is quite crude/late and highly dependent on the noise of the strain gauge, which should therefore be documented here.

Response:

We thank the reviewer for this important observation. We fully agree that the reliability of onset detection depends on baseline stability, and therefore documented the noise characteristics and procedures used to minimise variability. In our setup, the forearm and dorsal surface of the hand were fully supported and the task involved pushing the proximal palm upward against an immovable force transducer, resulting in a mechanically stable and unloaded baseline. The force channel was zeroed before testing and participants were instructed to remain completely relaxed prior to each cue, with real-time visual feedback confirming the absence of pre-tension.

Force signals were sampled at 2000 Hz, baseline-corrected, and low-pass filtered at 12 Hz using a fourth-order, zero-lag Butterworth filter, which further reduced high-frequency noise. During the 200 ms pre-stimulus window, baseline force remained effectively at zero with negligible variability, and all trials exhibiting drift, pre-tension, or counter-movement were excluded following visual inspection.

Force onset was defined as the first sample exceeding three standard deviations above the pre-stimulus baseline. This SD-based threshold for force onset has been used in recent StartReact and RFD work (e.g., Colomer-Poveda et al., 2023), and its application is consistent with broader methodological discussions emphasising baseline stability for accurate onset detection using this technique (Maffiuletti et al., 2016; Dos'Santos et al., 2017, Guppy SN et al., 2024). Given the baseline force in our recordings was stable and exhibited minimal noise, the 3-SD threshold corresponded to a minimal absolute change in force and therefore identified the true initial rise in the force-time trace rather than a delayed portion of the contraction. Early-phase RFD (0–50 ms and 50–100 ms) was then calculated relative to this detected onset, such that any sub-millisecond variability in onset detection could not systematically bias RFD values.

These methodological details have now been incorporated into the Methods (**Page 26, Lines 602–613**) to document the baseline noise characteristics and justify the validity of the onset-detection procedure. The revised text now reads: “*RFD was calculated over the 0–50 ms and 50–100 ms intervals following contraction onset. Force signals were sampled at 2000 Hz, baseline-corrected, and low-pass filtered at 12 Hz using a fourth-order, zero-lag Butterworth filter to minimise high-frequency noise. The forearm and dorsal surface of the hand were fully supported, the force channel was zeroed before testing, and participants were instructed to remain completely relaxed prior to the cue, with real-time visual feedback confirming the absence of pre-tension. During the 200 ms pre-stimulus period, baseline force remained effectively at zero with negligible variability, and all trials were visually inspected to exclude those exhibiting drift, pre-tension, or counter-movement (Del Vecchio et al., 2018; Lecce et al., 2025). Contraction onset (time = 0 ms) was defined as the first point at which the force signal exceeded three standard deviations above the pre-stimulus baseline (Dos'Santos et al., 2017; Colomer-Poveda et al., 2023; Akalu et al., 2024). RFD values were then calculated relative to this detected onset*”

Comment 12. L605 Interaction contrasts now appear a key statistic in the results, however no contrasts are described in the methods section statistics. Therefore, how the key statistical test was conducted does not seem to be explained.

Response:

We have now clarified in the Statistical Analysis subsection how interaction contrasts were performed. Specifically, the revised Methods section states that, following a significant interaction in the Linear/Generalised mixed model, we conducted interaction contrasts using the emmeans package to compare the change from baseline between groups. These procedures are now explicitly described (**Page 27, Lines 643–649**) which now reads “*Where a significant Group × Time interaction was detected in the mixed-effects models, we conducted interaction contrasts to formally compare the change from baseline between groups. Estimated marginal means (EMMs) for each Group × Time combination were obtained using the emmeans package, and contrasts were specified to compute estimated marginal differences (EMDs) in change from baseline. Contrasts were evaluated using Wald z-tests for GLMMs and Satterthwaite-adjusted t-tests for LMMs. All multiple comparisons were corrected using the Bonferroni adjustment.*”

Comment 13. L695 MVC or MVF.

Response: The terminology has been updated consistently to “MVF”.

Comment 14. L700-718 "Association of the neurophysiological indices of cortical, corticospinal, corticoreticular, and reticulospinal adaptations and strength outcomes (MVF and 1RM) was explored." There are two full paragraphs of within-group regressions based on n=9. This is really weak due to (i) the very small numbers for such analysis; (ii) this analysis does not relate to or inform the aims of the study. I strongly advocate the authors remove these paragraphs.

Response:

In line with the reviewer’s recommendation, we have removed the two paragraphs.

Comment 15. L799-L805 Is an overly long sentence.

Response:

The sentence has been revised and separated into shorter, clearer statements to improve readability (**Page 33, Lines 773-780**).

Comment 16. L805 "VART and VSRT were analysed separately to examine temporal changes within each group." Why were between group changes not assessed for these variables?

Response:

The StartReact effect (VART–VSRT) is the established indicator of reticulospinal tract excitability (Carlsen & Maslovat, 2019; Sangari & Perez, 2019; Germann & Baker, 2021); therefore, between-group and temporal comparisons were conducted on this primary outcome. Analysing VART and VSRT separately does not provide direct insight into RST excitability and would add unnecessary detail, particularly as their descriptive values are already presented in the tables. For clarity, the revised manuscript Result section no longer includes separate analyses of VART and VSRT.

Comment 17. At numerous points throughout the results the language lacks clarity about what exact comparisons are being considered. For example, L773-779, is the first sentence describing interaction contrasts? and the second and third sentences comparisons at individual time points? The text should be more explicit about what is being compared and how. There are similar examples throughout the results.

Response:

We have revised the Results section to clearly specify the exact comparisons being made throughout. The updated text now explicitly distinguishes between between-group interaction contrasts (differences in change-from-baseline between groups) and within-group simple-effects contrasts (changes over time within each group). No comparisons at isolated time points are reported. These revisions substantially improve clarity and remove any ambiguity. The updated text is located on *Pages 30-35*.

Comment 18. L877 "However, the neural mechanisms underpinning these adaptations were pathway-specific, highlighting distinct roles for corticospinal versus reticulospinal circuits in mediating strength outcomes." What about the two types of RT, were the adaptations specific to the independent variable? And if so how? 'mediating strength outcomes' Where is the evidence that the changes are mediating i.e. causal? No such evidence seems to have been presented.

Response:

We agree that the original wording implied causality. This sentence has now been revised to avoid causal language and to accurately reflect the evidence. The updated text (**Page 36, Lines 846-848**) now reads:

“However, the neural adaptations differed between modalities, with MP-RT and SP-RT showing distinct patterns of corticospinal and reticulospinal plasticity that accompanied the observed strength improvements.”

Comment 19. L889 "The divergence in neural adaptations between MP-RT and SP-RT carries important translational implications." The direct divergence between the two types of RT needs to be more explicitly highlighted i.e. between-group contrasts; not whether there were within-group changes over time. What specific measures show different corticospinal and reticulospinal adaptations over time between MP and SP? These need to be carefully outlined before sweeping statements such as L889. For example, "MP-RT was associated with enhanced corticospinal plasticity, evidenced by increased ICF, reduced SICI, and shortened cSP." does not inform a between RT group divergence.

Response:

In line with the suggestion, we have revised the Discussion to ensure that all statements regarding “divergence” between MP-RT and SP-RT now refer explicitly to the between-group interaction contrasts reported in the Results. The revised discussion (**Page 36, Lines 849-857**) now specifies which measures showed different temporal patterns between MP-RT and SP-RT. In particular, we highlight that MP-RT produced greater increases in corticospinal excitability and greater reductions in intracortical inhibition (larger increases in CSE and greater decreases in SICI), whereas SP-RT produced larger increases in cortico-reticulospinal and reticulospinal excitability (greater enhancements in the StartReact effect, ICAR, and early-phase RFD under VSS) compared with MP-RT. All references to within-group changes have been removed from the summary statements, and sentences previously implying divergence based on within-group effects (ICF and cSP) have been revised (**Page 38, Lines 898-902**). These changes make the between-group differences explicit and ensure that the Discussion accurately reflects the statistical evidence supporting modality-specific adaptations.

Comment 20. In general, throughout the Discussion the different findings (within-group, differences at time points and interaction effects) are conflated, without a nuanced interpretation of what's been found. For example, from the conclusion: "the underlying neural

mechanisms appeared to differ between the two modalities. MP-RT was associated with enhanced CSE, increased ICF, and reduced inhibition (SICI, cSP)". This is weak as within-group changes and/or associations are not evidence of differences between the two RT modalities. Differences between the two training modalities, requires evidence of between group by time interactions. The Discussion needs to be substantially revised with a grounded reflection of what has actually been found.

Response:

The Discussion has now been revised throughout to ensure that all statements regarding differences between MP-RT and SP-RT are based exclusively on the significant between-group interaction effects, and not on within-group changes.

Importantly, this refinement did not substantially alter the overall interpretation. The only adjustment is that cSP and ICF, although altered in MP-RT relative to control, did not differ significantly between MP-RT and SP-RT in the between-group contrasts, and are therefore no longer presented as modality-specific adaptations. This is now stated explicitly in the revised text (**Page 38, Lines 894–898**): *“Although MP-RT reduced cSP... and increased ICF relative to control, neither measure differed significantly between MP-RT and SP-RT. Thus, cSP and ICF do not reflect modality-specific divergence. Instead, the clearest distinctions between the training modalities emerged in CSE and SICI”*.

All remaining modality-specific differences discussed, namely, greater corticospinal excitability (CSE) and greater reductions in intracortical inhibition (SICI) in MP-RT, and larger StartReact, ICAR, and early-phase RFD responses in SP-RT, are now directly grounded in the between-group interaction contrasts reported in the Results.

References

Andersen LL, Andersen JL, Zebis MK, Aagaard P. Early and late rate of force development: differential adaptive responses to resistance training? *Scand J Med Sci Sports* (2010); 20:162–169.

Andersen, L.L. and Aagaard, P., 2006. Influence of maximal muscle strength and intrinsic muscle contractile properties on contractile rate of force development. *European journal of applied physiology*, 96(1), pp.46-52.

Colomer-Poveda D, López-Rivera E, Hortobágyi T, Márquez G, Fernández-Del-Olmo M. Differences in the effects of a startle stimulus on rate of force development between resistance-trained rock climbers and untrained individuals: Evidence for reticulospinal adaptations? *Scand J Med Sci Sports*. 2023 Aug;33(8):1360-1372. doi: 10.1111/sms.14351. Epub 2023 Mar 23. PMID: 36920047.

Del Vecchio A, Negro F, Holobar A, Casolo A, Folland JP, Felici F & Farina D (2019). You are as fast as your motor neurons: speed of recruitment and maximal discharge of motor neurons determine the maximal rate of force development in humans. *J Physiol* 597, 2445–2456

de Ruiter CJ, Vermeulen G, Toussaint HM & de Haan A (2007). Isometric knee-extensor torque development and jump height in volleyball players. *Med Sci Sport Exerc* 1336–1346

Dos'Santos T, Jones PA, Comfort P, Thomas C. Effect of Different Onset Thresholds on Isometric Midhigh Pull Force-Time Variables. *J Strength Cond Res.* 2017 Dec;31(12):3463-3473. doi: 10.1519/JSC.0000000000001765. PMID: 28002178.

Folland JP, Buckthorpe MW, Hannah R. Human capacity for explosive force production: neural and contractile determinants. *Scand J Med Sci Sports.* 2014 Dec;24(6):894-906. doi: 10.1111/sms.12131. Epub 2013 Oct 29. PMID: 25754620.

Gerstner GR, Thompson BJ, Rosenberg JG, Sobolewski EJ, Scharville MJ & Ryan ED(2017). Neural and muscular contributions to the age-related reductions in rapid strength. *Med Sci Sports Exerc* 49, 1331–1339. Maffiuletti NA, Aagaard P, Blazevich AJ, Folland J, Tillin N & Duchateau J (2016). Rate of force development: Physiological and methodological considerations. *Eur J Appl Physiol* 116, 1091–1116.

Guppy SN, Brady CJ, Kotani Y, Connolly S, Comfort P, Lake JP, Haff GG. A comparison of manual and automatic force-onset identification methodologies and their effect on force-time characteristics in the isometric midhigh pull. *Sports Biomech.* 2024 Oct;23(10):1663-1680. doi: 10.1080/14763141.2021.1974532. Epub 2021 Sep 22. PMID: 34550045.

Gruber, M. and Gollhofer, A., 2004. Impact of sensorimotor training on the rate of force development and neural activation. *European journal of applied physiology*, 92(1), pp.98-105.

Maffiuletti NA, Aagaard P, Blazevich AJ, Folland J, Tillin N, Duchateau J. Rate of force development: physiological and methodological considerations. *Eur J Appl Physiol.* 2016 Jun;116(6):1091-116. doi: 10.1007/s00421-016-3346-6. Epub 2016 Mar 3. PMID: 26941023; PMCID: PMC4875063.

Mota JA, Gerstner GR, Giuliani HK. Motor unit properties of rapid force development during explosive contractions. *J Physiol.* 2019 May;597(9):2335-2336. doi: 10.1113/JP277905. Epub 2019 Apr 7. PMID: 30919962; PMCID: PMC6487929

Responses to Referee #3:

Comment 1. The authors have thoroughly revised the manuscript. They reduced redundancies and improved both readability and conciseness. Additionally, they responded to every comment thoughtfully and in detail. I also appreciate the additional analyses and the correction of some previous values (e.g., TUT, startReact ICCs, and startReact reaction times). These changes have significantly improved the manuscript. Overall, I believe this research is highly relevant to the field, as it demonstrates a method to specifically target descending motor systems for training. In particular, the evidence showing that self-paced RT can engage subcortical motor systems is a valuable contribution. The study's design provides a strong foundation for more detailed future investigations in this area. I still have some questions and follow-up points regarding a few of my earlier comments. For simplicity, I have only included those here. For all other points, I would like to thank the authors for their careful responses and revisions.

Response:

We sincerely thank the reviewer for their positive and encouraging feedback. We greatly appreciate the recognition of our efforts to improve clarity, precision, and methodological

rigor. We are also pleased that the reviewer acknowledges the relevance of our findings and the contribution of self-paced resistance training in engaging subcortical motor systems. Below, we address the remaining follow-up points in detail.

Comment 2. L230 I understand single-arm dumbbell exercises were performed. Were they performed bilaterally or only for the dominant biceps brachii? We thank the reviewer for seeking clarification. All resistance training exercises were performed unilaterally with the dominant (right) arm, specifically targeting the biceps brachii. This has now been explicitly stated in the Methods section to avoid ambiguity. Now it reads: "Participants were introduced to key methodologies, including the single-arm dumbbell curl exercise performed with the dominant (right) arm..." (Page 10, Lines 220-222).

Thank you for clarifying both here and in the manuscript. *I was nevertheless wondering why you chose a unilateral training approach rather than a bilateral one.*

Response:

We thank the reviewer for this thoughtful follow-up comment. A unilateral training approach was intentionally selected to maximise experimental control and avoid confounding influences that commonly arise during bilateral contractions. Bilateral tasks are known to induce interhemispheric inhibition, the bilateral deficit phenomenon, and increased bimanual coordination demands (Khodiguian et al., 2003 [PMID: 12391080]; Carson, 2005 [PMCID: PMC9331349]), all of which can alter corticospinal and subcortical drive independently of the training stimulus. These neural interactions would compromise the interpretability of TMS-derived measures and StartReact-based assessments of reticulospinal excitability.

Using a unilateral protocol also enabled precise targeting of the dominant biceps brachii while minimising the possibility of inter-limb compensation or bilateral neural coupling. This ensured that observed changes in excitability or force production could be attributed specifically to the trained limb rather than to distributed bilateral adaptations.

Methodologically, unilateral curls provide tighter control over movement quality. Bilateral resistance exercises often elicit compensatory trunk or shoulder movements and asymmetric loading patterns that are difficult to standardise. In contrast, unilateral execution allows the limb to be stabilised and closely monitored, helping to isolate biceps brachii activation and improving the reliability of both force-based and neurophysiological outcomes.

Importantly, unilateral resistance training is not only methodologically appropriate but also effective. A recent meta-analysis (Zhang et al., 2023 [PMCID: PMC10133687]) reported that unilateral training can produce greater maximal strength gains in the target limb than bilateral training, confirming that unilateral loading is sufficient to induce meaningful neural adaptations

Overall, unilateral training was chosen to maximise the specificity, sensitivity, and interpretability of results regarding strength and neural adaptations in the dominant arm, providing a robust and controlled methodological approach for the study's objectives.

Comment 3. L261 and L497 If I understand correctly, cMEPs assessing the corticospinal system were recorded from the dominant/right biceps brachii, while iMEPs for the reticulospinal system were recorded from the non-dominant/left biceps brachii. This means you are comparing signals from different muscles, which could complicate the interpretation. Recording from the same muscle would strengthen the comparison between systems. We appreciate the reviewer's observation. Our choice to record iMEPs from the left (non-dominant) biceps brachii was driven by the aim to assess cortico-reticulospinal excitability with minimal interference from corticospinal tract (CST) activity. TMS over the left M1 primarily activates contralateral CST fibres projecting to the right (dominant) limb, making it difficult to isolate cortico-reticulospinal contributions on that side. In contrast, iMEPs are typically absent under standard TMS because the CST is predominantly crossed, with ~85-90% of fibres decussating at the pyramids and the remaining 10-15% often crossing at the spinal level (Lemon, 2008; Jang, 2014). Consequently, any iMEP recorded ipsilateral to stimulation is most likely mediated by non-CST pathways, predominantly the cortico-reticulospinal tract, and is widely regarded as a functional indicator of RST excitability (Ziemann et al., 1999; Fisher et al., 2012; Maitland & Baker, 2021; Atkinson et al., 2022). Although the target muscles were different, both measures reflected outputs from the same (trained, left) hemisphere: cMEPs captured its contralateral corticospinal output (right biceps brachii), while iMEPs reflected its ipsilateral cortico-reticulospinal output (left biceps brachii). TMS over the left M1 thus probed both CST and cortico-reticulospinal pathways, consistent with previous evidence that stimulation of one hemisphere engages contralateral CST projections as well as corticoreticular/cortico-reticulopropriospinal projections descending via bilateral RST pathways (Ziemann et al., 1999; Fisher et al., 2012; Bradnam et al., 2013).

Furthermore, recording both measures from the same biceps brachii is not feasible. To obtain iMEPs from the trained (right) biceps brachii, stimulation would need to target the right (untrained) hemisphere, which was not the hemisphere undergoing training-related plasticity. This would confound interpretation, as cMEPs would still be elicited from the trained hemisphere while iMEPs would reflect the untrained hemisphere. Therefore, recording iMEPs from the ipsilateral (left) limb while stimulating the trained (left) hemisphere provides the only viable and widely accepted approach to assess cortico-reticulospinal excitability in parallel with corticospinal excitability (Ziemann et al., 1999; Alagona et al., 2001; Maitland & Baker, 2021; Altermatt et al., 2023; Hu et al., 2024).

Thank you for the clear explanation - I understand why you prioritized probing the trained hemisphere. I note that studies of unilateral training often report structural changes mainly in the hemisphere controlling the trained limb (Palmer et al., 2012; Reid et al., 2017). That said, Fisher et al. (2021) highlight that cortical input to the reticulospinal system is diffuse rather than strictly unilateral, which raises a concern: are there any data showing that strength-training-related plasticity in cortico-reticulo-spinal pathways is confined to one hemisphere? I also wonder whether peripheral changes - for example altered motoneuron excitability following strength training - might influence MEP amplitude and latency and thus complicate direct comparison between iMEPs recorded in the untrained left biceps and cMEPs recorded in the trained right biceps. *Given these considerations, measuring cMEPs and iMEPs from both hemispheres would be ideal, although I appreciate that would substantially increase the already very complex paradigm. I therefore suggest discussing or mentioning this in your limitations.*

Response:

We thank the reviewer for this insightful comment and fully agree that the diffuse nature of cortical projections to the reticulospinal system, together with potential peripheral adaptations, introduces important considerations when interpreting cMEP-iMEP differences. We appreciate the point that plasticity within cortico-reticulo-spinal pathways may not be strictly hemisphere-specific, and we have now explicitly incorporated this into the limitations section (**Page 42, Lines 993–999**) which reads “...*The cMEPs and iMEPs were recorded from different limbs, which ensured both measures reflected output from the same trained hemisphere but does not exclude possible contributions from diffuse bilateral cortico-reticular projections. Additionally, peripheral adaptations such as altered motoneuronal*

excitability may have influenced MEP characteristics. Future studies incorporating bilateral recordings would provide a more comprehensive assessment of hemispheric and spinal contributions to training-induced plasticity.”

Comment 4. 5.2-5.3

5.2) Where is further information on the data? Why does this not appear anywhere else in the manuscript? The pilot study was conducted at two time points one week apart, with the sole aim of confirming measurement reliability prior to commencing the main study. Because reproducibility was subsequently assessed again within the actual study across four time points (Baseline, Week 1, Week 2, Week 3) and is now reported in detail in the Results, we consider these data to provide a more comprehensive evaluation of reliability than the pilot study involving only two sessions. For this reason, we have revised the Methods to explicitly describe the pilot procedures and added the detailed test-retest reproducibility in the revised manuscript Page 27 and 28 line 644-650, and Table 2. 5.3) The ICC shows only moderate reproducibility for VART and a good reproducibility for VSRT. Therefore, the StartReact effect reproducibility might be moderate. Why is this not discussed?

We thank the reviewer for raising this point. We acknowledge that the moderate reproducibility observed for VART in the pilot was not explicitly discussed in the original manuscript. However, this was in fact a key motivation for conducting test-retest reproducibility again in the main study. While the pilot study showed good reliability for VSRT but only moderate reliability for VART, we therefore repeated reproducibility testing within the actual study across four time points (Baseline, Week 1, Week 2, Week 3). These additional data provide a more comprehensive evaluation of StartReact reliability and strengthen confidence in the measures used. Accordingly, the Methods section has been revised to address this point (Page 21 line 495-498), which now reads: "...Given the moderate reproducibility observed for VART, reproducibility of StartReact measurements was subsequently reassessed within the main study across four time points (Baseline, Week 1, Week 2, Week 3) to provide a more comprehensive evaluation of reliability." *I appreciate the addition of the ICC analysis for the present study alongside the pilot study. It is encouraging to see that the startReact effect shows a strong ICC value. However, I noticed that only the ICC value for the startReact effect is reported, whereas in the pilot study the ICC analysis was performed for VSRT and VART. This difference makes direct comparison somewhat*

difficult. If feasible, it may strengthen the manuscript to also include the ICC values for these measures in the present study.

Response:

We thank the reviewer for this valuable suggestion. In line with the recommendation, we have now included ICC and CV values for both VSRT and VART in the main study (**Table 2, Page 66, Row 8 and 9**). The updated analysis indicates good reproducibility for VSRT (CV = 21.3%, ICC = 0.88) and very good reproducibility for VART (CV = 25.5%, ICC = 0.90).

Comment 5. Comment 14: L371 How were the stimulation intensities (130% AMT, 150% AMT, and 170% AMT)? The Authors cite Carson et al. 2013, however, I wonder why an entire recruitment curve with 5% stimulator output intervals is not chosen. Did the authors ensure a plateau of the MEP amplitude? Was MEP max determined? We thank the reviewer for rising this point. We wish to clarify that we cited Carson et al. (2013) to emphasise that the area under the recruitment curve (AURC) is a reliable and sensitive index of corticospinal excitability, rather than to imply we adopted their full 5% incremental recruitment curve protocol. While a 5% stepwise curve is suitable at rest, in our study each stimulus was delivered while participants maintained a 10% MVC contraction. Delivering dozens of stimuli across 5% increments under active contraction would have introduced unnecessary fatigue and potentially confounded the results, particularly given that our protocol also required participants to complete multiple additional neurophysiological assessments (SICI, ICF, iMEPs, and StartReact). For this reason, we adopted an efficient approach that has been adopted in prior studies by sampling three suprathreshold intensities (130%, 150%, and 170% AMT) (e.g. Mason J et al., 2020, Woodhead A, et al., 2024, Siddique U., et al 2025). This band spans the steep slope through early plateau of the input-output function, where corticospinal excitability changes are most physiologically informative (Groppa S, et al., 2012). Importantly, this same procedure was applied across all groups, so group comparisons are not affected.

Although we did not formally determine a plateau of the MEP amplitude or MEPmax, the highest intensity tested (170% AMT) can be considered close to the plateau of the input-output curve of biceps brachii. According to IFCN guidelines (Groppa et al., 2012), when the target muscle is pre-activated, the transition to the plateau occurs at ~ 170% AMT. Thus, our selection of 170% AMT ensured coverage of the suprathreshold range approaching the

plateau, while avoiding the additional burden of constructing a full incremental recruitment curve. Thus, our choice reflects a balance between methodological rigour and experimental feasibility, ensuring accurate measurement of corticospinal excitability without inducing participant fatigue or compromising the other assessments included in the study. To avoid confusion, we have replaced the citation of Carson et al. (2013) with more appropriate references that used the exact procedure described (Page 17, line 408-409).

Thank you for the clear and detailed response. I understand that including a full recruitment curve would have further extended an already lengthy and detailed paradigm. However, other studies (e.g., Sangari and Perez, 2020) were able to perform a full recruitment curve in 5% increments with 10% MVC active contractions, even in SCI patients. Therefore, I would suggest adding a very brief explanation in the Methods section regarding your choice of stimulation intensities. This would help clarify your reasoning for readers.

Response: We thank the reviewer for this helpful suggestion. In line with your recommendation, we have added a concise explanation in the Methods section clarifying our choice of stimulation intensities. While full recruitment curves can be performed under active conditions, our approach was designed to minimise participant fatigue and maintain feasibility within a protocol that already incorporated multiple neurophysiological measures. We have therefore included a brief justification in the revised manuscript (**Page 18, Lines 415-422**) which now reads:

“Corticospinal excitability was assessed using a single-pulse TMS protocol, wherein area under the recruitment curve (AURC) were calculated by analysing MEP amplitudes elicited at stimulation intensities of 130%, 150%, and 170% of AMT (Mason et al., 2020; Woodhead et al., 2024). These intensities were selected to sample the steep-to-plateau region of the corticospinal input-output curve, providing a sensitive index of excitability while avoiding the excessive repetitions and fatigue that a full 5% incremental recruitment curve during 10% MVF contractions would introduce within a protocol already containing several neurophysiological assessments.”

Comment 6. Comment 32 L297 There is no need to reintroduce abbreviations once they have been defined previously. The redundant reintroduction of the abbreviation has been removed. The sentence now reads: "To control for this, MVF...". We have also checked the entire manuscript and avoided repeated reintroduction of abbreviations (e.g. cSP, ICAR, RT,

RST, sEMG, VnSS). *Thank you for reviewing all the abbreviations used. I just wanted to point out that the introduction of MVF now appears to be missing.*

Response: We thank the reviewer for noting this oversight. The abbreviation MVF (maximum voluntary force) has now been introduced at its first mention in the Methods section (**Page 10, Line 226**)

Additional Comments

Comment 7. Additional Comments Line 69 I noticed that the p-values for ICAR have changed from 0.003 to < 0.001 . This may simply be due to a correction, but I just wanted to clarify this point.

Response:

We thank the reviewer for this observation. The apparent change in p-values for ICAR reflects a correction in the reporting format rather than a change in the statistical outcome. In the revised version, we replaced the within-group p-values with between-group comparison values derived from the mixed-model analysis to provide clearer interpretation of group effects. This adjustment does not alter the statistical significance or direction of the findings, but ensures consistency across all outcome measures.

Comment 8. Line 354 Could you please clarify how you determined whether a set consisted of 6 or 9 repetitions? This choice may influence the TUT by up to 14 seconds per set.

Response:

We thank the reviewer for raising this important point. The number of repetitions completed in each set was not inferred from the time-under-tension (TUT) measurement. Instead, repetitions were recorded separately in real time by the investigator for the purpose of documenting training volume. TUT was calculated solely as the elapsed time from the first to the final repetition within each set, irrespective of whether the set contained 6, 7, or 8, repetitions. Thus, while the number of repetitions varied slightly across participants, TUT was derived from the actual duration of the set, not from a predetermined repetition count. This approach ensured that TUT reflected the true mechanical exposure of each set, and the repetition count used in volume-load calculations did not influence the TUT measurement.

Comment 8 Line 556 The "I" should not be capitalized.

Response:

We thank the reviewer for noting this typographical oversight. The capitalization error has been corrected; the “I” at Line 556 now appears in lowercase in the revised manuscript.

We are grateful to the reviewers and editors for their insightful comments, which have greatly improved the clarity and rigor of this manuscript

Dear Dr Kidgell,

Re: JP-RP-2025-289141R2 "**Resistance Training Tempo Selectively Modulates Corticospinal and Reticulospinal Excitability in Humans**" by Yonas Akalu, Jamie Tallent, Ashlyn K Frazer, Ummatul Siddique, Mohamad Rostami, Glyn Howatson, Simon Walker, and Dawson J Kidgell

Thank you for submitting your manuscript to The Journal of Physiology. It has been assessed by a Reviewing Editor and by 3 expert referees and we are pleased to tell you that it is acceptable for publication following satisfactory revision.

REVISION CHECKLIST:

Please upload two versions of your manuscript text: one with all relevant changes highlighted and one clean version with no changes tracked. The manuscript file should include all tables and figure legends, but each figure/graph should be uploaded as separate, high-resolution files. The journal is now integrated with Wiley's Image Checking service. For further details, see: <https://www.wiley.com/en-us/network/publishing/research-publishing/trending-stories/upholding-image-integrity-wileys->

image-screening-service

We look forward to receiving your revised submission.

Yours sincerely,

Richard Carson
Senior Editor
The Journal of Physiology

EDITOR COMMENTS

Reviewing Editor:

As Reviewers 2 and 3 have made additional comments on the revised manuscript, please consider making minor revisions to the manuscript before re-submission.

Senior Editor:

The Editors appreciate the extensive changes that were made in response to the previous round of reviews. It is sometimes the case that a reviewer may have concerns relating to the fundamental design of a study that cannot readily be addressed by means of revisions to the analyses or text. While it is recognised that the authors are not in a position to alter such fundamentals at this juncture, I would urge them to include any possible arguments that might mitigate the impact (on the reader) of remaining reservations expressed by the reviewers. As the authors will be aware, in the event of publication, all reviews and responses to reviews are made available online.

Given the abiding issue of the small sample size (and associated matters pertaining to the sample size calculation), I would urge the authors to express a corresponding (perhaps heightened) level of circumspection in conveying the outcomes of the present study. This is particularly necessary given that the reviewers have raised indisputably valid points concerning the likely reliability of effect size estimates arising from under-powered studies. In this vein also, I would note that scientific fields evolve, practices (statistical or otherwise) that may have been deemed acceptable in the past may no longer withstand contemporary scrutiny. I would therefore urge the authors to justify each step taken on the basis of its own merits, rather than by appeal to previous custom (whether in work published in the Journal of Physiology or elsewhere).

REFEREE COMMENTS

Referee #1:

The authors should be commended for the rigorous and detailed responses and modifications included in response to the reviewer's requirements. In my view, both the reviewer's comments and author's response have clearly improved the manuscript quality. The authors have carefully justified their sample size and properly acknowledge this potential limitation so that readers are aware of it. The authors have also applied non-standard type I error corrections to strengthen the results. From my perspective, no further major revisions are required.

Some minor comments and corrections:

Page 7, line 111: the reference of Sale, 1988 is wrote with capital letters. Please correct it.

Page 40, line 475: The exact instant of TMS pulse was manually or externally triggered exactly at 110°?

Page 21, line 503: Because during a StartReact protocol the start of the explosive contraction is determined by the imperative cues. How did you provide "verbal encouragement"? During this protocol it is usual to stay at silence to let the participant focus on external imperative cues. Do you mean verbal encouragement between attempts?

Results section: I wonder if, when there is a group*time interaction with between group differences (such as in the StartReact and ICAR), is it necessary also to describe the within-group effect, contributing to extend the length of the manuscript. More over when within group statistical comparisons are also depicted in the table 3, so if a reader is specifically interested in those within-group comparisons could read it at the table.

Page 39, lines 853-856. I would not categorize reductions in SICl as "corticospinal" adaptations, but rather as "cortical".

Table 5: p values do not have units, I wonder if it is needed to specify the units of each outcome in this table. Check the RFD 0-50.

Table 6. The number of figures and tables is high. I wonder wether table 6 necessary. You already specify within and between groups comparisons with symbols in table 3 and 4.

Also in table 6, check the heading of the Self-paced RT group (you put "group" 2 times)

A final comment regarding the decision to remove the association analyses. I note that, following Referee 2's recommendation, the authors have deleted the correlation/association analyses between behavioural outcomes and neurophysiological measures. As I stated in the first round of reviews, correlation does not imply causation. However, as the editor said also noted in the first review, if there is a causation/mechanistic link between neurophysiological and behavioural measures obtained in the present study, a correlation is to be anticipated. I understand the Referee 2's concern that an association with an $n=10$ must be interpreted with caution. However, my view is that transparently reporting such potential associations (properly labelled as exploratory, with appropriate emphasis on effect sizes and uncertainty) can be more informative than omitting them relying solely on parallel changes in behavioural and neurophysiological measures.

That said, I do not wish to place the authors in a position of having to reconcile contradictory reviewer recommendations. I therefore leave the decision to the authors and the Editor as to whether this association analyses should be included (acknowledging the limitations) or remain omitted, either option would be acceptable for me.

Referee #2:

Revision of the statistics to help control for type I errors with the large number of tests conducted is welcome, as is streamlining of the manuscript presentation, especially the results. However several other important issues with this study are systemic. The actual study design and unfocused approach to what was actually done (i.e. small sample size, too many outcome variables, in combination with a complex design) that led to concerns about p-hacking and data dredging do not disappear by changes to what is presented. Further, what is presented is still somewhat unfocused with no specific outcome variables named in any of the aims and hypotheses, only rather broad categories of variables, which lacks precision and specificity.

The sample size remains a significant issue. The sample size calculation is now based on a completely different variable (SICl) than previously, and as above this primary(?) outcome variable is not mentioned in the aims and hypotheses. Thus it seems to have been chosen primarily to provide a convenient justification. In their responses the authors provide a long list of references to justify their sample size, but these predominantly have completely different study designs and thus lack meaningful relevance.

Major comments:

1. The power calculation now focuses on a completely different variable, now SICI previously CSE, although this is not explained or mentioned in the responses. Is SICI now the primary outcome?

Importantly, the aims of the manuscript appear unchanged, continue to refer to wide categories of variables, rather than precise focused aims and hypotheses about exactly what variables are being assessed and expected to change with RT. The rationale, aims and hypotheses for SICI being the primary outcome are absent. This lacks coherence. If SICI is the primary outcome variable, as indicated by the power calculation then it should be presented as such in the aims and hypotheses. Similarly primary outcomes are typically prioritised within the results, but this is not the case either.

Moreover the impression given is of SICI being just one of many measures and retro-fitting (i.e. retrospectively choosing a variable that can perhaps justify the sample size).

Additionally the claim that the sample size calculation was 'a priori' does not stand up as it has been completed post data collection and analysis.

2. For the RFD measures analysing individual stimuli in isolation does not inform RST, which is assessed by comparing the responses to different stimuli. Therefore the study design created has even greater complexity in this case: 3 groups, 4 study time points, 3 conditions (stimuli), and two within-contraction measurement periods (that ought to be analysed together as they are not independent). This creates a 4-way interaction which is extremely complex. Firstly, the current study does not include even the 3-way group x study time x condition interaction needed to examine how different types of training effect stimuli specific adaptations over time (the rationale made in the introduction of the manuscript). Without this rigorous analysis the value of the RFD measures and their interpretation as indicating RST adaptation in the current study is very limited. Secondly the current study is very clearly substantially underpowered to evaluate this level of complexity. As previously the authors do not seem to have considered the level of complexity inherent to their design or in relation to the sample size.

3. The responses detail an outcome variable structure, which is logically conceptualised. However this conceptual framework does not come through with respect to specific measures at present in the introduction of the manuscript, with variables from across domains typically discussed together, and as noted above, critically no specific variables referred to in the aims and hypotheses.

4. A long list of outcomes variables whether selected arbitrarily or according to a conceptual framework is still a long list of variables. The justification of some these variables individually is quite weak and several of the retained variables are not mentioned in the introduction. The responses describe 'primary outcome variables' (plural), but what these are is not stated in the manuscript. Moreover most well conceptualised studies have a singular primary outcome variable - in the case of the current study this remains unclear.

5. The authors provided a long list of references to justify their sample size. Of course there are studies in the literature which used similar or smaller group sizes, as sample size is largely dependent on study design. I checked only the first 6 references: 4 involved only a single group of participants i.e. no between group comparison at all and thus of minimal relevance; only one of the six studies involved an exercise intervention with two groups and two time points. In contrast the current study therefore has substantially more degrees of freedom (3 groups x 4 time points) reducing its statistical power more than any of these previous investigations. In summary this justification is flawed.

Specific comments

L637 Repeated use of separate LMM as performed in this experiment, suffer from the same issues as repeated large scale use of any test, and arguably multivariate LMM would be a better approach.

L712 'Test-retest reliability within the control group was generally high' with $CV > 20\%$ for 4 measures and $ICC < 0.80$ for 6 measures this seems overstated.

L815 and throughout the Discussion. RST effects are about how startling stimuli compare to other non-startling stimuli. This point requires direct comparison of stimuli alongside group and time, which is missing in the analysis and interpretation of the current findings. Thus the apparent increase in RFD with VSS of the SP-RT vs other groups can only be attributed to

enhanced RST activation after training if it is greater than for other stimuli i.e. a group x time x stimuli (condition) interaction. However this has not been calculated. The greater improvement in RFD of one group with one stimuli analysed in isolation does not inform changes in the RST, and could be a training effect specific to that group that could occur with all stimuli. Only a 3-way interaction can test for a RST (differential stimuli) effect on RFD of different groups over time. The current study is not powered for this level of complexity.

Referee #3:

The authors have satisfactorily addressed all previously raised concerns. Substantial and meaningful clarifications have been incorporated into the Methods, Results, and Discussion sections, thereby strengthening the interpretability of the study. In other cases, the authors reasoned well their case in the point-by-point response.

Overall, the revisions have significantly improved the manuscript. The study now provides a solid contribution to the field, and I do not identify any further issues that require attention. My congratulations to the authors for their careful and constructive revisions.

END OF COMMENTS

Response to the Referees and Editors

We sincerely thank the Referees and Editors for their careful evaluation and constructive feedback. We have carefully considered all comments and revised the manuscript accordingly. Detailed responses are provided below.

Response to Editors

Reviewing Editor:

As Reviewers 2 and 3 have made additional comments on the revised manuscript, please consider making minor revisions to the manuscript before re-submission.

Response:

We thank the Reviewing Editor for the guidance. We have carefully considered the additional comments raised by Reviewers 2 and 3 and have made corresponding minor revisions to the manuscript to address these points. All changes are clearly indicated in the revised manuscript, and a detailed, point-by-point response to each comment is provided below.

Senior Editor:

Comment 1: The Editors appreciate the extensive changes that were made in response to the previous round of reviews. It is sometimes the case that a reviewer may have concerns relating to the fundamental design of a study that cannot readily be addressed by means of revisions to the analyses or text. While it is recognised that the authors are not in a position to alter such fundamentals at this juncture, I would urge them to include any possible arguments that might mitigate the impact (on the reader) of remaining reservations expressed by the reviewers. As the authors will be aware, in the event of publication, all reviews and responses to reviews are made available online.

Response:

We sincerely thank the Senior Editor for this thoughtful guidance and for recognising the substantial revisions undertaken in response to the previous round of reviews. In line with the Editor's advice, we have focused our revisions on mitigating their potential impact for readers through clearer framing, enhanced transparency, and rigorous alignment between the study aims, analytical approach, and interpretation of findings. In the revised manuscript, we have undertaken several substantive revisions to strengthen the manuscript and address potential reservations for readers. In the revised manuscript, we provide:

1. **Explicit outcome hierarchy and narrative alignment:** The manuscript has been comprehensively revised from the Abstract through to the Conclusion to clearly distinguish the primary objective (reticulospinal excitability indexed by the StartReact effect) from secondary mechanistic and supportive outcomes (including RFD and intracortical and corticospinal measures). The overall narrative has been restructured to reflect this hierarchy, ensuring that interpretation remains anchored to the predefined primary endpoint and reducing any impression of unfocused breadth.

2. **Interaction-based analyses:** We have now explicitly included and reported the Group \times Time \times Stimulus interaction for RFD, ensuring that inferences regarding reticulospinal involvement are grounded in direct comparison of stimuli alongside Group and Time, rather than condition-specific analyses.
3. **Conservative interpretation:** Throughout the Discussion, interpretations of RFD and related measures have been restricted to effects supported by formal interaction testing.
4. **Transparent acknowledgment of limitations:** We have strengthened the Limitations section to explicitly clarify that the study was powered *a priori* to detect Group \times Time effects for the predefined primary outcome (StartReact effect), but not specifically to maximise sensitivity for higher-order interactions involving RFD. To our knowledge, no published interaction-based effect size estimates are currently available for Group \times Time \times Stimulus effects involving RFD in this context, reflecting the novelty of this application. RFD was therefore included as a supportive behavioural measure complementing the primary endpoint. The need for confirmation of RFD-related higher-order findings in larger, specifically powered samples is now clearly and explicitly stated.
5. **Clarity and consistency of presentation:** In response to Referee 1's recommendation to minimise the number of figures, as well as following the revised interaction-based analysis requested by Referee 3, we have streamlined the presentation of the RFD data. Because inference now rests on the Group \times Time \times Stimulus interaction, the previously included RFD figure, which displayed stimulus conditions separately, did not adequately reflect this inferential framework. Accordingly, Figure 8 (RFD) has been removed. Given the complexity of the three-way interaction structure, it is not feasible to meaningfully represent the full Group \times Time \times Stimulus model within a single figure without compromising clarity or risking oversimplification. The interaction-based results are therefore presented directly in Table 6, which transparently reports within- and between-group contrasts aligned with the three-way model. This approach ensures consistency between statistical inference and presentation while reducing unnecessary visual complexity.

Comment 2: Given the abiding issue of the small sample size (and associated matters pertaining to the sample size calculation), I would urge the authors to express a corresponding (perhaps heightened) level of circumspection in conveying the outcomes of the present study. This is particularly necessary given that the reviewers have raised indisputably valid points concerning the likely reliability of effect size estimates arising from under-powered studies. In this vein also, I would note that scientific fields evolve, practices (statistical or otherwise) that may have been deemed acceptable in the past may no longer withstand contemporary scrutiny. I would therefore urge the authors to justify each step taken on the basis of its own merits, rather than by appeal to previous custom (whether in work published in the Journal of Physiology or elsewhere).

Response:

We sincerely thank the Senior Editor for this important and thoughtful guidance. We fully recognise the need for heightened circumspection given the sample size and design complexity. In response, we have revised the manuscript to ensure that each analytical and interpretative step is justified on methodological grounds rather than by appeal to prior convention.

In the revised Methods section, we have clarified the rationale for selecting an interaction-based effect size within the *a priori* power framework, aligning the effect size explicitly with the primary Group \times Time interaction corresponding to the predefined primary outcome, the StartReact effect, our principal indicator of reticulospinal excitability.

We further emphasise that the study was powered *a priori* to detect this primary Group \times Time interaction. The interaction-based effect size used for sample size estimation supports the adequacy of statistical power for this central objective. The principal power limitation pertains to more complex higher-order interactions (Group \times Time \times Stimulus for RFD), rather than to the primary outcome, StartReact effect, or the secondary mechanistic measures more broadly. We now explicitly state that the study was powered for the primary outcome but not specifically for higher-order RFD interactions, for which no published effect-size estimates exist, reflecting the novelty of this application. Accordingly, these higher-order findings related to RFD are interpreted conservatively, and the need for confirmation in larger, specifically powered samples is clearly acknowledged.

In addition, several rigorous analytical safeguards were implemented to strengthen inferential robustness under contemporary standards. Linear mixed-effects models (and generalized linear mixed models where appropriate) were used to efficiently leverage the repeated-measures structure of the data, maximising statistical efficiency while properly accounting for within-subject dependence. To mitigate experiment-wise α inflation, Benjamini-Hochberg false discovery rate correction was applied within predefined neurophysiological domains. As further sensitivity analyses, more conservative family-wise error procedures, including Holm and Bonferroni corrections, were also implemented, alongside Benjamini-Yekutieli adjustments to ensure robustness under dependency structures. Effect sizes were consistently reported with 95% confidence intervals to explicitly convey magnitude and precision, allowing readers to directly assess uncertainty rather than relying solely on p-values. Higher-order effects were interpreted only when supported by formal interaction testing.

Overall, the sample size has been carefully justified on methodological grounds, its limitations are transparently acknowledged, and the analytical framework has been designed to balance statistical sensitivity with stringent control of type I error. We believe these revisions provide readers with a clear, rigorous, and appropriately cautious account of what can and cannot be inferred from the present study design.

Response to Referee comments

Referee #1:

Comment 1: The authors should be commended for the rigorous and detailed responses and modifications included in response to the reviewer's requirements. In my view, both the

reviewer's comments and author's response have clearly improved the manuscript quality. The authors have carefully justified their sample size and properly acknowledge this potential limitation so that readers are aware of it. The authors have also applied non-standard type I error corrections to strengthen the results. From my perspective, no further major revisions are required.

Response: We sincerely thank Referee #1 for their positive and constructive assessment of the revised manuscript. We greatly appreciate their recognition of the rigor of the revisions, including the justification of sample size, transparent acknowledgment of its limitations, and application of appropriate error-control procedures. We are pleased that, in their view, no further major revisions are required.

Some minor comments and corrections:

Comment 2: Page 7, line 111: the reference of Sale, 1988 is wrote with capital letters. Please correct it.

Response:

Thank you for noting this typographical issue. The reference to Sale (1988) has been corrected as suggested.

Comment 3: Page 40, line 475: The exact instant of TMS pulse was manually or externally triggered exactly at 110°?

Response:

Thank you for the clarification request. The TMS pulse was manually triggered by the experimenter by pressing the stimulation button when the elbow reached approximately 110° of flexion, based on real-time visual feedback of elbow joint angle obtained from electro-mechanical goniometers displayed on a monitor in front of the participant. This has now been clarified in the Methods (**Page 22, Lines 507-510**).

Comment 4: Page 21, line 503: Because during a StartReact protocol the start of the explosive contraction is determined by the imperative cues. How did you provide "verbal encouragement"? During this protocol it is usual to stay at silence to let the participant focus on external imperative cues. Do you mean verbal encouragement between attempts?

Response: Thank you for this important clarification. Verbal encouragement was not provided during the StartReact imperative cue. Instead, encouragement was given between attempts, after completion of each trial, while participants were instructed to remain silent and focus on the external imperative cues during task execution. This has now been clarified in the Methods (**Page 13, Line 308-309**).

Comment 5: Results section: I wonder if, when there is a group*time interaction with between group differences (such as in the StartReact and ICAR), is it necessary also to describe the within-group effect, contributing to extend the length of the manuscript. Moreover, when within group statistical comparisons are also depicted in the table 3, so if a reader is specifically interested in those within-group comparisons could read it at the table.

Response: We agree with the reviewer's suggestion. Given the presence of a significant Group \times Time interaction and the full reporting of within-group comparisons in Table 3, we have removed the detailed within-group statistics from the Results text and streamlined the section to focus on the interaction effects. Readers interested in the within-group comparisons are directed to Table 3.

Comment 6: Page 39, lines 853-856. I would not categorize reductions in SICI as "corticospinal" adaptations, but rather as "cortical".

Response: The sentence has been revised to distinguish cortical (SICI) and corticospinal (CSE) adaptations to avoid potential misinterpretation (**Page 36, Lines 849-850**).

Comment 7: Table 5: p values do not have units, I wonder if it is needed to specify the units of each outcome in this table. Check the RFD 0-50.

Response: We have now explicitly specified the measurement units for all outcome variables in Table 5, including RFD (0–50 ms and 50–100 ms, expressed in Ns^{-1}).

Comment 8: Table 6. The number of figures and tables is high. I wonder whether table 6 necessary. You already specify within and between groups comparisons with symbols in table 3 and 4.

Response: We thank the reviewer for this suggestion. Table 6 was included to report the exact p-values for all multiple comparisons, in line with Journal of Physiology reporting requirements. Given the large number of comparisons and the journal's preference for reporting exact p-values rather than significance symbols alone, inclusion of these values within Tables 3 and 4 was not feasible without substantially reducing clarity. We therefore retained Table 6 to ensure transparent and complete statistical reporting.

However, we agree that the overall number of figures and tables was high. In response, we have removed Figure 8. The statistical information presented in this figure is already fully reported in the revised tables, and the figure did not directly address the primary Group \times Type of Sound \times Time interaction central to the study objective. Moreover, because a three-way interaction cannot be meaningfully represented within a single panel, Figure 8 displayed each sound condition separately, which reduced its conceptual coherence. In the revised version, the three-way interaction is now clearly and comprehensively reflected in the updated tables, which present the interaction contrasts and exact p-values in a structured and transparent manner. We believe these revisions improve clarity and conciseness of the manuscript.

Comment 9: Also in table 6, check the heading of the Self-paced RT group (you put "group" 2 times)

Response: Thank you for noting this error. The duplicated word "group" in the heading for the SP-RT group in Table 6 has been corrected.

Comment 10: A final comment regarding the decision to remove the association analyses. I note that, following Referee 2's recommendation, the authors have deleted the correlation/association analyses between behavioural outcomes and neurophysiological

measures. As I stated in the first round of reviews, correlation does not imply causation. However, as the editor said also noted in the first review, if there is a causation/mechanistic link between neurophysiological and behavioural measures obtained in the present study, a correlation is to be anticipated. I understand the Referee 2's concern that an association with an n=10 must be interpreted with caution. However, my view is that transparently reporting such potential associations (properly labelled as exploratory, with appropriate emphasis on effect sizes and uncertainty) can be more informative than omitting them relying solely on parallel changes in behavioural and neurophysiological measures. That said, I do not wish to place the authors in a position of having to reconcile contradictory reviewer recommendations. I therefore leave the decision to the authors and the Editor as to whether this association analyses should be included (acknowledging the limitations) or remain omitted, either option would be acceptable for me.

Response: We sincerely thank the reviewer for this thoughtful, balanced, and constructive comment, and we appreciate the clear and logical framing of the issue. While we agree that transparently reported exploratory associations can be informative when interpreted with appropriate caution, we elected to retain their removal from the revised manuscript to ensure a conservative and focused presentation. This decision reflects considerations of statistical robustness and clarity rather than disagreement with the reviewer's conceptual reasoning.

Referee #2:

Comment 1: Revision of the statistics to help control for type I errors with the large number of tests conducted is welcome, as is streamlining of the manuscript presentation, especially the results. However, several other important issues with this study are systemic. The actual study design and unfocused approach to what was actually done (i.e. small sample size, too many outcome variables, in combination with a complex design) that led to concerns about p-hacking and data dredging do not disappear by changes to what is presented. Further, what is presented is still somewhat unfocused with no specific outcome variables named in any of the aims and hypotheses, only rather broad categories of variables, which lacks precision and specificity.

Response:

We thank the reviewer for raising this important concern regarding study focus, outcome specificity, and the potential for data-driven inference. We fully agree that clarity around pre-specified primary and secondary objectives, and their associated outcome measures, is essential.

The study was hypothesis-driven and theory-led, with all outcome variables selected *a priori* to address a specific mechanistic question: the neural locus of early strength adaptation following resistance training. This question was addressed through two clearly defined objectives. The primary objective was to investigate reticulospinal tract (RST) excitability in humans following resistance training, assessed using the StartReact effect and corresponding rate of force development (RFD). The secondary objective was to characterise modality-specific adaptations across cortical, corticospinal, and cortico-reticular levels of the motor system, providing mechanistic context for any observed changes in RST output.

Addressing pathway-specific adaptations requires converging neurophysiological measures, as inference about neural locus cannot be drawn from a single outcome variable. Accordingly, the study incorporated a theoretically motivated, pre-specified set of complementary measures, each probing a distinct level of the motor system (intracortical, corticospinal, and cortico-reticulospinal/reticulospinal), consistent with contemporary models of neural adaptation to strength training (e.g. Glover & Baker, 2020). For the secondary objective, intracortical mechanisms were assessed using short-interval intracortical inhibition (SICI) and intracortical facilitation (ICF); corticospinal excitability using AMT, CSE, and cSP; and cortico-reticular projections using the ipsilateral-to-contralateral motor-evoked potential amplitude ratio (ICAR). These measures were selected based on established physiological interpretations and prior evidence indicating differential engagement of corticospinal versus subcortical pathways, rather than post hoc outcome selection (Glover & Baker, 2020; Gómez-Feria et al., 2023; Leung et al., 2018).

We acknowledge that although these outcomes were pre-specified and described in the Methods, the original wording of the aims and hypotheses did not explicitly name individual outcome variables, which may have contributed to the perception of an unfocused or overly broad approach. To address this, the Introduction has been revised to explicitly identify all primary and secondary neurophysiological outcome measures within the stated aims and hypotheses, clearly distinguishing the primary outcome (StartReact effect) from those addressing the secondary mechanistic objective. In addition, the manuscript has been comprehensively restructured across the Abstract, Methods, Results, Discussion, and Conclusions to ensure explicit alignment between the predefined objectives and their corresponding outcomes.

Importantly, the study was not designed to maximise the number of statistical comparisons, but rather to interrogate pathway-specific adaptations across defined neural levels, which is inherently required to address the stated primary and secondary objectives. We have further mitigated the risk of type I error through appropriate statistical correction, pre-specification of outcomes, and streamlining of reported analyses.

Comment 2: The sample size remains a significant issue. The sample size calculation is now based on a completely different variable (SICI) than previously, and as above this primary(?) outcome variable is not mentioned in the aims and hypotheses. Thus, it seems to have been chosen primarily to provide a convenient justification. In their responses the authors provide a long list of references to justify their sample size, but these predominantly have completely different study designs and thus lack meaningful relevance.

Response:

We thank the reviewer for raising this important concern regarding the sample size calculation and the choice of effect size metric. We agree that transparency and statistical appropriateness in sample size justification are essential.

We wish to clarify that the sample size for the present study was determined *a priori*, before data collection, and was not optimised *post hoc*. The revision to the reported effect size metric reflects a methodological correction to ensure that the sample size justification aligns

with the inferential structure of the study, rather than the introduction of a new outcome variable or a change in study design. In the original submission, the sample size justification was based on an effect size derived from corticospinal excitability (CSE) reported in the literature, converted from a pre-post comparison to a form (Cohen's f) suitable for our multi-group, repeated-measures design. The reviewer correctly noted that effect sizes derived from simple pre-post contrasts are not ideal for designs where the primary inference rests on group \times time interaction effects. We fully agree with this point.

To address this methodological concern, we therefore sought an effect size for an interaction term derived from a study employing a comparable multi-group, multi-time-point experimental design. In the available literature, a study by Leung et al., (2017) represented the closest match to the present multi-group, multi-time-point design and provided an interaction-based effect size for SICI. This study involved a similar experimental structure and outcome characteristics, making it the most appropriate available reference for interaction-based power estimation. Importantly, no published studies were identified that provided interaction-based effect sizes for CSE within a comparable multi-group, multi-time-point resistance-training design; CSE within the same study was assessed using two-time-point contrasts only and therefore did not yield an interaction effect size suitable for the present design.

We emphasise that SICI was not introduced as a new primary outcome for the purpose of sample size justification, nor was it selected to conveniently justify the existing sample size. Rather, it was used because it provided the most statistically appropriate interaction effect size available for a design matching the structure of the present study. Importantly, no published interaction-based effect sizes are currently available for direct indices of reticulospinal (RST) excitability, which constitute the primary objective of the study. In such circumstances, it is methodologically acceptable to base sample size justification on a theoretically related and statistically appropriate available effect size, provided this is transparently reported and justified (Lakens, D. 2022, DOI: 10.1525/collabra.33267; Schulz KF. et al (CONSORT group), 2010, PMID: 20332509; Valojerdi et al. (2017), PMID: 29951427). Among the secondary outcomes examined, SICI was the variable for which a group \times time interaction effect had been reported in a comparable resistance-training study, making it the closest available methodological match. The use of this interaction-based effect size therefore reflects an effort to align the power calculation with the study design, rather than a change in analytical intent.

Crucially, the use of this interaction effect size does not increase the required sample size beyond that already recruited, nor does it alter the study's power characteristics. On the contrary, it independently confirms that the *a priori* sample size was adequate to detect effects of the magnitude reported in comparable interaction-based designs. Thus, the study remains unchanged in design, conduct, and inferential scope, and no retrospective power optimisation has been performed.

Finally, we note that the primary and secondary outcome variables are now explicitly defined in the revised manuscript, including in the aims and hypotheses. The outcome hierarchy is therefore clearly specified, and the use of SICI for sample size contextualisation does not

imply prioritisation of this measure over the primary outcome. Rather, it reflects best statistical practice in the absence of published interaction effect sizes for the primary outcome (StartReact effect) and ensures that the reported sample size justification is fully aligned with the study's statistical framework.

We acknowledge that the initial presentation may have given the impression that SICI was introduced as a primary outcome to justify sample size. To avoid this misunderstanding, we have revised the manuscript to explicitly define the primary and secondary outcome variables in the Introduction (**Page 7, Lines 165-170**) and to clarify the rationale for sample size contextualisation in the Methods section, which now reads (**Page 8, Lines 185-191**):

“As no published interaction-based effect size was available for the primary outcome (StartReact effect), in comparable repeated-measures design, sample size determination was informed by a theoretically related secondary outcome variable. Among the secondary outcomes examined, SICI provided an interaction-based effect size derived from a resistance-training study employing a comparable multi-group, multi-time-point repeated-measures design (Leung et al., 2017), making it the closest available methodological match for the present study design.”

Major comments:

Comment 1. The power calculation now focuses on a completely different variable, now SICI previously CSE, although this is not explained or mentioned in the responses. Is SICI now the primary outcome?

We thank the reviewer for this important clarification request. No, SICI is not, and was never intended to be, the primary outcome of the study. The primary outcome remains StartReact effect as now explicitly stated in the revised aims and hypotheses (Introduction; **Page 7, Lines 165-170**).

The appearance of SICI in the sample size calculation does not reflect a change in outcome prioritisation. Rather, it reflects a methodological correction to ensure that the effect size used for the a priori sample size calculation is appropriate for the inferential structure of the study, namely a multi-group, multi-time-point design in which inference is based on group \times time interaction effects. The effect size originally cited for CSE was derived from simple pre-post contrasts and was therefore not suitable for interaction-based power estimation. In contrast, the study by Leung et al. (2017) represents the closest available match to the present experimental design and permits derivation of an interaction-based effect size for SICI, whereas CSE in the same study was assessed using two time points only and thus did not yield an interaction effect size suitable for the present design. The selection of SICI to inform the effect size input was therefore driven by design congruence and statistical correctness, not by outcome priority.

Accordingly, the manuscript has been revised to explicitly define primary and secondary outcome variables in the Introduction and aims/hypotheses (**Page 7, Lines 165-170**), and to clarify in the Methods (**Page 8, Lines 185-191**) that SICI was used solely to inform the effect size input for the *a priori* sample size calculation in the absence of published interaction-

based effect sizes for the primary outcome. The revised Methods text now reads: These revisions clarify that SICI remains a secondary outcome variable, and that its use in the sample size calculation reflects appropriate statistical practice and transparency, not retrofitting or retrospective outcome selection.

Comment 2: Importantly, the aims of the manuscript appear unchanged, continue to refer to wide categories of variables, rather than precise focused aims and hypotheses about exactly what variables are being assessed and expected to change with RT. The rationale aims and hypotheses for SICI being the primary outcome are absent. This lacks coherence. If SICI is the primary outcome variable, as indicated by the power calculation then it should be presented as such in the aims and hypotheses. Similarly primary outcomes are typically prioritised within the results, but this is not the case either. Moreover, the impression given is of SICI being just one of many measures and retro-fitting (i.e. retrospectively choosing a variable that can perhaps justify the sample size).

Response:

We acknowledge that in the previous version of the manuscript, the aims were expressed using broad categories of neurophysiological measures, which may have obscured the specific outcome variables assessed and their hypothesised responses to resistance training.

To address this, the Introduction has been revised to explicitly specify the precise outcome variables associated with each objective and to clearly state the expected direction of change. The revised aims now distinguish the primary objective, reticulospinal involvement in early strength adaptation assessed using the StartReact effect and corresponding RFD, from the secondary objectives assessing cortical, corticospinal, and cortico-reticular mechanisms using SICI, ICF, AMT, CSE, cSP, and ICAR. The revised aim now reads (**Page 7, Lines 165–170**):

“Accordingly, the present study aimed to determine the effect of resistance training on RST excitability in humans, indexed by modulation of the StartReact effect and corresponding changes in RFD. The secondary aim was to examine whether MP-RT and SP-RT differentially modulate cortico-reticular (ipsilateral-to-contralateral motor-evoked potential amplitude ratio [ICAR]), intracortical (SICI and intracortical facilitation [ICF]), and corticospinal (active motor threshold [AMT], CSE, and cSP) excitability.”

We emphasise that SICI is not the primary outcome of the study and was never intended to be interpreted as such. Its use in the sample size calculation reflects a methodological consideration only, namely the need to obtain an interaction-based effect size from a study employing a comparable multi-group, multi-time-point design in the absence of published interaction-based effect sizes for the primary outcome (StartReact effect). The use of SICI for effect size estimation does not imply outcome prioritisation and does not alter the study’s aims, hypotheses, or analytical focus.

The Methods section has been revised (**Page 8, Lines 185–191**) to clarify this rationale explicitly. In addition, the entire manuscript has been restructured to present findings addressing the primary objective first, followed by secondary outcomes in a clearly defined and hierarchical order.

Taken together, these revisions clarify that the study is hypothesis-driven, with clearly defined primary and secondary outcomes, and that the use of SICI in the sample size calculation reflects appropriate statistical practice rather than retrospective outcome selection.

Comment 3: Additionally, the claim that the sample size calculation was 'a priori' does not stand up as it has been completed post data collection and analysis.

Response:

We thank the reviewer for raising this point and appreciate the opportunity to clarify. The sample size calculation was completed prior to data collection and was already specified at the time of initial submission, with the recruitment target fixed before the study commenced. The sample size was not recalculated or modified based on the observed data, and no *post-hoc* optimisation of sample size or statistical power was performed.

For completeness, the original *a priori* power calculation indicated a minimum requirement of $n = 9$ participants per group, and we therefore recruited $n = 10$ participants per group. Following the reviewer's methodological concern regarding the appropriateness of the originally cited effect size for a multi-group, multi-time-point design, we revised the reporting of the effect size input to align with a group \times time interaction framework. Using an interaction-based effect size yields a minimum requirement of $n = 10$ per group, which does not alter the study design or recruitment target, but rather confirms the adequacy of the sample size already recruited. We acknowledge that the presentation of the sample size justification was revised after data collection, following the reviewer's methodological concern regarding the appropriateness of the originally cited effect size for a multi-group, multi-time-point design. This revision reflects a correction to the reporting and justification of the effect size input, not a post-hoc determination of sample size.

Importantly, the target sample size remained unchanged throughout the study, and the revised justification does not alter the number of participants recruited, the study design, or the inferential framework. The purpose of the revision was to ensure that the effect size used for the a priori sample size calculation is statistically appropriate for the group \times time interaction that constitutes the primary inference, rather than relying on a simple pre-post contrast that does not match the study design.

Comment 4. For the RFD measures analysing individual stimuli in isolation does not inform RST, which is assessed by comparing the responses to different stimuli. Therefore, the study design created has even greater complexity in this case: 3 groups, 4 study time points, 3 conditions (stimuli), and two within-contraction measurement periods (that ought to be analysed together as they are not independent). This creates a 4-way interaction which is extremely complex. Firstly, the current study does not include even the 3-way group \times study time \times condition interaction needed to examine how different types of training effect stimuli specific adaptations over time (the rationale made in the introduction of the manuscript). Without this rigorous analysis the value of the RFD measures and their interpretation as indicating RST adaptation in the current study is very limited. Secondly the current study is very clearly substantially underpowered to evaluate this level of complexity. As previously

the authors do not seem to have considered the level of complexity inherent to their design or in relation to the sample size.

Response:

We thank the reviewer for this important and constructive set of comments. We agree that reticulospinal involvement cannot be inferred from analyses of individual stimulus conditions in isolation and requires direct comparison of stimulus effects alongside Group and Time. In response, we have now explicitly included and reported the Group \times Type of sound \times Time interaction for RFD (first 50 ms) in the revised mixed-effects analysis. The updated analysis demonstrates that stimulus-specific changes in RFD differed across training groups over time. Specifically, only the SP-RT group showed greater enhancement of RFD in response to the startling stimulus relative to non-startling stimuli (VnSS and VS), whereas no stimulus-specific modulation was observed in the MP-RT or Control groups. These stimulus comparisons alongside Group and Time are now clearly presented in the revised Results (**Page 31 and 32, Lines 737-759**), in Table 6 (**Table 6A: rows 7 and 8; Table 6B: rows 7 and 8**) and are interpreted in the revised Discussion (**Page 38, Lines 900-906**). The corresponding Group \times Type of sound \times Time interaction for RFD during the 50–100 ms interval was not statistically significant and is also reported in Results section (**Page 32 Lines 757-758**). Accordingly, the revised analysis directly addresses the reviewer's concern by formally testing the Group \times Type of sound \times Time interaction. For ease of reference, the revised Results and Discussion sections are reproduced below in response to **Specific Comment 3**".

With respect to model complexity, we agree that a four-way interaction (Group \times Types of sound \times Time \times Interval) is theoretically possible. However, the 0–50 ms and 50–100 ms RFD epochs represent physiologically distinct phases of force development and were defined *a priori* as separate outcomes. The earliest interval (0–50 ms) is most relevant for probing rapid, stimulus-evoked motor output and reticulospinal contributions, whereas the later interval reflects additional contribution of non-neural neuromuscular processes, such as e.g. fascicle shortening velocity and tendon stiffness. Treating interval as an additional repeated factor and modelling a four-way interaction would therefore not provide physiologically meaningful inference and could obscure interval-specific effects. For this reason, the intervals were analysed separately.

We acknowledge the reviewer's concern regarding statistical power for higher-order interactions. No predetermined effect-size estimates exist for a Group \times Types of sound \times Time interaction involving RFD, particularly in the context of reticulospinal function, as this represents a novel application. Accordingly, the study was not powered *a priori* to maximise sensitivity for this specific interaction. Importantly, the StartReact effect was the primary and established indicator of reticulospinal excitability, and the power analysis was conducted using a Group \times Time interaction framework based on a comparable interaction effect size relevant to this primary outcome. RFD was analysed as a supportive and supplementary measure, intended to provide convergent behavioural evidence alongside the StartReact effect rather than serving as an independent primary endpoint.

While mixed-effects models leverage within-subject comparisons across stimulus conditions and therefore improve statistical efficiency relative to purely between-subject designs, we have interpreted the RFD findings conservatively. To clarify this for readers, we have explicitly acknowledged this limitation in the manuscript (**Page 41 and 42, Lines 980–986**), stating:

“Although the study was powered a priori to detect training-related changes in reticulospinal excitability using a Group × Time interaction framework, with the StartReact effect as the primary indicator, it was not specifically powered to maximise sensitivity for higher-order Group × Time × Stimulus interactions involving RFD, as no prior effect-size estimates exist for such designs. Accordingly, future studies specifically powered to detect higher-order interactions will be important to confirm and extend these findings.”

Comment 5. The responses detail an outcome variable structure, which is logically conceptualised. However, this conceptual framework does not come through with respect to specific measures at present in the introduction of the manuscript, with variables from across domains typically discussed together, and as noted above, critically no specific variables referred to in the aims and hypotheses.

Response:

We thank the reviewer for this constructive comment. We agree that, in the earlier version of the manuscript, the conceptual framework underpinning the outcome variable structure was not sufficiently explicit in the Introduction, with measures from different neural domains discussed together and not clearly mapped onto the stated aims and hypotheses.

To address this, we have revised the Introduction to explicitly link the conceptual framework to specific, named outcome measures within the aims and hypotheses (Introduction; **Page 7, Lines 165-170**). The revised aims now clearly define two objectives and read:

“...Accordingly, the present study aimed to determine the effect of resistance training on RST excitability in humans, indexed by modulation of the StartReact effect and corresponding changes in RFD. The secondary aim was to examine whether MP-RT and SP-RT differentially modulate cortico-reticular (ipsilateral to contralateral motor-evoked potential amplitude ratio (ICAR), intracortical (SICI and intracortical facilitation [ICF]), and corticospinal (active motor threshold [AMT], CSE and cSP) excitability”.

The hypotheses have been correspondingly revised to explicitly state the expected direction of change for these specific measures, namely preferential enhancement of cortico-reticular and reticulospinal outcomes following self-paced resistance training, and predominant modulation of corticospinal measures following metronome-paced training. These revisions ensure that the conceptual framework is now clearly articulated through explicitly named outcome variables, with each measure directly aligned to a defined neural pathway and hypothesis.

Comment 6. A long list of outcomes variables whether selected arbitrarily or according to a conceptual framework is still a long list of variables. The justification of some these variables individually is quite weak and several of the retained variables are not mentioned in the

introduction. The responses describe 'primary outcome variables' (plural), but what these are is not stated in the manuscript. Moreover, most well conceptualised studies have a singular primary outcome variable - in the case of the current study this remains unclear.

Response:

We thank the reviewer for this important comment regarding outcome number, justification, and outcome hierarchy. We agree that, in the earlier version of the manuscript, the distinction between primary and secondary outcomes and the rationale for individual measures were not articulated with sufficient clarity.

The present study is centred on a single primary objective: to determine whether reticulospinal tract (RST) involvement contributes to early strength adaptation following resistance training, particularly in the context of different training modalities. Consistent with this objective, the study was designed around a single primary outcome domain, namely reticulospinal system involvement in early strength adaptation, with one primary outcome variable: the StartReact effect supported by early-phase RFD as a convergent behavioural index of reticulospinal output. This primary outcome is now clearly stated in the Introduction and aims/hypotheses (*Page7, Lines 165-170*). We acknowledge that referring to “primary outcome variables” in the plural was imprecise, and we have corrected to reflect a single primary outcome.

All remaining neurophysiological variables are explicitly designated as secondary, mechanistic outcomes. Their inclusion is not arbitrary, nor intended to increase the breadth of hypothesis testing, but is necessary to contextualise the primary RST findings within the broader and unresolved literature on the neural locus of early strength adaptation. As outlined in the revised Introduction, previous studies, particularly those using self-paced resistance training have demonstrated strength gains in the absence of consistent changes in corticospinal excitability, suggesting that non-corticospinal pathways may mediate these adaptations. This uncertainty in the site of neural adaptation, and its apparent dependence on training modality, provides the physiological rationale for concurrently assessing cortical (SICI, ICF), corticospinal (AMT, CSE, cSP), and cortico-reticular (ICAR) measures.

To address the reviewer’s concern directly, we have revised the manuscript to ensure that all retained outcome variables are explicitly introduced and justified in the Introduction, each linked to a specific neural domain and mechanistic role.

Finally, we have revised the manuscript throughout (from Abstract to Conclusion) to reflect this hierarchy, presenting findings related to the primary RST objective first, followed by secondary outcomes that provide mechanistic context. These revisions clarify the study focus, establish a coherent outcome hierarchy, and ensure that the number and selection of outcome measures are aligned with the central mechanistic question rather than reflecting exploratory breadth.

Comment 7. The authors provided a long list of references to justify their sample size. Of course there are studies in the literature which used similar or smaller group sizes, as sample size is largely dependent on study design. I checked only the first 6 references: 4 involved

only a single group of participants i.e. no between group comparison at all and thus of minimal relevance; only one of the six studies involved an exercise intervention with two groups and two time points. In contrast the current study therefore has substantially more degrees of freedom (3 groups x 4 time points) reducing its statistical power more than any of these previous investigations. In summary this justification is flawed.

Response: We thank the reviewer for this careful evaluation and agree that sample size adequacy is inherently dependent on study design complexity. We also agree that direct numerical comparisons of sample size across studies with substantially different designs (e.g. single-group or two-time-point studies) are not sufficient, on their own, to justify sample size adequacy in the present multi-group, multi-time-point design.

The intention of citing previous studies was not to suggest that comparable sample sizes are automatically appropriate across designs of differing constraints. Rather, these references were included to contextualise the practical constraints and prevailing sample sizes within human neurophysiology and resistance-training research, including studies published in *Journal of Physiology* and other high-quality journals, where intensive neurophysiological protocols, repeated testing, and complex experimental paradigms frequently limit larger sample sizes.

Importantly, we fully acknowledge, as the reviewer correctly notes, that the present study involves greater degrees of freedom (3 groups \times 4 time points), which places higher demands on statistical power than simpler designs. For this reason, we did not rely on comparisons with prior studies as the primary justification for sample size adequacy. Instead, we have adopted a statistically appropriate analytical framework tailored to the complexity of the design, including linear mixed-effects modelling, which efficiently handles repeated measures, and hierarchical variance structures, and is more powerful and flexible than traditional repeated-measures ANOVA approaches commonly used in earlier works.

Moreover, as detailed in our revised sample size justification, we have grounded the power contextualisation in an interaction-based effect size derived from the closest available multi-group, multi-time-point study, rather than from simpler single-group or pre-post designs. This revised approach directly addresses the concern regarding design complexity by grounding the power justification in a study with a comparable multi-group, multi-time-point interaction structure.

Specific comments

Comment 1. L637 Repeated use of separate LMM as performed in this experiment, suffer from the same issues as repeated large-scale use of any test, and arguably multivariate LMM would be a better approach.

Response:

We thank the reviewer for this thoughtful methodological suggestion. We agree that, in principle, multivariate linear mixed models can be advantageous when analysing multiple correlated outcomes simultaneously. However, we respectfully submit that the use of

separate, pre-specified linear mixed-effects models (LMMs) is methodologically appropriate and justified for the aims, structure, and inferential framework of the present study.

First, the outcome variables examined here represent distinct neurophysiological constructs arising from different levels of the motor system (intracortical, corticospinal, cortico-reticular, and reticulospinal). These measures differ substantially in scale, distributional properties, and physiological interpretation (paired-pulse TMS ratios, MEP amplitudes, CSP durations, StartReact latencies, RFD intervals). Modelling them jointly in a single multivariate framework would therefore conflate mechanistically distinct processes and complicate interpretation, without necessarily yielding clearer inference regarding pathway-specific adaptations.

Second, the primary inferential interest of the study lies in outcome-specific group \times time interactions, rather than in estimating covariance structures between outcomes. Separate LMMs directly target these interaction effects while appropriately accounting for within-subject correlation via random effects.

Third, multivariate LMMs impose substantially greater demands on sample size and model stability, particularly when estimating cross-outcome covariance matrices in designs with multiple groups and repeated measures. Given the modest sample sizes typical of invasive and technically demanding neurophysiological studies, such models can be underpowered, numerically unstable, or overly sensitive to modelling assumptions, potentially reducing rather than enhancing inferential reliability.

Importantly, the concerns associated with repeated testing across separate models were explicitly addressed through pre-specification of outcomes, clear distinction between primary and secondary measures, and appropriate control of type I error. As such, the use of separate LMMs does not constitute unstructured multiple testing, but rather a set of theory-driven, outcome-specific analyses, each aligned with a defined hypothesis.

For these reasons, while we acknowledge the theoretical appeal of multivariate LMMs, we believe that the chosen approach of separate, hypothesis-driven LMMs is statistically sound, physiologically interpretable, and consistent with best practice for complex longitudinal neurophysiological data.

Comment 2. L712 "Test-retest reliability within the control group was generally high" with CV>20% for 4 measures and ICC<0.80 for 6 measures this seems overstated.

Response:

We thank the reviewer for this important observation. We agree that the phrase “generally high” may overstate the reliability profile. In response, we have revised the wording in the manuscript to provide a more accurate and nuanced description of the reliability findings. The revised sentence now reads (**Page 29, lines 692-694**):

“Test-retest reliability within the control group varied across measures, with several outcomes demonstrating good-to-excellent reliability ($ICC \geq 0.80$), whereas others showed moderate reliability based on ICC and CV estimates.”

Comment 3. L815 and throughout the Discussion. RST effects are about how startling stimuli compare to other non-startling stimuli. This point requires direct comparison of stimuli alongside group and time, which is missing in the analysis and interpretation of the current findings. Thus the apparent increase in RFD with VSS of the SP-RT vs other groups can only be attributed to enhanced RST activation after training if it is greater than for other stimuli i.e. a group x time x stimuli (condition) interaction. However, this has not been calculated. The greater improvement in RFD of one group with one stimuli analysed in isolation does not inform changes in the RST and could be a training effect specific to that group that could occur with all stimuli. Only a 3-way interaction can test for a RST (differential stimuli) effect on RFD of different groups over time. The current study is not powered for this level of complexity.

Response:

We thank the reviewer for this important and constructive set of comments. As outlined in our response to Major Comment 4, we have revised both the analysis and interpretation to ensure that stimulus-specific effects are evaluated through direct comparison of stimuli alongside Group and Time. Specifically, we have now explicitly included the Group \times Types of sound \times Time interaction in the mixed-effects analysis. The revised analysis demonstrates that changes in RFD (0–50 ms) across time differed between groups as a function of stimulus condition. Importantly, stimulus effects were evaluated within this three-way interaction framework rather than as isolated stimulus-specific analyses. Accordingly, interpretation of enhanced RST involvement is now based exclusively on differential stimulus effects supported by the Group \times Types of sound \times Time interaction. The revised Results section now reads (**Page 31 and 32, Lines 737-759**):

“Analysis of RFD during the first 50 ms revealed a significant Group \times Type of Sound \times Time interaction ($\chi^2(12) = 25.20, p = 0.0179, \eta^2G = 0.052, 95\% CI [0.023, 0.081]$).....Within-group interaction contrasts demonstrated that in the SP-RT group the increase in RFD from baseline was greater for VSS compared with both VnSS and VS at Week 1 (VSS vs VnSS: $EMD = 0.64, SE = 0.18, z = 3.53, p < 0.001$; VSS vs VS: $EMD = 0.61, SE = 0.18, z = 3.34, p = 0.0017$), Week 2 ($EMD = 0.88, SE = 0.18, z = 4.83, p < 0.001$; $EMD = 0.60, SE = 0.18, z = 3.28, p = 0.002$), and Week 3 ($EMD = 0.76, SE = 0.18, z = 4.15, p < 0.001$; $EMD = 0.67, SE = 0.18, z = 3.70, p < 0.001$). No stimulus-specific differences were detected within the MP-RT or Control groups at any time point (all $p \geq 0.378$).

Between-group interaction contrasts showed that for the $\Delta VSS - \Delta VnSS$ comparison, SP-RT exhibited greater enhancement than Control at Week 2 ($EMD = 0.80, SE = 0.26, z = 3.11, p = 0.00560$) and Week 3 ($EMD = 0.68, SE = 0.26, z = 2.63, p = 0.0256$), and greater enhancement than MP-RT at Week 2 ($EMD = 0.68, SE = 0.26, z = 2.64, p = 0.0246$). For the $\Delta VSS - \Delta VS$ contrast at Week 3, SP-RT also demonstrated greater enhancement than Control ($EMD = 0.63, SE = 0.26, z = 2.44, p = 0.0448$) and MP-RT ($EMD = 0.62, SE = 0.26, z = 2.43, p = 0.0453$). No other between-group differences in stimulus-specific change in RFD were observed (all $p \geq 0.0659$) (Table 4).

Analysis of RFD during the 50–100 ms interval did not reveal a significant Group × Time × Type of Sound interaction ($\chi^2(12) = 8.12, p = 0.776, \eta^2G = 0.06, 95\% CI [0.00, 0.12]$).....”

The Discussion section has likewise been revised (**Page 38, Lines 900–907**) to ensure that interpretation of the RFD findings is grounded in direct comparison of stimuli alongside Group and Time within the three-way interaction framework. The results of this interaction-based analysis, directly comparing startling and non-startling stimuli across groups and time, are reported in Table 6 (**Page 64, Table 6A: rows 7 and 8; Table 6B: rows 7 and 8**).

We acknowledge the reviewer’s concern regarding statistical power for this level of complexity. While the study was not powered *a priori* specifically to maximise sensitivity for higher-order Group × Types of sound × Time interactions involving RFD, the StartReact effect was the predefined primary indicator of reticulospinal excitability, and the power analysis was conducted using a Group × Time interaction framework relevant to this primary endpoint. RFD was analysed as a complementary behavioural measure intended to provide convergent evidence alongside the primary StartReact findings. We have also explicitly acknowledged this limitation in the manuscript (**Page 41 and 42, Lines 980–986**) to ensure transparency for readers.

Considering the reviewer’s emphasis on direct evaluation of the Group × Types of sound × Time interaction, we have removed the previously included RFD graphs. These figures presented stimulus conditions separately and therefore did not adequately reflect the three-way interaction framework required to support inference regarding reticulospinal involvement. Given the complexity of the interaction, it is not feasible to meaningfully represent the full Group × Types of sound × Time structure within a single figure without compromising clarity or risking misinterpretation. Accordingly, we now present the interaction-based results directly in Table 6 (Table 6A: rows 7 and 8 for within-group contrasts; Table 6B: rows 7 and 8 for between-group contrasts), which clearly report stimulus comparisons alongside Group and Time within the three-way interaction model.

Referee #3:

The authors have satisfactorily addressed all previously raised concerns. Substantial and meaningful clarifications have been incorporated into the Methods, Results, and Discussion sections, thereby strengthening the interpretability of the study. In other cases, the authors reasoned well their case in the point-by-point response. Overall, the revisions have significantly improved the manuscript. The study now provides a solid contribution to the field, and I do not identify any further issues that require attention. My congratulations to the authors for their careful and constructive revisions.

Response:

We sincerely thank Referee #3 for their careful evaluation and generous comments. We greatly appreciate their constructive feedback, which has substantially contributed to improving the clarity, interpretability, and overall quality of the manuscript. We are grateful for their assessment that the revisions have strengthened the study and that no further issues require attention.

Finally, we sincerely thank the Referees and Editors for their careful evaluation and highly constructive feedback, which has substantially strengthened the manuscript. We are grateful for the opportunity to revise and further improve this work.

The Authors

Dear Associate Professor Kidgell,

Re: JP-RP-2026-289141R3 "**Resistance Training Tempo Selectively Modulates Corticospinal and Reticulospinal Excitability in Humans**" by Yonas Akalu, Jamie Tallent, Ashlyn K Frazer, Ummatul Siddique, Mohamad Rostami, Glyn Howatson, Simon Walker, and Dawson J Kidgell

We are pleased to tell you that your paper has been accepted for publication in The Journal of Physiology.

Yours sincerely,

Richard Carson
Senior Editor
The Journal of Physiology

IMPORTANT POINTS TO NOTE FOLLOWING ACCEPTANCE OF YOUR PAPER:

- **IMPORTANT NOTICE ABOUT OPEN ACCESS:** To assist authors whose funding agencies mandate immediate public access to published research findings, The Journal of Physiology allows authors to pay an Open Access (OA) fee to have their papers made freely available immediately on publication.

The Corresponding Author will receive an email from Wiley with details on how to register or log in to Wiley Authors where you will be able to place an order.

- You can check if your funder or institution has a Wiley Open Access Account here:
<https://authors.wiley.com/author-resources/Journal-Authors/open-access/author-compliance-tool.html>

- You can help your research get the attention it deserves! Check out Wiley's free Promotion Guide for best-practice recommendations for promoting your work at: www.wileyauthors.com/eoo/guide. You can learn more about Wiley Editing Services which offers professional video, design, and writing services to create shareable video abstracts, infographics, conference posters, lay summaries, and research news stories for your research at: www.wileyauthors.com/eoo/promotion.

- If you would like to receive our 'Research Roundup', a monthly newsletter highlighting the cutting-edge research published in The Physiological Society's family of journals (The Journal of Physiology, Experimental Physiology, Physiological Reports, The Journal of Nutritional Physiology and The Journal of Precision Medicine: Health and Disease), please click this link, fill in your name and email address and select 'Research Roundup':
<https://www.physoc.org/journals-and-media/membernews>

EDITOR COMMENTS

Reviewing Editor:
The authors have satisfactorily addressed the comments made by the senior editor.